# An intralayer microcircuit in the temporal association cortex underlies sensory-induced escape in mice

He Li [1,2,3,4,7], Jiajia Chen [1,2,3,4,7], Wen Zhong [5,8] ✉, Na Lian[1,2,3,4], Yumei Huang[1,2,3,4], Linhui Yao[1,2,3,4], Peiran Yin[5], Ziyi Xu[1,2,3,4], Xiaoxia Qin[6], Jie Tan[5], Yingying Zeng[5], Jinhua Liu[6] & Zhongju Xiao [1,2,3,4,6,8] ✉

A central goal in neuroscience is to clarify how neural circuits translate sensory input into adaptive behaviours. Although unisensory evoked escape circuits in mice are well defined, it remains unclear whether a single nucleus contains specialized sensory, sensory–motor decision, and motor command neurons for escapes driven by distinct sensory cues, and how these neurons form functional microcircuits. Using multiple sensory stimuli in mice, we identified the temporal association cortex (TeA) as a critical escape hub. Combining in vivo electrophysiology, optogenetics and chemogenetics, we characterized three distinct neuron subtypes within TeA layer 5 (L5) CaMKII neurons that correspond to these three functional classes. Intratelencephalic (IT) neurons serve as sensory–motor decision neurons, while layer matched pyramidal tract (PT) neurons projecting to the dorsal periaqueductal grey (dPAG) act as motor command neurons. We reveal a laminar IT–PT microcircuit that converts sensory input into sensory-motor decisions and commands for escape locomotion.

Animals' perceptions of their external and internal environments determine and modulate their behaviours, as evidenced by studies exploring sensory modalities such as auditory[1–6], visual[7–11], olfactory[12], and somatosensory stimuli[4,5], all of which can elicit innate flight responses such as escape, which are essential for animals to survive in the face of life-threatening environmental cues[13]. A critical function of neural centres is to establish neuronal circuits that perceive these cues, process them as sensory signals, make decisions, transform these decisions into motor commands, and subsequently initiate appropriate defensive behaviours[14]. The neuronal circuits in mice that underlie the detection of and escape from threats have been

extensively investigated recently. However, whether a specific neural nucleus acts as a central hub to coordinate cross-sensory escape behaviours remains unclear. Such a nucleus might integrate inputs from sensory and decision-making neurons to orchestrate motor commands, potentially within a localized microcircuit.

A widely accepted hypothesis is that sensory regions process sensory information[1,3]; motor regions generate motor signals (e.g., in circuits related to innate escape behaviours, which ultimately project to the deep periaqueductal grey (dPAG)[1,3,15]; and higher-order cortical areas (such as the association cortex), striatum, superior colliculus, and various midbrain regions generate decision-related information,

[1]Department of Physiology, School of Basic Medical Sciences, Southern Medical University, Guangzhou, Guangdong, China. [2]Key Laboratory of Psychiatric Disorders of Guangdong Province, Southern Medical University, Guangzhou, Guangdong, China. [3]Guangdong-Hong Kong-Macao Greater Bay Area Center for Brain Science and Brain-Inspired Intelligence, Southern Medical University, Guangzhou, Guangdong, China. [4]Key Laboratory of Mental Health of the Ministry of Education, Southern Medical University, Guangzhou, Guangdong, China. [5]School of Traditional Chinese Medicine, Guangdong Basic Research Center of Excellence for Integrated Traditional and Western Medicine for Qingzhi Diseases, Southern Medical University, Guangzhou, Guangdong, China. [6]The Seventh Affiliated Hospital, Southern Medical University, Foshan, Guangdong, China. [7]These authors contributed equally: He Li, Jiajia Chen. [8]These authors jointly supervised this work: Wen Zhong, Zhongju Xiao. ✉e-mail: zhong1981@smu.edu.cn; xiaozj@smu.edu.cn

facilitating the transformation of sensory signals into motor signals[8,16–20]. However, sensory regions also encode decision-making and reward-related information[21–24], and motor-related signals have been detected in nonmotor-related nuclei[25–30]. These observations suggest that sensory neurons, sensory–motor decision neurons, and motor command neurons capable of directly converting sensory input into motor output may all be present within some nuclei of the mouse brain. As a region that receives sensory signals and connects to motor-related areas, the association cortex appears to be the most likely candidate for fulfilling this function.

The temporal association cortex (TeA), which is part of the association cortex, is a high-level nucleus that receives diverse sensory inputs[31], processing auditory[32], visual[33–35], and tactile[36] signals through projections from different sensory nuclei and enables the integration of multimodal sensory information[37–39]. Previous research has revealed the role of the TeA in processing auditory signals[40] and influencing both innate[41] and learned behaviours[40,42]. Moreover, the TeA sends projections to the dPAG, a region that controls defensive behaviours[31,41]. While emerging evidence has implicated the TeA in behavioural responses such as the retrieval of pups in maternal mice (an innate care behaviour)[41] and auditory-cued fear conditioning (learned behaviour)[40,42], its specific role in the defensive behaviour circuit remains poorly defined. This critical knowledge gap, coupled with the robust anatomical connectivity of the TeA to key command neurons, renders this region a highly compelling and potentially pivotal node worthy of in-depth investigation. We hypothesize that the TeA likely serves not only as a simple relay station but also as a critical hub coordinating diverse sensory-induced escape behaviours. It may integrate sensory information and participate in driving downstream motor command pathways, constituting a microcomputational hub that encompasses sensory integration, decision-making, and motor command functions. It plays an indispensable role in triggering adaptive escape behaviours.

In a mouse model of flight behaviours driven by sound, light, and air puffs, along with a combination of in vivo and in vitro electrophysiological recordings and chemogenetic and optogenetic interventions, our study elucidated the contribution of the TeA–dPAG pathway to sensory-induced escape behaviours and characterized the features of neurons within the TeA nucleus that are responsible for sensory response, sensorimotor decision-making, and motor command. Furthermore, information on diverse sensory stimuli is integrated into layer 5 (L5) intratelencephalic (IT) neurons as running signals; these IT neurons activate L5a critical pyramidal tract (PT) neurons, which project to the dPAG, and, together, these neurons form an intralayer IT–PT microcircuit for sensory–motor decision-making and inducing running behaviours.

## Results

### The TeA is necessary for sensory-induced escape

We explored the stimulus conditions required to induce escape behaviours in mice exposed to three types of aversive sensory stimuli—sound, light, and air puffs—in an open-field escape model. The mice freely explored a custom, opaque acrylic box with side holes for the speaker, an LED, and a silicone tube, as well as central openings to prevent prolonged stillness. When the mice were located in one compartment of the experimental chamber, the random application of sound, light, or air puff stimuli at their current position invariably caused them to move away from the stimulus source to the opposite compartment, thus resulting in escape behaviour. No measurable freezing behaviour was detected either during stimulus presentation or during the execution of escape movements[1] (Fig. 1a). Such escape behaviours were detected in all the experimental mice, whose running trajectories were recorded with a high-definition infrared camera positioned above the box. The data were used to create running speed heatmaps for each stimulus (Fig. 1b) and average traces for each

unisensory stimulus (Fig. 1c). The sensory stimulus robustly evoked escape behaviour. The latency of running initiation following stimulation was highly temporally locked ($1.84 \pm 0.36$ s), in stark contrast to the broadly distributed latencies of spontaneous running during the baseline period (Fig. 1d). This pronounced difference in distribution was quantified by comparing their cumulative distribution functions (CDFs) and was statistically confirmed (Fig. 1e). Based on this clear separation, we applied a conservative response window of [0, 2.4 s] poststimulus to classify evoked events (see Methods).

Among the stimuli, sound was the most effective at inducing flight behaviours, followed by air puffs and light (Fig. 1f). The latency between the onset of the sound stimulus and the flight behaviours was the shortest (Fig. 1g). The stimulation intensity of the sound, light, or air puffs was selected to evoke similar peak flight speeds across the three stimuli (Fig. 1h). The flight duration following sound stimulation was the shortest, followed by that resulting from light and air puff stimulation (Fig. 1i).

We transitioned from the open-field model to a head-fixed, turntable running model to facilitate interventions in the TeA, observe corresponding behavioural changes in the mice, and record neuronal responses. This model also allowed the running speeds of the mice to be recorded and a more detailed analysis of their movement characteristics in subsequent studies (Fig. 1j). After training, the head-fixed mice (different from those used in the open-field escape model) were able to freely walk or run on a turntable. An optical velocity sensor (blue module) placed beneath the disc recorded the rotational speed, which was used to calculate the running speed of the mice. The stimuli (sound, light, and air puffs) were delivered 10 cm in front of the mice, with the type of sensory stimulus pseudorandomly selected to minimize the variability in mouse movement induction (Fig. 1j). Example speed heatmaps of mouse flight induced by the three types of stimuli (Fig. 1k, left panel) showed significant differences after classification (Fig. 1k, right panel), which are more clearly observed in the averaged traces (Fig. 1l). The sensory stimulus elicited a highly time-locked escape response. The distribution of poststimulus running latencies differed significantly from the broad distribution of spontaneous running behaviours at baseline (Fig. 1m), confirming the evoked nature of the behaviour. The poststimulus responses were tightly concentrated, as reflected in the steep CDF (Fig. 1m, insert). Applying the objectively defined response window [0, 1.7 s], we found that the stimulus successfully evoked running in a significant majority of trials.

Among the stimuli, sound was the most effective at inducing flight behaviours, followed by air puffs and light (Fig. 1n). The latency between the onset of the sound stimulation and the flight behaviours was the shortest (Fig. 1o), whereas the peak flight speeds induced by sound, light, and air puffs were not significantly different (Fig. 1p). Moreover, the flight duration in response to sound stimulation was the shortest, followed by that in response to light and air puff stimulation (Fig. 1q). A comparative analysis of locomotor parameters, including probability, latency, peak speed, and duration, between the open-field running paradigm and the turntable-induced running model revealed convergent movement patterns in the mice, suggesting that distinct sensory modalities elicit fundamentally similar kinematic signatures during stimulus-evoked flight.

We bilaterally injected the adeno-associated virus (AAV)-hSyn-hM4D(Gi)-mCherry into the TeA, driving the expression of the inhibitory muscarinic designer receptor hM4D(Gi) in local cells to validate our hypothesis that the TeA is critical for encoding flight behaviours triggered by diverse sensory stimuli. Intraperitoneal (i.p.) injections of clozapine-N-oxide (CNO) activated hM4D(Gi) and inhibited neuronal firing[43–46] (Fig. 1r), and sound-, light-, and air puff-induced flight behaviours were completely blocked (Fig. 1s). In contrast, i.p. injections of CNO had a minimal effect on control animals expressing only AAV-hSyn-mCherry without hM4D(Gi) (Fig. 1t). Second, since CNO i.p. injections still represent a chemical intervention, optogenetic

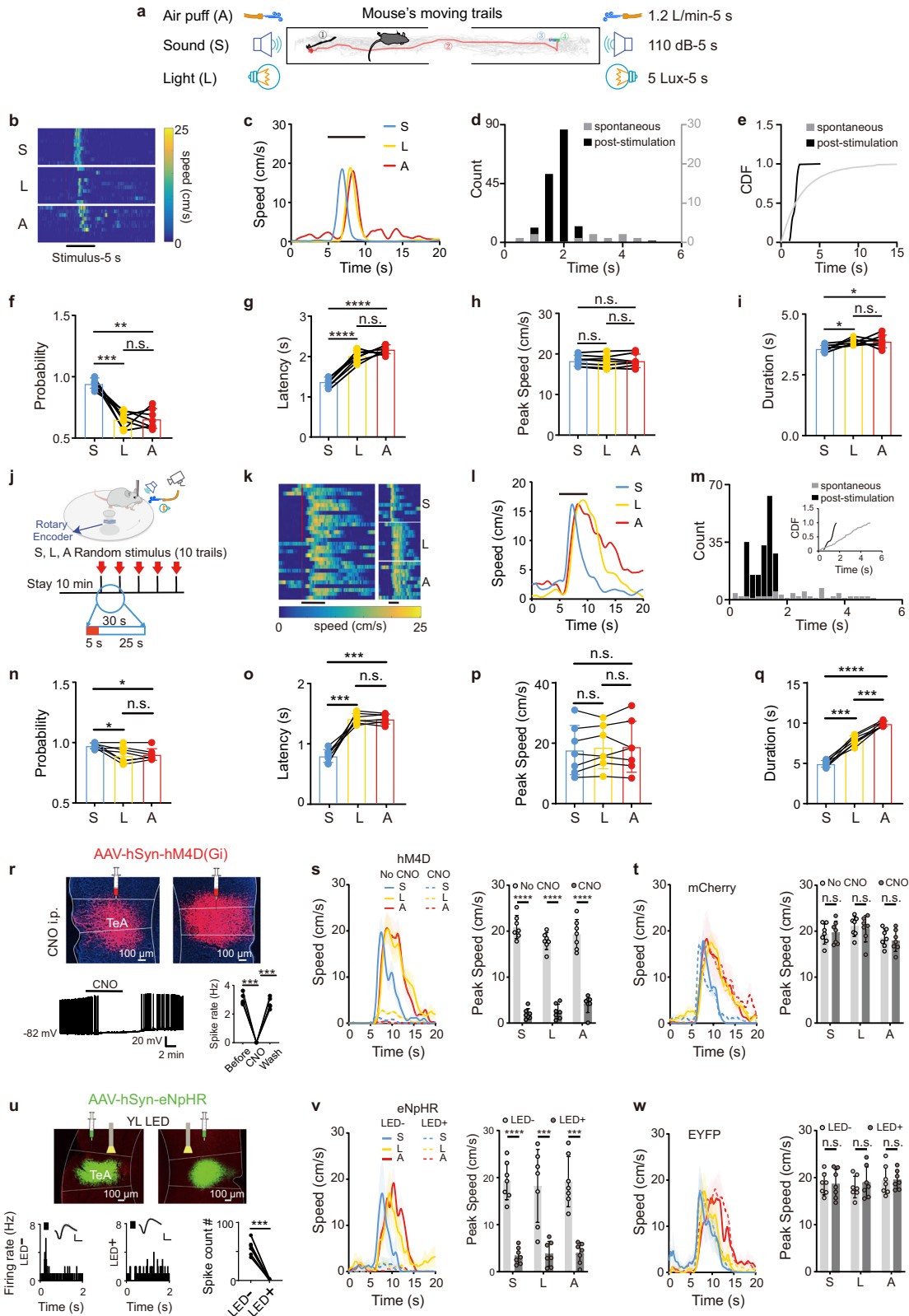

interventions were performed to further confirm our findings. We performed bilateral injections of AAV-hSyn-eNpHR3.0-EYFP into the TeA to broadly express eNpHR3.0. Yellow light was then applied to suppress the firing activity of individual TeA cells (Fig. 1u). The bilateral application of yellow light effectively blocked the flight behaviours driven by the three sensory stimuli (Fig. 1v), whereas no blockade was

observed in the control group injected with the hSyn-EYFP virus (Fig. 1w).

The results from chemogenetic and optogenetic interventions collectively confirm that the TeA is a critical node in the neural pathway mediating flight behaviours induced by diverse sensory stimuli.

**Fig. 1 | The TeA controls flight behaviours. a** Schematic of the open-field model of flight induced by S, L, or A stimuli. **b** Example flight speed heatmap. **c** Representative plot of the average flight speed (10 trials). **d** Intervals for spontaneous running and response latencies for evoked running. $P < 0.001$, Two-sided Kolmogorov–Smirnov test. **e** Latency CDFs. **f–i** Probability (**f**), Latency (**g**), Peak speed (**h**), and Duration (**i**) of flight. ($n = 7$ animals. $*P < 0.05$, $**P < 0.01$, $***P < 0.001$, $****P < 0.0001$, one-way RM ANOVA with Bonferroni post hoc correction). **j** Turntable running model. **k** Example speed heatmaps to randomly interleaved (left) or grouped (right) stimuli. **l** Representative plot of average flight speed (10 trials). **m** The same as **d** and **e**. $P < 0.001$, Two-sided Kolmogorov–Smirnov test. **n–q** Same as (**f–i**) but for turntable running model. ($n = 7$ animals. $*P < 0.05$, $***P < 0.001$, one-way RM ANOVA with Bonferroni post hoc correction). **r** Top, bilateral TeA viral expression. Bottom left, raw traces of current-clamp recordings

from an hM4D(Gi)-expressing TeA cell in a slice preparation. Bottom right, CNO reversibly suppresses spontaneous firing. ($n = 5$ cells from 2 mice. $***P < 0.001$, one-way RM ANOVA with Bonferroni post hoc correction). **s, t** Left, example flight speed traces before (CNO-) and after CNO (i.p., CNO + ) in hM4D (**s**) or mCherry (**t**)-expressing animals. Right, peak speed induced by S, L, and A before and after the CNO injection. ($n = 7$ animals each group, $****P < 0.0001$, two-sided Paired t-test). **u** Bilateral TeA viral expression (top) and PSTHs of a TeA neuron to noise (80 dB SPL) without/with yellow light LED (bottom). Calibration: 50 μV, 0.5 ms. ($n = 7$ cells from 7 mice, $***P < 0.001$, two-sided Paired t-test). **v, w** As in (**s**) and (**t**), but for animals expressing eNpHR (**v**) or EYFP (**w**). ($n = 7$ animals each group, $***P < 0.001$, $****P < 0.0001$, two-sided Paired t-test). Data are presented as mean ± SD. (Source data are provided as a Source Data file; see Supplementary Data 1 for detailed statistics).

## The TeA–dPAG circuit contributes to both sensory-evoked and spontaneous running

Previous studies have established an association between the dPAG and flight behaviours[15]. We speculate that the inhibition of TeA neurons, which blocks the flight behaviours driven by the three sensory stimuli described above, may act through projections from the TeA to the dPAG.

Although previous studies have shown that TeA neurons project to the dPAG[31,41], the details remain largely unknown. We characterized these projections further by injecting AAV-hSyn-EYFP into the TeA and observed high levels of green fluorescence in dPAG fibres (Fig. 2a). We next confirmed the projections of the TeA to the dPAG by injecting AAVretro-hSyn-mCherry into the dPAG, which retrogradely labelled TeA cells specifically in layer 5a (L5a) (Fig. 2b). These cells projected to the dPAG (TeA$_{dPAG}$) from the rostral to caudal direction (Fig. 2c), and the number of cells exhibited a Gaussian distribution (Fig. 2d). Different types of inhibitory neurons in the TeA[47], such as somatostatin (SOM), vasoactive intestinal peptide (VIP), and parvalbumin (PV) neurons, are involved in processing sensory stimuli. We injected AAVretro-hSyn-EYFP into the dPAG of wild-type mice along with AAVretro-CaMKIIα (glutamatergic)-mCherry, as well as into the dPAG of *SOM-, VIP-,* and *PV-Cre* mice crossed with *Ai14* mice to determine the specific neurons projecting to the dPAG (Supplementary Fig. 1a). The majority of TeA L5a neurons projecting to the dPAG were CaMKII-positive, with a colocalization rate of 72.4%. In contrast, TeA$_{dPAG}$ cells did not significantly overlap with inhibitory neurons (0.65% SOM-positive, 0.37% VIP-positive, and 0% PV-positive) (Supplementary Fig. 1b and c). Notably, the TeA region lacked PV neurons. As noted in previous studies[48], the *PV-IRES-Cre* mouse line exhibits low efficiency in labelling PV-INs in the association cortex, which may be a common characteristic across regions in the association cortex.

We determined where the dPAG neurons receiving projections from the TeA ($_{TeA}$dPAG neurons) are distributed by injecting AAV-Flp into the TeA and AAV-fDIO-EYFP into the dPAG to label $_{TeA}$dPAG neurons (Fig. 2e). The $_{TeA}$dPAG neurons were distributed in all regions of the dPAG from the rostral end to the caudal end, and their numbers similarly followed a Gaussian distribution (Fig. 2e, f). As an approach to further clarify their types, we used the Flp + fDIO virus to label the $_{TeA}$dPAG neurons of wild-type mice and three transgenic mouse strains in which different inhibitory neurons (SOM, VIP, and PV) are labelled (Supplementary Fig. 1d). The $_{TeA}$dPAG neurons were mainly CaMKII-positive (colocalization rate with CaMKII of 58%). A small percentage (5%) were SOM cells, whereas no VIP (0%) or PV (0%) neurons were observed (Supplementary Fig. 1e and f). The low proportion of inhibitory neurons aligns with the sparse distribution of these neurons in the dPAG. SOM inhibitory neurons were the most abundant among the three inhibitory neurons and were distributed primarily in the dorsomedial and intermediate parts of the dPAG (Supplementary Fig. 1g), whereas fewer VIP inhibitory neurons were scattered throughout the

entire dPAG (Supplementary Fig. 1h), and PV inhibitory neurons were not present (Supplementary Fig. 1i).

As described above, the $_{TeA}$dPAG cells were primarily CaMKII-positive, while a small percentage were inhibitory cells (Supplementary Fig. 1f). We locally injected Cre (AAV-Cre-EYFP, TeA) + DIO (AAV-DIO-CaMKIIα-hM4D(Gi)-mCherry, dPAG) to specifically inhibit CaMKII-positive $_{TeA}$dPAG neurons (Fig. 3a). Subsequent i.p. injections of CNO specifically activated hM4D expression in CaMKII-positive dPAG cells (Fig. 3b) and blocked the flight behaviours of the mice. Similarly, no blockade was observed in the control group injected with the virus lacking hM4D (Fig. 3c, d).

Unsurprisingly, blocking neurons in the dPAG inhibited escape behaviours, as the PAG is a central hub for controlling defensive behaviours[8]; specifically, the dPAG contains "flight" neurons that directly control defensive running behaviours[49,50]. However, the cerebral cortex, particularly layer 5, is capable of monitoring the activity of neurons associated with running[26]. Direct activation of cortical layer 5 neurons can drive running behaviours in mice[1], and these neurons likely project to the PAG[3]. We hypothesize not only that dPAG neurons directly influence behaviour but also that TeA$_{dPAG}$ neurons may be involved in the direct control of running behaviours. The CaMKII-positive TeA neurons that projected to the dPAG were mostly from L5a (Fig. 2b, c). We verified whether TeA$_{dPAG}$ neurons directly influence running behaviours by injecting AAVretro-CaMKIIα-eNpHR3.0-EYFP into the dPAG (Fig. 3e) to express eNpHR3.0 in CaMKII TeA$_{dPAG}$ cells (Fig. 3f). Bilateral stimulation with yellow light specifically inhibited glutamatergic TeA$_{dPAG}$ neurons, which also blocked flight behaviours, whereas no blockade was observed in the control group injected with the eNpHR-free virus (Fig. 3g and h).

The above results indicated that TeA$_{dPAG}$ neurons are necessary to induce escape from diverse sensory threats, such as sound, light, and air puff stimuli. Contextual escape at least includes contextual signalling and locomotion (running). We conducted additional experiments in an open-field environment to further validate the ability of TeA$_{dPAG}$ neurons to control running behaviours. We observed that the blockade of both TeA$_{dPAG}$ (Fig. 3i, j) and $_{TeA}$dPAG (Fig. 3m, n) neurons separately affected the ability of the mice to freely explore the open field (Fig. 3k, o). Moreover, the inhibition of TeA$_{dPAG}$ or $_{TeA}$dPAG neurons induced bradykinesia, as indicated by significant decreases in total distance travelled and movement speed (Fig. 3l, p, upper panels), and akinesia, as indicated by a substantial increase in immobility (Fig. 3l, p, lower left panels). However, the inhibition of these neurons did not affect the time mice spent exploring the central zone (Fig. 3l, p, lower right panels), indicating that anxiety-like behaviours were not affected. Similarly, no blockade was observed in the control group injected with the hM4D-free virus. These results indicate that the TeA–dPAG neural circuit is a key node that positively modulates general locomotor activity.

Taken together, the findings of these experiments indicate that CaMKII-positive projections from the TeA L5a to the dPAG regulate running behaviours.

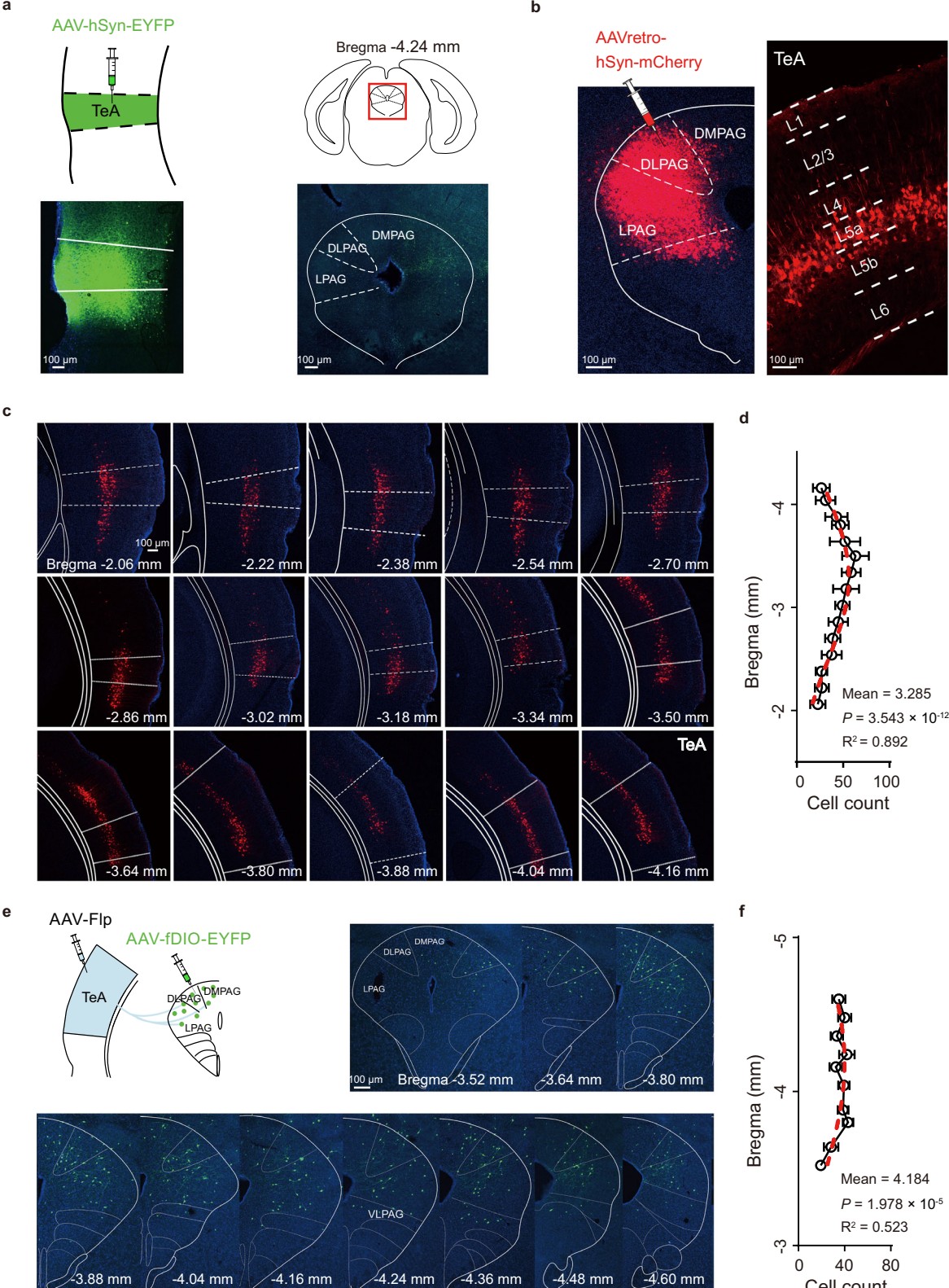

**Fig. 2 | TeA layer 5a cells project to the dPAG. a** Left, schematic of virus injection into the TeA and subsequent virus expression. Right, fluorescently labelled axons in target regions of the dPAG. **b** Left, site of AAVretro injection in the dPAG. Right, mCherry-labelled TeA$_{dPAG}$ neurons. **c, d** Images showing mCherry-labelled TeA$_{dPAG}$ neurons and average cell counts from the rostral to caudal directions. Each image provides the coordinates for the section relative to the bregma hereafter. The red dashed line is the Gaussian fit ($n = 3$ animals, $P < 0.0001$, F-test). **e, f** Schematic of virus injection into the TeA and dPAG, images of EYFP-labelled $_{TeA}$dPAG neurons, and average cell counts from the rostral to caudal directions. The red dashed line is the Gaussian fit ($n = 3$ animals, $P < 0.0001$, F-test). Data are presented as mean ± SD. (Source data are provided as a Source Data file; see Supplementary Data 1 for detailed statistics).

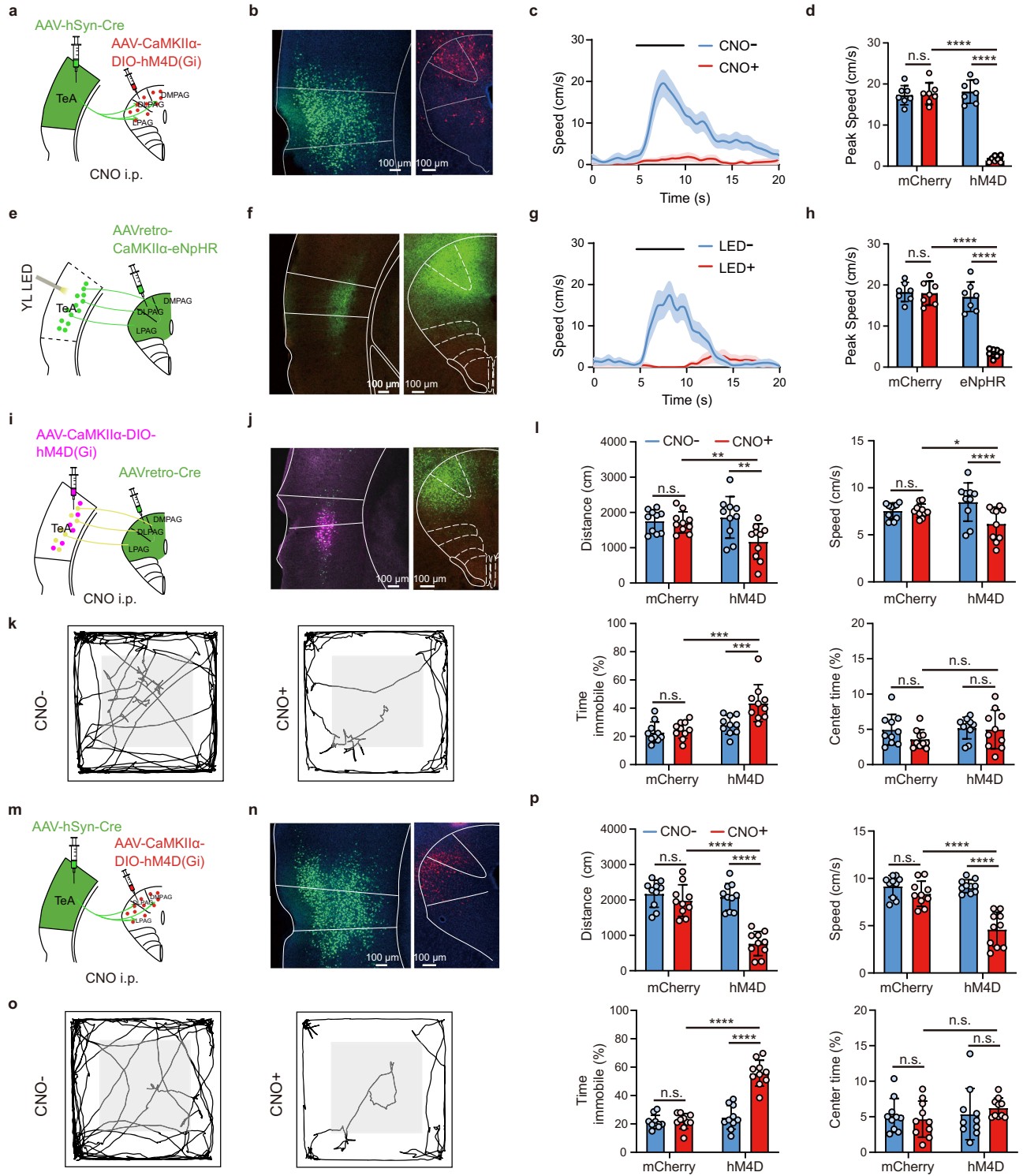

**Fig. 3 | A TeA−dPAG circuit determines running behaviours. a−h** Blockade of the TeA−dPAG pathway in mice running on the turntable: ₜₑₐdPAG CaMKII neurons (**a−d**) and TeA_dPAG CaMKII neurons (**e−h**). **a, e** Virus injection site and chemogenetic (CNO) and optogenetic (yellow light LED) interventions. **b, f** Virus expression. **c, g** Representative flight speed trace following random presentation of Sound, Light, and Air puff stimuli before (CNO- or LED-) and after (CNO+ or LED + ) interventions in hM4D(Gi)- or eNpHR-expressing animals. Thick lines: average flight speed (30 trials). **d, h** Peak flight speeds before vs after CNO or LED intervention. (*n* = 7 animals/group, ****$P$ < 0.0001, two-way ANOVA with the Bonferroni post hoc correction). **i** Virus injection strategy for TeA_dPAG CaMKII neurons blockade. **j** Virus expression for (**i**). **k** Representative traces before (CNO-) and after (CNO + )

intraperitoneal (i.p.) injections of CNO in hM4D(Gi)-expressing animals. **l** Performance of hM4D-expressing and control mice in the open field test (**i**): distance travelled, speed, immobility time, and time spent exploring the centre zone before (CNO-) and after (CNO + ) intraperitoneal injections of CNO. (*n* = 10 animals/group, *$P$ < 0.05, **$P$ < 0.01, ***$P$ < 0.001, ****$P$ < 0.0001, two-way ANOVA with the Bonferroni post hoc correction). **m−o** The same as in (**i−k**) for ₜₑₐdPAG CaMKII neurons blockade. **p** Same as (**l**), but for (**m**). (*n* = 10 animals/group, ****$P$ < 0.0001, two-way ANOVA with the Bonferroni post hoc correction). **c, g** Data are presented as mean ± SEM; in all other panels, data are presented as mean ± SD. (Source data are provided as a Source Data file; see Supplementary Data 1 for detailed statistics).

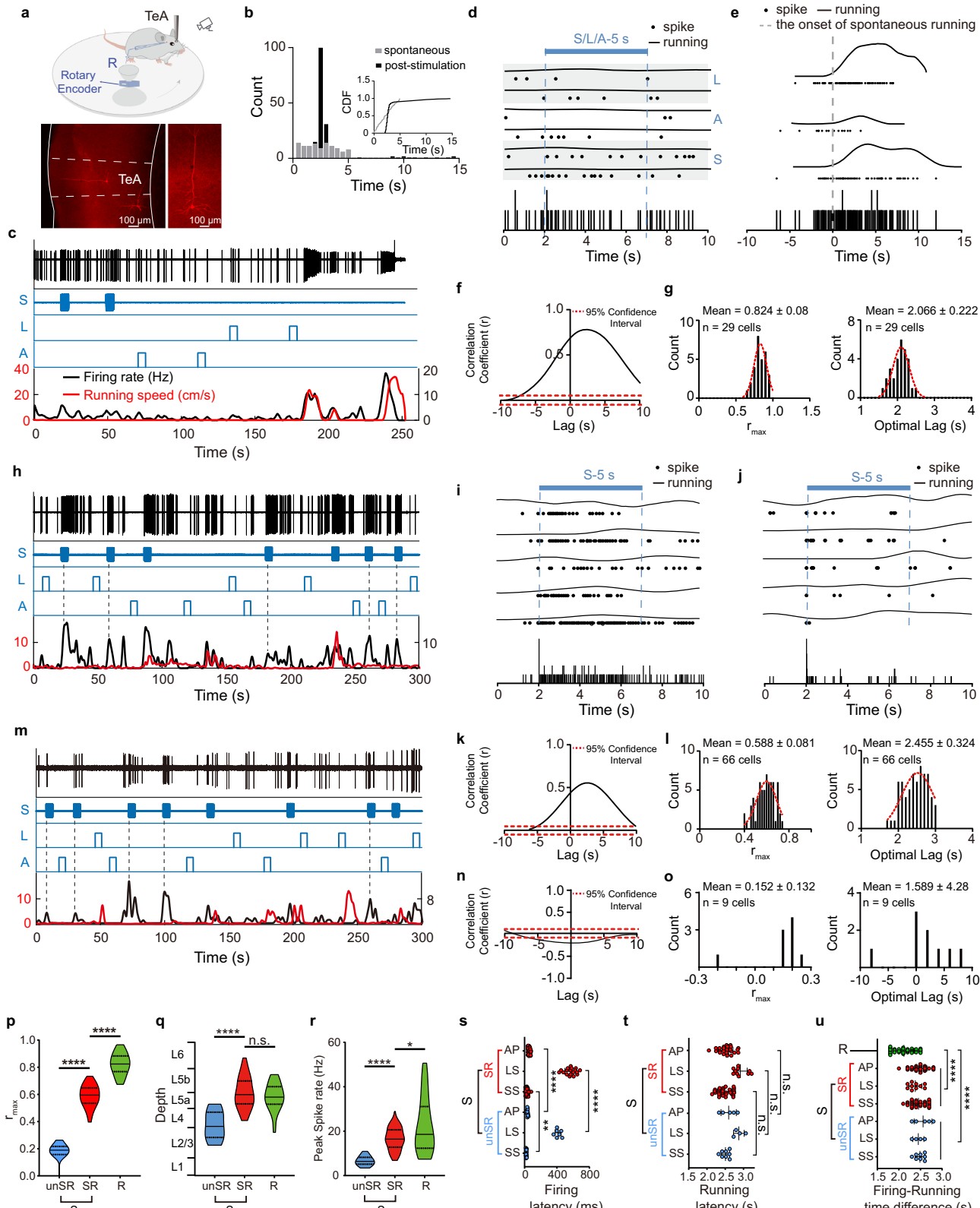

## Identification and characterization of three functional neuron types in the TeA region

The TeA, a region that both receives and integrates diverse types of sensory information, modulates behavioural output. Thus, it is likely to play a pivotal role in the transformation of sensory inputs into behaviours. We conducted in vivo, single-cell, loose-patch recordings in running animals to simultaneously monitor the firing activity of TeA neurons and the running speed of the animals (Fig. 4a, top panel). The recorded neurons were labelled with biocytin for localization (Fig. 4a, bottom panel). The stimulation paradigms employed for this analysis involved sound, light, and air puff stimuli, which differed from the parameters used in the behavioural experiments (sound: 50 dB, 12 kHz, and 1 s or 5 s; light: 2 lux and 1 s or 5 s; air puff: 0.8 L/min and 1 s or 5 s). The stimulus intensities were reduced because, despite the use of head

**Fig. 4 | Sensation-related and running-related neurons in the TeA. a** Top, schematic of the recording setup (R: pipette) during delivery of sound, light, and air-puff stimuli. Running speed was monitored via a rotary encoder. Bottom, a recorded running-related neuron (R-neuron) labelled with biocytin. **b** Similar to Fig. 1m, analysis of turntable running model in (**a**). $P < 0.0001$, two-sided Kolmogorov–Smirnov test. **c** Characteristics of a representative R-neuron in TeA. **d** Top, raster plots and running speed for individual trials in response to three sensory stimuli in (**c**). Bottom, PSTH of firing activity. **e** Similar to (**d**), responses to spontaneous running events. **f** CCF analysis of (**c**), assessed by permutation test (1000 circular shuffles of speed). $r_{max}$ was significant ($P < 0.001$, two-sided). **g** Gaussian-fitted distributions of CCF parameters from all significant TeA R-neurons. ($P < 0.0001$, F test). **h** A representative SR-neuron. Dashed lines mark sensory stimuli that did not induce running, used to analyze stimulus-firing correlation. **i** Same as (**c**) but for example SR-neuron (**h**) responses to sound. **j** Similar

to (**i**), but for example unSR-neuron (**m**). **k** Similar to (**f**), analysis of (**h**) ($P < 0.01$, two-sided permutation test). **l** Similar to (**g**), analysis of all recorded SR-neurons. ($P < 0.0001$, F test). **m** A representative unSR-neuron. **n** Similar to (**k**), analysis of (**m**) ($P < 10^{-6}$, two-side permutation test). **o** Similar to (**l**), analysis of all recorded unSR-neurons. **p–r** Summary of the $r_{max}$ (**p**), depth (**q**) and peak spike rate (**r**). (*$P < 0.05$, ****$P < 0.0001$, one-way ANOVA with the Tukey's correction). **s–t** Summary of the stimulus–firing latency (**s**) and stimulus–running latency (**t**) for each type of neuron. (**$P < 0.01$, ****$P < 0.0001$, two-sided Mann–Whitney U-test or two-sample t-test). **u** Firing–running time difference. (****$P < 0.0001$, Kruskal–Wallis one–way ANOVA with the Bonferroni correction). Violin plots with inner box plot elements: quartiles (dashed), min/max (edges), mean (center solid line). Data are presented as mean ± SD. (Source data are provided as a Source Data file; see Supplementary Data 1 for detailed statistics).

fixation in the turntable model, high-speed locomotion compromised the stability of the in vivo electrophysiological recordings, and thus maintaining consistent single-unit isolation over extended periods was challenging. Reducing the stimulus intensity decreased both the efficacy and temporal precision of the evoked running. The latency distribution was broader, and the defined response window was consequently wider [0, 3.4 s] (Fig. 4b). At a reduced stimulus intensity, running was still evoked with a probability (51.58%) significantly exceeding spontaneous levels, and the behaviour occurred with a defined latency (2.574 ± 0.259 s), further confirming a time-locked relationship.

We recorded action potentials (APs) from a total of 104 single neurons and recorded the generation of flight behaviours in the mice during neuronal firing. These neurons were classified into two categories based on their responses to sensory stimuli and the relationship of their firing to running speed.

We identified running-related neurons (R-neurons) whose firing appeared to be associated with running rather than with sensory stimuli and whose firing preceded running (Fig. 4c). We begun by confirmed that these neurons did not respond to sensory stimuli. Based on the sensory stimulus–running time window of mice running on the turntable, we selected instances with a failure to elicit running behaviour. Alignment of the timing of each sensory stimulus revealed no response in the example neuron to any stimulus (Fig. 4d). We further examined the relationship between the firing rate and running by analysing spontaneous running events without sensory stimuli, aligning them to run onset and inspecting activity before running began. As shown in Fig. 4e, a consistent increase in the firing rate was observed before the onset of the spontaneous running, and the firing–running time window was [-1.995–0 s]. Based on this criterion, we analyzed all neurons recorded from the TeA region to identify R-neurons. (Supplementary Fig. 2a and 2b). The firing–running time window [−3.034–0 s] was determined by analyzing the peri-event time histogram (PETH) (see Methods).

We performed a cross-correlation function (CCF) analysis on the firing rate and running speed of the example R-neuron (Fig. 4c) to explore the potential correlation between neuronal firing and running (Fig. 4f). The CCF indicated that the strongest correlation between neuronal firing and running speed was observed when the speed trace was shifted forwards in time. The maximum correlation coefficient ($r_{max} = 0.778$) was observed to form a stable plateau for a time lag ranging from $\tau = 2.1$ s to $\tau = 2.5$ s, indicating a positive correlation between the two variables. The peak exhibited a broad temporal width, with a full width at half maximum (FWHM) of 10.588 s. The distribution of the optimal correlation time lag, which extended to 0.4 s, suggests a sustained relationship between neural activity and running speed parameters, as opposed to one precisely time-locked to a single instant. This result suggests that spiking may encode running speed, with the firing rate being continuously modulated during the running preparation and execution phases to

represent speed information. The CCF results for all recorded R-neurons (Supplementary Fig. 2d) also revealed a significant and relatively stable positive correlation ($r_{max} = 0.824 ± 0.08$) between firing rate and running speed (Fig. 4g, left panel). The optimal time lag was centred at -2.066 ± 0.222 s (Fig. 4g, right panel), and this value remained within the firing–running time window (−3.034–0 s). These results indicate that changes in firing activity occur prior to behavioural changes, supporting the functional role of spiking in encoding behaviour and suggesting that their relationship is characterized by a relatively stable temporal order.

The second category of neurons was defined as sensory-related neurons (S-neurons) (Fig. 4h). Similarly, based on the sensory stimulus–running time window defined using the turntable model, we selected instances where the sensory stimulus failed to elicit a running event. Aligning the time points of each sensory stimulus revealed that the example neuron (Fig. 4h) consistently responded to the sound stimuli (Fig. 4i). Analysis of the population peri-stimulus time histograms (PSTHs) for all recorded S-neurons under different sensory stimuli revealed that the response latency to sound stimuli was with 10-48.38 ms, to light stimuli was within 375.602–648.251 ms, and to air puff stimuli was within 29.65–71.763 ms (Supplementary Fig. 2c).

After confirming that these neurons responded to sensory stimuli, we further investigated the relationship between neuronal firing and running. We performed a CCF analysis of the correlations between the firing rates and running speeds for all the S-neurons. The results showed that among these neurons, a subset exhibited similarities to R-neurons, such that advancing the running speed curve resulted in the highest level of alignment between neuronal firing and running speed (Supplementary Fig. 2e). Figure 4h presents an example from this neuronal subset, where the highest correlation coefficient ($r_{max} = 0.540$) showed a stable peak within a time delay ranging from $\tau = 2.5$ s to $\tau = 2.8$ s (Fig. 4k), indicating a positive correlation between the two variables.

The CCF analysis of this neuronal subset (Supplementary Fig. 2e) further demonstrated a significant and relatively stable positive correlation ($r_{max} = 0.588 ± 0.081$) between firing and running speed (Fig. 4l, left panel), with optimal time lags concentrated within the range of 2.455 ± 0.324 s (Fig. 4l, right panel). We defined this group of neurons, which respond to sensory stimuli while also exhibiting firing rates correlated with running speed, as SR-neurons. In contrast to SR-neurons, another group of neurons, which were capable of responding to sensory stimuli (Fig. 4m, j), showed no correlation or a significant but weak correlation ($r_{max} < 0.3$) between the firing rate and running speed based on CCF results (Fig. 4n, o and Supplementary Fig. 2f). This group was defined as unSR-neurons. We recorded neuronal responses to sound, light, and air puff stimuli in both SR-neurons and unSR-neurons (Supplementary Fig. 3). Comparison across groups revealed differences in the $r_{max}$ between firing rate and running speed among the three neuron types (Fig. 4p). R-neurons exhibited the highest mean

($r_{max}$ = 0.824 ± 0.08), followed by SR neurons ($r_{max}$ = 0.588 ± 0.081), while unSR neurons had the lowest mean $r_{max}$ ($r_{max}$ = 0.152 ± 0.132).

The distribution of all 104 neurons in the TeA relative to the cortical surface is shown in Fig. 4q. TeA R-neurons and SR-neurons were predominantly concentrated in L5, whereas unSR-neurons, which were recorded less frequently, were located primarily in layers 2 to 5. Notable differences in the firing rates were observed among the three types of neurons (Fig. 4r); specifically, R-neurons had the highest firing rates, followed by SR-neurons and unSR-neurons. The firing latency of S-neurons to each individual sensory stimulus (sound, light, and air puff) was calculated. The sensory response latencies of SR-neurons were significantly longer than those of unSR-neurons (Fig. 4s). The behavioural latency in response to stimulus presentation was also calculated and named the running latency. This latency did not differ between SR- and unSR-neurons under identical stimulation conditions (Fig. 4t). Since R-neurons did not produce APs in response to stimulation with the three stimuli, the difference in time between the onset of behaviour-related firing and the onset of running was calculated (see the Methods) and referred to as the difference in the firing–running time. Similarly, the difference in firing–running time for S-neurons was calculated as the difference in time between the onset of related firing and the onset of running (Fig. 4u). The differences in the firing–running time between R-neurons (2.117 ± 0.24 s), and SR-neurons (2.449 ± 0.18 s) or unSR-neurons (2.5 ± 0.15 s) were significant.

Based on these results, we hypothesized that at least three types of neurons are involved in information processing from sensation to running behaviour. unSR-neurons respond to sensory stimuli and are localized upstream of SR-neurons. The SR-neurons translate sensory information into a running signal and relay this signal to R-neurons, which then are responsible for driving running behaviours. R-neurons might be TeA$_{dPAG}$ neurons, whereas SR-neurons might be those that receive projections from input sources, hereafter referred to as $_{Sens}$TeA neurons. Both types of neurons are located in L5, but they play different roles in the sensory-to-motor transformation process.

## Contextual signals delivered to the TeA modulate sensory-induced escape

The TeA receives extensive input from sensory projections[31,37,41,51–54]. We investigated the characteristics of $_{Sens}$TeA neurons, which we proposed receive contextual signals from sensory sources and modulate sensory-induced escape, to test the hypothesis described in the previous subsection.

We injected AAVretro-hSyn-mCherry into the TeA to determine the sources and distribution of projections to this region (Fig. 5a, left panel). Abundant cells expressing mCherry were observed in the auditory cortex (AC: Au1, AuD, and AuV), somatosensory cortex (S1), visual cortex (V1 and V2L), dorsal lateral geniculate nucleus (DLG), and medial geniculate body (MGB) (Fig. 5a, middle and right panel). Neurons originating from the cortex were located mainly in L2/3 and L5, consistent with the findings of a previous report[31].

We focused on five areas—the AC, S1, V2L, MGB, and DLG—and locally injected AAV-hSyn-Cre into each area to further elucidate the specific locations of the projections from different sources to the TeA (Supplementary Fig. 4a). Projections from AC and S1 were prominent and concentrated mainly in layers 2–5 of the TeA. Although the V2L, MGB, and DLG also projected to the TeA, these projections were not as robust (Supplementary Fig. 4b). For more precise visualization, we injected AAV-hSyn-Cre into the AC, S1, V2L, MGB, and DLG of different animals individually and Cre-dependent AAV-hSyn-DIO-mCherry into the TeA to label TeA neurons. Thus, TeA neurons receiving projections from different sensory areas ($_{Sens}$TeA; e.g., $_{AC}$TeA, $_{S1}$TeA, $_{VC}$TeA, $_{MGB}$TeA, and $_{DLG}$TeA) were labelled separately (Fig. 5b). Projections to the TeA neurons were located primarily in L5, with a small number distributed in other layers, whereas projections from the MGB were located mainly in L4 (Fig. 5c). A statistical analysis of the fluorescence

intensity curves from different projection areas to the TeA matched these results (Fig. 5c).

We also investigated the neuronal types of TeA neurons receiving sensory projections ($_{Sens}$TeA neurons). AAV-Cre was injected into different sensory cortices, and AAV-hSyn-DIO-mCherry was injected into the TeA to label neurons receiving distinct sensory inputs in the TeA (Supplementary Fig. 4c). Direct identification of these neurons was performed via GAD67 (a marker for GABAergic neurons) immunostaining (Supplementary Fig. 4d). The results revealed a low proportion of GABAergic neurons among these $_{Sens}$TeA neurons. The colocalization rates were 0% for the AC, 0% for S1, 0% for V2L, 0.40% for the MGB, and 0.90% for the DLG projections (Supplementary Fig. 4e). Since the cerebral cortex is primarily composed of glutamatergic and GABAergic neurons, these findings suggest that the $_{Sens}$TeA neurons receiving projections from different sensory areas, are mostly glutamatergic neurons.

We also attempted parallel labeling using a dual-virus strategy (CaMKIIα-DIO-mCherry and hSyn-fDIO-EYFP) (Supplementary Fig. 4f, g). We observed that the colocalization ratio of EYFP (green, broad spectrum) and mCherry (expression in magenta, glutamatergic) signals in labeled TeA neurons varied depending on the input region, with rates of AC: 50%, S1: 60%, V2L: 66%, MGB: 48%, and DLG: 51% (Supplementary Fig. 4e). The low overlap rate is likely attributable to differences in the expression efficiency of the two viral vectors. Therefore, based on the conclusive evidence from GAD67 staining (showing a near absence of GABAergic neurons) and to circumvent the technical limitations of dual-virus co-labeling efficiency, we selected the broadly expressed AAV-hSyn promoter for all subsequent labeling and manipulations. This approach allows us to confidently and comprehensively target the $_{Sens}$TeA neuron population, which we have established as predominantly glutamatergic.

Next, we sought to determine whether different sensory inputs converge to the same cells in the TeA. We injected viruses (AAV-hSyn-Cre and AAV-hSyn-Flp) into pairs of sensory areas (AC and S1, AC and V2L, and S1 and V2L) and a mixture of viruses (AAV-hSyn-DIO-mCherry and AAV-hSyn-fDIO-EYFP) into the TeA (Fig. 5d). Subsequently, Cre-expressing TeA neurons exhibited red fluorescence (expression in magenta), whereas Flp-expressing neurons exhibited green fluorescence. Although TeA neurons receiving projections from different pairs of sensory areas presented different rates of overlap (Fig. 5e, f), the differences between projections within each pair of sensory areas (magenta vs. green) were not significant, indicating that TeA neurons may receive multiple sensory inputs indistinguishably.

Although TeA neurons seemed to receive multiple types of sensory input, including auditory (Au1, AuD, AuV, and MGB), visual (V1, V2L, and DLG), and somatosensory air puff (S1) inputs, we still believe that different types of sensory information in the TeA exert varying degrees of influence on driving running behaviours. Therefore, we locally injected anterograde transsynaptic AAV-hSyn-Cre into the AC, S1, V2L, MGB, and DLG areas of the animals and AAV-hSyn-hM4D(Gi)-DIO-mCherry into the TeA. CNO was injected i.p. to selectively inhibit the activity of these targeted $_{Sens}$TeA neurons (Fig. 5g). As an example, we show hM4D-mCherry expression in $_{AC}$TeA neurons (Fig. 5h). When activated by CNO, $_{AC}$TeA neurons were inhibited, and both the running speed and the probability of running behaviours driven by the three sensory stimuli decreased (Fig. 5i). We systematically assessed the behavioural effects of inhibiting $_{Sens}$TeA neurons defined by inputs from the AC, MGB, S1, V2L, and DLG on the peak running speed and the probability of inducing running behaviours. Chemogenetic inhibition of these neuronal populations significantly reduced the probability of sound-induced flight behaviours. This effect was more pronounced when the TeA neurons targeted by AC and MGB projections were inhibited (Fig. 5j). In addition, inhibition of TeA neurons receiving inputs from V2L, DLG, or AC significantly affected light-induced flight behaviours (Fig. 5k), while inhibition of those receiving S1 input

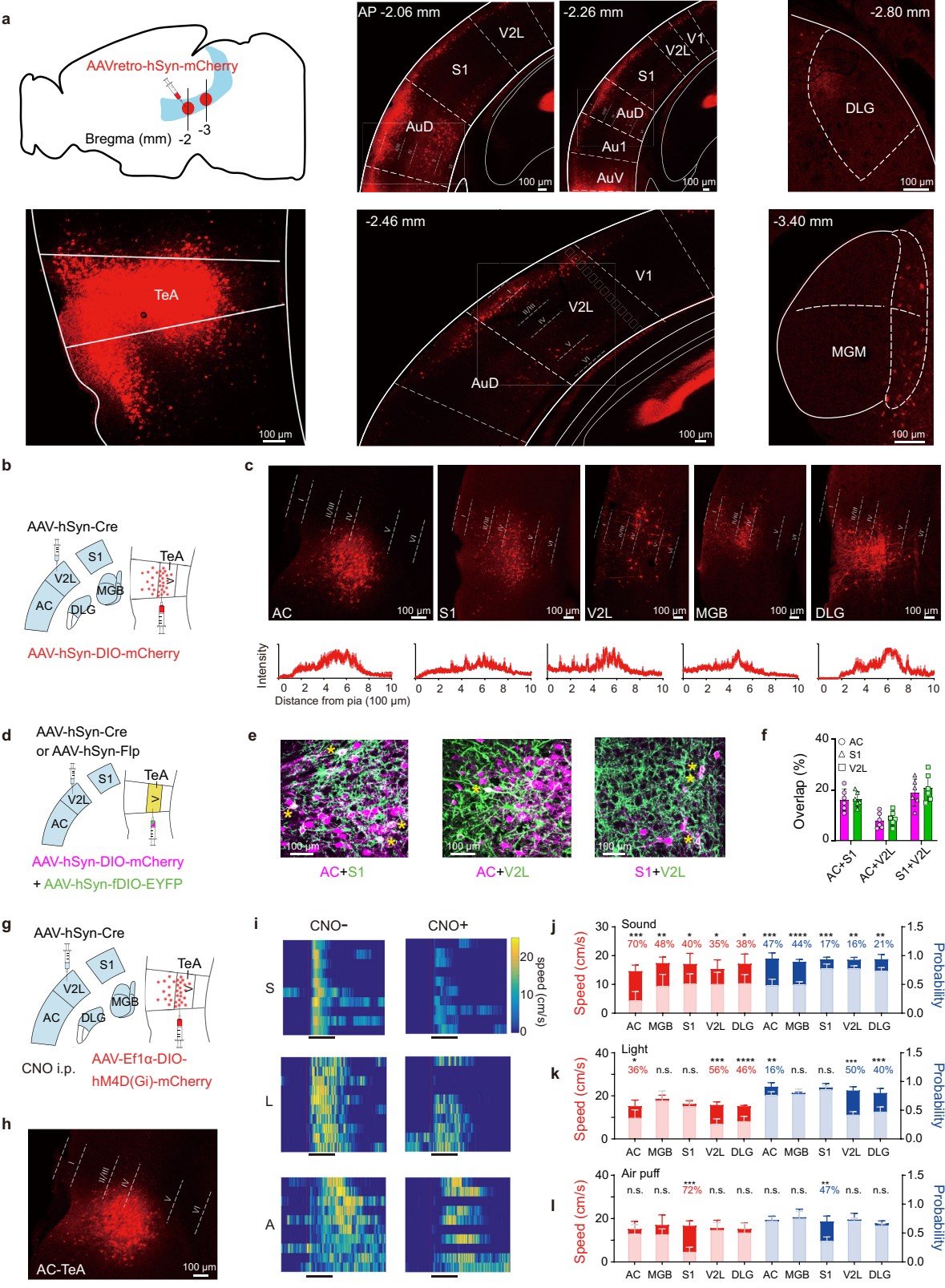

significantly affected air puff-induced flight behaviours (Fig. 5l). These results demonstrate that $_{Sens}$TeA neurons exhibit modality-specific functional roles in driving flight behaviours. While TeA neurons receiving AC or MGB inputs are crucial for sound-induced flight, those receiving V2L/DLG inputs are crucial for light-induced flight, and those receiving S1 inputs are crucial for air puff - induced flight. This input-defined functional specificity ensures appropriate behavioural

responses to distinct threats. Despite this specificity, all three sensory inputs contributed to the overall regulation of flight behaviours. The presence of bisensory responsive neurons partly explains the influence of different stimuli on movement speed (Fig. 5e and f), although the impact of multisensory integration cannot be excluded.

Theoretically, motor behaviours induced by multisensory stimuli should be more robust than those caused by unisensory stimuli. The

**Fig. 5 | Diverse sensory input characteristics of the TeA. a** Viral injection (TeA, left) and expression sites: cortical (middle, AC, S1, and V2L) and subcortical (right, DLG and MGB). **b, d, g** Viral injection sites targeting sensory source regions (AC, S1, V2L, MGB, or DLG) and the TeA. **c** Top, distribution of ₛₑₙₛTeA neurons targeted by distinct sensory source regions. Bottom, analysis of the fluorescence intensity. *n* = 4, 5, 5, 5, and 3 slices from 3, 4, 3, 4, and 4 animals, respectively. **e** TeA neurons receiving multisensory inputs. Panel **e** shows mCherry expression in magenta for red-green discrimination. **f** Percentage of TeA neurons (**d**) with input overlap from paired source regions. (*n* = 6 slices / 3 animals per group). **h** Representative TeA

neurons receiving AC projections. **i** Running speed heatmaps of a representative animal before (CNO-) and after (CNO + ) i.p. injections of CNO (black bar: 5 s stimulus duration). **j–l** Peak speed (red) and probability (blue) of running induced by sound (**j**), light (**k**), and air puff (**l**) stimuli before (dark) and after (undertint) the i.p. injection of CNO. Numbers indicate the percentage reduction in the peak speed and running probability for CNO+ animals relative to those for CNO- animals. (*n* = 5 animals for each sensory area, \**P* < 0.05, \*\**P* < 0.01, \*\*\**P* < 0.001, \*\*\*\**P* < 0.0001, two-sided Paired t-test). Data are presented as mean ± SD. (Source data are provided as a Source Data file; see Supplementary Data 1 for detailed statistics).

mice were presented with one, two, or three different types of sensory stimuli simultaneously after training to visually observe the behavioural responses induced by different types of stimuli (Supplementary Fig. 5a). We calibrated the intensity of the sounds, lights, and air puffs through multiple tests to ensure that each stimulus alone did not significantly drive running behaviours in the mice, which was confirmed by the low running speeds and low probabilities of successfully inducing running behaviours (Supplementary Fig. 5b). Then, we combined different types of sensory stimuli to produce multisensory stimuli. Compared with the unisensory stimuli, the multisensory stimuli significantly drove running behaviours (Supplementary Fig. 5c, d), with a significantly higher probability (Supplementary Fig. 5e) and peak speed (Supplementary Fig. 5f). Nevertheless, although the speed of the running behaviours induced by multisensory stimuli was notably faster than that of the running behaviours induced by the unisensory stimuli, the difference between combinations of two and three sensory stimuli was not significant (Supplementary Fig. 5f). These results indicate that multisensory inputs contribute to the overall modulation of running behaviours.

The integration of multisensory sensory information likely relies on the weights assigned to different sensations. The related stimuli can also vary in terms of characteristics such as the type, intensity, and duration of a single sensation. In this context, understanding which factors influence the corresponding weights in multimodal sensory integration is important. For auditory stimuli, these factors include frequency, intensity, and duration. Mouse escape behaviour is related to the perceived level of danger[6,8], and in this context, higher-intensity auditory stimuli drive faster animal running behaviours[1]. Therefore, we focused on auditory stimuli with different parameters to further explore the mechanisms underlying unisensory stimulus integration.

We systematically altered the sound stimulus frequency, intensity, and duration to observe the corresponding effects on running latency, duration, and peak speed. We found that changes in frequency had minimal effects on the latency, duration, and peak speed of the induced running behaviours (Supplementary Fig. 5g, j–l, left panel). In addition, increasing the duration of the sound increased only the duration of running (Supplementary Fig. 5h, j–l, middle panel). The sound intensity, however, significantly impacted running latency, duration, and peak speed (Supplementary Fig. 5i, j–l, right panel). The results of these experiments indicated that auditory stimulus integration depends more on the intensity and duration of the stimulus than on the specific (frequency) characteristics.

## TeA SR-neurons to R-neurons microcircuits
Based on the abovementioned hypothesis, we next checked whether SR-neurons translate sensory information into a running signal and relay it to R-neurons to initiate flight behaviours (running).

We investigated the properties of ₛₑₙₛTeA and TeA_{dPAG} neurons. Dual injections with AAV-hSyn-Cre in individual sensory regions (AC, S1, VC, MGB, or DLG) and AAV-hSyn-DIO-EYFP in the TeA (Fig. 6a, left panel) were performed to label ₛₑₙₛTeA neurons, namely, _{AC}TeA, _{S1}TeA, _{VC}TeA, _{MGB}TeA, and _{DLG}TeA neurons, in green. Concurrently, AAVretro-hSyn-mCherry was injected into the dPAG to label TeA_{dPAG} neurons, visualized in magenta (Fig. 6a). The TeA_{dPAG} neurons were located in L5a of the TeA, whereas the ₛₑₙₛTeA neurons were located in L2–5b,

with differences reflecting different input sources. We characterized the morphology of these two types of neurons. Performing three-dimensional reconstructions in regions with high viral expression intensity (Fig. 6a) posed certain challenges. For single-cell 3D reconstruction, we preferentially selected images containing only one isolated neuron (Supplementary Fig. 6a and c). A secondary strategy was to choose regions with relatively sparse viral labeling (Supplementary Fig. 6b and d, the white dashed boxes in the left panels). When two or more neurons were located in close proximity, it often became difficult to trace dendritic arbors accurately in two-dimensional planes. In such cases, we prioritized 3D reconstruction, rotating the model to broadly assess neuronal morphology, and made efforts to select neurons whose structures remained discernible after rotation (as indicated by the white arrows in the right panels of Supplementary Fig. 6b and d). Following these selection criteria, five cells from each group were ultimately chosen for reconstruction. The reconstruction results revealed that both neuronal types exhibited a pyramidal morphology characterized by an apical dendritic tuft and basal dendrites around the soma. Nevertheless, ₛₑₙₛTeA neurons receiving sensory inputs (Fig. 6a, green; 6b) had significantly more basal dendrites around the soma and broader apical tufts (thick-tufted) than TeA_{dPAG} neurons, which displayed a relatively slender apical tuft (slender-tufted) (Fig. 6c). A 3D reconstruction of dendritic branches via Sholl analysis revealed significant differences in dendritic branching between the two types of neurons, particularly at the base of the soma and the distal dendritic branches of the terminal cluster. ₛₑₙₛTeA neurons were thick-tufted, whereas TeA_{dPAG} neurons were slender-tufted (Fig. 6d, e). Additionally, the soma diameter of ₛₑₙₛTeA neurons was slightly larger than that of TeA_{dPAG} neurons (Fig. 6f). These differences suggest that the two types of neurons may serve as distinct information channels, mediating different functions in sensation–behaviour transformation. We postulate that ₛₑₙₛTeA neurons in L5 integrate sensory information to drive the activity of L5a TeA_{dPAG} neurons.

Do ₛₑₙₛTeA neurons synapse with TeA_{dPAG} neurons? We separately injected anterograde AAV-hSyn-Flp into the sensory regions (AC, S1, and V2L) and AAV-hSyn-fDIO-mCherry + AAV-hSyn-DIO-EYFP into the TeA. Concurrently, we injected AAVretro-CMV-WGA (wheat germ agglutinin)-Cre into the dPAG. Figure 6g illustrates the principle of virus labelling, wherein WGA-infected neurons retrogradely and transsynaptically infect upstream neurons. The results revealed the presence of merged neurons (white) in TeA L5, with colocalization rates of 23.54% for AC, 15.07% for S1, and 12.99% for V2L projections (Fig. 6h and Supplementary Movie 1; see the Methods). These results indicate that ₛₑₙₛTeA neurons directly project to TeA_{dPAG} neurons.

We determined the function of this direct projection by injecting AAVretro-hSyn-mCherry in the dPAG to retrogradely label TeA_{dPAG} neurons while injecting AAV-hSyn-Cre in the AC or S1 and AAV-hSyn-DIO-hChR2-EYFP in the TeA, which enabled ₛₑₙₛTeA neurons to express ChR2. We subsequently performed in vitro whole-cell recordings of TeA neurons (Fig. 6i). After successful patch clamp sealing, we checked the firing characteristics of the cells by holding them at different currents. In response to a current pulse, ₛₑₙₛTeA neurons displayed a brief burst (often observed as an AP doublet) followed by repetitive single nonadapting APs (Fig. 6j, top panel). In contrast, TeA_{dPAG} neurons behaved as regular spiking neurons and showed pronounced changes

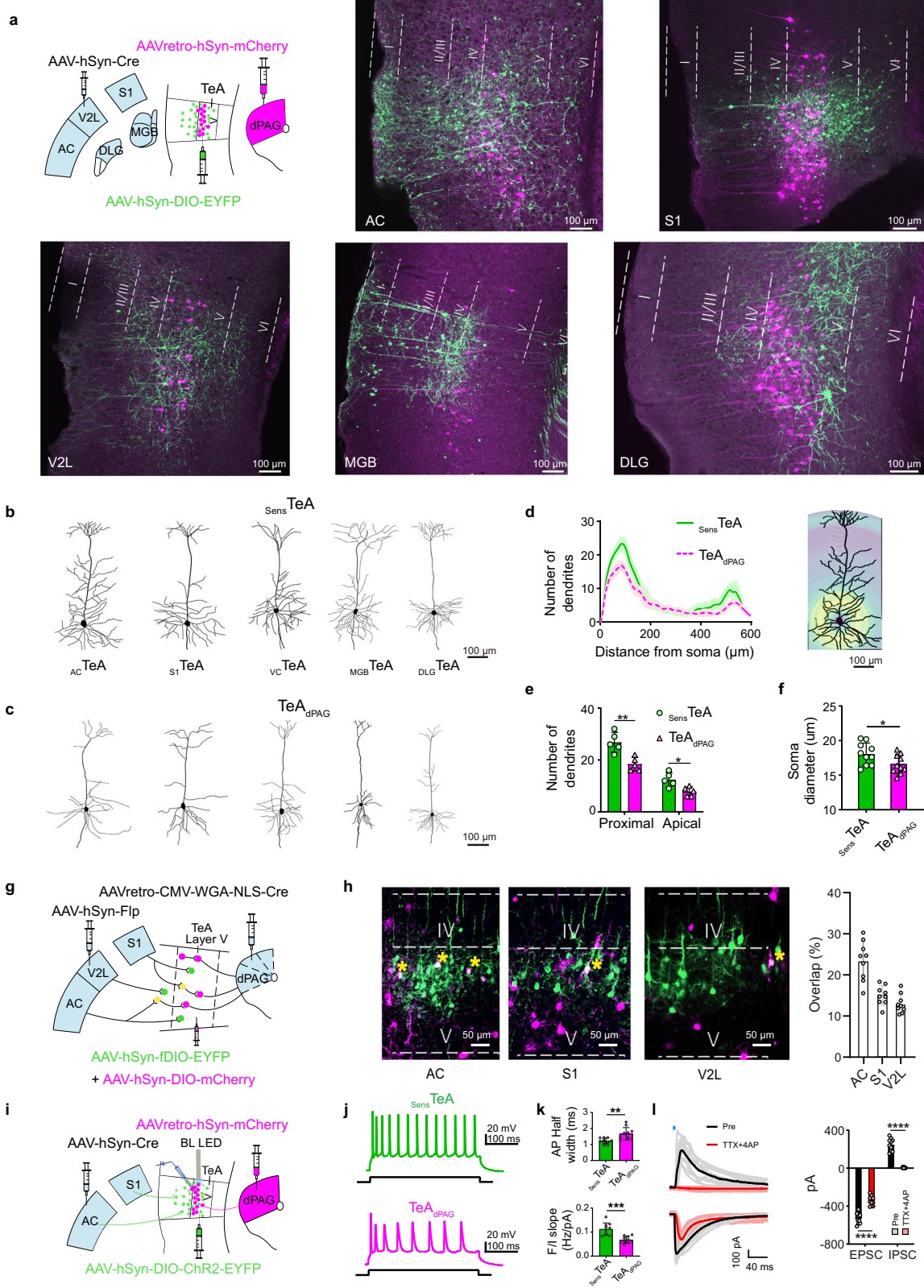

in the firing rate in response to a depolarizing current pulse (Fig. 6j, bottom panel). Additionally, compared with the TeA$_{dPAG}$ neurons, the $_{Sens}$TeA neurons had a smaller AP half-peak width (Fig. 6k, top panel). When we compared the repeated firing characteristics of the two types of neurons, we found that the F/I (firing rate/current) slope of the $_{Sens}$TeA neurons was greater (Fig. 6k, bottom panel). Then, we optogenetically activated the $_{Sens}$TeA neurons with blue light (5 ms pulses

delivered by a 5 mW LED with a frequency of 0.05 Hz)[40], and excitatory postsynaptic currents (EPSCs) and inhibitory postsynaptic currents (IPSCs) were recorded from the TeA$_{dPAG}$ neurons (Fig. 6l). The IPSCs completely disappeared with the addition of tetrodotoxin (TTX) + 4-aminopyridine (4AP), whereas the EPSCs were not blocked but were weakened (Fig. 6l). These findings suggest that $_{Sens}$TeA neurons form direct monosynaptic excitatory connections onto TeA$_{dPAG}$ neurons.

**Fig. 6** | $_{Sens}$**TeA and TeA$_{dPAG}$ neurons. a** Top left, schematic of the virus injection in the AC, S1, V2L, MGB, DLG, TeA, and dPAG. Images showing $_{Sens}$TeA neurons (EYFP, green) in L2-5b, including $_{AC}$TeA, $_{S1}$TeA, $_{V2L}$TeA, $_{MGB}$TeA, and $_{DLG}$TeA, and TeA$_{dPAG}$ neurons in L5a (mCherry, expression in magenta) in the TeA. **b, c** Morphologies of representative $_{AC}$TeA, $_{S1}$TeA, $_{V2L}$TeA, $_{MGB}$TeA, and $_{DLG}$TeA neurons in L5 (**b**) and TeA$_{dPAG}$ neurons in L5a (**c**). **d** Left, group-averaged Sholl analysis of L5 $_{Sens}$TeA and TeA$_{dPAG}$ neurons ($n = 5$ cells from 5 slices each). Right, representative Sholl analysis of an L5 $_{Sens}$TeA neuron. **e, f** Dendrite number along the somatodendritic axis (**e**) and soma diameter (**f**) of L5 $_{Sens}$TeA versus TeA$_{dPAG}$ neurons. ($n = 5$ cells for both (**e**), $n = 10$ cells for each group (**f**), *$P < 0.05$, **$P < 0.01$, two-sided two-sample t-test). **g** Schematic of the virus injection sites. **h** Left, images showing $_{Sens}$TeA neurons (EYFP), TeA$_{dPAG}$ neurons (mCherry, expression in magenta) and neurons labelled in the upper layer (magenta or white). Right, percentage of $_{Sens}$TeA neurons (white)

directly connected to TeA$_{dPAG}$ neurons among all labelled $_{Sens}$TeA neurons. ($n = 9$ slices from 3 animals per group). **i** Schematic of virus **i**njection and experimental protocol. **j** Firing patterns of $_{Sens}$TeA and TeA$_{dPAG}$ neurons in response to 240 pA depolarizing current. **k** Top, comparison of action potential half widths. Bottom, F/I (firing rate/current) slope. ($n = 10$ cells from 10 slices, **$P < 0.01$, ***$P < 0.001$, two-sided two-sample t-test). **l** Left, IPSCs and EPSCs in a TeA$_{dPAG}$ neuron evoked by optogenetic stimulation of ChR2-expressing $_{AC}$TeA or $_{S1}$TeA inputs. Right, LED-evoked currents before and after infusion of TTX (1 µM) + 4-AP (1 mM). ($n = 10$ cells from 10 slices, ****$P < 0.0001$, two-sided Paired t-test). Data were obtained from both the right and left hemispheres. **d** Data are presented as mean ± SEM; in all other panels, data are presented as mean ± SD. (Source data are provided as a Source Data file; see Supplementary Data 1 for detailed statistics).

## Activating $_{Sens}$TeA neurons drives TeA$_{dPAG}$ neurons, whereas activating TeA$_{dPAG}$ neurons triggers running

Thus far, we have identified two types of neurons, $_{Sens}$TeA and TeA$_{dPAG}$, based on their projections, morphology and electrophysiology, as well as SR-neurons and R-neurons, based on their function, firing and relationship to running behaviours. We hypothesize that SR-neurons and R-neurons may correspond to $_{Sens}$TeA and TeA$_{dPAG}$ neurons, respectively, and that SR-neurons may activate R-neurons. One group of mice received injections of AAV-hSyn-Cre into the AC and AAV-hSyn-DIO-hChR2-EYFP into the TeA (Fig. 7a), whereas another group received injections of AAVretro-CaMKIIα-hChR2-EYFP into the dPAG (Fig. 7b) to selectively label and express ChR2 in $_{Sens}$TeA and TeA$_{dPAG}$ neurons. The control group received injections of AAV-hSyn-DIO-EYFP into the TeA to rule out direct effects of the virus and confirm that the effects were due to ChR2. Upon 20 Hz blue light stimulation (15 mW, 5 s), $_{AC}$TeA-ChR2$^+$ neurons responded consistently to each stimulus that induced running behaviours, in which the speed increased from baseline values to a plateau with repeated light stimulation (Fig. 7c). We then performed in vivo recordings in the TeA to identify neurons that were activated by this pathway but were not directly expressing ChR2. We identified the locally recorded, light-responsive, ChR2-negative neurons as AC-TeA-ChR2$^-$ responsive neurons (hereafter referred to as AC-TeA-ChR2$^-$ neurons). These cells represent the functional postsynaptic targets of the AC to TeA pathway in our experimental paradigm. These AC-TeA-ChR2$^-$ neurons also responded to photostimulation. Their firing rates gradually increased across successive light pulses until reaching a plateau (Fig. 7d). This pattern was mirrored in the animal's behavioural output: the evoked running speed also increased progressively across trials before stabilizing (Fig. 7d). In contrast, the firing of the presynaptic $_{AC}$TeA-ChR2$^+$ neurons remained stable throughout the stimulation train (Fig. 7c), indicating that the observed facilitation originated within the local TeA circuit. The observed facilitation of both postsynaptic firing and behaviour suggests that the AC to TeA pathway engages a local microcircuit that exhibits short-term plasticity. Crucially, this microcircuit is positioned to directly influence running behaviour. Stimulation of TeA$_{dPAG}$-ChR2$^+$ neurons with each stimulus resulted in a firing response, and the firing rate was correlated with the speed of the induced running behaviour (Fig. 7e), whereas no response was recorded from TeA-ChR2$^-$ neurons (neurons upstream of TeA$_{dPAG}$ neurons, equivalent to $_{Sens}$TeA neurons) in TeA$_{dPAG}$-ChR2$^+$ mice. The firing rates of $_{AC}$TeA-ChR2$^+$ and TeA$_{dPAG}$-ChR2$^+$ neurons were significantly greater than those of AC-TeA-ChR2$^-$ neurons (Fig. 7f). Moreover, at the same stimulation frequency, the speed of the running behaviours induced by TeA$_{dPAG}$ neuronal activation was significantly faster than that induced by the activation of $_{AC}$TeA neurons (Fig. 7g). The firing patterns of the three neuronal types after exposure to blue light were strongly correlated with the running speed (Fig. 7c–e, correlation coefficient r in Fig. 7h). The average latency of the firing of $_{AC}$TeA-ChR2$^+$ and TeA$_{dPAG}$-ChR2$^+$ neurons was 4.36 ± 0.574 ms and 4.157 ± 0.476 ms, respectively, indicating direct effects of blue light

stimulation on ChR2$^+$ neurons (Fig. 7i). In contrast, the latency of the firing of AC-TeA-ChR2$^-$ neurons was 407.8 ± 60 ms, indicating that this firing was a result of synaptic activation from $_{AC}$TeA-ChR2$^+$ neurons following blue light stimulation (Fig. 7i). The difference in latency between these two types of neurons was similar to the difference in firing–running time between R-neurons and SR-neurons (332 ms; Fig. 4u). Additionally, the average difference in time between the firing of $_{AC}$TeA-ChR2$^+$ neurons and the initiation of animal running was 2.419 ± 0.184 s, which was longer than that for AC-TeA-ChR2$^-$ neurons (2.075 ± 0.202 s) and TeA$_{dPAG}$-ChR2$^+$ neurons (2.078 ± 0.148 s) (Fig. 7j). On the other hand, SR-neurons and R-neurons also exhibited a difference in time between firing and running (Fig. 4u), and the difference in time between these two neuronal types (332 ms) closely matched that between the $_{AC}$TeA-ChR2$^+$ neurons and the TeA$_{dPAG}$-ChR2$^+$ neurons (341 ms) (Fig. 7j). Together, these results suggest that $_{Sens}$TeA neurons represent a subset of SR-neurons, whereas TeA$_{dPAG}$ neurons represent a subset of R-neurons, although the possibility of multisynaptic neural circuits involving SR-neurons cannot be excluded. Notably, we observed a highly similar time difference between firing and the initiation of running in AC-TeA-ChR2$^-$ neurons (2.075 ± 0.202 s) and TeA$_{dPAG}$-ChR2$^+$ neurons (2.078 ± 0.148 s) (Fig. 7j). Combined with the finding that the facilitatory profile of AC-TeA-ChR2$^-$ neuron firing precisely matched the facilitation of running speed evoked by $_{AC}$TeA-ChR2$^+$ stimulation positions the activity of AC-TeA-ChR2$^-$ neurons as a direct predictor of behavioural plasticity. The convergence of these congruent temporal and dynamic response properties provides evidence that AC-TeA-ChR2$^-$ neurons are functionally equivalent to, and likely encompass, the TeA$_{dPAG}$ neuron population that directly governs running behaviour.

Since the activation of both $_{AC}$TeA ($_{Sens}$TeA) and TeA$_{dPAG}$ neurons can induce running behaviours in mice, the inferred delay between the two suggests that TeA$_{dPAG}$ neurons are downstream of $_{AC}$TeA ($_{Sens}$TeA) neurons. The synaptic relationship between these two types of neurons is likely the neural circuit mediating the transformation of sensory input into behaviour. We injected AAV-hSyn-Flp into the sensory cortex (AC or S1), AAV-hSyn-fDIO-ChR2-EYFP + AAV-CaMKII-DIO-hM4D(Gi)-mCherry into the TeA, and AAVretro-hSyn-Cre into the dPAG to further investigate this hypothesis. This labelling strategy allowed us to label $_{AC}$TeA neurons and induce the expression of ChR2 while hM4D(Gi) TeA$_{dPAG}$ cells were selectively blocked after an i.p. injection of CNO (Fig. 7k). Subsequently, 20 Hz blue light stimulation of the TeA drove running behaviours in the mice. However, when we specifically blocked TeA$_{dPAG}$ cells, the ability to induce running in mice was abolished (Fig. 7l, m). Further analysis of the data across multiple animals further confirmed that after the TeA–dPAG circuit was blocked, the activation of TeA neurons, which are responsible for sensory input, could no longer drive running behaviours in mice (Fig. 7n). This observation suggests that the connection between $_{Sens}$TeA and TeA$_{dPAG}$ neurons is monosynaptic and supports the hypothesis that $_{Sens}$TeA neurons (namely, SR-neurons) transform sensory inputs into running signals and activate TeA$_{dPAG}$ neurons

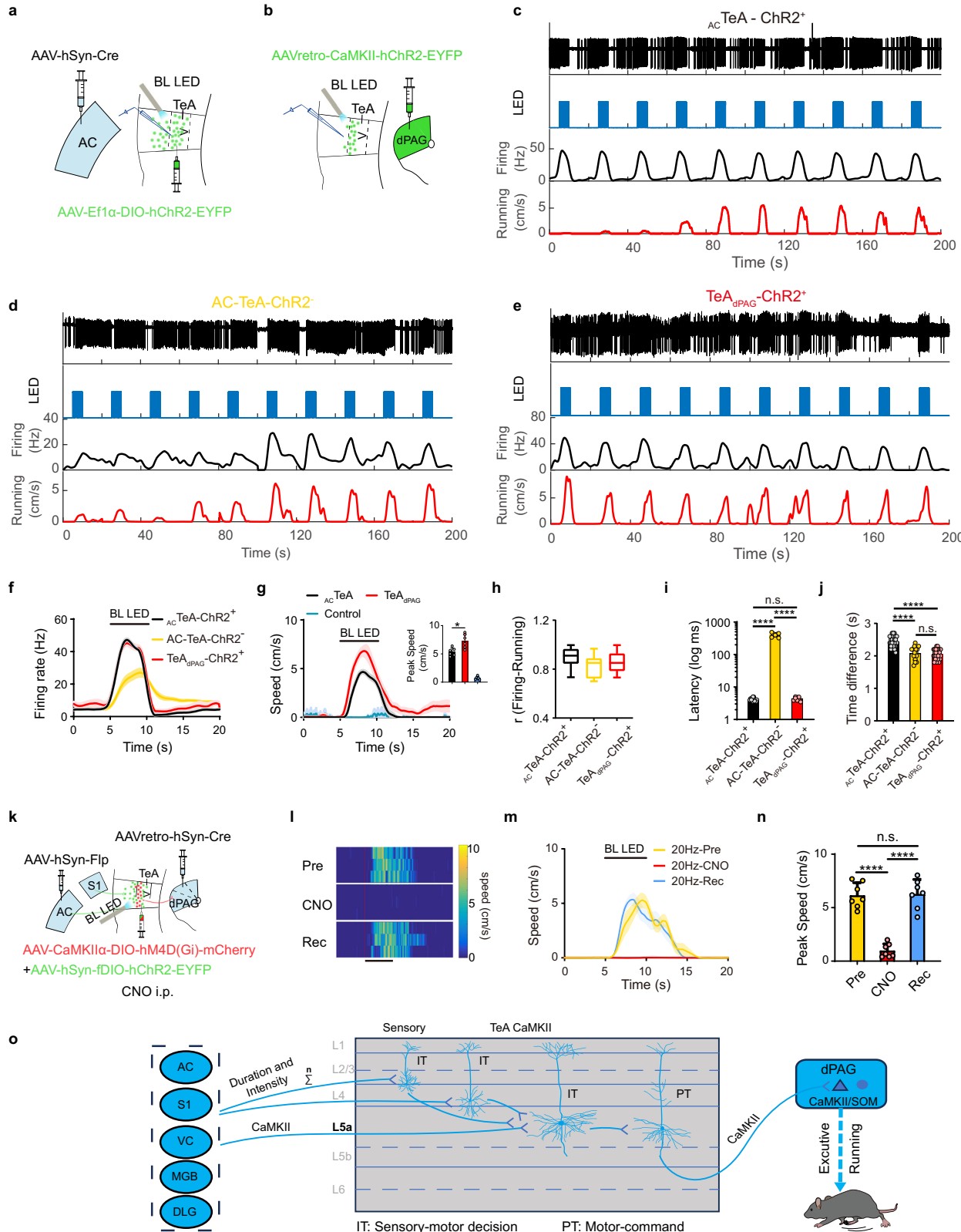

(namely, R-neurons) through weak monosynaptic connections. We surmise that this weak connection requires an extended period of temporal summation to sufficiently excite the downstream neuron, which in turn drives running behaviours. These results further confirm that the activation of the TeA−dPAG neural circuit is sufficient to elicit running behaviour. Given its capacity to integrate multisensory inputs,

this circuit is proposed to function as a critical pathway orchestrating adaptive escape behaviour (Fig. 7b, e, k−n).

Taken together, the above results reveal a microcircuit within TeA L5 wherein different types of sensory information (such as auditory, visual, and somatosensory information) encoded by neurons from the cortex (e.g., A1, V1, and S1) and subcortex (e.g., MGB and DLG) serve as

**Fig. 7 | Firing and running evoked by the activation of $_{AC}$TeA and TeA$_{dPAG}$ neurons. a, b** Schematic of virus injection and experimental protocol. **c–e** Firing rates of $_{AC}$TeA neurons expressing ChR2 (**c**), AC-TeA neurons not expressing ChR2 (**d**) and TeA$_{dPAG}$ neurons expressing ChR2 (**e**) that responded to the speed of running behaviours evoked by blue light LED stimulation (20 Hz, 5 s, 10 trials). **f, g** Firing rate (**f**, $n$ = 12, 7, and 8 cells, $P < 0.0001$, one-way ANOVA with the Tukey's post hoc correction.) and running speed (**g**, $n$ = 6 animals/group, $*P < 0.05$, one-way ANOVA with the Dunnett T3 post hoc correction). **h** Correlations between the firing rate and running speed. $_{AC}$TeA-ChR2$^+$: 0.8980 ± 0.0737; AC-TeA-ChR2: 0.8301 ± 0.0879; TeA$_{dPAG}$-ChR2$^+$: 0.8537 ± 0.0792; n = 12, 7, and 8 cells, respectively. Upper and lower quartiles (box plots), min and max values (whiskers), mean (center line). **i** Firing latency. ($n$ = 12, 7, and 8 cells, $****P < 0.0001$, one-way ANOVA with the Dunnett T3 post hoc correction). **j** Difference in time between firing events and

running events. ($n$ = 12, 7, and 8 cells, $****P < 0.0001$, one-way ANOVA with the Bonferroni post hoc correction). $_{AC}$TeA-ChR2$^+$: 2.419 ± 0.184; AC-TeA-ChR2: 2.075 ± 0.202; TeA$_{dPAG}$-ChR2$^+$: 2.078 ± 0.148. **k** Virus injection and chemogenetic (CNO) and optogenetic (LED) interventions. $_{AC}$TeA and $_{S1}$TeA neurons expressing ChR2-EYFP; TeA$_{dPAG}$ neurons expressing hM4D(Gi). **l, m** Representative LED light (20 Hz, black bar: 5 s stimulus duration)-induced speed heatmap (**l**) and running speed traces (**m**) before (Pre), during (CNO), and recovery (Rec) from the effect of i.p. injections of CNO. **n** Induced peak speed for the Pre, CNO, and Rec periods. ($n$ = 7 animals, $****P < 0.0001$, one-way repeated-measures ANOVA with the Bonferroni post hoc correction). **o** L5 IT and PT neurons in the TeA microcircuit underlying sensory-induced escape. **f, g, m** Data are presented as mean ± SEM. **i, j, n** Data are presented as mean ± SD. (Source data are provided as a Source Data file; see Supplementary Data 1 for detailed statistics).

inputs and are potentially integrated and converted into running signals, which then activate R-neurons to produce running behaviours in the animal (Fig. 7o). The $_{Sens}$TeA CaMKII neurons in L5, which can receive different kinds of sensory inputs and have a broad apical tuft and relatively large number of basal dendrites, respond to sensory stimuli, and their firing rate is correlated with running speed. In contrast, the firing rate of L5a TeA$_{dPAG}$ neurons, which have a narrow apical dendrite tuft and few basal dendrites, is correlated with running speed, and these neurons can be activated by L5 $_{Sens}$TeA neurons. The activation of TeA$_{dPAG}$ neurons drives dPAG neurons and triggers running in the mice. Therefore, $_{TeA}$dPAG neurons serve as executors of running commands from TeA$_{dPAG}$ neurons, functioning as neurons controlling running (behaviour or motor activity). Alternatively, TeA$_{dPAG}$ neurons can be considered running (behaviour or motor) command neurons, whereas $_{Sens}$TeA neurons in L5 process diverse types of sensory information into running signals, acting as sensory running (behaviour or motor) decision neurons.

## Discussion

Flight is the innate ability of prey animals to escape imminent danger[55,56]. It has been extensively studied as an experimental model for exploring underlying emotional responses[57], learning and memory[58], decision[59], reward[60], and addiction[61]. Accordingly, flight behaviours are triggered by stimuli associated mainly with unisensory fear and aversion, including loud noises[1–5], looming sounds[6], looming sights[7–11], wind[4], and aversive odours[12]. Numerous issues related to flight can be investigated with the unisensory escape model, including sensation, sensory integration[62], sensory decision[63], motor encoding[8], motor commands[64], and motor execution[49]. In this study, we selected a turntable-based running model employing multiple aversive sensory stimuli, such as pure tones[62,63], white light flashes[65], and air puffs[5], to answer a series of fundamental scientific questions: which neural nucleus serves as the key node regulating escape responses triggered by multiple sensory stimuli? What are the characteristics of the sensory neurons, sensory–motor decision neurons, and motor command neurons? How do these neurons assemble into a microcircuit underlying escape in response to diverse sensory stimuli?

### The TeA as a critical node in circuits regulating escape behaviour

The TeA is capable of receiving diverse sensory inputs, including auditory, visual, and somatosensory inputs, from the cortex (e.g., AC, VC, and SC) and subcortex (e.g., MGB and DLG), which converge onto the TeA in L2–5 (Fig. 5a–c). This finding aligns with the results of previous studies on anatomic projections[31,33,37,41,51–54]. Our results indicate the existence of TeA neurons that respond to two or even three different types of sensory stimuli. Blocking unisensory input also affects the speed at which other sensory inputs induce behaviours, and the effect of unisensory input on behaviour is determined by the pattern specificity of the corresponding neuronal population's response (Fig. 5j–l). However, chemogenetic and optogenetic inhibition of TeA

completely blocked the flight behaviours induced by diverse sensory stimuli (Fig. 1r–w). The critical role of the TeA in this process was further underscored by the specific inhibition of the TeA–dPAG circuit, which also abolished escape responses (Figs. 3a–h and 7k–n). Conversely, its sufficiency in driving running was demonstrated by the direct activation of this circuit (Fig. 7b), positioning it as a key efferent pathway for escape. Importantly, the finding that inhibition of the TeA–dPAG circuit reduces spontaneous locomotion (Fig. 3i–p) suggests that its function extends beyond threat-evoked escape to include a more general role in modulating locomotor activity. This finding positions the TeA–dPAG pathway not only as a dedicated escape circuit but also as a key regulator of motor output that can be powerfully engaged by threatening stimuli. In addition to the SC[66,67] and prefrontal cortex, our work identifies a brain region and establishes a model for quantitative and qualitative studies of multisensory integration and behaviour initiation.

Our data reveal that sensory stimulation in mice triggers robust escape running but with a significant delay of 1–3 s (Figs. 1d, g, m, o and 4b, t). Although decreasing the stimulus intensity prolongs the response latency (Fig. 4b), this latency is markedly longer than that of subcortically controlled startle reflexes, such as the auditory startle response, where rapid reactions typically occur on a millisecond timescale[68,69]. A delay of several seconds suggests a decision-making process involving more complex neural integration rather than simple, reflexive, immediate escape. Our observations revealed that the mice were not unresponsive during this period. Although they did not exhibit freezing behaviour, we observed brief vigilant and orienting behaviours (e.g., head turning and ear twitching). This finding indicates that the sensory stimulus likely triggers a cascade of neural processes and behaviours before culminating in full-flight escape. We quantified the frequency and latency of these alert behaviours. They occurred with a high probability (probability of head turning in the open field escape model: 89.02% ± 15.35%; probability of sound-induced ear twitch in the turntable running model: 96% ± 8.944%) and a short latency (latency of head turning: 298.1 ± 94.7 ms; latency of ear twitch: 194.6 ± 53.6 ms).

Therefore, we propose that the identified TeA pathway is not the sole initiator of all sensory-evoked behaviours in this paradigm but may play a more specific role in the final commitment to and execution of coordinated escape locomotion. The initial, faster components of the reaction (e.g., vigilance and orientation) are likely mediated by parallel subcortical pathways that process the threat rapidly and prime the animal for action. The longer latency of the TeA-driven escape is consistent with the polysynaptic nature of this cortico-amygdalar pathway, which may allow for more integrated processing of sensory context before triggering a major locomotor action[70]. In this model, the TeA pathway acts as a critical gate, translating the integrated assessment of a threat into the decisive motor program of running to shelter.

This behavioural sequence suggests the existence of a parallel processing architecture within the brain. We speculate that a fast

subcortical pathway (e.g., potentially involving the superior colliculus and/or amygdala) might be responsible for the initial rapid vigilance and orienting responses to such aversive stimuli. Simultaneously, the slower integration process we observed in the TeA might be crucial for constructing a more complex threat representation—perhaps by integrating the sensory stimulus with contextual information—and ultimately contributes to the decision to coordinate and execute a coordinated escape plan rather than merely eliciting a simple startle response. Thus, our study suggests that the TeA is not the exclusive pathway for initiating perceptually guided behaviour but is a key node involved in the translation of integrated threat signals into sustained escape motor commands.

## Cells in the TeA transform sensory signals into behavioural commands

Certain brain regions, including the SC[71], prefrontal cortex[59,72,73], amygdala[74,75], tail of the striatum[6], and medial septum[4], are involved in determining and modulating running behaviours. Previous studies have focused mainly on the role of neural circuits, where different functions are typically performed by distinct neural nuclei, forming pathways. However, whether a single neural nucleus in the mouse brain can directly convert sensory input into motor output, particularly the distributions of sensory neurons, sensory–motor decision neurons, and motor command neurons at the cellular level, remains unknown.

The neurons we refer to as unSR-neurons exhibit firing responses to all three unisensory stimuli (Supplementary Fig. 2c), but show either no correlation or a significant but weak correlation ($r_{max} < 0.3$) between their firing rate and running speed (Fig. 4m–o, Supplementary Fig. 2f). This response type is consistent with that reported for most neurons found in the auditory[40], visual[65], and somatosensory systems[76]. Therefore, these neurons function as sensory neurons (Fig. 4m, Supplementary Fig. 3e-h).

Sensory–motor decision cells are the SR-neurons with thick-tufted dendrites (Fig. 6a–f) located in L5a and b (Fig. 4q), exhibit firing rates that respond to sensory stimuli (Supplementary Fig. 2c) and correlate with running speed (Fig. 4h, k, l, and Supplementary Fig. 2e). When repeatedly activated, these neurons can cumulatively drive the running behaviours of the animals through temporal summation (Fig. 7c). Therefore, they can be called sensation–behaviour (running) decision neurons. Since the SR-neurons are downstream of the unSR-neurons (Fig. 4s), they can also be called sensory running (motor or behaviour) decision neurons (Fig. 4h, and Supplementary Fig. 3a–d).

Motor command cells, i.e., the R-neurons in TeA L5a, are innervated by sensory running (motor or behaviour) decision neurons (Fig. 6g–j) and in turn project to the dPAG and trigger running behaviours (Fig. 7e). Since R-neurons' firing correlates with running speed (Fig. 4c, e, f, g, Supplementary Fig. 2b and d) but does not respond to sensory stimuli (Fig. 4d, and Supplementary Fig. 2a), more importantly, because these neurons can directly drive running behavior (Fig. 7d, e), they are defined as running (motor or behaviour) command neurons. This study reports the observation of these three distinct types of neurons, thereby providing a framework and methodologies for future motor and behaviour research.

SR-neurons are connected monosynaptically to (Fig. 6g–l) and initiate running behaviours through R-neurons (Fig. 7a–n). Both SR-neurons ($2.449 \pm 0.18$ s) and R-neurons ($2.117 \pm 0.24$ s) exhibited long firing–running time differences (Fig. 4u). Among them, the difference in firing–running time following the activation of $_{AC}$TeA-ChR2$^+$ neurons ($2.419 \pm 0.184$ s) was similar to that following the activation of SR neurons. The difference in the firing–running time of AC-TeA-ChR2$^-$ neurons ($2.075 \pm 0.202$ s) or TeA$_{dPAG}$-ChR2$^+$ neurons ($2.078 \pm 0.148$ s) was similar to that of R-neurons (Fig. 7j). Based on the difference in firing–running time, the monosynaptic processing time was estimated to be 332 ms from SR-neurons to R-neurons (Fig. 4u) and 341 ms from $_{AC}$TeA-ChR2$^+$ neurons to TeA$_{dPAG}$-ChR2$^+$ neurons (Fig. 7j). In contrast,

the latency from the blue light-induced synaptic activation of $_{AC}$TeA-ChR2$^+$ neurons to the spiking of AC-TeA-ChR2$^-$ neurons was slightly longer ($407.8 \pm 60$ ms) (Fig. 7i). This process has been confirmed to trigger running behaviours through a monosynaptic connection (Fig. 7k–n). These findings suggest that the projection from sensory–decision neurons to motor–command neurons in the TeA requires prolonged and gradual accumulation. As shown in Fig. 7c-d, although the firing of neurons responded stably to sensory stimuli or light, motor activity was not induced immediately at the start of stimulation. Instead, multiple stimuli were required to achieve cumulative effects for initiating running behaviours.

Taken together, these findings suggest that sensory–decision $_{Sens}$TeA neuronal projections to motor–command TeA$_{dPAG}$ neurons require prolonged temporal summation to sufficiently depolarize and activate TeA$_{dPAG}$ neuronal populations.

## A noncanonical IT–PT microcircuit in TeA L5

Based on their long-range projections, cortical pyramidal neurons are broadly categorized into two major types: IT neurons, which project only within the telencephalon (neocortex, striatum, amygdala, and claustrum), and PT neurons, which project to subcortical structures such as the pons and thalamus[77,78]. Running (motor) command neurons, i.e., TeA$_{dPAG}$ and CaMKII R-neurons (Supplementary Fig. 1a–c), project to the dPAG (Fig. 2; and Supplementary Fig. 1d–f) and should traditionally be defined as PT neurons. In general, classical PT neurons are located in L5 and are characterized by thick-tufted dendrites, large bodies and high firing rates[78,79]. However, TeA PT neurons are located in L5a (Figs. 2b, c; 4q; and 6c) and present distinct morphological features (slender-tufted dendrites, small bodies) (Fig. 6a–f) and a relatively low firing rate (Figs. 4r and 6j, k). In contrast, $_{Sens}$TeA neurons receiving projections from multiple sensory areas of the neocortex and subcortex (Fig. 5b, c, and h; Fig. 6a; Supplementary Fig. 6a and b) could be categorized as IT neurons. TeA IT neurons include two subtypes: TeA sensory (unSR) neurons with low firing rates (Fig. 4r) located in L2/3 and L4, and sensory running (motor) decision (SR) neurons located specifically in L5 (Fig. 4q), which is consistent with the typical distribution of IT neurons in L2–6[78]. Confusingly, however, IT neurons possess slender-tufted dendrites and a smaller body than PT neurons[77,79,80], whereas TeA IT neurons, especially the SR-neurons in L5 (Fig. 4q; Fig. 6a), are large cells with thick-tufted dendrites and a high firing rate (Fig. 6a–f, j, and k). L2/3 IT neurons send dense projections to IT neurons in L5a/b with interlaminar axons[81,82]. Thus, we propose that unSR-neurons (Fig. 4m, Supplementary Fig. 3e–h) deliver sensory information to SR-neurons (Fig. 4h, Supplementary Fig. 3a–d). In fact, the sensory response latencies of unSR-neurons are shorter than those of SR-neurons (Fig. 4s), indicating that TeA L2/3-4 IT neurons are upstream of TeA L5 IT neurons (Fig. 4q).

As PT neurons receive extensive inputs from IT neurons originating from multiple layers[83,84], we observed a connection between TeA L5a PT neurons (R-neurons) and TeA L5 IT neurons (SR-neurons) (Fig. 6g and h; and Supplementary Movie 1). This connection results in the activation of R-neurons (Fig. 6i and j) with a long latency (~400 ms) (Fig. 7i) through accumulated stimulation (temporal summation) (Fig. 7c and d). These observations further suggest that the projections from SR-neurons to R-neurons require prolonged temporal integration to elicit spiking activity in the TeA–dPAG circuit. Based on previous reports indicating that deep-layer excitatory neurons in the human temporal cortex can be classified by their dendritic orientation, with each neuron receiving rare but strong axonal inputs (up to 50 synapses) among thousands of weak connections, we hypothesize that functional connectivity from SR-neurons to R-neurons in mice may require hundreds of milliseconds of weak synaptic integration[85]. In this study, we present a noncanonical microcircuit between IT and PT neurons in the TeA L5 that processes diverse forms of sensory information into running (behaviour or motor) commands.

## Limitations and future directions

This study utilized an AAV-based anterograde tracing strategy to label input neurons projecting from other sensory cortices to the TeA, as well as to label output neurons projecting from the TeA to the dPAG. However, we must consider the potential limitations of this method. In particular, certain AAV serotypes are known to possess retrograde tracing capabilities and exhibit varying degrees of transsynaptic spread[86]. Although we selected serotypes designed to prioritize anterograde spread (e.g., AAV1), the possibility of retrograde labelling cannot be entirely ruled out.

The effect of this potential retrograde labelling on the TeA–dPAG circuit is likely minimal. Our preliminary retrograde tracing experiments revealed that after AAV-retro was injected into the TeA, no labelled neuronal cell bodies were observed in the dPAG, whereas other regions (such as the AC and S1) were clearly visible (Fig. 5). Furthermore, based on the Allen Brain Atlas database, currently, no evidence supports the existence of neurons in the dPAG that project directly to the TeA.

However, when TeA neurons receive inputs from other cortical areas, the risk of retrograde labelling cannot be ignored because of the extensive reciprocal projections between the TeA and other cortical regions. When a Cre-DIO strategy is used, in addition to labelling TeA neurons that receive projections from other cortical areas, labelling TeA neurons that project to other cortical regions is possible. Nevertheless, the comparative analysis revealed that when we injected retrograde tracers into A1, retrogradely labelled neurons were observed in the TeA. The morphology of these retrogradely labelled neurons (with only the soma clearly labelled) differed from that of the TeA neurons receiving AC projections labelled using our Cre + DIO method (Fig. 6a). Therefore, the neurons we labelled should still primarily be those receiving projections from other cortical areas.

Second, in terms of functional impact, let us use the validation of AC projections to the TeA in our experiments as an example. A more complex loop (e.g., TeA→AC→other nuclei) could theoretically contribute to running behaviour. However, our experimental evidence does not support its functional necessity. As shown in Fig. 7k–n, activating the AC→TeA input fails to induce running if the TeA→dPAG output is blocked. These findings suggest that even if such alternative loops exist, their capacity to drive behaviour is negligible without the key TeA→dPAG link. Additionally, our results in Fig. 6i–l confirm that TeA neurons receiving AC projections send direct projections to TeA$_{dPAG}$ neurons. Thus, the loop we originally investigated remains valid.

Therefore, although the theoretical risks mentioned above exist, we believe that this limitation does not undermine the core conclusions of this study. The key finding is that a specific population within the TeA region, defined by the viral strategy (i.e., neurons functionally connected to other cortices), is functionally distinct from other neurons. Regardless of whether this population is absolutely pure, the discovery of this functional heterogeneity is robust. Furthermore, all optogenetic manipulations (inhibition or excitation) and subsequent behavioural experiments in this study targeted the TeA region with light stimulation. Therefore, light stimulation affected all ChR2$^+$ neurons simultaneously, regardless of how they were labelled. The behavioural effects observed were the net result of coactivation or coinhibition of the entire labelled neuronal population (which may include both cell types mentioned above). Our conclusion that manipulating this population is sufficient to influence behaviour remains valid.

Naturally, employing more specific systems in the future (such as rabies virus-mediated trans-monosynaptic tracing or dual-labelling strategies) will enable precise dissection of the AC–TeA circuit composition and differentiation of the independent functional contributions of distinct subpopulations.

While our study establishes a functional distinction between sensory-related neurons (S-neurons) and running-related neurons (R-

neurons) in TeA, some limitations should be considered. To isolate sensory-evoked activity, we strictly selected trials without overt locomotion (running speed <0.5 cm/s). However, as noted in the preceding discussion, sensory stimuli can elicit brief orienting behaviors such as head turns or ear twitches. Although our experimental setup and analysis focused on dissociating neural activity from gross locomotion, we did not employ high-resolution, multi-parametric behavioral tracking (e.g., videography or pose estimation) to systematically quantify these subtle movements. Consequently, we cannot fully exclude the possibility that the sensory-driven activity we report is partially related to, or modulated by, such unmeasured, stimulus-induced behavioral changes. This represents a common and recognized challenge in studying neural coding in awake, behaving animals. Future work that combines dense electrophysiological or imaging recordings with comprehensive behavioral phenotyping will be crucial to further disentangle pure sensory signals from activity related to specific motor outputs or internal states.

## Methods

### Animals

Male and female C57BL/6 J (Laboratory Animal Center of Southern Medical University, Guangzhou, China), *SOM-Cre* (RRID: IMSR_JAX: 013044), *Vip-Cre* (RRID: IMSR_JAX: 031628), *PV-Cre* (RRID: IMSR_JAX: 008069), and *Ai14* (Cre-dependent tdTomato reporter line RRID: IMSR_JAX: 007914) (Model Organisms Center, Shanghai) mice aged 4–12 weeks were used in this study. All the Cre driver lines were crossed with Ai14 mice. The mice were housed in a vivarium with the temperature controlled at 21–25 °C, humidity controlled at 50–60%, and a 12-h day/night cycle (lights on at 8 am). Food and water were available ad libitum. The experiments were conducted during the day phase. Implanted animals were habituated to handling by the researcher cupping the mouse in his or her hand for 5 min per day for 2 days before the start of the experiment. All animal procedures were conducted in accordance with the Regulations on the Management of Laboratory Animals (China) and were approved by the Animal Care and Use Committee of Southern Medical University (L2017207).

### Viral vectors

For retrograde monosynaptic tracing, we used AAV2/retro-hSyn-mCherry (5.53 × 10$^{12}$ vg/mL, BrainVTA), AAV2/retro-hSyn-Cre (2.2×10$^{12}$ vg/mL, BrainVTA), AAV2/retro-hSyn-EYFP (5.18 × 10$^{12}$ vg/mL, BrainVTA), AAV2/retro-CaMKII-mCherry (5.17 × 10$^{12}$ vg/mL, BrainVTA), and AAV2/retro-CMV-WGA-NLS-Cre (2.6 × 10$^{13}$ vg/mL, OBiO).

For anterograde tracing, we used AAV2/9-hSyn-EYFP (4.6 × 10$^{12}$ vg/mL, BrainVTA), AAV2/1-Cre-EYFP (4.6×10$^{12}$ vg/mL, BrainVTA), AAV2/9-hSyn-mCherry (5.84 × 10$^{12}$ vg/mL, BrainVTA), AAV2/1-hSyn-Flp (1.02 × 10$^{13}$ vg/mL, BrainVTA), AAV2/9-hSyn-fDIO-EYFP (5.27 × 10$^{12}$ vg/mL, BrainVTA), AAV2/1-hSyn-Cre (1.16 × 10$^{13}$ vg/mL, BrainVTA), AAV2/9-hSyn-DIO-mCherry (5.82 × 10$^{12}$ vg/mL, BrainVTA), AAV2/9-CaMKIIα-DIO-mCherry (5.72 × 10$^{12}$ vg/mL, BrainVTA), AAV2/9-hSyn-fDIO-EYFP (5.27 × 10$^{12}$ vg/mL, BrainVTA), and AAV2/9-hSyn-DIO-EYFP (5.05 × 10$^{12}$ vg/mL, BrainVTA). AAV2/1 was used for anterograde monosynaptic tracing[86–88].

For optogenetic activation and inhibition, we used AAV2/9-hSyn-eNpHR3.0-EYFP (4.47 × 10$^{12}$ vg/mL, OBiO), AAV2/retro-CaMKIIα-eNpHR3.0-EYFP (4 × 10$^{12}$ vg/mL, OBiO), AAV2/9-Ef1α-DIO-hChR2(H134R)-EYFP (2.15 × 10$^{13}$ vg/mL, OBiO), AAV2/retro-CaMKIIα-hChR2(H134R)-EYFP (8.07 × 10$^{12}$ vg/mL, BrainVTA), and AAV2/9-hSyn-fDIO-hChR2-EYFP (5.25 × 10$^{12}$ vg/mL, BrainVTA). For the empty vector control group, we used AAV2/9-hSyn-EYFP (4.6 × 10$^{12}$ vg/mL, BrainVTA) and AAV2/retro-CaMKIIα-EYFP (3.8 × 10$^{12}$ vg/mL, BrainVTA).

For chemogenetic activation, we used AAV2/9-hSyn-hM4D(Gi)-mCherry (2.5 × 10$^{12}$ vg/mL, OBiO), AAV2/9-Ef1α-DIO-hM4D(Gi)-mCherry (5.18 × 10$^{12}$ vg/mL, BrainVTA), and AAV2/9-CaMKIIα-DIO-hM4D(Gi)-mCherry (5.5 × 10$^{12}$ vg/mL, BrainVTA). For the empty vector

control group, we used AAV2/9-hSyn-mCherry ($1 \times 10^{12}$ vg/mL, OBiO), AAV2/9-Ef1α-DIO-mCherry ($5.13 \times 10^{12}$ vg/ml, BrainVTA), and AAV2/9-CaMKIIα-DIO-mCherry ($5.1 \times 10^{12}$ vg/ml, BrainVTA).

## Animal preparation

Animal preparation was performed in a relatively sterile room at a temperature of 23–26 °C.

**Virus injection.** Four- to five-week-old mice were anaesthetized with sodium pentobarbital (60–70 mg/kg, 15 mg/ml, i.p., Sigma, USA) and positioned in a stereotaxic frame (RWD Life Science, China). Body temperature was maintained at 37 °C with a heating pad. After a median incision was made, small skull openings ($\leq 0.5$ mm$^2$) were generated over the target sites with a miniature handheld cranial drill (RWD Life Science, China) to allow virus injection through a glass micropipette (tip diameter: 20–30 μm; Drummond Scientific, USA) driven by a microinjector (KD Scientific, USA). A 40–100 nl volume of the virus (depending on the desired expression strength and viral titre) was injected at a rate of 30 nl/minute.

The stereotaxic coordinates (posterior to the bregma, lateral to the midline, and below the brain surface, in mm) for the different target sites were as follows: TeA: −2 to −3, 4.7, 0.5; A1: −3.0, 4.2, 0.5; S1: −1.7, 3.0, 0.6; V2L: −4.0, 3.4, 0.5; MGB: −2.8, 2.0, 3.0; DLG: −2.3, 2.2, 2.3; and dPAG: −4.0, 0.2, 2.2. The head was laterally rotated for precise injection at 80° into the TeA and A1. The glass micropipette was laterally rotated 15° for injection into the V2L. For injection into S1 and the MGB, DLG, and dPAG, the animal's head was fixed as described by The Allen Institute (2011). After the injection, the virus needle was retracted to a depth of ~100 μm at the injection point and left in place for 5–10 minutes to allow the virus to fully spread. Medical sutures were used to close the wounds, and erythromycin ointment was applied to prevent infection. After the mice woke, they were returned to the animal facility for feeding. The mice were collected for the corresponding experiments 4 weeks after the virus injection.

**Head fixation.** Except for the animals designated for the open-field experiment, all the mice underwent head-fixation surgery. At least three days before the experiments, the mice were deeply anaesthetized with pentobarbital sodium (60–70 mg/kg, i.p., Sigma, USA) and fixed onto a stereotaxic instrument. Lidocaine was injected locally under the scalp, the skull was exposed, and the exudate was removed. A tripod-shaped head nail was attached to the skull with dental cement.

**Electrophysiological recording window.** For in vivo electrophysiological recordings in the TeA, some of the head-fixed mice simultaneously underwent craniotomy over the TeA with preservation of the dura. Vaseline was applied to cover the surface of the dura.

**Cannula implantation.** For optical stimulation of the TeA, one or two optic fibre cannulas (0.2 mm, 0.37 NA core diameter; Newdoon, China) with inner cores were embedded into the skull above the unilateral or bilateral TeA with dental cement during head-fixation surgery. The cannulas were anchored in a horizontal position within the TeA at a depth of 300 μm from the dura. After the operation, the mice were allowed to recover in their cages, and antibiotic ointment was applied to the wounds to prevent infection. Cannulas were not implanted in mice in which in vivo electrophysiological recordings were to be obtained at the same time as optogenetic activation during the experiment. Instead, an internal injection tube connected to the silicone hose or optical fibre jumpers connected to the optical fibre core pin was directly fixed to a micropropulsor (NARISHIGE, MN-151), which was slowly advanced to the anchor point in a horizontal position within the TeA at depths of 400–500 μm.

**Running training.** After three days of recovery (but prior to the electrophysiological recordings or behavioural experiments), the head-fixed mice were placed on a circular plate (diameter, 20 cm) to adapt to free running under head fixation conditions, in which the mice were head fixed by screwing their tripod pins to a metal pole. The height of the disc was adjusted so that the mice could adapt to free running on the disc while the head remained stable. Following three consecutive days of daily training (1–2 hours/day), 95% of the animals successfully adapted to the head-fixed experimental paradigm, demonstrating stable, self-initiated locomotion on the rotary disc platform with coordinated limb movements. Animals that failed to maintain such locomotor stability (5%), characterized by limb coordination deficits or complete immobility, were excluded from subsequent recordings and analysis to ensure experimental consistency.

**Preparation for the recording sessions.** The electrophysiological recordings and behavioural tests were all performed in a soundproof room at a temperature of 23–26 °C. Individual recording sessions lasted for no more than 2 h. The animal was administered 5% sucrose drops through a pipette every hour. Some animals were subjected to more than one session of recording, separated by at least 1 day.

## Sensory stimulation

Sound (pure tone, white noise), light (white light flash), and air puff stimuli were used, each of which was controlled by a Tucker–Davis Technologies System 3 (TDT3). The control programs were written with RPvdsEx software and loaded into the multifunction processor (RX6) of the TDT3 system. The three types of sensory stimuli were presented in a pseudorandom order. Each stimulus was repeated only 10 times to prevent habituation in mice, which could diminish behavioural responses, while also ensuring the validity of our results.

**Sound stimuli** were generated with the TDT3 system. The synthesized signals produced by RX6 (characterized as sine waves starting at the zero phase and a 5 ms rise/fall time) were amplified by an electrostatic speaker driver (ED1) and delivered through a loudspeaker (ES1, frequency range 2–110 kHz) placed 10 cm in front of the animal. The intensities of the sounds were controlled by a programmable attenuator (PA5). Before the experiments were performed, the loudspeaker was calibrated with an amplifier (Bruker and Kjaer 2610) connected to 1/4- and 1/8-inch microphones (Bruker and Kjaer 4135). All tone amplitudes are reported as the sound pressure level (SPL; 0 dB = 20 μPa). The sound parameters, such as the duration, frequency, and intensity, were controlled with an RP2.1 real-time processor controlled by BrainWare software. Pure tones were used with durations of 1, 5, and 8 s; frequencies of 6.5, 12, and 18.5 kHz; and intensities of 50, 80, and 110 dB SPL were used to drive running behaviours in the animals. White noise stimuli (2–50 kHz, 50 ms duration, and a 5 ms rise/fall time) were generated to examine the effects of drugs and viruses on AP recordings. Noise bursts varying in amplitude (0–90 dB SPL, 10 dB steps) were produced during the amplitude scan (A-scan). The sound parameters of the A-scans were controlled by BrainWare software through a computer and were presented pseudorandomly at a rate of 1/2 s. The A-scans were repeated 20–30 times to obtain a set of peri-stimulus time histograms (PSTHs).

**Light stimuli** were delivered through LED lamp beads driven by the RP2.1 real-time processor of the TDT3 system. The parameters of the white light flash, including the duration (5 s) and intensity (2 or 5 lux; calibration with KOMAX, Germany), were regulated through the RP2.1 real-time processor controlled by BrainWare software.

**Air puff stimuli** were generated through a microvalve (Kamoer-KVP04, China) controlled by an RP2.1 real-time processor. The air puff was presented through a tube connected to the microvalve placed 10 cm away from the animal. The air puff stimulation parameters

included a duration of 5 s and an air intensity of 0.8 L/min or 1.2 L/min (calibrated with Darhor-LZB-2, USA).

For the multimodal sensory integration experiments, the intensities of the sound (30 dB SPL), light (1 lux), and air puff (0.4 L/min) stimuli were selected through multiple tests to ensure that none of them could significantly drive mouse running behaviours, while the stimulus duration was 5 s. We used the following stimulus parameters to record the firing of a TeA neuron: sound, 50 dB, 12 kHz, for 1 s or 5 s; light, 2 lux for 1 s or 5 s; and air, 0.8 L/min for 1 s or 5 s.

### Manipulation

**Chemogenetic manipulation.** Mice that expressed hM4D(Gi) were intraperitoneally injected with CNO (0.33 mg/ml, 1 mg/kg). Prior to the formal experiments, a high-concentration stock solution of CNO (10 mg/ml) was prepared by dissolving CNO powder (BrainVTA, China) in sterile dimethyl sulfoxide (DMSO) (HY-Y0320; Med-Chem Express, USA). Briefly, sterile DMSO was added to the tube containing the CNO powder. The tube was tightly capped and vortexed vigorously for 30–60 seconds to achieve initial dispersion of the powder. The tube was then subjected to sonication in an ultrasonicator for 5–10 minutes to facilitate complete dissolution. The tube was inspected visually to confirm complete dissolution, indicated by a clear and transparent appearance without any undissolved particles at the bottom or on the walls of the tube. If the particles remained, the cycle of vortexing and sonication was repeated until the solution was entirely clear. Once fully dissolved, the stock solution was immediately aliquoted into single-use portions to avoid repeated freeze–thaw cycles and stored at −20 °C until use. The stock solution was thawed and diluted in sterile physiological saline (0.9% NaCl) to the desired working concentration (0.33 mg/ml) immediately prior to intraperitoneal (i.p.) injection into the animals.

**Optogenetic manipulation.** For optogenetic manipulation, an optical fibre jumper connected to an LED source (blue light 473 nm, yellow light 593.5 nm; THINKERTECH, China) was inserted at the target site at the proper depth for hChR2- or eNpHR3.0-expressing mice through the optic fibre cannula. The optic fibre tip was covered with black tape to prevent light leakage. No hChR2- or eNpHR3.0-expressing neural fibres or structures other than the target structure were present in the light pathway within 800 μm from the end of the optic fibre to ensure the specificity of the optogenetic stimulation. The axis of the light path was the same as the central axis of the optic fibre, and the NA of the fibre affected the illumination angle[40]. After each experiment, the brains were removed, sectioned and imaged under a confocal microscope to confirm the expression of hChR2-mCherry or eNpHR3.0-EYFP. Data from animals for which the injections were unsuccessful were excluded from further analysis.

### Running behaviour test

**Open-field escape model.** A custom-made rectangular chamber was constructed from opaque acrylic material with dimensions of 50 cm (length) × 6 cm (width) × 15 cm (height). Two circular openings (4 cm in diameter) were made in the lower portions of both sidewalls to place high-frequency speakers for auditory stimulation, LED bulbs for light stimulation, and silicone tubing for air puff delivery. Additional 5 cm-wide apertures were placed at the mid-section of both sidewalls to prevent the mice from remaining stationary in a specific area of the chamber for extended periods. For the auditory stimulation experiments, high-frequency speakers were installed on both sides of the chamber. A high-definition infrared camera (LRCP10620, 20 FPS, China) mounted above the chamber allowed real-time monitoring of the locomotor patterns and positional changes of the mice from outside the shielded testing environment. The two speakers were independently controlled, allowing the experimenters to deliver auditory stimuli along specific directions through camera-guided position

tracking using specialized imaging software. The chamber was elevated 30 cm above ground level to prevent animals from escaping through the lateral openings.

During the experimental procedures, the mice were gently grasped by the tail with forceps and positioned at the central opening as video recording commenced. Following a 10-minute acclimatization period, during which the mice typically occupied the end of either the left or right chamber, auditory stimulation was delivered through the speaker adjacent to the animal's position to elicit contralateral movement. The light and air puff stimulation protocols involved similar operational principles, with LED bulbs and silicone tubing replacing the speakers at the corresponding positions. The stimulation timing was randomized according to the real-time locomotor status of the animal, with each mouse performing 10 trials per stimulation type. Notably, no freezing behaviours were observed in the mice either prior to movement initiation or during sensory stimulation.

**Turntable running model.** After adapting to the head-fixing running paradigm, the mice could run freely on a plastic circular plate (diameter, 20 cm), under which a rotary encoder (E2-US Digital, 20 kHz, USA) was mounted on the stubble to record the turning speed. An external high-definition infrared camera (LRCP10620, 20 FPS, China) positioned ~15 cm above the plate was used to monitor the behaviour of the animals; both the behaviour and the turning speed were monitored in real time. All the data were digitized and stored on a computer for online and offline analyses. Unless specified otherwise, all in vivo electrophysiology, optogenetic, and chemogenetic behavioural experiments were performed with a head-fixed turntable model.

The running behaviour tests and in vivo cellular recordings were performed in a soundproof room at 23–26 °C. Sound, light, or air puff stimuli were delivered from a loudspeaker, an LED lamp, or a silicone hose placed 10 cm in front of the animal, respectively. During the experiments, all the equipment (the stimulation software BrainWare, the speed recording software US Digital, and the camera recording software used to collect the data) was triggered by the RP2.1 real-time processor with a custom-made program. The three sensory stimuli were presented in a pseudorandom manner with a 25-s interstimulus spacing, and this process was repeated 10 times. Each recording session lasted 15 minutes.

The open-field test was conducted during the night cycle, beginning 90 min after the lights were turned off and after an acclimation period of at least 2 h to the testing room. A 3-day window was maintained between the open-field tests to avoid intertest effects. The testing room was dimly lit by a lamp with a luminosity between 5 and 20 lux. The open-field test serves as a standard procedure for evaluating spontaneous movement and curiosity-driven actions in rodents. Experiments were conducted in a rectangular enclosure (50 cm length × 50 cm width × 40 cm height) partitioned into core and edge zones through the VisuTrack behavioural analysis platform (Shanghai XinRuan Information Technology Co., Ltd., China). The animals were carefully picked up by the tail from their housing units and centrally positioned within the testing area. Data acquisition commenced automatically upon the detection of the first motion-triggered sensor interruption and then continued continuously over a 5-minute period. The subsequent analysis of rodent navigation paths was conducted using the VisuTrack software suite.

### Electrophysiological recordings

**In vivo cellular recordings[40,89].** The running-trained mice were head fixed by screwing their tripod pins into a metal pole so that they could run freely on the running turntable, and the Vaseline and dura were removed. A patch pipette (tip diameter: 1.5 μm, resistance: 5–8 MΩ) filled with artificial cerebrospinal fluid (ACSF, containing (in mM) 126 NaCl, 2.5 KCl, 1.25 NaH$_2$PO$_4$, 26 NaHCO$_3$, 1 MgCl$_2$.6H$_2$O, 2 CaCl$_2$, 2 sodium pyruvate, and 10 glucose; pH 7.35–7.45; osmolarity

290–310 mOsm/kg) and controlled by a micromanipulator (Siskiyou, USA) was embedded into the TeA horizontally for in vivo cellular recordings. A slight positive pressure (0.5–1 psi) was applied before the pipette was embedded to prevent the tip from clogging. The pipette was quickly advanced to a depth of 100 μm and then further advanced in 1 μm steps. When the impedance changed to 5–10 MΩ, the pressure was switched to negative (−0.3 psi) until a loose seal was achieved (30–100 MΩ), indicating that the cellular attachment was suitable for recording spikes.

During the experiment, all the equipment (the stimulation software BrainWare, the speed recording software US Digital, and the camera recording software used to collect the data) was triggered by the RP2.1 real-time processor with a custom-made program to record the neuronal activity and the running speed and running state of the mouse while it ran on the turntable. Cellular recordings were performed in voltage-clamp mode (Vcmd = 0 mV) using a MultiClamp 700B amplifier (Axon, USA). The signals were bandpass filtered at 300–3000 Hz and sampled at 20 kHz. Clampex software was used to detect and record the original signal, and Clampfit and BrainWare software were used to observe and record neuronal firing (action potentials). Following the establishment of a seal resistance ranging from 0.2 to 1 GΩ[90], the amplifier was switched from voltage-clamp to current-clamp mode. A positive direct current (3–10 nA, 1 Hz) was applied for 20 minutes to iontophoretically deliver biocytin for an assessment of neuronal morphology. The biocytin tracer was prepared as a 1% (wt/vol) solution to enable the subsequent visualization of neuronal morphology. A total of 20 mice were used for the intracellular recordings, with an average of 6–10 neurons recorded per mouse.

We primarily employed biocytin labelling to localize the recorded neurons and obtain their coordinates during recording; however, the labelling efficiency was suboptimal. Accordingly, for a subset of neurons, an alternative approach was adopted. Following the electrophysiological recordings, glass micropipettes with larger tip diameters than those of standard recording electrodes were filled with an eosin dye solution. This dye was then pressure-ejected (~2 psi) at the recording site. Subsequent frozen sectioning and histological examinations allowed us to confirm whether the recording site was located within the TeA.

**Brain slice electrophysiology.** The mice were anaesthetized with urethane[40,91]. After the mouse brains were rapidly dissected, 300-μm-thick coronal slices containing the TeA from the virus-infected hemisphere were prepared with a vibratome (Leica, VT1200S) in ice-cold cutting solution (60 mM NaCl, 3 mM KCl, 1.25 mM NaH$_2$PO$_4$, 25 mM NaHCO$_3$, 115 mM sucrose, 10 mM glucose, 7 mM MgCl$_2$, 0.5 mM CaCl$_2$; 300–305 mOsm/L; pH 7.4). Slices were transferred to the holding chamber and incubated in fresh ACSF at 34 °C to recover for 30 min. The slices were incubated at room temperature before recording. The cutting solution and ACSF were continuously bubbled with 95% O$_2$ and 5% CO$_2$. Slices were submerged in the recording chamber and perfused with ACSF at 3–4 ml/min using a pump (longer pump). Whole-cell patch-clamp recordings were performed under an upright microscope (Eclipse FN1, Nikon). Patch pipettes were filled with a K⁺-based internal solution (containing 140 mM K⁺-gluconate, 9 mM HEPES, 5 mM EGTA, 4 mM Mg-ATP, 0.3 mM GTP, 4.5 mM MgCl$_2$, and 4.4 mM phosphocreatine sodium; pH 7.3; 295 mOsm). The spatial expression patterns of ChR2-EYFP and hSyn-mCherry were examined in each slice under a fluorescence microscope before recording; only slices in which the expression patterns were observed in the correct locations were used for further recording. In whole-cell current-clamp mode, the voltage response was recorded from the cells by injecting a step current (amplitude range −100 pA to 620 pA, step duration 300 ms, and interval 20 pA). EPSCs and IPSCs were recorded by clamping the cell membrane potential at −70 and 0 mV, respectively. Light-evoked EPSCs and IPSCs were recorded by delivering 5 ms of 5 mW blue LED light (473 nm, Thorlabs) through the objective to the TeA. Light-evoked PSCs were recorded before and after perfusion with 1 μM of the sodium channel blocker TTX and 1 mM of the potassium channel blocker 4-AP to measure the responses of different synapses.

Brain slices were prepared similarly, and whole-cell current-clamp recordings were obtained from neurons expressing hM4D(Gi) to test its efficacy. Spontaneous spikes were recorded from neurons expressing hM4D(Gi) before and after perfusion with CNO (10 μM) and after CNO wash out.

## Immunohistochemistry

After the behavioural experiments were performed, the mice were deeply anaesthetized with pentobarbital sodium (60–70 mg/kg, i.p.) and sequentially perfused with saline and 4% (wt/vol) PFA (pH 7.4). The brains were subsequently removed and postfixed with 4% PFA at 4 °C overnight. After the brains were cryoprotected with 30% (wt/vol) sucrose, coronal sections (40 or 100 μm) were cut on a cryostat (Leica CM1860, Germany).

For immunofluorescence staining, the brain sections were then washed with PBS (15 min, 3 times), incubated with 0.3% (v/v) Triton X-100 for 1 h, blocked with 5% normal goat serum (NGS, Boster) for 1 h at room temperature, and then incubated with primary antibodies, including an anti-CaMKII monoclonal rabbit antibody (1:200; ab52476; Abcam, USA), anti-GAD67 monoclonal mouse antibody (1:200; MAB5406; Millipore, USA) or streptavidin-Cy3 (1:200; 438315; Thermo Fisher Scientific, USA), in 5% NGS overnight at 4 °C. For anti-CaMKII immunofluorescence staining, sections were incubated with the corresponding fluorophore-conjugated secondary antibody, goat anti-rabbit IgG Alexa Fluor 647 (1:500; A21244; Invitrogen™, USA), in PBS for 2 h at room temperature. Sections stained with the GAD67 antibody were incubated with goat anti-mouse IgG (H + L) cross-adsorbed secondary antibody [Alexa Fluor 488 (1:500; A32723, Invitrogen™, USA)], followed by washes with PBS (10 min, 3 times).

Finally, all the sections were mounted onto microscope slides and covered with coverslips along with an anti-fade reagent containing DAPI (S2110; Solarbio, Beijing).

## Image acquisition and quantification

The fluorescence signals were visualized with a laser scanning confocal microscope (A1R, Nikon) and analysed using ImageJ 1.4 (NIH) or NIS-Elements software. All images of each brain were captured with the same settings. Confocal images were acquired, and cells were counted using ImageJ. The imaging settings were kept constant for all groups of sections. The fluorescence intensity of the presynaptic terminals was quantified with ImageJ. All fluorescence images were obtained by capturing serial z-stack images through the 10X, 20X, or 40X objectives of the confocal microscope, and the cell morphology was reconstructed in volume view (A1R, Nikon). Confocal image stacks were imported into FIJI software. Background noise was reduced using the "Subtract Background" function, and contrast was appropriately enhanced to improve neurite visibility. The soma was identified either in the maximum intensity projection or in the optical slice where it appeared largest and most clearly defined. With the "Freehand selection" tool, the somatic contour was carefully traced, and its cross-sectional area (in μm²) was measured using the "Measure" command. The "Simple Neurite Tracer" plugin was used for semiautomated 3D reconstruction of dendritic and axonal processes. Tracing began at the soma, and paths were manually followed through the z-stack by placing nodes along the neurites. The plugin interpolates between nodes to create a 3D structure. Tracing continued until all visible neuronal processes were fully reconstructed. Sholl analysis was performed using the "Sholl Analysis" plugin. The plugin calculates the number of intersections between the neurites and each concentric circle, generating a profile of the dendritic density as a function of radial distance from the soma. All measured data, including the somal area and number of Sholl intersections, were exported for further statistical

analysis and graphing using the appropriate software (e.g., GraphPad Prism, SPSS).

In accordance with the Allen Brain Atlas, each brain slice was converted to an 8-bit image by manually drawing selection contours that contained the brain region of interest. The fluorescence density was subsequently calculated by summing the grey value of each pixel using a blind method and dividing the sum by the number of pixels. The axonal density of each brain structure was normalized to the mean fluorescence density of $_{Sens}$TeA neurons and TeA$_{dPAG}$ neurons. Using standard histological methods and confocal microscopy imaging, we also validated the location of the optical fibre tips for all optogenetic behavioural experiments in this study.

### Data collection and statistical analysis

BrainWare software for the TDT3 system and Clampfit10.2 software for the Axon amplifier were used to export and analyse the recorded electrophysiological data. The custom software Animal.exe (Shanghai Jiliang Company) was used to analyse and export the recorded video data. In addition to these data, all experimental data, including the number of APs per unit time (firing rate), firing duration, firing latency, and AP waveforms and the speed, latency, and duration of the mouse running behaviours, were extracted and analysed offline with MATLAB 2016b (MathWorks) software. Microsoft Excel 2016, Origin 8.0, and MATLAB 2016b were used for data processing and further analysis. We analysed the data with custom software (MATLAB, MathWorks). The recorded running speeds were aggregated and processed for each animal individually using custom MATLAB code. The 30 stimulus–movement trials per animal were categorized by stimulus type, resulting in three groups: 10 sound–speed curves, 10 light–speed curves, and 10 air puff–speed curves. Each set of curves was then aligned to the stimulus onset and averaged, producing a mean movement speed curve for each animal in response to each type of sensory stimulus. The average time-dependent speed profile and neuronal firing rate curve were smoothed with a smoothing function (smooth) in MATLAB. Specifically, we used MATLAB's built-in function yy = smooth(y), which applies a moving average filter (window width = 0.5 s) to the column vector y and returns a smoothed column vector yy of the same length. we applied a compensatory delay to the latency measurements to correct for the effects of data smoothing on the calculated motion onset latency (i.e., advanced by -0.2 s).

During the behavioural tests, the rotation speed of the turntable was measured with a rotary encoder and recorded in real time[89]. We defined "running behaviour" as movements in which the running speed of the mouse exceeded 0.5 cm/s, and the onset of the running behaviour was defined as the point at which this speed threshold was first exceeded. The onset of sensory-evoked flight was defined as the time point when the speed exceeded the baseline level (mean speed during the 5 s prestimulus period) by 2 standard deviations (SDs). Neural data were analysed with custom MATLAB (MathWorks) scripts. A mean spike density function was constructed for each neuron by applying a Gaussian kernel ($\sigma = 10$ ms) to each spike.

### Definitions of the Response Windows for Evoked Behaviours.

We defined a response window for each experimental paradigm based on a comparative analysis of pre- and poststimulus running latencies to objectively distinguish sensory-evoked escape behaviour from spontaneous running. We confirmed that the distribution of running latencies following sensory stimulation significantly differed from the distribution of spontaneous running intervals during the pre-stimulus baseline period using a two-sample Kolmogorov–Smirnov test. This significant difference confirmed that the poststimulus running events constituted a distinct, time-locked population. Given this clear separation, we defined the response window to capture the specific temporal characteristics of the evoked response. The upper bound of the window was set to the 99th percentile of the poststimulus-evoked

latency distribution. This approach ensures that the window is tailored to the actual temporal profile of the stimulus-locked behaviour, providing a sensitive measure of elicitation efficacy. The spontaneous latency distribution was used as a null model to statistically confirm the specificity of the evoked behaviour within the defined window. We performed a binomial test to confirm that these responses were specifically evoked by the stimulus rather than occurring by chance. For all three experimental models, binomial tests were conducted against the null hypothesis that poststimulus running occurred at the spontaneous rate. The tests revealed that the number of runs within the defined response windows significantly exceeded the number expected by chance ($P < 10^{-4}$ in each model), providing strong statistical evidence that running was specifically evoked by the sensory stimulus.

**Neuronal Classification.** Recorded neurons were classified based on their responses to sensory stimuli and the correlation of their spiking activity with running speed. To dissociate sensory responses from running-related activity, analyses of sensory responses were performed exclusively using trials in which sensory stimulation failed to elicit running behavior.

1.  Assessment of Sensory Responses.

    Trial Selection: To isolate pure sensory responses, we analyzed only trials where sensory stimulation did not successfully evoke running. A trial was defined as a "non-running" trial if the running speed remained below a threshold of 0.5 cm/s throughout the posts-timulus-running window [0, 3.4 s].

    Response Definition: For each neuron and each sensory modality (sound, light, air puff), a stimulus-aligned peri-stimulus time histogram (PSTH; bin size = 10 ms) was constructed. A neuron was considered responsive to a given stimulus if the mean firing rate within the post-stimulus response window [0, 1] s showed a significant increase compared to the mean firing rate during a 2-s pre-stimulus baseline window (Wilcoxon signed-rank test, $P < 0.05$). The response threshold was defined as the mean baseline rate plus two times the standard deviation (baseline + 2 × SD). The mean baseline rate and its standard deviation (SD) were calculated from a 2 s pre-stimulus baseline window.

    Definition of S-neurons: Neurons that exhibited a significant response to at least one type of sensory stimulus were classified as sensory-related neurons (S-neurons). Neurons not meeting this criterion proceeded to the assessment for running-related activity.

2.  Assessment of Running Correlation.

    Trial Selection: Running-related activity was assessed using all spontaneous running events (i.e., running initiated in the absence of any sensory stimulus). Run onset was defined as the time point when running speed first exceeded a threshold of 0.5 cm/s.

    Response Definition: Running onset times were aligned to time zero for each neuron. Peri-event time histograms (PETHs) were constructed (bin size = 10 ms) centered on running onset. A neuron was considered to exhibit a significant running-related change in activity if its mean firing rate during the defined pre-running window showed a significant increase compared to the mean firing rate during a quiescent baseline period (Wilcoxon signed-rank test, $P < 0.05$). The activity threshold for defining the onset of this change was set as the mean baseline firing rate plus two times its standard deviation (baseline + 2 × SD). The mean baseline rate and standard deviation were calculated from periods when the mouse was not running.

3.  Final Classification

    Cross-Correlation Function (CCF) Analysis: For each neuron, the CCF between its firing rate and running speed was computed over a

time lag range of [−10, 10] s with 100 ms bins. The maximum correlation coefficient ($r_{max}$) and its corresponding time lag (τ) were recorded.

Significance Testing: The statistical significance of the observed correlation was assessed using a permutation test. A null distribution was generated by performing F = 1000 random circular shuffles of the running speed trace relative to the firing rate. The correlation was deemed statistically significant if the absolute value of the observed |$r_{max}$| exceeded the 95th percentile of this null distribution ($P < 0.05$).

Neurons were categorically assigned as follows: R-neurons (Running-related neurons): Neurons that were not classified as S-neurons (i.e., no significant sensory response) but showed a significant positive correlation ($P < 0.05$) between their firing rate and running speed. SR-neurons: Neurons that were classified as S-neurons (significant sensory response) and also exhibited a significant positive correlation ($P < 0.05$) between their firing rate and running speed. unSR-neurons: Neurons that were classified as S-neurons either show no significant correlation between firing rate and running speed ($P \geq 0.05$), or exhibit a correlation ($P < 0.05$) with a maximum correlation coefficient $r_{max} < 0.3$.

### Heatmap analysis of R-neurons and S-neurons in response to different sensor stimuli

To clearly visualize the response patterns of different neuronal populations to sensory stimuli, we converted raw raster plot data into firing rate heatmaps. Each row represents the average activity of a single neuron across all trials. All recorded neurons in the TeA region were presented with three types of sensory stimuli: auditory (sound), visual (light), and tactile (air puff), and corresponding heatmaps were generated for each stimulus modality. Only trials without running were included in the analysis. Non-running trials were defined based on a stimulus-running window: trials with no running events within 3.4 s after stimulus onset were considered unrelated to the stimulus, as determined by prior analysis of stimulus-running latency. The time bin size was set to 0.1 s to balance temporal resolution and statistical stability. Stimulus parameters were as follows: stimulus onset at 2.0 s, stimulus offset at 7.0 s, stimulus duration of 5.0 s, and the analysis time window spanned 0 to 10 s (aligned to stimulus onset at 2.0 s).

For each neuron i, the average firing rate in time bin j was calculated as:

$$Frequency_{ij} = SpikeCoun_{ij}/(N_t \cdot \Delta t)$$

where $SpikeCount_{ij}$ is the total number of spikes of neuron i in time bin j across all non-running trials, $N_i$ is the number of non-running trials for neuron i, and $\Delta t$ is the bin width (0.1 s).

Heatmaps were generated using the hot colourmap, ranging from black (no firing) to white (highest firing rate). Each heatmap was internally normalized by min-max scaling to preserve relative firing patterns and enhance visualization. This method was applied to analyse the response properties of different neuronal types in TeA: 29 R-neurons showed no clear response to any of the three sensory stimuli, whereas 66 SR-neurons and 9 unSR-neurons exhibited clear stimulus-locked increases in activity for all three sensory modalities, demonstrating distinct sensory-evoked responses in the sensory neuron populations.

To examine the relationship between spontaneous running and neural activity in R-neurons, we generated heatmaps of peri-event time histograms (PETHs) aligned to the onset of spontaneous running events. Each row represents the average activity of a single neuron across all trials. For each R-neuron, spontaneous running onsets were detected throughout the recording session using a speed threshold method (running onset defined as the time point when speed first exceeded 0.5 cm/s. Running-aligned neural activity was analyzed as follows: For each spontaneous running event, we extracted neural firing data from 10 seconds before to 10 seconds after running onset (time zero). Only running events that met the following criteria were included: no other running events occurred within 10 seconds before the onset, and running speed reached at least 0.5 cm/s during the event.

The average firing rate for each neuron i in time bin j was calculated across all qualified spontaneous-running trials using the same method as described for the sensory-stimulus analysis (see above). Heatmaps were generated using the "hot" colourmap (black to white), with each row representing one R-neuron's average running-aligned firing pattern across trials. Each heatmap was internally normalized using min–max scaling to preserve relative firing patterns while enabling visual comparison across neurons. This analysis was performed separately for the 29 R-neurons that showed no sensory-evoked responses, allowing us to examine whether these neurons exhibited running-related activity during spontaneous locomotion.

### Quantification of Response Latency

The onset of each sensory stimulus was aligned to time 2.0 s. Spike times from all trials for a given stimulus condition were pooled. Peri-stimulus time histograms (PSTHs) were constructed with a bin width of 10 ms to visualize the temporal dynamics of neuronal firing rates in response to each stimulus. The neuronal response latency was determined for each effective stimulus using a threshold-based method. The mean firing rate (baseline rate) and its standard deviation (SD) were calculated from a 2 s pre-stimulus epoch. The response threshold was defined as the baseline rate plus two times the standard deviation (baseline + 2 × SD). The response latency was identified as the first bin in the poststimulus period (starting from time zero) where the firing rate in the PSTH exceeded this threshold. Only neurons that showed a significant excitatory response to at least one stimulus type were included in the latency analysis.

### Quantification of Running-Related Firing Activity

We defined a running epoch to identify neurons whose firing was modulated by spontaneous running. Spontaneous running onsets were aligned to time zero. Peri-event time histograms (PETHs) were constructed for each neuron using a 10-ms bin width centred on the running onset. The baseline firing rate was defined as the average firing rate when the mouse was not running. The standard deviation (SD) of the firing rate during this baseline period was also calculated. The onset of running-related neuronal activity was determined using a threshold-crossing method. The threshold was set as the baseline mean firing rate plus two times its standard deviation (Baseline + 2 × SD). The start of the running-related activity window was identified as the first time bin where the firing rate in the PETH exceeded this threshold. The temporal window of running-related activity was then defined as the period from this identified start time until movement onset (time zero). This window represents the period of a significant increase in the firing rate preceding observable running.

### Statistical analysis and reproducibility

All key experiments were independently repeated at least three times with similar results. For representative micrographs shown in the figures, each image is representative of at least three independent experiments. The sample size (n) for all quantitative data is defined in the figure legends and represents biological replicates. This refers either to the number of individual animals or to the number of cells sampled from distinct animals, as explicitly stated in each legend. The number of biological replicates and statistical details for each figure panel are provided in the figure legends and/or Supplementary Data 1. OriginPro 2017 (OriginLab Corporation) or GraphPad Prism 10 (GraphPad Software) was used for statistical analysis and graphing. Shapiro−Wilk test was first applied to examine whether samples had a normal distribution. Student's t tests and one-way repeated-measures

or two-way ANOVA were used to evaluate statistical significance, unless stated otherwise. The Mann–Whitney U, Kolmogorov–Smirnov or Kruskal–Wallis test was used if the data were not normally distributed. In the figures, significance levels are indicated as $*P < 0.05$, $**P < 0.01$, $***P < 0.001$, and $****P < 0.0001$, and n.s. indicates a non-significant difference. The results are presented as the means ± SDs, unless specified otherwise.

## Reporting summary

Further information on research design is available in the Nature Portfolio Reporting Summary linked to this article.

## Data availability

All relevant data are included within the paper and its Supplementary Information files. Source data are provided with this paper.

## Code availability

We used MATLAB 2016b to write custom code for subsequent data processing. The MATLAB code for processing the neural, behavioural, and electrophysiology data is available at https://github.com/LiHe0606/Animal-speed-and-Neuron-firing-1-XiaoLab.

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

## Acknowledgements

This work was supported by grants from the National Natural Science Foundation of China (Grant Nos. 32371044 and 32070994), the Regional Joint Fund Key Projects of Guangdong Province (Grant No. 2022B1515120088) and the Key Research and Development Plan of Guangzhou Science and Technology Plan Project (Grant No. 2023B03J1337). We acknowledge American Journal Experts LLC for their professional language editing and proofreading services.

## Author contributions

Z.X. and W.Z. conceived and supervised the study. H.L., J.C., W.Z., N.L., Y.H., L.Y., and P.Y. performed all of the experiments. H.L., J.C., W.Z., L.Y., Y.Z. and J.T. contributed to data collection. H.L., J.C., Z.X., X.Q., P.Y., J.T., Y.Z. and J.L. analysed the data. Z.X., W.Z., H.L., and J.C. wrote the manuscript. H.L. and J.C. contributed equally to this work.

## Competing interests

The Authors declare no competing interests.
