## [Transparent Peer Review file · Nature Communications]

An intralayer microcircuit in the temporal association cortex underlies sensory-induced escape in mice

Corresponding Author: Dr Wen Zhong

Version 0:

Reviewer comments:

Reviewer #1

(Remarks to the Author)

Review on "Cells and their coding for running, backing away, rebound 1 running, stopping" by Chen et al. (NCOMMS-24-78436-T).

General comments on the two companion papers:

This paper has been submitted as a companion paper (NCOMMS-24-78434-T) by Zhongju Xiao and Wen Zhong. The two papers investigate neuronal circuits driving different defensive behaviors (running away, stopping, backing away and rebound running) in the dorsal periaqueductal grey (dPAG), temporal association areas (TeA) and superior colliculus (SC). The authors use a combination of viral tracing, projection selective neuronal manipulation and single-unit extracellular recordings in head-fixed behaving mice to define the neuronal circuits driving defensive behaviors.

The two papers cover two distinct aspects of the circuitry with only little overlap. The first paper (Chen et al., NCOMMS-24-78436-T) describes the role of two pathways, TeA-to-dPAG and SC-to-dPACG in driving running and backing away, involving two distinct cell populations in the dPAG. The second paper (Li et al., NCOMMS-24-78434-T) investigates in more details the TeA neuronal circuits involved in integrating sensory cues from different modality into motor action (running) showing that L5 IT cells in TeA integrate sensory information from sensory cortical areas and thalamic nuclei and drive L5PT cells projecting to dPAG to initiate running.

Both papers present exciting and new findings that are largely supported by an impressive set of experimental data using state of the art methodologies. I find particularly attractive the combination of cell-type- and projection- specific manipulations combined with single-cell electrophysiology that provides a straightforward functional characterization of the neuronal circuits involved in simple innate behaviors.

On a less positive note, I found both papers not easy to fully grasp due to lack of clarity. This comes both from a purely formal aspect with the writing of the paper – some sentences are just difficult to understand (starting with the title!) - and from some lack of explanation about the experimental procedures and data analysis.

Both papers would also require a better and more throughout analysis of the neuronal spiking data using more standard approaches to disentangle what part of the activity is related to sensory cues and movements (not only running speed). It would be good to homogenize as much as possible the analyses across the two papers, in particular for the neuronal activity (see my comments below).

In addition, I have a series of more specific concerns and questions for each paper that I detail below for this paper (Chen et al. NCOMMS-24-78436-T):

Major comments:

- It is really not clear in this paper whether we are dealing with spontaneous or sensory-evoked running behavior. Line 98 seems to indicate that sensory stimuli are presented but there is no indication of sensory stimuli in the figures (1a-b and 2a-b) and no further mention of sensory stimuli in the rest of the paper. If running is largely evoked by sensory stimuli, that should be taken into account in the analyses: the sensory-evoked responses might become a confounding factor for the correlation between neuronal activity and running, and a more careful analysis of sensory-evoked and running related activity need to be carried out – in line with what is done in the second paper.

- Overall, I am not convinced by the analysis of the neuronal activity and the correlation of firing rate with running speed. My understanding is that the correlation is computed after realigning the neuronal activity and the running speed curves for each event (running bout), which will likely artificially maximize the correlation and does not seem right to me. If the increase in neuronal activity does not reliably correlate with running at short and fixed latencies across trials, one cannot conclude that this neuron encodes running speed. The time-lag between the neuronal activity and the running speed could be measured in a more unbiased and simpler way by computing the cross-correlogram and defining the time-lag at the peak of the cross-correlogram. Then this specific time-lag should be used to fit the one-phase function only for the neurons, if any, with strong significant correlation. My impression when looking at the data presented is that the activity of TeA neurons at single cell level might be a good predictor of running onset time (initiation) but is a poor predictor running speed (due to important trial-to-trial variability). A better approach would be to use a more standard Generalized Linear Model (GLM) approach (fitting neuronal activity using running speed as regressor) or decoding approach (predict running speed from neuronal activity). Altogether I believe that a simpler and more standard analysis using event triggered PSTHs aligned to different event times (running onset, stimulus onset) and ROC analysis would be just as good. It would also help to clarify how running-responsive cells are defined. In any case, the authors should show single-neuron examples and grand-average PSTHs to be show overall time course of the neuronal activity relative to different events (onset of running, sensory cues...)

- I have also a general conceptual problem with the time delays between neuronal activity and movement onset time reported in this (and the companion) paper. In a similar study, Evans et al. (Nature 2018 <https://doi.org/10.1038/s41586-018-0244-6> see figure 3) report running latencies in the range of 0.2 s when stimulating dPAG neurons and 0.5 s when stimulating SC. In the current paper, stimulation of TeA or dPAG neurons also seems to evoke running with short latencies (based on the example sessions displayed in the figures). However, the time differences between running and firing rate reported (figure 2c) are really long (about 0.8 s for dPAG neurons and 2 s for TeA neurons on average) which seems incompatible with the claim of a direct monosynaptic connection from TeA to dPAG driving running.

- Anatomy of the recorded neurons (Figure 1f-i): this part would also need clarification. There is no explanation about how the coordinates of each recorded neuron are determined to obtain the distribution map in panel 1g. Figure 1g should indicate orientation of the map (DV/AP) and a scale bar. I find really odd that R-Neurons are so well confined within TeA area considering that movements affect neuronal activity everywhere in the brain (see Steinmetz et al, Nature 2019 and Stringer et al., Science 2019 just as examples). Looking at the image in panel 1f, it seems clear that the anatomy of the recorded neurons could only partially be recovered. That is certainly enough to identify the neurons as pyramidal cells and their respective location, but I do not think the reconstructions in panel 1i are meaningful.

- Provide more details about the extracellular recording procedure. It seems that the recordings in TeA are performed by inserting the pipette orthogonal relative to the cortex surface, which suppose a close to horizontal angle. How is it achieved practically speaking (size of the craniotomy, recording chamber, reference electrode, how is the brain protected during the recording)? Also describe the protocol for juxtacellular labelling.

- line 158-163: this sentence does not make sense to me.

- line 302-303, figure 4g: Backing and stopping should be dissociated. Interestingly, the duration of 'backing' appears to be constant, regardless of the duration of the opto-stimulation, but the duration of 'stopping' lasts as long as SC is stimulated!

- line 360-363: This is a reasonable assumption but it is not shown in the current study. That could go to the Discussion.

- Provide more details about the extracellular recording procedure. It seems that the recordings in TeA are performed by inserting the pipette orthogonal relative to the cortex surface, which suppose a close to horizontal angle. How is it achieve practically speaking?

Minor comments:

- First paragraph of Results (lines 89-94) should move to the Introduction.

- Line 210: it should be specified that the AAV-Cre virus used here is an anterograde transsynaptic serotype that will drive expression of Cre in the post-synaptic neurons receiving inputs from TeA.

- Line 281, figure 4e, f: I do not see any sign of 'backing away' when the stimulus is withdrawn?!

- Clearly indicate on each figure panel when is the stimulus presented (ex: figure 3 a-e2 right panel, 4f, 4g2 and h2 ...)

Sylvain Crochet

(Please do leave my name apparent to the authors)

Reviewer #2

(Remarks to the Author)

The study by He et al. aims to investigate the role of the TeA (temporal association area) as a hub for multisensory integration driving escape behavior through microcircuit computations. While the breadth and depth of the experiments—encompassing behavior, anatomy, physiology, and causal manipulations—are impressive, and some findings do indeed support the authors' claims, the overall narrative suffers from significant shortcomings.

The manuscript is plagued by numerous inconsistencies in experimental design, logical reasoning, and the interpretation of results. The narrative lacks cohesion, with critical connections between key experiments and conclusions often left underdeveloped or unclear. Instead of building a compelling and logical story, the sheer volume of data and experiments overwhelm and obscure the central message, detracting from the paper's impact.

Although a few specific examples are highlighted below, these issues are pervasive throughout the manuscript. As it stands, the paper is disjointed and challenging to follow, making it difficult to discern a coherent scientific narrative. A thorough and thoughtful revision is required to address these issues before the manuscript can be considered suitable for peer review.

Below are a few examples of major concerns with the manuscript (minor comments are far too numerous to list):

The central theme of “running” as highlighted in the title and abstract feels misaligned with the actual findings, which seem to emphasize sensory-evoked escape behavior instead.

Furthermore, the broader significance of the study is poorly articulated. The authors must clarify why this research is interesting and important within the larger context of behavioral neuroscience.

The text is frequently unclear, with superficial or anecdotal explanations.

There are innumerable inconsistencies between figure labeling and the main text, making interpretation challenging.

The English language must undergo significant editing for grammar, style, and overall readability to meet minimal academic standards.

Some core experimental designs lack clarity and justification. For instance, the claim that AAV-Cre (source) and DIO-virus (target) represent transsynaptic labeling is fundamentally questionable. This combination typically labels neurons projecting from the target to the source region, not synaptically connected downstream cells. This issue alone casts doubt on the interpretation and validity of conclusions drawn from these experiments.

The framing suggests a focus on multisensory integration, yet the experiments alternate inconsistently between unisensory and multisensory paradigms, leaving the narrative fragmented.

The analysis of interneuron anatomy and connectivity feels redundant and poorly integrated into the overarching story.

These problems, combined with the sheer volume of data and experiments, result in a manuscript that is difficult to follow and comprehend. The connections between experiments, results, and conclusions are weak, and the narrative lacks cohesion.

In summary, the manuscript requires substantial revision to address these conceptual, technical, and narrative flaws. Without these changes, it remains disjointed and exceedingly difficult to read as a coherent scientific manuscript.

Reviewer #3

(Remarks to the Author)

This manuscript by Li et. al presents the interesting and somewhat provocative claim that multi-sensory integration, decision-making, and generation of motor command to initiate walking all occur within an intra-laminar circuit in layer 5 of temporal association cortex (TeA). The work does a careful anatomical characterization of the projection from TeA to periaqueductal grey (PAG) and a series of manipulations of this pathway, with increasing specificity as the manuscript progresses.

However, the only readout is stimulus-evoked locomotion. Locomotion in other contexts would need to be assayed to determine specificity. Furthermore, analyses of locomotor responses to each of the stimuli are not sufficiently presented to interpret the effects. More generally, the results are sloppily presented, with inconsistencies in presentation, a lack of justification for selection of data for illustrations, a lack of detail needed to interpret statistical comparisons, and numerous saturated anatomical images. The intersectional anatomical characterization of input in Figure 3 is valuable, but for a more specialized audience. Key experiments for necessity are missing, which would rule out other pathways, such as ACC, sensory-to-PAG, mesencephalic locomotor region, or cortico-striatal pathways. Along these lines, more broadly, many circuits and pathways are known to be involved in sensory-driven locomotion and thus the results of this paper are a modest advance that would require more careful experiments, analysis, presentation, and interpretation to be persuasive.

Major concerns:

Organization of the results lacks introductory and other transitional sentences, so it is not clear how one section relates to the next. For example, first the first two sections with headers ‘Open field running model.’ And ‘turntable running model’ describe two behaviors but it is not clear where things are going. Are both going to be used later? What is the purpose of the open field model given the turntable model is described as having some advantages but no disadvantages.

The framing around multisensory may create confusion, since the paper does not look at multisensory integration (e.g. stimuli/cues with components/features from multiple senses). Rather, it presents sound, light, or air puff on separate trials. As such, they are studying a set of related associations for the 3 senses in isolation. From this perspective, there are gross overstatements of the results, such as this: “With the above experiments, we have established that the TeA-dPAG pathway is the neural circuit associated with multisensory-induced movement.” And “Therefore, the TeA must have the function of integrating various sensory information.” This is partially addressed by experiments in Figure 4g-l; most of the paper is about uni-sensory motor responses and framing should be adjusted, accordingly.

The descriptions of analyses and selection of data to be presented are not clearly motivated or described to give confidence in what is being compared and why.

The results are sloppily presented, with inconsistencies in presentation, a lack of justification for selection of data for

illustrations, and a lack of detail needed to interpret statistical comparisons. Numerous examples are given below in 'minor concerns' section, as well as a few 'big ones' here in the major concerns section.

It is not clear if 'Silencing TeA blocks running' experiments are done in the freely moving or head-fixed paradigm. Similarly, it is not clear in panel l and m of Figure 1 if these speed calculations are done for sound, light, or air puff. There is a similar issue with the optogenetic experiments, where it is not clear if the running under investigation is triggered by sound, light, or air puff.

Physiological recording for the CNO experiments in Figure 1 are lacking, so it is not clear what effect the CNO has had on TeA and other structures.

AAVretro is known to leak into neighboring cells. It would be helpful to confirm the projection from TeA layer 5a to PAG using a confirmatory method, such as pseudorabies or fluoro-gold.

Some images are saturated (e.g. 1k,q; 2a1; 3a,c; e1b; e2g; e3a) or show signs of injection damage (2a2), raising concerns about data quality. Images should be adjusted so they are not saturated and extent of tissue damage at injection site should be discussed and/or addressed as needed with new experiments.

It seems possible that the CNO and optogenetic experiments in Figures 1 and 2 are a 'hammer' and are shutting down locomotion and behavior rather broadly, rather than indicating the specific multimodal integration and transformation to a motor signal that the paper suggests. It would be helpful to determine if locomotion in other contexts (e.g. plus maze exploration) is impacted. Also, CNO injections into virus control animals would be needed to mitigate e.g. tissue damage concerns.

In Figure 2, the responses to the 3 different stimuli should be parsed (walking to sound, light, and airpuff).

The characterization of neuronal responses in TeA to running versus sound stimuli, although intriguing, do not demonstrate much about the integration process. Furthermore, running related signals are extremely prominent across neocortex, and thus not surprising or very informative regarding this study.

The critical experiment of silencing the TeA-dPAG neurons and determining if that blocks sound/light/air-puff induced locomotion appears to be missing. This is the experiment that would test if those neurons are necessary for the behavior. Without this experiment, it seems likely that many other pathways are also involved and/or sufficient.

Minor concerns:

Figure 1f shows peak running speed and the authors claim the stimulation intensities were carefully chosen to evoke similar running speeds. It would also help to see average, rather than peak, running speed, given that this peak running speed may be insensitive to differences in evoked running patterns by the stimuli used.

How were 10 trials shown in 1c selected? What do these plots look like for different animals and single trial?

It is not clear in Fig. 1d-f how the stats were done. There are 7 animals and 7 data points, suggesting each point is an animal. This should be stated. The type of post hoc test is not stated. And the F values are reported as 2,18 for degrees of freedom. How do these numbers relate to the 7 mice and 7 data points?

Why are walk bout durations not shown for the freely-moving task in Figure 1?

It is not clear if it is the same, or different, 7 animals used in the freely moving versus head-fixed behavior.

This statement "However, the characteristics of the running, such as the probability, delay, and peak speed (Fig. 1j), were not significantly different between the turntable model and the open field model (Fig. 1c-f), except for the running duration." Is not justified based on what is shown in panel 'j', at least as described. Again, how the F values were determined, and what the groups were, etc., is not clear.

The co-localization rate reported in Fig. S1 is ambiguous; they should report both false positive and false negative rates (i.e. were there any retrogradely labeled cells that were negative for camKII/SOM/VIP/and PV? Also, why was AAVretro used for the CaMKII labeling?

The purpose of this sentence is unclear: "Due to the scarcity of cholinergic and dopaminergic neurons in the cortex, we did not perform corresponding tests on these neuron types." Why are cholinergic and dopaminergic neurons singled out for mention? And in fact there are cholinergic neurons in neocortex (Von Engelhardt, J., Eliava, M., Meyer, A. H., Rozov, A., & Monyer, H. (2007). Functional characterization of intrinsic cholinergic interneurons in the cortex. *Journal of Neuroscience*, 27(21), 5633-5642.).

Statistical quantification of significant reduction for Figure 4d-f is lacking.

Experiments in Figure 4m-r would benefit from a CNO or optogenetic manipulation. More broadly, some form of psychometric manipulation to understand how graded responses varies with graded manipulation would strengthen the manuscript.

Reviewer #4

(Remarks to the Author)

Li et al. 2025 - An intra-layer microcircuit for multisensory-motor decision and running

I congratulate the authors on a powerful manuscript. It is thought-provoking and reads nicely with a logical structure. Included are an awe-inspiring array of experiments. Particularly noteworthy are the number of methodological approaches (pharmacological, chemogenetic, and optogenetic) to address the same research question. This contributes to the convincing nature of the manuscript. The figures include appropriate panels showing the data clearly, clear micrographs of the injections and expression patterns, and the legends are complete with relevant statistics and descriptions.

The overall claim of the paper can be argued to be quite extraordinary. A single piece of association cortex (TeA) is responsible for routing and integrating not just one, but multiple sensory signals to evoke running behavior, thus encompassing a significant part of the sensorimotor pathway.

I have two main comments, several additional comments, and minor (editorial) remarks. I hope that by addressing these

major comments, the manuscript could present its findings more cohesively while maintaining a balanced and scientifically rigorous narrative. I look forward to seeing this exciting work refined and published.

1. Clarification of Running Behavior Context and Interpretation.

I had a hard time understanding how the authors think about running behavior. The introduction starts with reviewing innate defense and escape behaviors that include running. From lines 55-63 it appears that running is equated with escape behaviors – while escape behavior consists of more complex sequences of behavioral elements including running. The results section talks about the running with the TeA circuit being almost directly responsible for the motor command signals being sent. Then in the discussion, the authors present the sensory stimuli as more neutral than the aversive stimuli used on other studies. However, 110 (!) dB white noise auditory stimuli, a bright inescapable light and an air puff in the eye (classic aversive stimulus) can NOT be called neutral. These are classic aversive stimuli. Moreover, it becomes completely puzzling to me how the authors think of (the adaptive role of) systematic running following neutral stimuli. Without any context dependence such a proposed circuit would be highly disadvantageous behavior (i.e. starting to run with the onset of any neutral stimulus). On top of this the dPAG is of course pivotal in innate defense and escape behaviors, including running but also freezing.

The methods lack a lot of detail on the running behavior in the freely moving scenario. When was the sensory stimulus presented and where, on different sides? Did the behavior depend on the current location of the animal? Was the mouse always running away from the stimulated side and express thigmotactic behavior? What was considered the onset of running? Did the mice freeze before running?

Line 55: Running is usually chosen... I would argue that this is not true and depends on the research. In perceptual decision-making licking, nose poking or forepaw movements are often used as behavioral indices in multisensory decision making tasks. In the context of fear, innate and defense research, freezing is also often used. In sum, the authors need to argue better if and why they think of running as different from escape behavior or not. This brings me to a related point. Brain-wide activity changes associated with the behavioral state of mice have been shown throughout literature; most prominently in (Stringer et al. 2019), and across most studies studies now studying awake behavior in mice. Neurons throughout cortex, and in particular deeper layers correlate with locomotion state and running speed (Niell & Stryker 2010, Saleem et al. 2013, Steinmetz et al. 2019, Musall et al. 2019, Vinck et al. 2015). An alternative interpretation of the signal of deep TeA layers is thus that these neurons correlate with running as any other part of mouse cortex (Fig. 7d supports this – nonChR2 cell modulation of TeA activity when running) and rather than motor command signals, that their outputs reflect the threat-relevance of sensory stimuli to dPAG. Perhaps the authors can discuss to what extent the data support a direct role in motor initiation versus alternative interpretations that consider the context of the behavior, such as that TeA provides the contextual signals necessary to signal a situation in which escape behavior is necessary. To further clarify, I suggest:

-Providing additional context about the adaptive relevance of the observed behavior.

-Expanding the methods to include critical details about the running behavior in freely moving animals, such as:

- How and where stimuli were presented.
- Whether the animals' location influenced their responses.
- Onset criteria for running and whether the animals exhibited freezing behavior beforehand.

2. Latency and Circuit Dynamics

The authors present a circuit in which sensory signals coming from respective cortical sensory systems project to TeA. After an intralaminar projection, the outputs of TeA project to dPAG responsible for the running behavior. The latencies of the proposed pathway, tens of milliseconds, is very different from the timescale at which the behavior is elicited (Fig. 1e, 1j). Why does it take 2 seconds to start to run? Are the animals freezing first? Coming back to the first point, is this escape behavior? Methods details are lacking here. Line 1291: What is the smoothing kernel size? How was latency to start of running estimated? How does the smoothing kernel affect latency estimates?

From Figure 1b it appears that the running speed in the freely moving context is not a smoothly increasing function, but rather a small bout in the middle of stimulus presentation. Why do the animals run only the middle second? Then the dynamics in the head fixed scenario, Fig. 1h, are different but still have onset latencies of 1 sec or more. How do the authors account for the difference between the output of TeA (100ms?) and running behavior (1-2 sec?)? Is dPAG integrating over time? Moreover, some neural latencies do not match with the proposed circuit. The latencies to light stimulation for example are not in line with the suggested cortical circuit: VC to TeA, where VC responds at 50-60 ms latencies, but TeA around 400 (Fig. 5h).

I suggest therefore greater clarity on:

- Specify how running onset latency was measured (e.g., smoothing kernel size, its effect on estimates).
- Address why the running speed in Fig. 1b peaks during the middle second of stimulus presentation.
- Reconcile neural response latencies (e.g., VC to TeA at 50-60 ms vs. 400 ms for TeA, Fig. 5h) with the behavioral timescale. If the authors maintain that the pathways depends on weak connections that need 100s of milliseconds to accumulate, they need to argue how this system is robust to noise, evolving temporal dynamics of the brain and changing behavioral objectives.

Minor comments

- I was confused by the usage of multimodal stimuli in the manuscript. The authors also use multisensory and multimodal to refer to effects and situations in which different sensory modalities alone can produce similar effects (e.g. line 200). As the manuscript also involves more classical multisensory experiments with simultaneous sensory stimulation, this can be confusing. My suggestion is to reserve multimodal/multisensory for bimodal or trimodal stimulation. I don't see what is lost if the term multisensory would be reserved for experiments involving simultaneous sensory stimulation. E.g. line 218: Multisensory inputs converge to TeA Sensory inputs converge to TeA.

- There are several conclusions in the results sections that are not (yet) supported by the data. I strongly suggest that the authors adjust their claims to precisely match the data shown. For example:
- Line 318: "These results indicate..." It has not been shown that the multimodal integration (here I am specifically referring to the synergistic interactions between the modalities) that induces running behavior is taking place in TeA. Only that pathways from early sensory cortex to TeA are involved in eliciting running behavior, and (Fig 4) that multimodal stimuli can interact to produce running behavior. The circuits supporting the integration of different sensory modalities that underlie the change in elicited behavior could still be elsewhere.
- Line 338: "These experiments indicate..." This statement is problematic. First it makes a claim about the whole research field of multimodal sensory integration, whereas (1) no multimodal stimuli were tested, only sound, and (2) one limited behavior was tested. Second, the statement generalizes to other modalities not tested.
- Line 533: It believe it was not specifically shown that this weak connection requires an extended time to sufficiently excite downstream neurons (perhaps move to discussion of proposed circuit).
- Line 406-413: formulate as hypothesis. Only part of the claims here is shown.
- Line 376: Biologists love categorization, but in this scenario, it is unclear whether it is justified and what is gained. Do the authors believe the categorization into 14 neuron types is biologically important and future research should focus on the differences between subtypes? We know from earlier work (Raposo et al. 2014) that neurons in higher order mouse cortex show mixed selectivity that defy categorization. It is likely that the reported sensory subtypes come from continuous distributions. It is hard to judge, the Methods lack any description of how significant responsiveness was calculated, which statistics were used. If similar to running analyses the standard deviation of the baseline was used, which time frames served as baseline? Moreover, how was running and sensory responsiveness dissociated? (If sensory stimulation causes the mice to run, then to what does the neuron respond?). The presented single neurons are not convincing and highlight the problem. For example, the neuron in Ext. Data Fig 4c, is it responding to light stimulation or running? And the neuron in Ext. Data Fig 4d, responding to air puffs, or increasing its firing rate upon running? Misinterpreting behavioral state related signals as and dissociating them from sensory signals has recently been an important topic in (multi)sensory research (Zagha et al. 2022, Bimbard et al. 2024, Oude Lohuis et al. 2024) Supporting the statement that there are meaningful categories of multisensory responsive neurons would thus require showing distributions of a metric of sensory responsiveness + low-latency sensory-evoked responses that are independent of behavioral state.

Further notes:

- Line 64: two sensory information, information has no plural. Similarly, line 221, various information..
- Line 158: retrospectively -> retrogradely?
- Fig. 1k, inset of waveform missing for Mus+? 7 TeA cells in 3 animals. Why 7 animals for muscimol, but only 3 animals have cells recorded? Some more clarity in the methods about how many cells were recorded across animals and whether there were exclusion criteria would help interpretability.
- From line 192: consistency, why PAG and not dPAG?
- Line 281, 'distinguish' is a verb, not a noun
- Line 290, an article is missing 'AC-TeA projection loop'
- Figure 3, legend missing for panel f
- Fig. 3d: how anterograde? Or retrograde?
- Fig 3j: The resolution is too poor to judge the overlap in expression. Make sure the resolution is sufficiently high and/or include a closeup.
- Line 324: 'Even within a single modality'... this is hardly a surprising statement with rich worlds of sensory signals in each modality.
- Line 357: Specify for the lazy reader how this stimulation intensity compares to earlier experiments and why this intensity was used.
- Line 363: 'Connections', poor word usage as it might be interpreted anatomically and can be described sticking closer to the results: 'whose firing rate correlated with running speed'
- Line 401: S-neuron, here and below I suggest to use plural (S-neurons) or S-neuron type.
- Line 411: 'correspond to' -> activate?
- Line 441: the virus was retrograde, but was not injected retrograde.
- Line 477. I confused the results of ChR2- cells in the experimental groups with the control animals. Perhaps the authors can include additional phrasing that help distinguish the behavioral results of the optogenetic control cohort with the distinct analysis of ChR2 pos and neg cells.
- Fig 6h: the presence of at least one merged neuron can be observed, but a quantification is missing, similar to earlier in the manuscript (Fig 3k).
- Line 564: adjust word order.
- Line 578: What are the differences between the TeA circuit and SC circuit. Do they overlap, play a role in different circumstances? How come TeA inhibition can block the running behavior with intact SC circuit?
- Line 591: 'involved in'... Involved in what?
- Line 586-592: I find the positioning of this study in the literature a bit off. In comparison to the cited references, this study also epitomizes on circuit mapping techniques to discover pathways and cell types involved in specific sensorimotor transformations. No proper cellular mechanisms are described. Moreover, the phrasing 'special technics' (techniques) suggest a newly introduced technique whereas the study uses commonly available tools and techniques, albeit a comparably sophisticated level of circuit investigation.
- Line 881: degrees -> percentage?

Methods:

- Line 1000: a retro-pseudotyped virus was used for anterograde tracing?
- Line 1074: What does 'maintaining a stable state on the recording platform' mean? Were the mice spontaneously running?

What percentage of animals does 'most animals' mean and what was done if animals did not 'maintain a stable state'?

Line 1289: What does randomized batch processing refer to?

Line 1276: No fiber photometry data is presented in the study.

Line 1296: 'produced apparent' -> 'were'?

Line 1129: Pharmacal -> pharmacological? + revise sentence grammar

Line 1152: 800 mm -> um?

Line 1156: Were animals excluded based on expression? If so, on what criteria and how many?

Line 1383: 'The ordinates.' What about the ordinates?

Line 1315: What is meant by that the raw data is available in the main text?

Line 1321: Where is the code available?

Version 1:

Reviewer comments:

Reviewer #1

(Remarks to the Author)

This revised version of the manuscript has been greatly improved in terms of readability and clarity, although the wording of some sentences remains a little odd. The current version presents a clear and elegant description of a cortical circuit transforming sensory input into the decision to initiate running (escape). Unfortunately, despite this improvement in the form, many of my concerns about the robustness of the data and analyses remain. In fact, the authors have largely ignored many of my concerns on these points. Another problem is that the "Methods" section still lacks essential information or provides incomplete information, making it difficult to truly assess the relevance of the analyses and results.

Here are examples of critical issues that have not been carefully addressed:

- It is still unclear to me whether the authors have adequately addressed the issue of disentangling the sensory- and movement-related neuronal activity (juxtacellular recordings). I understand that a complex approach like GLM might not be suited for the type of data in this study, with low number of trials but at least the sensory-evoked response should be assessed from purely sensory trials (no evoked running) and the running-related activity should be assessed from spontaneous running not preceded by any sensory stimulus. And the significance of the responses should be assessed for well-defined time-windows and appropriate statistical test. PSTHs aligned to sensory stimulus AND running onset times should be presented for single neurons and groups of neurons (See my Major comments 4 and 5 from the previous version).

- I am also still concerned by the possibility to achieve good-quality 3D single-cell reconstructions (Figure 6b-c) from high-density labelling obtained from viral expression (Figure 6a). Again, no information provided in the Methods section.

- According to the rebuttal of the other paper (NCOMMS-24-78436A) the anatomical location of the recorded neurons was achieved either through single-cell juxtacellular staining with biocytin (no example picture provided in this paper!) or by injection of a dye at the same location after the recording. There is no mention of that in the Method section!

- There still is not explanation about how Chr2+ neurons were identified. There should be clear criteria to assign neurons as Chr2+ or Chr2-. Also, it is (almost) correct to assume that Chr2+ neurons are either AC-TeA or TeA-PAG neurons depending on the labeling procedure but it is absolutely not correct to assign Chr2- neurons to any category!

Other less critical issues:

- Regarding response latencies, in their rebuttal the authors claim that "The latencies of sensory-induced behaviours are almost universally on the order of seconds[REFs 1-3]". This is absolutely wrong, in many goal-directed behaviors in head-fixed mice, response latencies are in the range of 100-300 ms! Also, the study by Evans et al. (Nature 2018 <https://doi.org/10.1038/s41586-018-0244-6> cited by the authors, REF 2) reports response latencies of 0.5 s to sound or high-contrast visual stimulus (see figure 1). Considering the high-response probability in the current study (Figure 1k), one could expect much shorter latencies. Which questions the way response probability, peak speed and latency were computed: ideally, one should define a response window during which one may consider that the running onset is indeed initiated in response to the sensory stimulus and not just a spontaneous running event (see my Major comments 1 and 3 from the previous version).

- From the revised version, I understood that in fact, each sensory stimulus is only presented 10 times, which does not provide many trials to analyze. Was it to avoid habituation and decrease in behavioral response (see my Major comment 2 from the previous version)?

- Importantly, the extracellular recording experiments were conducted with lower-intensity sensory stimuli (to avoid large and fast behavioral responses). And indeed, when looking at example recordings, it appears that the sensory stimuli rarely evoke running and running appears more like spontaneous than sensory-driven events. The behavioral response (running probability and latency) during the recording session should have been quantified and reported. If the sensory stimuli do not evoke running with significantly higher probability than spontaneous running, it does limit the relevance of the extracellular recordings for this study.

- The correlation between neuronal activity and running speed would be better assessed by simply computing the cross-correlogram between the two, which would provide both peak correlation and time lag.
- The text mention 29 R-Neurons, 66 SR-Neurons and 9 unSR-Neurons (104 cells) but figure 4h-j shows more many unSR-Neurons (at least 15) and different number of cells from panel to panel (see unSR/LS).
- How many mice were used for juxtacellular recordings? How many neurons / mouse were recorded ?
- In figure 6g, the example images show 1 or 2 double-labeled neurons (yellow) for single-labeled neurons (green or red). That does not really fit with the reported % of overlap (60% for AC).
- Results figure 6l: there seems to be a scaling mismatch between the left and the right panels: the grand-average traces for EPSCs on the left peak at about 200 pA (black) and 125 pA (red) whereas the bar graph on the right indicate averaged values of 500 pA (black) and 300 pA (red) ?! (see my Minor comments from the previous version).

Other not-addressed comments from the previous version:

- An important claim is that TeA is critical to elicit running in response to the sensory-cue. And indeed, TeA inactivation blocks running induced by all 3 sensory stimuli. But the authors now also show that TeA inactivation has an effect per se on spontaneous locomotor activity, i.e. TeA inactivation reduces locomotion thus questioning a specific role of TeA in sensory-evoked running.
- Extended Figure 1f: I am really surprised by the distribution of VIP+ and PV+ cells in TeA. I would have expected much less VIP+ cells (especially in deep layers) and much more many PV+ cells. See <https://www.frontiersin.org/journals/neural-circuits/articles/10.3389/fncir.2021.781928/full> for a possible explanation. By the way, the JAX reference of the PV-Cre line (Methods, line 977) is missing.
- Methods, line 1143: Provide more details about the CNO solution and preparation. How was CNO dissolved?
- Methods, line 1202-1203: By definition in voltage-clamp mode, the Vm is clamped to a given value! What was this value during the recordings?
- Last page: it would help to have the abbreviations in alphabetical order.

Sylvain Crochet

Reviewer #4

(Remarks to the Author)

I want to thank the authors for the considerable amount of work that has improved the manuscript in significant ways. The points were addressed with attention, references to other work, the companion paper and explanation of reasoning. The manuscript is improved with additional controls, panels and supplementary data to aid interpretation of the data. I maintain that with revision the manuscript offers exciting data for the neuroscience community to move forward.

I have 2 major comments and a few minor comments. Line numbers are for the markup document.

1) Onset latency to running

Although the authors provide more information about the neural and behavioral latencies, and discussion of their relationships, I am still puzzled by the proposed circuit and behavior.

Line 204 says "triggered immediate escape behaviors", but the data show a latency of 1-3 seconds until mice start running to the other side. Auditory startle responses are within 50-100 ms, so this onset latency can hardly be called immediate.

In the rebuttal, the authors say that the mice often changed their behavior after the aversive stimuli: "Furthermore, our experimental video recordings revealed that mice typically exhibit brief vigilance (e.g., head turning, ear twitching) after sensory stimulation, followed by the initiation of running, rather than entering a motionless state."

I believe these observations are instrumental. They show that running speed is an incomplete description of the behavior and that changes in behavior can be observed before any changes in locomotion. This means that in parallel to the proposed circuit there are likely pathways that process the sensory stimuli and participate in motor programs involved in vigilance and orientation behaviors.

I strongly suggest the authors to discuss (1) whether they consistently observe other behavioral changes before locomotion onset and this explains the latency to running, and (2) whether they believe that sensory to motor processing in TeA is central to all sensory-induced behaviors observed in this paradigm, or specific to locomotion (with faster circuits operating in parallel to this slow synaptic integration in TeA to induce locomotion?).

2) TeA-dPAG specificity to sensory-induced flight behavior

The inclusion of new results regarding the role of the proposed circuit in an open field arena are powerful, but bring new interpretations to the table. The fact that inhibition of the TeA-dPAG neurons also affects non-sensory induced locomotion generally degrades the claim that this pathway "integrates information from diverse sensory stimuli into running signals for inducing escape" (from Abstract).

These results need to be discussed in the main text and – based also on the opinion of the other reviewers – the authors

should attenuate the claim that the TeAdPAG is specifically involved in the integration of sensory stimuli to evoke flight throughout the manuscript.

Minor comments:

- Line 2953, smoothing: what is the unit for the window width of 5?
- The exclusion criteria and fraction of data excluded should be included in the manuscript for transparency and reproducibility (based on replies to my comments on line 1585-1589, 1629-1634 in rebuttal).
- Line 1593 in rebuttal: The updated methods regarding the 'batch processing' are still quite unclear to me. Data was 'pooled' across what? Why was data processed in 'randomized batches'? Was the running speed data simply aligned to stimulus onset and averaged across stimulus conditions?
- Line 1440: 'extractive', what does this mean?

Version 2:

Reviewer comments:

Reviewer #1

(Remarks to the Author)

In this revised manuscript, the authors have addressed most of my comments. But there remains a few changes to be made before this paper can be accepted for publication:

- I would suggest a more specific title that includes explicit reference to the TeA and mouse model.
- line 286-287: "Three types of neurons in the TeA may regulate signalling that running behaviour" ???
- Figure 4 : report real mean or median values, not the ones computed from Gaussian fitting.
- Lines 381-397: I maintain that it is not correct to realign each running event to the preceding neuronal event with variable delays! This section is flawed and unnecessary, and should be removed. The report of the max correlation coefficients (r_{max}) and lags from the cross-correlations is sufficient.
- As indicated by the authors "the low overlap rate between hSyn and CaMKII could result from differences in the efficiency of the expression of the two types of viruses". Indeed, the images in the Extended Data Figure 4d clearly show a significant number pyramidal cells only labeled with EYFP (green, hSyn promoter) and not by mCherry (CaMKIIa promoter). Therefore, I do not think that this part of the data is really informative.
- Lines 506-508: Incorrect phrasing. In these experiments the inhibitory receptor (hM4D-Gi) is expressed in TeA neurons receiving inputs from sensory cortical or thalamic areas. So technically, the CNO will not block the sensory inputs from cortex or thalamus but instead block the activity downstream to SENS-TeA neurons.
- Lines 510-511: No. These results show that SENS-TeA neurons are somewhat modality specific (at least partially).
- Lines 570-584: In the end only 5 cells of each group were reconstructed. The authors should clearly indicate that they could only reconstruct the dendritic arborization from a small subset of neurons that were isolated enough. The authors should include the images provided in the rebuttal (line 157-158) as Extended Figure.
- Lines 634-636 and Figure 7: Considering ChR2- cells in the experiments labelling the cells receiving inputs from AC, it is quite speculative to consider them as TeA-dPAG neurons. It would be more rigorous and less confusing to call them 'AC-TeA-ChR2-'

Sylvain Crochet

Reviewer #4

(Remarks to the Author)

This revised version of the manuscript presents clear improvements. However, the manuscript remains problematic in the disentanglement of sensory and motor related signals. In sum, I am not convinced about the presence of clear sensory-evoked activity, which presents a key aspect of the proposed model where within TeA diverse sensory signals are transformed into motor command signals.

First, the new raster plots are unclear to me. For example:

Extended Data Fig. 2 | S- and R-neurons in the TeA. a, Responses of the R-neurons to three types of sensory stimuli: auditory (sound), visual (light), and tactile (air puff). Top panel, Population raster plot and the corresponding PSTH of firing activity. Bottom panel, Superimposed running speed curves from all trials ($n = 156$ trials). The blue bars indicate the stimulus period (duration: 5 s).

In the legend it says response of the R-neurons and Population raster plot, but these must be individual example neurons (not a population of neurons) with each row representing individual trials, is that correct? A figure that shows the responses across all neurons of a certain category is then lacking. (If it is not, and each row is the mean activity across trials for one neuron and it is indeed a population plot, then there is just one spike for some neurons, and the legend is incorrect as these must be different numbers of trials for neurons recorded on different days.)

Extended data figure 2f and 2g show 2 example S neurons that do not show stimulus-locked onset latency, arguing against strongly sensory driven activity. Similarly, Extended Data Fig 3 shows only example neurons and only for 1-3 individual stimulus presentations, too little to judge stimulus-locked sensory-evoked activity that is independent of behavioral changes. I understand from the Results section that the identification of SR and S neuron is assessed based on trials without any movement, but shockingly I can not find a good explanation of how S, SR and unSR neurons were categorized in the Methods section, including this information about trial selection.

By now, claiming strictly sensory evoked activity in awake behaving mice is not easy, and the default null hypothesis is widespread activity changes related to running, but also whisking, postural changes and other facial motion indices.

I am happy to see the discussion. Given that the stimuli used cause behaviors other than locomotion such as head twitching and orienting behaviors at least in the open area, it would be a natural interpretation that the activity observed is related to non-locomotion related behavioral changes.

In sum, with more and more methods and new figures coming to the table in this MS, I am not assured by the presented unconvincing example neurons that there are clear auditory, tactile and visual responses.

A convincing result for me would be a simple heatmap where each row is the activity of one S neuron (66 SR and 9 unSR cells) aligned to stimulus onset and averaged for all trials without locomotion. This should show clear stimulus locked increase in activity across both populations.

Additionally, it would be even more convincing to show that these responses also occur in the absence of other movements, or at least have different temporal dynamics. However, given the high rate of sound-induced behavioral changes reported in the discussion there might be too few trials without any sensory-induced behavioral changes.

In that case it is warranted that the authors add a statement in the Discussion that they can not fully exclude that the reported sensory-related activity is (partially) related to observed movements other than locomotion.

Dear Reviewer,

The line numbers referenced in our point-by-point responses correspond to the
"Revised manuscript without track changes-He Li.pdf" file.

**Reviewer #1-Li (Remarks to the Author):**

**General comments on the two companion papers:**

**This paper has been submitted as a companion paper (NCOMMS-24-78434-T) by**
**Zhongju Xiao and Wen Zhong. The two papers investigate neuronal circuits**
**driving different defensive behaviors (running away, stopping, backing away and**
**rebound running) in the dorsal periaqueductal grey (dPAG), temporal association**
**areas (TeA) and superior colliculus (SC). The authors use a combination of viral**
**tracing, projection selective neuronal manipulation and single-unit extracellular**
**recordings in head-fixed behaving mice to define the neuronal circuits driving**
**defensive behaviors.**

**The two papers cover two distinct aspects of the circuitry with only little overlap.**
**The first paper (Chen et al., NCOMMS-24-78436-T) describes the role of two**
**pathways, TeA-to-dPAG and SC-to-dPACG in driving running and backing away,**
**involving two distinct cell populations in the dPAG. The second paper (Li et al.,**
**NCOMMS-24-78434-T) investigates in more details the TeA neuronal circuits**
**involved in integrating sensory cues from different modality into motor action**

(running) showing that L5 IT cells in TeA integrate sensory information from
sensory cortical areas and thalamic nuclei and drive L5PT cells projecting to
dPAG to initiate running.

Both papers present exciting and new findings that are largely supported by an
impressive set of experimental data using state of the art methodologies. I find
particularly attractive the combination of cell-type- and projection- specific
manipulations combined with single-cell electrophysiology that provides a
straightforward functional characterization of the neuronal circuits involved in
simple innate behaviors.

On a less positive note, I found both papers not easy to fully grasp due to lack of
clarity. This comes both from a purely formal aspect with the writing of the paper
– some sentences are just difficult to understand (starting with the title!) - and
from some lack of explanation about the experimental procedures and data
analysis.

Both papers would also require a better and more throughout analysis of the
neuronal spiking data using more standard approaches to disentangle what part
of the activity is related to sensory cues and movements (not only running speed).

It would be good to homogenize as much as possible the analyses across the two
papers, in particular for the neuronal activity (see my comments below).

We sincerely appreciate the reviewers' time and insightful comments, which have
significantly improved our manuscript. On the basis of your feedback, we have
restructured the overall framework of the manuscript, clarified key concepts, and

improved the logical flow.

The entire manuscript has been professionally edited by American Journal Experts LLC.

Your insightful suggestions have significantly improved the coherence, logical
consistency, and conceptual clarity of the paper, resulting in better readability and
impact. Furthermore, both the semantics and grammar have been substantially refined.

We have supplemented additional data analysis based on the questions you raised. Both
articles adopted a unified processing method when analyzing whether neurons are
related to motor or sensory functions, further clarifying the types of neurons. All points
have been addressed in the revised version. Regarding the specific point-by-point
responses, they have already been addressed in the replies to another manuscript (Chen
et al. NCOMMS-24-78436-T).

Dear Reviewer,

The line numbers referenced in our point-by-point responses correspond to the
"Revised manuscript without track changes-He Li.pdf" file.

**Reviewer #2-Li (Remarks to the Author):**

**The study by He et al. aims to investigate the role of the TeA (temporal association**
**area) as a hub for multisensory integration driving escape behavior through**

**microcircuit computations. While the breadth and depth of the experiments—**
**encompassing behavior, anatomy, physiology, and causal manipulations—are**
**impressive, and some findings do indeed support the authors' claims, the overall**
**narrative suffers from significant shortcomings.**

**The manuscript is plagued by numerous inconsistencies in experimental design,**
**logical reasoning, and the interpretation of results. The narrative lacks cohesion,**
**with critical connections between key experiments and conclusions often left**
**underdeveloped or unclear. Instead of building a compelling and logical story, the**
**sheer volume of data and experiments overwhelm and obscure the central message,**
**detracting from the paper's impact.**

**Although a few specific examples are highlighted below, these issues are pervasive**
**throughout the manuscript. As it stands, the paper is disjointed and challenging to**
**follow, making it difficult to discern a coherent scientific narrative. A thorough**
**and thoughtful revision is required to address these issues before the manuscript**
**can be considered suitable for peer review.**

We sincerely appreciate your valuable comments and suggestions. On the basis of your
feedback, we have restructured the overall framework of the manuscript, clarified key
concepts, and improved the logical flow. Additionally, we have redesigned the figures,
provided new experimental data, and conducted additional data analysis. The entire
manuscript has been thoroughly revised and polished by professional English language
editors. Your insightful suggestions have significantly improved the coherence, logical

consistency, and conceptual clarity of the paper, resulting in better readability and
impact. Furthermore, both the semantics and grammar have been substantially refined.

**Below are a few examples of major concerns with the manuscript (minor**
**comments are far too numerous to list):**

**-The central theme of “running” as highlighted in the title and abstract feels**
**misaligned with the actual findings, which seem to emphasize sensory-evoked**
**escape behavior instead.**

We sincerely appreciate your insightful comments. Your feedback has helped us
understand that simply equating defensive escape with running is inaccurate and that
this conceptual ambiguity would confuse readers, negatively affecting the paper's
readability and appeal. On the basis of your suggestions, we have revised the conceptual
framework of the entire manuscript and thoroughly rewritten the text, with particular
emphasis on the Title, Abstract, Introduction, and Discussion sections. These
modifications have significantly improved the manuscript's impact, clarity, and overall
readability.

**-Some core experimental designs lack clarity and justification. For instance, the**
**claim that AAV-Cre (source) and DIO-virus (target) represent transsynaptic**
**labeling is fundamentally questionable. This combination typically labels neurons**
**projecting from the target to the source region, not synaptically connected**

**downstream cells. This issue alone casts doubt on the interpretation and validity**
**of conclusions drawn from these experiments.**

Thank you for your insightful comment. We acknowledge that the experimental design
required further clarification, and so the manuscript now contains detailed explanations
with supporting references “*Among them, AAV2/1 was used for anterograde*
*monosynaptic tracing*⁸²⁻⁸⁴.” [Methods (line 806)] As you correctly noted, early
applications of the AAV-Cre and DIO-virus systems were primarily used to label
neurons projecting from the target to the source region. However, emerging evidence
has demonstrated that certain AAV serotypes, particularly AAV1, exhibit anterograde
transsynaptic transduction properties¹⁻⁵. Notably, Zingg et al.⁶ systematically
demonstrated that AAV1 and AAV9 are capable of anterograde transsynaptic spread. In
their study, delivery of AAV1-Cre to presynaptic neurons robustly and selectively drove
Cre-dependent reporter expression in postsynaptic targets, allowing both anatomical
tracing and functional manipulation of input-defined pathways. This established AAV1
as a validated anterograde transsynaptic tracer, a role that has been subsequently
leveraged by multiple studies to identify neuronal subpopulations receiving direct
inputs (e.g., mapping Y-nucleus neurons receiving X-derived projections).

Consistent with the results of that last study, our vendor also recommended AAV1 for
anterograde transsynaptic labelling. In our experiments, injection of AAV2-1-Cre into
TeA (presynaptic site) combined with injection of DIO virus in the dPAG (postsynaptic

target) successfully labelled T_{eAdPAG} neurons (see Fig. 3b1-b2 and c1-c2), confirming
the efficacy of the system for pathway-specific labelling. We appreciate your critique
and hope that the clarified methodology and cited literature adequately address your
concerns. Please let us know if further elaboration would be helpful.

**Fig. 3 | A TeA to dPAG circuit determines running behaviours. b-c, Blockade of the TeA-dPAG**

pathway in turntable model mice: T_{eAdPAG} neurons (b), and T_{eAdPAG} CaMKII neurons (c). 1, Viral

injection site and chemogenetic (CNO) and optogenetic (LED) interventions. 2, Viral expression.

**-Furthermore, the broader significance of the study is poorly articulated. The**

**authors must clarify why this research is interesting and important within the**

**larger context of behavioral neuroscience.**

We sincerely appreciate your valuable feedback. Indeed, our initial conceptual

framework was unclear, as it erroneously equated defensive escape with running and

misattributed the multisensory integration observed in the behavioural experiments to

integration in the TeA. These errors inadvertently affected the importance of our study

in elucidating the relevant sensory-to-motor information processing pathway and its

underlying cellular mechanisms. We have thoroughly revised the Introduction and

Discussion sections to better contextualize our findings within the field of behavioural
neuroscience. We are grateful for your insightful comments, which have helped us in
correcting these issues and in improving the scientific rigor of the manuscript.

**-The text is frequently unclear, with superficial or anecdotal explanations.**

**-There are innumerable inconsistencies between figure labeling and the main text,**
**making interpretation challenging.**

**-The English language must undergo significant editing for grammar, style, and**
**overall readability to meet minimal academic standards.**

As you rightly pointed out, our manuscript indeed suffers from the issues you identified,
primarily stemming from conceptual ambiguities—notably, the oversimplification of
defensive escape as running and the inaccurate interpretation of experimental results
where behavioural multisensory integration was incorrectly extended to integration
within the TeA. Additionally, we acknowledge the grammatical, stylistic, and overall
readability shortcomings in the original text. With your invaluable guidance, we have
thoroughly revised the manuscript to: 1. Clarify and refine the core concepts underlying
the research; 2. Restructure the logical flow of the arguments; and 3. Correct all
inaccuracies regarding results interpretation. Moreover, we have engaged professional
editors to improve language quality, including grammar, syntax, and overall readability.
These comprehensive improvements have significantly elevated the manuscript's
scientific rigor and presentation quality. We deeply appreciate your expert suggestions,

which have made these essential revisions possible.

**-The framing suggests a focus on multisensory integration, yet the experiments**
**alternate inconsistently between unisensory and multisensory paradigms, leaving**
**the narrative fragmented.**

Thank you very much for your suggestion. During this revision, we also realized that
our description of the ability of the TeA to perform multisensory integration was
inappropriate. Therefore, we have revised the corresponding content in the main text of
the manuscript. For example, we have moved the text “multisensory integration” and
“Auditory sensory integration depends on the intensity and duration of the stimuli” to
relatively unimportant parts of the manuscript, although we wanted to delete them. We
greatly appreciate your suggestion and apologize for our previous lack of precision in
the statement.

**-The analysis of interneuron anatomy and connectivity feels redundant and poorly**
**integrated into the overarching story.**

Per your suggestion, in the revised version, we have adjusted the narrative logic. We
moved the results investigating the TeA-dPAG projections from the original
Supplementary Fig. 1 and 2 to Fig. 2, whereas the exploration of cell types in the TeA-
dPAG circuit have been moved from the original Supplementary Fig. 1 and 2 to

Supplementary Fig. 1.

This adjustment allows the description of the TeA-dPAG projection relationship to be
interspersed with our exploration of the cell types in the TeA-dPAG circuit, creating a
mutually complementary flow of the narrative. At the same time, it prevents readers
from deviating from our main points while preserving logic.

**References:**

Oh, S. W. *et al.* A mesoscale connectome of the mouse brain. *Nature* **508**, 207-
214,(2014).

Salegio, E. A. *et al.* Axonal transport of adeno-associated viral vectors is
serotype-dependent. *Gene Ther* **20**, 348-352,(2013).

Hutson, T. H., Kathe, C. & Moon, L. D. Trans-neuronal transduction of spinal
neurons following cortical injection and anterograde axonal transport of a
bicistronic AAV1 vector. *Gene Ther* **23**, 231-236,(2016).

Harris, J. A., Oh, S. W. & Zeng, H. Adeno-associated viral vectors for
anterograde axonal tracing with fluorescent proteins in nontransgenic and cre
driver mice. *Curr Protoc Neurosci* **Chapter 1**, Unit 1.20.21-18,(2012).

Aschauer, D. F., Kreuz, S. & Rumpel, S. Analysis of transduction efficiency,
tropism and axonal transport of AAV serotypes 1, 2, 5, 6, 8 and 9 in the mouse
brain. *PLoS One* **8**, e76310,(2013).

Zingg, B. *et al.* AAV-Mediated Anterograde Transsynaptic Tagging: Mapping

Corticocollicular Input-Defined Neural Pathways for Defense Behaviors.

*Neuron* **93**, 33-47,(2017).

Dear Reviewer,

The line numbers referenced in our point-by-point responses correspond to the

"Revised manuscript without track changes-He Li.pdf" file.

**Reviewer #3-Li (Remarks to the Author):**

**This manuscript by Li et. al presents the interesting and somewhat provocative**

**claim that multi-sensory integration, decision-making, and generation of motor**

**command to initiate walking all occur within an intra-laminar circuit in layer 5 of**

**temporal association cortex (TeA). The work does a careful anatomical**

**characterization of the projection from TeA to periaqueductal grey (PAG) and a**

**series of manipulations of this pathway, with increasing specificity as the**

**manuscript progresses. However, the only readout is stimulus-evoked locomotion.**

**Locomotion in other contexts would need to be assayed to determine specificity.**

**Furthermore, analyses of locomotor responses to each of the stimuli are not**

**sufficiently presented to interpret the effects. More generally, the results are**

**sloppily presented, with inconsistencies in presentation, a lack of justification for**

**selection of data for illustrations, a lack of detail needed to interpret statistical**

comparisons, and numerous saturated anatomical images. The intersectional
anatomical characterization of input in Figure 3 is valuable, but for a more
specialized audience. Key experiments for necessity are missing, which would rule
out other pathways, such as ACC, sensory-to-PAG, mesencephalic locomotor
region, or cortico-striatal pathways. Along these lines, more broadly, many circuits
and pathways are known to be involved in sensory-driven locomotion and thus the
results of this paper are a modest advance that would require more careful
experiments, analysis, presentation, and interpretation to be persuasive.

**Major concerns:**

**-Organization of the results lacks introductory and other transitional sentences,**
**so it is not clear how one section relates to the next. For example, first the first two**
**sections with headers ‘Open field running model.’ And ‘turntable running model’**
**describe two behaviors but it is not clear where things are going. Are both going**
**to be used later? What is the purpose of the open field model given the turntable**
**model is described as having some advantages but no disadvantages.**

Indeed, our text in this part of the manuscript is unclear. Under open-field conditions,
we induced defensive escape behaviours in mice using three types of sensory stimuli.
To facilitate localized brain intervention and record neuronal activity, we needed to
immobilize the animal's head while allowing it to run, so we adopted a turntable running
model. The comparison between the two setups was intended to validate the use of the

turntable running model for interpreting defensive escape behaviours observed under
freely moving conditions. This part of the statement was also included in the Results
section “*To facilitate interventions in the TeA and observe corresponding behavioural*
*changes in the mice, as well as to record neuronal responses, we transitioned from the*
*open-field model to a head-fixed, turntable running model.*” [Results (line 124-126)]

**-The framing around multisensory may create confusion, since the paper does not**
**look at multisensory integration (e.g. stimuli/cues with components/features from**
**multiple senses). Rather, it presents sound, light, or air puff on separate trials. As**
**such, they are studying a set of related associations for the 3 senses in isolation.**
**From this perspective, there are gross overstatements of the results, such as this:**
**“With the above experiments, we have established that the TeA-dPAG pathway is**
**the neural circuit associated with multisensory-induced movement.” And**
**“Therefore, the TeA must have the function of integrating various sensory**
**information.” This is partially addressed by experiments in Figure 4g-l; most of**
**the paper is about uni-sensory motor responses and framing should be adjusted,**
**accordingly.**

Thank you very much for your comments. Your feedback highlighted that our writing
could have led readers to confuse single sensory stimulation with multisensory
integration behaviours while mistakenly attributing the observed multisensory
integration effects observed in the experimental behaviours to the TeA. We have

removed the section describing the latter while acknowledging that our experimental
design using separate, unimodal stimuli (sound, light, air puffs) does not actually test
multisensory integration but rather examines parallel processing of distinct sensory
modalities.

**-The descriptions of analyses and selection of data to be presented are not clearly**
**motivated or described to give confidence in what is being compared and why.**

**-The results are sloppily presented, with inconsistencies in presentation, a lack of**
**justification for selection of data for illustrations, and a lack of detail needed to**
**interpret statistical comparisons.**

Thank you for your feedback, which has highlighted critical conceptual, technical, and
narrative shortcomings in the original manuscript. To address these issues, we have
undertaken a comprehensive revision: we have restructured the logical flow of the
entire article to clarify key concepts; we have redrawn the figures and provided details
on additional experimental investigations. We have conducted novel data analyses to
improve the methodological rigor of the manuscript, and following a full rewrite of the
manuscript, the paper has been submitted to a professional English editing service to
improve the linguistic precision of the text. For example, in the revised manuscript, we
made significant modifications to the experiments under “*Silencing TeA blocks sensory-*
*induced flight behaviour.*” In this study, we employed pharmacological, chemogenetic,
and optogenetic methods to inhibit neuronal activity in the bilateral TeA and observed

changes in flight behaviours induced by different sensory stimuli. To ensure consistency
 in the results, we have revised the presentation of the original Fig. 1, which separately
 illustrates the changes in flight behaviour induced by three types of sensory stimuli
 under each silencing method. Additionally, to provide further details on the statistical
 analysis, we explicitly state in the figure legend that one-way repeated-measures
 ANOVA with Bonferroni post hoc correction was performed, along with the specific p
 values and degrees of freedom.

**Fig. 1 | The TeA as a controller of flight behaviours.** i, Top, slices containing the bilateral TeA

showing the spread of muscimol. Scale bar, 100 μ m, unless otherwise specified. Bottom,

Peristimulus time histograms (PSTHs, 100 trials) for responses of a TeA neuron to noise (80 dB

SPL) before (Mus-) and after muscimol injections (Mus+). Statistics for 7 TeA cells (at depths of
250–800 μm) in 7 animals. Calibration: 50 μV , 0.5 ms. $t_6 = 9.927$, $P = 6 \times 10^{-5}$. Paired t test. **m**,
Example flight speed trace. Thick lines: averages (10 trials). **n**, Peak speeds. S, $F_{2,12} = 854.417$, $P <$
10^{-8} . L, $F_{2,12} = 488.007$, $P < 10^{-8}$. A, $F_{2,12} = 366.002$, $P < 10^{-8}$. One-way repeated-measures ANOVA
with Bonferroni post hoc correction. $n = 7$ animals. **o**, Top, slices containing the bilateral TeA
showing viral expression. Bottom left, raw traces of current-clamp recordings from an hM4D(Gi)-
expressing TeA cell in a slice preparation. Bottom right, average spontaneous spike frequencies
before and after brain perfusion with CNO and after CNO washout. $F_{2,8} = 260.973$, $P < 10^{-8}$. One-
way repeated-measures ANOVA with Bonferroni post hoc correction, $n = 5$ cells from two mice. **p**,
Left, example flight speed traces before (CNO-) and 30 min after CNO intraperitoneal injection (i.p.,
CNO+) in hM4D(Gi)-expressing animals. Right, peak speed induced by S ($t_6 = 18.192$, $P = 2 \times 10^{-6}$),
L ($t_6 = 13.191$, $P = 1.2 \times 10^{-5}$), and A (A: $t_6 = 14.407$, $P = 7 \times 10^{-6}$) before and after CNO injection.
Paired t test. $n = 7$ animals. **q**, Left, Example flight speed traces before (CNO-) and 30 min after
CNO intraperitoneal injection (i.p., CNO+) in mCherry control animals. Right, peak speed induced
by S ($t_6 = -0.523$, $P = 0.620$), L ($t_6 = 1.127$, $P = 0.303$), and A (A: $t_6 = 0.922$, $P = 0.392$) before and
after CNO injection. Paired t test. $n = 7$ animals. **r**, Top, bilateral TeA slices showing viral expression.
Bottom, PSTHs (100 trials) for responses of a TeA neuron to noise (80 dB SPL) without (LED-) and
with (LED+) LED illumination. Data correspond to 7 TeA cells (at depths of 250–800 μm) from 7
animals. Calibration: 50 μV , 0.5 ms. $t_6 = 10.405$, $P = 1.41 \times 10^{-4}$. Paired t test. **s**, Left, Example flight
speed traces without (LED-) and with (LED+) LED illumination for eNpHR-expressing animals.
Right, peak speed induced by S ($t_6 = 12.179$, $P = 1.9 \times 10^{-5}$), L ($t_6 = 6.209$, $P = 0.001$), and A (A: t_6
$= 8.175$, $P = 1.8 \times 10^{-4}$) without and with LED illumination. Paired t test. $n = 7$ animals. **t**, Left,

example flight speed traces obtained without (LED-) and with (LED+) LED illumination for EGFP
control animals. Right, peak speed induced by S ($t_6 = 18.192$, $P = 2 \times 10^{-6}$), L ($t_6 = 13.191$, $P =$
1.2×10^{-5}), and A (A: $t_6 = 14.407$, $P = 7 \times 10^{-6}$) without and with LED illumination. Paired t test. n =
7 animals.

**-Numerous examples are given below in ‘minor concerns’ section, as well as a few**
**‘big ones’ here in the major concerns section.**

**-It is not clear if ‘Silencing TeA blocks running’ experiments are done in the freely**
**moving or head-fixed paradigm. Similarly, it is not clear in panel l and m of Figure**
**1 if these speed calculations are done for sound, light, or air puff. There is a smillar**
**issue with the optogenetic experiments, where is is not clear if the running under**
**investigation is triggered by sound, light, or air puff.**

We sincerely appreciate your careful reading and insightful questions regarding our
experimental paradigms. We would like to clarify the following important
methodological details: The experiments in the “*Silencing TeA blocks sensory-induced*
*flight behaviour.*” section (including both muscimol silencing, shown in Fig. 1m and n,
and optogenetic inhibition, shown in Fig. 1s and t) were indeed conducted with the
head-fixed turntable paradigm. As explained in our response to previous comments, we
transitioned all subsequent experiments to the turntable model after initial validation in
an open field setting, owing to the advantages of the former for precise neural
manipulation and recording. For all the silencing experiments (muscimol,

chemogenetic, and optogenetic), we systematically recorded and analysed the
responses to all three sensory modalities (sound, light, and air puff) separately. These
stimulus-specific results are presented in Fig. 11-t.

**-Physiological recording for the CNO experiments in Figure 1 are lacking, so it is**
**not clear what effect the CNO has had on TeA and other structures.**

Thank you for your valuable feedback. In response, we have added additional
information to Fig. 1 (see Fig. 1o).

**Fig. 1 | The TeA as a controller of flight behaviours. o**, Top, slices containing the bilateral TeA
showing viral expression. Bottom left, raw traces of current-clamp recordings from an hM4D(Gi)-
expressing TeA cell in a slice preparation. Bottom right, average spontaneous spike frequencies
before and after brain perfusion with CNO and after CNO washout. $F_{2,8} = 260.973$, $P < 10^{-8}$. One-
way repeated-measures ANOVA with Bonferroni post hoc correction, $n = 5$ cells from two mice.

**-AAVretro is known to leak into neighboring cells. It would be helpful to confirm**
**the projection from TeA layer 5a to PAG using a confirmatory method, such as**

**pseudorabies or fluoro-gold.**

We acknowledge the concern regarding the leakage of AAVretro into neighbouring cells.

We fully appreciate your concerns and agree that the use of complementary methods

would improve our validation of the TeA-dPAG circuit. In fact, multiple findings in our

manuscript consistently support the hypothesis that TeA layer 5a neurons project to the

dPAG, including Fig. 3d2 and e2, Fig. 6a and h, and Extended Data Fig. 1b. In direct

response to your suggestion, we have performed additional verification experiments by

injecting the retrograde tracer CTB into the dPAG. The results (see Figure below)

confirm our original findings, again demonstrating the projection of TeA layer 5a

neurons to the dPAG.

**Fig. 3 | A TeA to dPAG circuit determines running behaviours. d,** Blockade of the TeA_{dPAG}

CaMKII neurons in turntable model mice: **1,** Viral injection site and chemogenetic (CNO) and

optogenetic (LED) interventions. **2,** Viral expression. **e,** Blockade of TeA_{dPAG} CaMKII neurons. **1,**

Virus injection site and chemogenetic (CNO) intervention. **2,** Viral expression.

**Figure 6 | Sensory TeA and TeAdPAG neurons.** **a**, Top left, schematic of virus injection in the AC, S1,

V2L, MGB, DLG, TeA, and dPAG. Images showing Sensory TeA neurons (EYFP, green) in L2-5b,

including AC TeA, S1 TeA, V2L TeA, MGB TeA, and DLG TeA, and TeAdPAG neurons in L5a (mCherry, red)

in the TeA. **g**, Schematics of the virus injection sites. **h**, Left, images showing Sensory TeA neurons

(EYFP), TeAdPAG neurons and neurons labelled in the upper layer (mCherry).

**Extended Data Fig. 1 | Cells in the TeA and dPAG that comprise the TeA-dPAG circuit. b,**

Images showing mCherry- and EYFP-labelled TeA_{dPAG} neurons in wild-type mice (top) and Cre

transgenic mice (bottom). Yellow reflects labelling overlap in the neurons.

**a,** Injection site for CTB 488 in the dPAG. **b,** Images showing CTB 488-labelled TeA_{dPAG} neurons.

-Some images are saturated (e.g. 1k, q; 2a1; 3a, c; e1b; e2g; e3a) or show signs of

injection damage (2a2), raising concerns about data quality. Images should be

**adjusted so they are not saturated and extent of tissue damage at injection site**
**should be discussed and/or addressed as needed with new experiments.**

We sincerely appreciate your careful examination of our images and your constructive
suggestions for improvement. We have carefully reviewed all the listed images (Figs.
1k and q, 2a1, and 3a, c, e1b, e2g, and e3a) and have adjusted the exposure levels to
eliminate saturation while maintaining the necessary signal clarity. All saturated images
or those showing signs of injection damage have been replaced in the revised
manuscript. Regarding the latter images, we would like to clarify that this damage was
not caused by the mouse intracranial cannula implantation surgeries. Instead, it resulted
from the removal of the cannulas after the experiments to verify their placement
accuracy. Moreover, the extent of this damage was relatively minor. We also noted this
damage ourselves. To further confirm whether blocking TeA_{dPAG} neurons affects
sensory stimulus-induced flight behaviour, we conducted additional experiments,
illustrated in Fig. 3d, where we bilaterally injected optogenetic viruses into the dPAG
and delivered yellow LED light to the bilateral TeA to specifically inhibit these neurons.
The results are consistent with those in Fig. 3a. Therefore, we believe that the
experimental findings in Fig. 3a are reliable.

Figure 1

Figure 2

Figure 3

Figure 5

Figure S1

Figure S3

**Fig. 1 i**, Slices containing the bilateral TeA showing the spread of muscimol. Scale bar, 100 μ m,
unless otherwise specified. **r**, Bilateral TeA slices showing viral expression. **Fig. 2 b**, Left, injection
site for AAVretro in the dPAG. Right, mCherry-labelled TeA_{dPAG} neurons. **Fig. 3 a**, Blockade of the
fibres from the TeA to the dPAG in turntable model mice. **1**, Viral injection site and chemogenetic
(CNO) and optogenetic (LED) interventions. **2**, Viral expression. **Fig. 5 a**, Viral injection site in the
TeA (left) and expression in target cortical (middle, AC, S1, and V2L) and subcortical regions (right,
DLG, and MGB). **c**, Distribution of TeA neurons receiving projections from different sensory
(_{Sens}TeA) source regions. **Extended Data Fig. 1 i**, PV neurons in the PAG. **Extended Data Fig. 3**
**a**, Injection sites of AAV2/9-hSyn-mCherry into the AC, S1, V2L, MGB, and DLG.

**-It seems possible that the CNO and optogenetic experiments in Figures 1 and 2**
**are a ‘hammer’ and are shutting to locomotion and behavior rather broadly,**
**rather than indicating the specific multimodal integration and transformation to**
**a motor signal that the paper suggests. It would be helpful to determine if**
**locomotion in other contexts (e.g. plus maze exploration) is impacted. Also, CNO**
**injections into virus control animals would be needed to mitigate e.g. tissue**
**damage concerns.**

We sincerely thank you for your suggestion and have incorporated it into our study.
Specifically, we conducted an open field test while performing chemogenetic inhibition
to selectively block TeA_{dPAG} and TeAdPAG neurons and examined the effects of this
blockade on spontaneous locomotor activity. By blocking TeA_{dPAG} (Fig. 3 e1 and e2)

and TeA_{dPAG} neurons (Fig. 3 f1 and f2) separately, we observed that both manipulations
 affected the ability of the mice to freely explore (Fig. 3e3 and f3). The results revealed
 that inhibiting TeA_{dPAG} or TeA_{dPAG} neurons induced bradykinesia, as indicated by
 significant decreases in both total distance moved and movement speed (Fig. 3e4 and
 f4, upper panels), and akinesia, as indicated by a sharp increase in immobility (Fig. 3e4
 and f4, lower panels, left). These results indicate that CaMKII-positive projections
 through TeA layer 5a to the dPAG determine running behaviour. Furthermore, we have
 now included the CNO control figure that was initially omitted for brevity.

**Fig. 3 | A TeA to dPAG circuit determines running behaviours. e**, Blockade of TeA_{dPAG} CaMKII

neurons. **1**, Virus injection site and chemogenetic (CNO) intervention. **2**, Viral expression. **3**,

Representative traces before (CNO-) and after (CNO+) intraperitoneal (i.p.) injections in

hM4D(Gi)-expressing animals. **4**, Performance of hM4D and control mice in the open field test:
distance ($F_{1,18} = 5.192$, $P = 0.0351$), speed ($F_{1,18} = 15.68$, $P = 0.0009$), immobility time ($F_{1,18} =$
8.107 , $P = 0.0107$), and time spent exploring the centre zone ($F_{1,18} = 0.9949$, $P = 0.3318$) before
(CNO-) and after (CNO+) CNO intraperitoneal injections. Two-way ANOVA with Bonferroni post
hoc correction. $n = 10$ animals/group. **f**, Blockade of T_e ADPAG CaMKII neurons. **1**, Virus injection
site and chemogenetic (CNO) intervention. **2**, Viral expression. **3**, Representative traces before
(CNO-) and after (CNO+) CNO intraperitoneal (i.p.) injection in hM4D(Gi)-expressing animals. **4**,
Performance of hM4D and control mice in the open field test: distance ($F_{1,18} = 27.70$, $P < 10^{-4}$),
speed $F_{1,18} = 51.17$, $P < 10^{-4}$), immobility time ($F_{1,18} = 78.19$, $P < 10^{-4}$), and time spent exploring
the centre zone ($F_{1,18} = 0.7787$, $P = 0.3892$) before (CNO-) and after (CNO+) intraperitoneal
injection. Two-way ANOVA with Bonferroni post hoc correction. $n = 10$ animals/group.

**-In Figure 2, the responses to the 3 different stimuli should be parsed (walking to**
**sound, light, and airpuff).**

We appreciate your suggestion to parse the responses to the different sensory stimuli.
We would like to clarify our presentation strategy. As noted by the reviewer, we have
already presented separate analyses for each sensory modality (sound, light, air puffs)
in Fig. 1m, n, p, q, s, and t. The data shown in these figure panels demonstrate that TeA-
dPAG inhibition blocks flight behaviours induced by all three stimulus types similarly,
with no significant differences observed among the modalities. Given the consistent
effects across all sensory stimuli (i.e., complete blockade of flight behaviour regardless

of modality), we sought to maintain a focus on the core finding that TeA-dPAG circuit
 activity is essential for sensory-induced flight regardless of sensory modality and thus
 avoided the redundant presentation of identical effects across multiple panels.

**Fig. 1 | The TeA as a controller of flight behaviours.** m, Example flight speed trace. Thick lines:

averages (10 trials). n, Peak speeds. S, $F_{2,12} = 854.417, P < 10^{-8}$. L, $F_{2,12} = 488.007, P < 10^{-8}$. A,

$F_{2,12} = 366.002, P < 10^{-8}$. One-way repeated-measures ANOVA with Bonferroni post hoc correction.

n = 7 animals. p, Left, example flight speed traces before (CNO-) and 30 min after CNO

intraperitoneal injection (i.p., CNO+) in hM4D(Gi)-expressing animals. Right, peak speed induced

by S ($t_6 = 18.192, P = 2 \times 10^{-6}$), L ($t_6 = 13.191, P = 1.2 \times 10^{-5}$), and A (A: $t_6 = 14.407, P = 7 \times 10^{-6}$)
before and after CNO injection. Paired t test. n = 7 animals. **q**, Left, Example flight speed traces
before (CNO-) and 30 min after CNO intraperitoneal injection (i.p., CNO+) in mCherry control
animals. Right, peak speed induced by S ($t_6 = -0.523, P = 0.620$), L ($t_6 = 1.127, P = 0.303$), and A
(A: $t_6 = 0.922, P = 0.392$) before and after CNO injection. Paired t test. n = 7 animals. **s**, Left,
Example flight speed traces without (LED-) and with (LED+) LED illumination for eNpHR-
expressing animals. Right, peak speed induced by S ($t_6 = 12.179, P = 1.9 \times 10^{-5}$), L ($t_6 = 6.209, P =$
0.001), and A (A: $t_6 = 8.175, P = 1.8 \times 10^{-4}$) without and with LED illumination. Paired t test. n = 7
animals. **t**, Left, example flight speed traces obtained without (LED-) and with (LED+) LED
illumination for EGFP control animals. Right, peak speed induced by S ($t_6 = 18.192, P = 2 \times 10^{-6}$),
L ($t_6 = 13.191, P = 1.2 \times 10^{-5}$), and A (A: $t_6 = 14.407, P = 7 \times 10^{-6}$) without and with LED illumination.
Paired t test. n = 7 animals.

**-The characterization of neuronal responses in TeA to running versus sound**
**stimuli, although intriguing, do not demonstrate much about the integration**
**process. Furthermore, running related signals are extremely prominent across**
**neocortex, and thus not surprising or very informative regarding this study.**

We sincerely appreciate your insightful critique regarding the interpretation of our
neuronal response data. We have removed all claims about the role of the TeA in
multisensory integration. We initially considered removing this section, but given its
potential implications regarding multisensory integration behaviours, we have decided

to retain it as supplementary information.

While we acknowledge that the mechanisms underlying running-related activity are
widespread throughout the neocortex (as demonstrated in Steinmetz et al.¹ and Stringer
et al.²), our study reveals novel insights into the functional heterogeneity of neurons in
neocortical TeA layer 5a. Additionally, although running-related signals are very
prominent in the neocortex, their characteristics are not necessarily correlated
specifically with the the animal's running speed. Furthermore, the results of
pharmacological, chemogenetic, and optogenetic interventions collectively confirmed
that the TeA is a key hub for flight behaviours induced by various sensory stimuli.
Importantly, we identified both running-related neurons (R-neurons; Fig. 4b) and
sensory-related neurons (S-neurons; Fig. 4c-d) in this area. A distinct population of S-
neurons demonstrated firing rates that were correlated with running speed (SR-neurons;
Fig. 4b) while simultaneously responding to sensory stimuli. These neuronal subtypes
are predominantly localized in L5a and appear to integrate different sensory
information into running signals, ultimately driving running behaviour. To our
knowledge, this represents the first demonstration of such functional specialization in
TeA L5a, suggesting that this region plays a pivotal role in sensory–motor
transformation.

**Fig. 4 | Sensation-related and running-related neurons in the TeA.** a, Recording setup. R,

recording pipette. Sound, light, and air puff stimuli were delivered to the mouse. The running speed

was monitored in real time by the rotary encoder. b-d, Example characteristics for three types of

TeA neuron with firing rates (black), sound, light, and air puff waveforms (blue), and running speed

(red). (b), A running-related neuron (R-neuron). Right panel, top, peristimulus time histogram

(PSTH) of the neuron during presentation of S, L, and A. Right panel, bottom, firing rate plotted

versus the original running speed. Insert, discharge events and running events extracted following

adjustment of the running speed data to align the firing and running speed peaks. (c), A sensory-

related neuron (S-neuron) whose firing rate was correlated with running speed (SR-neuron;

responds to sound stimulation (SS)). (d), A sensory-related neuron whose firing rate was

uncorrelated with running speed (unSR; responds to SS). The discharge characteristics are plotted

as the firing rate (black). The ordinates for firing rate and running speed are on the right and left,

respectively.

**-The critical experiment of silencing the TeA-dPAG neurons and determining if**
**that blocks sound/light/air-puff induced locomotion appears to be missing. This is**
**the experiment that would test if those neurons are necessary for the behavior.**
**Without this experiment, it seems likely that many other pathways are also**
**involved and/or sufficient.**

We apologize for not providing the detailed experimental procedures underlying the
results in Fig. 2 in the original manuscript. However, we did present them in the original
Fig. 2 (now reorganized as Fig. 3; please refer to “*To investigate the role of the TeA-*
*dPAG pathway in controlling flight behaviours, we injected the AAV-hSyn-hM4D(Gi)-*
*mCherry virus into the bilateral TeA to express hM4D in TeA cells (Fig. 3a1). Local*
*injection of CNO (with 5% BDA) into the dPAG blocked the TeA-dPAG projection fibres*
*(Fig. 3a2), resulting in a significant cessation of the flight behaviours induced by the*
*three different stimuli. In contrast, local injection of CNO had a minimal effect on*
*control animals expressing only AAV-hSyn-mCherry without hM4D(Gi) (Fig. 3 a3 and*
*a4). To specifically block the TeA-dPAG pathway, we locally injected Cre (AAV-Cre-*
*EYFP, TeA) + DIO (AAV-DIO-hM4D(Gi)-mCherry, dPAG) (Fig. 3b1), which led to*
*hM4D expression in $T_{eA}dPAG$ cells (Fig. 3b2). Intraperitoneal injections of CNO*
*specifically inhibited $T_{eA}dPAG$ cells and blocked the flight behaviours induced by the*
*three sensory stimuli, whereas no blockade was observed in the control group injected*
*with hM4D-free virus (Fig. 3b3 and b4).” [Results (line 219-231)] and the updated Fig.*

3). Additionally, to further clarify whether the TeA-dPAG circuit is related to running,
 we conducted an open field test in which chemogenetic inhibition was used to
 selectively block TeAdPAG and TeAdPAG neurons and examine the effects of this
 blockade on spontaneous locomotor activity. By blocking TeAdPAG (Fig. 3 e1 and e2)
 and TeAdPAG neurons (Fig. 3 g1 and g2) separately, we observed that both
 manipulations affected the ability of the mice to freely explore (Fig. 3e3 and f3). The
 results revealed that inhibiting TeAdPAG or TeAdPAG neurons induced bradykinesia, as
 indicated by significant decreases in both total distance moved and movement speed
 (Fig. 3e4 and f4, upper panels), and akinesia, as indicated by a sharp increase in
 immobility (Fig. 3e4 and f4, lower panels, left). These results indicate that CaMKII-
 positive projections through TeA L5a to the dPAG determine running behaviours.

**Fig. 3 | A TeA to dPAG circuit determines running behaviours. a-d**, Blockade of the TeA-dPAG

pathway in turntable model mice: fibres from the TeA to the dPAG (**a**), TeAdPAG neurons (**b**),

TeAdPAG CaMKII neurons (**c**), and TeAdPAG CaMKII neurons (**d**). **1**, Viral injection site and

chemogenetic (CNO) and optogenetic (LED) interventions. **2**, Viral expression. **3**, Example flight

speed trace following random S, L, and A presentation before (CNO- or LED-) and after (CNO+ or

LED+) interventions in hM4D(Gi)- or eNpHR-expressing animals. Thick lines: average flight speed

(30 trials). **4**, Peak flight speeds for **a** ($F_{1,12} = 46.54, P < 10^{-4}$), **b** ($F_{1,12} = 69.73, P < 10^{-4}$), **c** ($F_{1,12} =$

$112.0, P < 10^{-4}$), and **d** ($F_{1,12} = 62.16, P < 10^{-4}$), two-way ANOVA with Bonferroni post hoc

correction. $n = 7$ animals/group. **e**, Blockade of TeAdPAG CaMKII neurons. **1**, Virus injection site

and chemogenetic (CNO) intervention. **2**, Viral expression. **3**, Representative traces before (CNO-

and after (CNO+) intraperitoneal (i.p.) injections in hM4D(Gi)-expressing animals. **4**, Performance

of hM4D and control mice in the open field test: distance ($F_{1,18} = 5.192, P = 0.0351$), speed ($F_{1,18} =$

$15.68, P = 0.0009$), immobility time ($F_{1,18} = 8.107, P = 0.0107$), and time spent exploring the centre

zone $F_{1,18} = 0.9949$, $P = 0.3318$) before (CNO-) and after (CNO+) CNO intraperitoneal injections.
Two-way ANOVA with Bonferroni post hoc correction. n = 10 animals/group. **f**, Blockade of
$T_{eA}dPAG$ CaMKII neurons. **1**, Virus injection site and chemogenetic (CNO) intervention. **2**, Viral
expression. **3**, Representative traces before (CNO-) and after (CNO+) CNO intraperitoneal (i.p.)
injection in hM4D(Gi)-expressing animals. **4**, Performance of hM4D and control mice in the open
field test: distance ($F_{1,18} = 27.70$, $P < 10^{-4}$), speed $F_{1,18} = 51.17$, $P < 10^{-4}$), immobility time ($F_{1,18} =$
78.19 , $P < 10^{-4}$), and time spent exploring the centre zone ($F_{1,18} = 0.7787$, $P = 0.3892$) before (CNO-)
and after (CNO+) intraperitoneal injection. Two-way ANOVA with Bonferroni post hoc correction.
n = 10 animals/group.

**Minor concerns:**

**-Figure 1f shows peak running speed and the authors claim the stimulation**
**intensities were carefully chosen to evoke similar running speeds. It would also**
**help to see average, rather than peak, running speed, given that this peak running**
**speed may be insensitive to differences in evoked running patterns by the stimuli**
**used.**

We sincerely appreciate this suggestion. In our preliminary analyses, we systematically
evaluated multiple kinematic parameters, including average speed, acceleration, and
total distance. However, we ultimately selected peak speed as our primary metric on
the basis of two key observations:

1. Neural correlations: We found that instantaneous running speed showed the
strongest correlation with neuronal firing rate (Chen et al., Figs. 1-2), suggesting that
peak speed best reflects the neural dynamics of interest.

2. Field standard: This choice aligns with established practices in comparable studies
employing similar behavioural paradigms, including Evans et al.³, Xiong et al.⁴, and
Wang et al.⁵.

To ensure robustness, we complemented this analysis with a quantification of running
duration (Fig. 1g). We believe that this approach provides the most biologically
meaningful representation of locomotor capability across behavioural states.

**Fig. 1 | The TeA as a controller of flight behaviours.** g, duration ($F_{2,12} = 4.191, P = 0.042$) of flight.

One-way repeated-measures ANOVA with Bonferroni post hoc correction. $n = 7$ animals. * $P <$

0.05 , ** $P < 0.01$, *** $P < 0.001$, **** $P < 0.0001$, *n.s.* not significant.

**-How were 10 trials shown in 1c selected? What does these plots look like for**
**different animals and single trial?**

We realize that our original description may have been unclear. Fig. 1c actually displays
the average movement trajectories from 10 stimulus repetitions for the representative

animal shown in Fig. 1b. These trajectories were originally positioned below Fig. 1b
but were moved to Fig. 1c to improve the layout of the figure.

**Fig. 1 | The TeA as a controller of flight behaviours. b**, Example flight speed raster. Black bar:

Stimulus presentation (5 s). Coloured bar: flight speed. **c**, Example average flight speed (10 trials).

**-It is not clear in Fig. 1d-f how the stas were done. There are 7 animals and 7 data**
**points, suggesting each point is an animal. This should be stated. The type of post**
**hoc test is not stated. And the F values are reported as 2,18 for degrees of freedom.**

**How do these numbers relate to the 7 mice and 7 data points?**

Thank you for your question, which helped us realize not only that our explanation
unclear but also that we used incorrect statistical methods. In the revised version, we
have improved the related descriptions in both the figure legend and methods section
and report the results of one-way repeated-measures ANOVA with Bonferroni post hoc
correction for statistical analysis. Accordingly, we have systematically reviewed and
corrected the statistical methods throughout the manuscript. We are grateful for this
important critique, which has significantly improved the rigor and transparency of our
statistical reporting. The revised manuscript now provides complete documentation of

all analyses to enable full evaluation of our findings.

**-Why are walk bout durations not shown for the freely-moving task in Figure 1?**

We have included additional data in this figure to address this point (see Fig. 1g).

**Fig. 1 | The TeA as a controller of flight behaviours. g**, duration ($F_{2,12} = 4.191, P = 0.042$) of flight.

One-way repeated-measures ANOVA with Bonferroni post hoc correction. $n = 7$ animals. * $P <$

0.05 , ** $P < 0.01$, *** $P < 0.001$, **** $P < 0.0001$, *n.s.* not significant.

**-It is not clear if it is the same, or different, 7 animals used in the freely moving**

**versus head-fixed behavior.**

Thank you for highlighting this important point. We now clarify in the revised

manuscript that the freely moving and head-fixed experiments were conducted using

different cohorts of animals (7 mice per condition) to avoid potential confounding.

**-This statement “However, the characteristics of the running, such as the**

**probability, delay, and peak speed (Fig. 1j), were not significantly different**
**between the turntable model and the open field model (Fig. 1c-f), except for the**
**running duration.” Is not justified based on what is shown in panel ‘j’, at least as**
**described. Again, how the F values were determined, and what the groups were,**
**etc., is not clear.**

We sincerely appreciate your careful reading and constructive critiques regarding our
statistical reporting and interpretation. We acknowledge the confusion caused by our
original statement and have implemented the following revisions. We now present a
more precise and data-driven description; in both behavioural paradigms (open field
and turntable), the presentation of sound stimuli consistently resulted in the highest
probability of eliciting escape behaviours, the shortest response latencies, and the
shortest running durations, while no significant differences were observed in peak
running speeds among the sound, light, and air puff stimuli.

**-The co-localization rate reported in Fig. S1 is ambiguous; they should report both**
**false positive and false negative rates (i.e. were there any retrogradely labeled cells**
**that were negative for camKII/SOM/VIP/and PV? Also, why was AAVretro used**
**for the CaMKII labeling?**

Thank you for your excellent question regarding our use of AAVretro for CaMKII
labelling. Below, we explain our experimental rationale in detail.

Initially, for cell type identification, we planned to use a single-virus approach
(AAVretro-hSyn-EYFP) combined with immunohistochemistry. This strategy would
have involved retrograde labelling of hSyn-expressing neurons followed by anti-
CaMKII staining to quantify the proportion of glutamatergic neurons, similar to the
methodology presented in the original Extended Fig. 2 b-c.
However, after extensive testing of multiple anti-CaMKII antibodies, we observed that
while these antibodies showed excellent labelling efficiency throughout the thalamic
regions (original Extended Fig. 2c), their labelling in cortical areas was less satisfactory.
As shown in the attached figure, the antibody labelling was predominantly localized to
the superficial cortical layers and showed strong edge effects, whereas deeper cortical
layers presented significantly weaker staining.

a

723 **Merge** **hSyn** **Anti-CaMKII**
**a**, Images showing the anti-CaMKII-EYFP-labelled neurons, and hSyn-mCherry-labelled TeA_{dPAG}
neurons. Scale bar, 100 μm.

An alternative approach for neuronal subtype identification would be use of the
Cre/loxP system, involving the injection of AAV-DIO-mCherry into the TeA of
CamKII α -Cre mice to specifically label glutamatergic neurons combined with
AAVretro-hSyn-EYFP injections in the dPAG to examine neuronal colocalization

patterns. Unfortunately, CamKII α -Cre mice are not available in our laboratory.

Therefore, we opted for a CaMKII promoter-driven viral vector, which is a well-

established and widely adopted tool for specific neuronal labelling. To validate

inhibitory neuronal populations, we used Cre \times Ai14 mice in combination with

AAVretro-hSyn-EYFP.

To address your first question concerning the validation of CaMKII neuronal subtypes

in the TeA, we employed an alternative experimental approach. Specifically, we

injected AAVretro-hSyn-EYFP together with AAVretro-CaMKII α (glutamatergic)-

mCherry into wild-type mice. Four weeks postinjection, we performed

immunofluorescence staining with an alternative anti-CaMKII antibody to quantify the

proportion of glutamatergic neurons relative to virally labelled neurons.

While this new antibody-based system yielded improved staining results relative to our

previous attempts, some challenges persisted. Notably, as shown in the attached figure,

the distribution of labelled neurons in the cortex remained uneven relative to reference

data from the Allen Brain Atlas. This observation suggests potential regional variability

in antibody penetration or expression efficiency, particularly in deeper cortical layers.

b

Anti-CaMKII

c

Allen Brain ISH for CaMKII2a

**b**, Images showing the staining results of anti-CaMKII antibody in TeA. Scale bar, 100 µm. **c**,

Images showing the labeling results of CaMKII in the TeA using in ISH from the Allen Brain Atlas.

Scale bar, 100 µm.

Our immunohistochemical analysis of six sections from each of three mice revealed

that 53% of TeA_{dPAG} hSyn-positive neurons were labelled with the anti-CaMKII

antibody, whereas 55% of TeA_{dPAG} CaMKII-positive neurons exhibited anti-CaMKII

immunoreactivity. Notably, 73% of TeA_{dPAG} hSyn-positive neurons exhibited

colabeling with CaMKII, representing a lower colocalization rate than the 86.59%

reported in our original manuscript. This difference likely reflects our adjustment of the

viral titres, as in our previous experiment, we employed AAV_{retro}-CaMKII-mCherry

(8.07×10^{12} vg/mL, BrainVTA) at a higher titre than that used for AAV_{retro}-hSyn-EYFP

(5.18×10^{12} vg/mL, BrainVTA), whereas in the current study, we employed AAV_{retro}-

CaMKII-mCherry (5.17×10^{12} g/mL, BrainVTA) at a comparable titre to that of
AAVretro-hSyn-EYFP. Importantly, the anti-CaMKII antibody successfully labelled
more than 50% of AAVretro-CaMKII-mCherry-targeted neurons without evidence of
overlabelling artefacts related to the viral titre, suggesting that virus-mediated labelling
is more efficient than antibody-based detection for this neuronal population.

d

**d**, Images showing EYFP- and mCherry- labelled TeA_{dPAG} neurons, and the anti-CaMKII-labelled
neurons in wild-type mice. Scale bar, 100 μ m.

For inhibitory neuron validation, we injected AAVretro-hSyn-EYFP into wild-type
mice and performed immunofluorescence staining with an anti-GAD67 antibody four
769 weeks postinjection to quantify the proportion of inhibitory neurons among the virus-
770 labelled neurons. Analysis of 9 brain sections (3 sections per mouse from 3 mice)
revealed a colabelling rate of only 3.82% between virus-labelled hSyn-positive neurons
and GAD67-immunoreactive neurons. This exceptionally low positive rate strongly
suggests that the vast majority of TeA L5a neurons projecting to the dPAG are
glutamatergic in nature. The minimal overlap observed between virus labelling and
GABAergic markers provides compelling evidence for the predominantly excitatory
character of this neural pathway.

e

Merge

hSyn

Anti-GAD67

e, Images showing hSyn-mCherry -labelled TeA_{dPAG} neurons, and anti-GAD67-labelled neurons.

Scale bar, 100 µm.

**-The purpose of this sentence is unclear: “Due to the scarcity of cholinergic and**
**dopaminergic neurons in the cortex, we did not perform corresponding tests on**
**these neuron types.” Why are cholinergic and dopaminergic neurons singled out**
**for mention? And in fact there are cholinergic neurons in neocortex (Von**
**Engelhardt, J., Eliava, M., Meyer, A. H., Rozov, A., & Monyer, H. (2007).**
**Functional characterization of intrinsic cholinergic interneurons in the cortex.**
**Journal of Neuroscience, 27(21), 5633-5642.).**

We sincerely appreciate your insightful comments regarding cholinergic neurons in the
cortex. We apologize for our initial incorrect statement about their absence in cortical
layers. Upon reviewing the manuscript you have cited, we recognize that while
cholinergic neurons are present in the neocortex, they are predominantly localized to
L2/3, a distribution distinct from our focus on L5a neurons projecting to the dPAG.
After careful consideration, we have decided not to pursue whether the TeA L5a-dPAG

projection contains cholinergic neurons in the current study.
 Regarding your question about why we specifically mentioned cholinergic and
 dopaminergic neurons, this reflected our initial exploratory efforts using antibody
 staining to examine these populations in the TeA, although we acknowledge that the
 limitations of this approach in cortical tissue led to unreliable results. We agree that
 discussing these specific neuron types without definitive data could be potentially
 distracting; therefore, in our revised manuscript, we have removed this speculative
 content to maintain a focus on our core findings.

**-Statistical quantification of significant reduction for Figure 4d-f is lacking.**

Thank you for your valuable suggestion. We have added details regarding the statistical
 analysis for quantifying the reduction in the figure legend (Fig. 5j-l), including details
 on the specific tests used, sample sizes, and significance thresholds.

**Fig. 5 | Diverse sensory input characteristics of the TeA. j-l, Peak speed (red) and probability**

(blue) of running induced by sound (**j**), light (**k**), and air puff (**l**) stimulation before (dark) and after
(undertint) CNO i.p. injection, n = 5 animals for each sensory area. Numbers: percentage reduction
in the peak speed and running probability for CNO+ animals relative to CNO- animals. **j-speed:**
AC: $t_4 = 10.88$, $P = 4 \times 10^{-4}$; MGB: $t_4 = 8.648$, $P = 0.001$; S1: $t_4 = 4.181$, $P = 0.014$; V2L: $t_4 = -$
3.701 , $P = 0.021$; DLG: $t_4 = 4.480$, $P = 0.011$. **j-probability:** AC: $t_4 = 10.59$, $P = 5 \times 10^{-4}$; MGB: t_4
$= 21.78$, $P < 10^{-4}$; S1: $t_4 = 9.914$, $P = 6 \times 10^{-4}$; V2L: $t_4 = 8.257$, $P = 0.001$; DLG: $t_4 = 5.568$, $P =$
0.005 . **k-speed:** AC: $t_4 = 4.351$, $P = 0.012$; MGB: $t_4 = 0.937$, $P = 0.402$; S1: $t_4 = 1.580$, $P = 0.189$;
V2L: $t_4 = 13.97$, $P = 2 \times 10^{-4}$; DLG: $t_4 = 46.96$, $P < 10^{-4}$. **k-probability:** AC: $t_4 = 4.851$, $P = 0.008$;
MGB: $t_4 = 1$, $P = 0.374$; S1: $t_4 = 1.183$, $P = 0.302$; V2L: $t_4 = 14.85$, $P = 1 \times 10^{-4}$; DLG: $t_4 = 9.408$,
$P = 7 \times 10^{-4}$. **l-speed:** AC: $t_4 = 1.344$, $P = 0.250$; MGB: $t_4 = 2.087$, $P = 0.105$; S1: $t_4 = 12.08$, $P =$
3×10^{-4} ; V2L: $t_4 = 0.560$, $P = 0.606$; DLG: $t_4 = 1.335$, $P = 0.253$. **l-probability:** AC: $t_4 = 0.703$, P
$= 0.521$; MGB: $t_4 = 0.222$, $P = 0.835$; S1: $t_4 = 7.882$, $P = 0.001$; V2L: $t_4 = 0.247$, $P = 0.817$; DLG:
$t_4 = 1.855$, $P = 0.137$. Paired t test.

**-Experiments in Figure 4m-r would benefit from a CNO or optogenetic**
**manipulation. More broadly, some form of psychometric manipulation to**
**understand how graded responses varies with graded manipulation would**
**strengthen the manuscript.**

Thank you very much for your suggestions. On the basis of your earlier feedback, we
fully recognize the mistake of extrapolating the multisensory integration observed in
behavioural experiments to imply the integration function of the TeA. While we initially

considered removing this section entirely, we decided to retain it as supplementary
information, as it may still provide some useful insights into multisensory integration
behaviour.

**References:**

Steinmetz, N. A., Zatzka-Haas, P., Carandini, M. & Harris, K. D. Distributed
coding of choice, action and engagement across the mouse brain. *Nature* **576**,
266-273,(2019).

Stringer, C. *et al.* Spontaneous behaviors drive multidimensional, brainwide
activity. *Science* **364**, 255,(2019).

Evans, D. A. *et al.* A synaptic threshold mechanism for computing escape
decisions. *Nature* **558**, 590-594,(2018).

Xiong, X. R. *et al.* Auditory cortex controls sound-driven innate defense
behaviour through corticofugal projections to inferior colliculus. *Nat Commun*
**6**, 7224,(2015).

Wang, H. *et al.* Direct auditory cortical input to the lateral periaqueductal gray
controls sound-driven defensive behavior. *PLoS Biol* **17**, e3000417,(2019).

**J.C., H.L. et al. “Neural circuits and speed coding of running, rebound running
and backing away behaviours” . Submitted as a companion paper.**

Dear Reviewer,

The line numbers referenced in our point-by-point responses correspond to the
"Revised manuscript without track changes-He Li.pdf" file.

**Reviewer #4 (Remarks to the Author):**

**Li et al. 2025 - An intra-layer microcircuit for multisensory-motor decision and**
**running**

**I congratulate the authors on a powerful manuscript. It is thought-provoking and**
**reads nicely with a logical structure. Included are an awe-inspiring array of**
**experiments. Particularly noteworthy are the number of methodological**
**approaches (pharmacological, chemogenetic, and optogenetic) to address the same**
**research question. This contributes to the convincing nature of the manuscript.**
**The figures include appropriate panels showing the data clearly, clear**
**micrographs of the injections and expression patterns, and the legends are**
**complete with relevant statistics and descriptions.**

**The overall claim of the paper can be argued to be quite extraordinary. A single**
**piece of association cortex (TeA) is responsible for routing and integrating not just**
**one, but multiple sensory signals to evoke running behavior, thus encompassing a**
**significant part of the sensorimotor pathway.**

**I have two main comments, several additional comments, and minor (editorial)**
**remarks. I hope that by addressing these major comments, the manuscript could**

**present its findings more cohesively while maintaining a balanced and**
**scientifically rigorous narrative. I look forward to seeing this exciting work refined**
**and published.**

**1. Clarification of Running Behavior Context and Interpretation.**

**I had a hard time understanding how the authors think about running behavior.**
**The introduction starts with reviewing innate defense and escape behaviors that**
**include running. From lines 55-63 it appears that running is equated with escape**
**behaviors – while escape behavior consists of more complex sequences of**
**behavioral elements including running. The results section talks about the running**
**with the TeA circuit being almost directly responsible for the motor command**
**signals being sent.**

Thank you very much for your valuable comments. Your insights have helped us gain
a clearer understanding of the precise meaning of "escape behaviours" allowing us to
properly distinguish between "flight" and "running." Accordingly, we have thoroughly
revised the manuscript to reflect these conceptual distinctions more accurately
throughout the text. In this revised version, we describe a more precise behavioural
classification: all sensory-induced locomotion is now consistently referred to as "flight
behaviour", whereas the specific motor output mediated by the TeA-dPAG circuit is
distinctly characterized as "running". This distinction reflects our improved
understanding of these behavioural categories, with "flight" representing broader
defensive responses to threat stimuli and "running" describing the specific locomotor

component controlled by the abovementioned neural pathway.

**-Then in the discussion, the authors present the sensory stimuli as more neutral**
**than the aversive stimuli used on other studies. However, 110 (!) dB white noise**
**auditory stimuli, a bright inescapable light and an air puff in the eye (classic**
**aversive stimulus) can NOT be called neutral. These are classic aversive stimuli.**
**Moreover, it becomes completely puzzling to me how the authors think of (the**
**adaptive role of) systematic running following neutral stimuli. Without any**
**context dependence such a proposed circuit would be highly disadvantageous**
**behavior (i.e. starting to run with the onset of any neutral stimulus). On top of this**
**the dPAG is of course pivotal in innate defense and escape behaviors, including**
**running but also freezing.**

We apologize for the confusion caused by the inaccurate terminology used in our
original manuscript. Our use of the term "neutral stimulus" was initially intended to
emphasize that the observed behaviour was specifically running rather than an escape
behaviour induced by aversive stimuli. However, in the current revision, we have
clearly separated and refined our descriptions. We now explicitly refer to the stimuli as
"aversive", as our experiments indeed employed a sensory-induced flight model in the
mice. These changes ensure precise alignment between our methodological approach
and the behavioural paradigm we investigated.

-The methods lack a lot of detail on the running behavior in the freely moving
scenario. When was the sensory stimulus presented and where, on different sides?
Did the behavior depend on the current location of the animal? Was the mouse
always running away from the stimulated side and express thigmotactic behavior?
What was considered the onset of running? Did the mice freeze before running?

We sincerely appreciate your insightful comments. In response to your valuable
suggestions, we have thoroughly revised both the Methods and Results sections of our
manuscript to incorporate these important details. The updated version now provides
comprehensive descriptions addressing all the points you raised, ensuring a clearer
presentation of our experimental procedures and findings. Below is a detailed schematic
diagram of our open field escape model setup:

**-Line 55: Running is usually chosen... I would argue that this is not true and**
**depends on the research. In perceptual decision-making licking, nose poking or**
**forepaw movements are often used as behavioral indices in multisensory decision**
**making tasks. In the context of fear, innate and defense research, freezing is also**
**often used.**

We would like to clarify that our study employed a sensory-induced flight behaviour
model involving mice. We fully acknowledge that it was conceptually inappropriate to
equate defensive flight behaviours with simple running, despite the superficial
similarities in their locomotor manifestations. The two behaviours clearly represent
distinct categories with different underlying mechanisms and ethological implications.
In the revised manuscript, we have carefully corrected this terminology throughout the
text to ensure precise behavioural classification. Specifically, we now consistently use
"flight" to describe the defensive escape response and reserve "running" to refer to the
specific locomotor component mediated by the TeA-dPAG circuit.

**-In sum, the authors need to argue better if and why they think of running as**
**different from escape behavior or not.**

**-This brings me to a related point. Brain-wide activity changes associated with the**
**behavioral state of mice have been shown throughout literature; most prominently**
**in (Stringer et al. 2019), and across most studies studies now studying awake**
**behavior in mice. Neurons throughout cortex, and in particular deeper layers**

**correlate with locomotion state and running speed (Niell & Stryker 2010, Saleem**
**et al. 2013, Steinmetz et al. 2019, Musall et al. 2019, Vinck et al. 2015). An**
**alternative interpretation of the signal of deep TeA layers is thus that these neurons**
**correlate with running as any other part of mouse cortex (Fig. 7d supports this –**
**nonChR2 cell modulation of TeA activity when running) and rather than motor**
**command signals, that their outputs reflect the threat-relevance of sensory stimuli**
**to dPAG. Perhaps the authors can discuss to what extent the data support a direct**
**role in motor initiation versus alternative interpretations that consider the context**
**of the behavior, such as that TeA provides the contextual signals necessary to signal**
**a situation in which escape behavior is necessary.**

We sincerely appreciate your insightful comments regarding the distinction between
running and escape behaviour. After careful consideration, we would like to clarify our
conceptual framework. “Escape behaviour” represents a complete process from
sensation to movement, whereas “running” is a specific manifestation of this process.
Our findings demonstrate that TeA_{dPAG} neurons and $TeAdPAG$ neurons directly control
running behaviours, as evidenced by both the activation patterns of these neurons (Fig.
7) and their correlations with running speed (Chen et al.). While these neurons receive
threat-related sensory inputs from $_{sens}TeA$ neurons, we classify TeA_{dPAG} neurons as
motor command neurons on the basis of two key observations: (1) they directly control
running behaviours, and (2) their firing patterns encode running speed rather than
sensory information. The speed-dependent firing of both TeA_{dPAG} and $TeAdPAG$ neurons

(Chen et al.) further supports their role in motor execution rather than sensory
processing. Therefore, we maintain that TeA_{dPAG} neurons function as the final motor
command executors within this circuit.

**To further clarify, I suggest:**

**-Providing additional context about the adaptive relevance of the observed**
**behavior.**

**-Expanding the methods to include critical details about the running behavior in**
**freely moving animals, such as:**

**• How and where stimuli were presented.**

**• Whether the animals' location influenced their responses.**

**• Onset criteria for running and whether the animals exhibited freezing behavior**
**beforehand.**

We confirm that all the methodological details mentioned have been thoroughly
incorporated into the revised manuscript. Furthermore, each of the specific concerns
raised has been systematically addressed in our previous point-by-point responses. The
current version of the manuscript now provides comprehensive documentation of these
experimental details, ensuring full transparency and reproducibility of our findings.

Regarding the question of whether the latency period might include freezing behaviours,
we provide the following explanation. In this study, using a classic sensory stimulus-
induced escape model, we did not observe typical freezing behaviours (such as

complete immobility, cessation of locomotion, immobility accompanied by shaking, or
halting during an intermediate action state) either before or during the animal's flight
behaviours. In a previous study referenced in our manuscript that employs similar
experimental models¹, the authors explicitly stated that no freezing was observed in the
latency period during sound-induced escape behaviours; instead, escape was triggered
directly. Furthermore, our experimental video recordings revealed that mice typically
exhibit brief vigilance (e.g., head turning, ear twitching) after sensory stimulation,
followed by the initiation of running, rather than entering a motionless state.

Freezing behaviour is defined as a “learned, fear-related defensive behaviour” that may
manifest as immobility, cessation of locomotion, or immobility accompanied by
shaking. However, the brief pauses observed during the latency period in our study are
more consistent with vigilance or a state of decision-making regarding movement rather
than the active motor suppression characteristic of freezing.

We observed a significant time difference between neuronal firing in the TeA and dPAG
and the onset of running behaviours (as shown in Figs. 1-2 of Chen et al.), which aligns
with the theory that the activity of “running units” alone is insufficient to directly trigger
the cessation of behaviours. Therefore, brief pauses during the latency period are more
likely to reflect neural signal integration or motor preparation rather than active
freezing-induced suppression.

**2. Latency and Circuit Dynamics**

**-The authors present a circuit in which sensory signals coming from respective**
**cortical sensory systems project to TeA. After an intralaminar projection, the**
**outputs of TeA project to dPAG responsible for the running behavior. The latencies**
**of the proposed pathway, tens of milliseconds, is very different from the timescale**
**at which the behavior is elicited (Fig. 1e, 1j). Why does it take 2 seconds to start to**
**run? Are the animals freezing first? Coming back to the first point, is this escape**
**behavior? Methods details are lacking here. Line 1291: What is the smoothing**
**kernel size? How was latency to start of running estimated? How does the**
**smoothing kernel affect latency estimates?**

The latencies of sensory-induced behaviours are almost universally on the order of
seconds¹⁻³, which contrasts sharply with the much faster response times of sensory
neurons. Sensory-driven behaviours require at least three processing stages: sensory
detection, sensory-to-behaviour encoding, and motor command execution. However,
the identities of the neurons that mediate sensory-to-behaviour encoding and motor
execution, as well as their connectivity, remain unclear. To our knowledge, Evans et al.²
(Fig. 1c, right panel; 1b, bottom) demonstrated that the synapses between the superior
colliculus (SC) and periaqueductal grey (PAG) driving running behaviours are slow-
acting and require prolonged latencies (Fig. 2d and g). Our study was specifically
designed to investigate how sensory-induced running behaviours emerge through these
stages (sensory detection-sensory to behaviour encoding-motor execution). However,

neither our work nor that of Evans et al.⁹ has resolved the mechanistic basis for these
long latencies.

Notably, our second manuscript (Chen et al., Fig. 3j and k) reveals that the latency of
sensory stimulus-induced flight behaviour in animals is not stable but instead decreases
gradually as running speed increases. Under high-speed running conditions, the latency
remains very low. Additionally, as shown in Figs. 1 and 2 of Chen et al., both TeA and
dPAG running-related neurons show significant time differences between neuronal
firing and the running behaviours of the animals. Integrating our findings from the
companion manuscript (Chen et al., Fig. 2h) with prior literature⁴, we note that dPAG
neurons exhibit an average delay of ~ 0.8 s between firing onset and running initiation.
Thus, sensory-induced defensive flight or, more broadly, dPAG-initiated behaviours
under natural conditions (i.e., without artificial interventions such as high-frequency
optogenetic activation) inherently require prolonged transmission times across these
neural stages.

In the revised manuscript, we have provided a detailed explanation regarding how to
smoothly record latency during running behaviours.

Evans et al. Nature 2018—Figure 1. Escape behaviour during threats of varying intensity. c,

Single trial traces from one mouse escaping to different contrast spots (left) and sound (right).

**Evans et al. Nature 2018—Figure 2. Encoding of threat and escape behaviour in the superior**

**colliculus and periaqueductal gray. d, Average calcium response for active dPAG cells, aligned**

**to escape and sorted by onset (57/138 cells, N=3, 55 trials). g, Same as d for dmSC (177/218 active**

**cells, N=8, 111 trials; mean onset= -1.51±0.17s, P=3.5×10⁻¹² Wilcoxon test comparison with 0s).**

[FIGURE REDACTED]

**Chen et al.—Figure 2 h, Distribution of time differences for dPAG neurons (black) and TeA**

**neurons (gray), originally shown in Fig. 1o. Gaussian fitting curve for dPAG; R² = 0.983, P =**

**1.5×10⁻⁴. Figure 3 j, Running latency versus St. Fre. (every 10 trials, error bars: SDs). k, Running**

**latency versus running speed.**

**-From Figure 1b it appears that the running speed in the freely moving context is**

**not a smoothly increasing function, but rather a small bout in the middle of**

**stimulus presentation. Why do the animals run only the middle second? Then the**
**dynamics in the head fixed scenario, Fig. 1h, are different but still have onset**
**latencies of 1 sec or more.**

We sincerely appreciate your intriguing suggestion. We hypothesize that this represents
an adaptive motor pattern for responding to sudden threats; in naturalistic free-field
conditions, escape behaviours triggered by sensory or fear stimuli drive animals to
rapidly reach safety, causing the initiation of running to decouple from the duration of
the stimulus. As the perceived threat diminishes following successful escape, the
running speed correspondingly decreases. To address your second question further, we
would like to highlight a critical distinction between the behavioural paradigms: in the
head-fixed scenario, the mice are physically constrained on a treadmill and thus cannot
rapidly escape from potential threats as they would in free-field conditions. This
physical limitation results in significantly prolonged running durations under the head-
fixed paradigm. Moreover, as demonstrated in Fig. 4 of the study by Chen et al., the
running duration was positively correlated with stimulus duration within certain
parameter values under the head-fixed condition. Additionally, our data in
Supplementary Fig. 4 further illustrate the dependence of the animal's running duration
on the stimulation duration. These differences underscore how physical constraints
fundamentally alter the temporal dynamics of the threat response behaviours between
these experimental models.

[FIGURE REDACTED]

**Chen et al.—Figure 4 | SC_{dPAG} stimulation. h**, Behavioural responses to BL stimulation for
durations of 5, 10, and 15 s and a frequency of 40 Hz in the TeA. Panels 1-4: (1) Movement speed
heatmap, (2) average speed versus time, (3) response duration versus stimulation duration, (4)
statistical comparison of the peak speed for backing away or rebound running behaviours. $F_{(2, 8)} =$
0.212, $P = 0.813$. One-way repeated-measures ANOVA with Bonferroni post hoc correction (error
bars: SDs).

**Extended Data Fig. 4 | Multisensory integration. h**, Example average speed (10 trials) in response
to sounds with different durations (in s). **k**, duration (Freq: $F_{2,12} = 1.485$, $P = 0.265$. Dur: $F_{2,12} =$
197.738, $P < 10^{-6}$. Int: $F_{2,12} = 195.134$, $P < 10^{-6}$.) of running induced by sound stimuli of different
durations (Dur). One-way repeated-measures ANOVA with Bonferroni post hoc correction; $n = 7$
animals.

**-How do the authors account for the difference between the output of TeA (100ms?)**
**and running behavior (1-2 sec?)? Is dPAG integrating over time? Moreover, some**
**neural latencies do not match with the proposed circuit. The latencies to light**
**stimulation for example are not in line with the suggested cortical circuit: VC to**

**TeA, where VC responds at 50-60 ms latencies, but TeA around 400 (Fig. 5h).**

Regarding the temporal discrepancy between neuronal activation and behavioural onset,
we acknowledge that while the TeA responds to sensory signals within milliseconds
(Fig. 4h), the latency for the production of flight behaviours is significantly longer (1–
2 s). This delay likely reflects both the inherent response time of dPAG-mediated motor
initiation and a potential temporal integration process for sensory signals (Fig. 7c and
1126 d). Our data reveal an interesting parallel: by recording neuronal discharge and
1127 activating ChR2⁺-AC TeA SR neurons and ChR2⁺-TeA_{dPAG}, a long-latency monosynaptic
processing from SR neurons to R neurons was observed (approximately 332 ms, as
shown in Fig. 4j, or 343 ms, as shown in Fig. 7j). While we initially considered
alternative polysynaptic pathways, the complete behavioural blockade of TeA_{dPAG}
neurons upon chemogenic inhibition (Fig. 7k-n) confirmed that _{Sens}TeA neurons must
act through this direct pathway to trigger running behaviours. These findings suggest a
slow integration process in which sensory signals gradually accumulate in TeA_{dPAG}
neurons before the latter reach the firing threshold. Although we have not fully
elucidated the underlying mechanism, recent work^{2,5} supports our hypothesis that the
connection between _{Sens}TeA and TeA_{dPAG} neurons in the mouse TeA may involve weak
synapses requiring hundreds of milliseconds of temporal summation for effective signal
transmission.

Regarding your comment on "the difference between the output of TeA (~100ms?) and

running behavior (1-2 sec?)," we would like to clarify that, as shown in Fig. 1 and 2 of
Chen et al., both TeA and dPAG running-related neurons exhibit a significant time
difference between neuronal firing and the animal's running behaviour. This
observation suggests that the neural activity in these regions precedes and drives the
running output, which is consistent with the role of these neurons in sensorimotor
integration during flight responses. The time difference may reflect the time required
for signal propagation through downstream circuits or the integration of sensory inputs
before sustained running behaviours are initiated. Furthermore, in Chen et al. (Fig. 3i),
we further clarified that the TeA-dPAG transmission delay closely matches the latency
observed between TeA and dPAG neuronal activity, as shown in Fig. 2. This delay is
attributable to the monosynaptic connection from the TeA to the dPAG. This finding
reinforces the conclusion that the TeA directly and rapidly drives activity in dPAG
neurons during flight behaviours, with temporal dynamics governed by single-synaptic
transmission rather than polysynaptic circuits.

Figure 4

Figure 7

**Fig. 4 h**, Summary of the stimulus–firing latency for each type of neuron. unSR-SS: 21.76 ± 3.46
 1157 ms, SR-SS: 31.86 ± 13.19 ms, $U = 192.5$, $z = -3.217$, $P = 0.001$, Mann–Whitney U test. unSR-LS:
 414.8 ± 37.13 ms, SR-LS: 568.3 ± 70.11 ms, $t_{23} = -5.456$, $P = 1.5 \times 10^{-5}$, two-sample t test. unSR-
 AP: 37.62 ± 5.71 ms, SR-AP: 56.87 ± 11.24 ms, $t_{55} = -5.715$, $P < 10^{-8}$, two-sample t test. i, Summary
 of the stimulus–running latency for each type of neuron. unSR-SS: 2.51 ± 0.11 s, SR-SS: $2.49 \pm$
 0.18 s, $t_{38} = 0.315$, $P = 0.754$, two-sample t test. unSR-LS: 2.87 ± 0.15 s, SR-LS: 2.92 ± 0.19 s, t_{12}
 $= -0.402$, $P = 0.695$, two-sample t test. unSR-AP: 2.60 ± 0.21 s, SR-AP: 2.52 ± 0.19 s, $t_{32} = 0.774$,
 $P = 0.445$, two-sample t test. j, Summary of the firing–running time difference for each type of
 neuron. unSR-SS: 2.49 ± 0.11 s, unSR-LS: 2.45 ± 0.13 s, unSR-AP: 2.56 ± 0.21 s, SR-SS: $2.45 \pm$

0.18 s, SR-LS: 2.38 ± 0.15 s, SR-AP: 2.47 ± 0.19 s, R: 2.12 ± 0.24 s. $K-W = 47.16$, $P < 10^{-8}$, Kruskal–
Wallis one–way ANOVA with Bonferroni post hoc correction. **Fig. 7 c-d**, Firing rates of $_{AC}TeA$
neurons expressing ChR2 (**c**), TeA_{dPAG} neurons not expressing ChR2 (**d**) that respond to the speed
of running behaviours evoked by blue LED light stimulation (20 Hz, 5 s, 10 trials). **j**, Time
difference between firing and running events. $F_{2,72} = 45.16$, $P < 10^{-8}$, one-way ANOVA with
Bonferroni post hoc correction. $ChR2^+_{-AC}TeA$: 2.483 ± 0.1661 , $ChR2^-TeA_{dPAG}$: 2.075 ± 0.2017 ,
$ChR2^+TeA_{dPAG}$: 2.059 ± 0.1474 , $n = 12, 7, 8$ cells. **k**, Virus injection and chemogenetic (CNO) and
optogenetic (LED) interventions. $_{AC}TeA$ and $_{S1}TeA$ neurons expressing ChR2-EYFP; TeA_{dPAG}
neurons expressing hM4D(Gi). **l, m**, Example LED light (20 Hz, 5 s, 9 trials)-induced speed raster
(**l**) and running speed trace (**m**) (10 trials) before (Pre), during (CNO), and recovery (Rec) from the
effect of CNO i.p. injections **n**, Induced peak speed for the Pre, CNO, and Rec periods. Pre: 6.149
± 1.179 cm/s, CNO: 0.9806 ± 0.6357 cm/s, Rec: 6.27 ± 1.366 cm/s, $F_{2,12} = 142.444$, $P < 10^{-6}$, one-
way repeated-measures ANOVA with Bonferroni post hoc correction. $n = 7$ animals.

[FIGURE REDACTED]

**Chen et al.—Figure 2 h**, Distribution of time differences for dPAG neurons (black) and TeA
neurons (gray), originally shown in Fig. 1o. Gaussian fitting curve for dPAG; $R^2 = 0.983$, $P =$
1.5×10^{-4} . **Figure 3 i**, In vivo loose-patch recordings from a TeA CaMKII α neuron expressing ChR2
($ChR2^+$, left top) (**a**) and a dPAG neuron not expressing ChR2 ($ChR2^-$) ($_{TeA\ fibre}dPAG$) (left bottom)

(error bars: SDs) (b). The light stimulation pulse duration was 10 ms (scale bar, 100 ms, 50 pA).

Right, the latency between the onset of photoactivation and the generation of neuronal action
potentials.

**I suggest therefore greater clarity on:**

• **Specify how running onset latency was measured (e.g., smoothing kernel size,**
**its effect on estimates).**

• **Address why the running speed in Fig. 1b peaks during the middle second of**
**stimulus presentation.**

• **Reconcile neural response latencies (e.g., VC to TeA at 50-60 ms vs. 400 ms for**
**TeA, Fig. 5h) with the behavioral timescale. If the authors maintain that the**
**pathways depends on weak connections that need 100s of milliseconds to**
**accumulate, they need to argue how this system is robust to noise, evolving**
**temporal dynamics of the brain and changing behavioral objectives.**

We confirm that the specific questions you have raised above have been thoroughly
addressed in our preceding responses. We sincerely appreciate your insightful queries,
which have significantly contributed to improving the clarity and rigor of our
manuscript.

**Minor comments**

- I was confused by the usage of multimodal stimuli in the manuscript. The authors
also use multisensory and multimodal to refer to effects and situations in which
different sensory modalities alone can produce similar effects (e.g. line 200). As the
manuscript also involves more classical multisensory experiments with
simultaneous sensory stimulation, this can be confusing. My suggestion is to
reserve multimodal/multisensory for bimodal or trimodal stimulation. I don't see
what is lost if the term multisensory would be reserved for experiments involving
simultaneous sensory stimulation. E.g. line 218: Multisensory inputs converge to
TeA Sensory inputs converge to TeA.

We sincerely appreciate your valuable suggestion and fully agree with your assessment.
Upon re-evaluation, we recognize that the term "multisensory stimulation" should only
be applied to the experiments presented in Extended Fig. 4a-f, where combinations of
sensory modalities were systematically tested. While we employed three distinct
sensory stimuli (sound, light, and air puffs) throughout the study, these stimuli were
applied in isolation across separate experiments rather than as simultaneous
multisensory inputs. To improve the precision of the concepts described in the text, we
have comprehensively reviewed all the multisensory-related descriptions in the
manuscript and have made corresponding revisions to clarify this distinction.

**Extended Data Fig. 4 | Multisensory integration. a-f**, Paradigm of the running model (a) with
 subthreshold or suprathreshold unisensory (b, sound: 30 dB, 12 k; light: 1 lux and air puff: 0.4 L/min)
 or multisensory stimuli in different combinations for inducing running shown as rasters (c) and
 running speed traces (d, 5 trials) and corresponding plots of the probability of running (e, $F_{2,12} =$
 176.448, $P < 10^{-6}$) and peak speed (f, $F_{2,12} = 35.425$, $P = 9 \times 10^{-6}$). One-way repeated-measures
 ANOVA with Bonferroni post hoc correction. $n = 7$ animals.

- There are several conclusions in the results sections that are not (yet) supported
 by the data. I strongly suggest that the authors adjust their claims to precisely
 match the data shown. For example:

- Line 318: “These results indicate...” It has not been shown that the multimodal
 integration (here I am specifically referring to the synergistic interactions between
 the modalities) that induces running behavior is taking place in TeA. Only that
 pathways from early sensory cortex to TeA are involved in eliciting running
 behavior, and (Fig 4) that multimodal stimuli can interact to produce running
 behavior. The circuits supporting the integration of different sensory modalities
 that underlie the change in elicited behavior could still be elsewhere.

Thank you for your valuable suggestion, which we fully agree with. We have revised
the relevant statement in the main text to “*These results indicate that multisensory*
*inputs contribute to the overall modulation of running behaviours.*” [Results (line 458-
459)] This change ensures that the text is precisely aligned with the experimental design
and terminology, as only the experiments in Extended Fig. 4a-f explicitly involved
concurrent multisensory stimulation.

- **Line 338: “These experiments indicate...” This statement is problematic. First it**
**makes a claim about the whole research field of multimodal sensory integration,**
**whereas (1) no multimodal stimuli were tested, only sound, and (2) one limited**
**behavior was tested. Second, the statement generalizes to other modalities not**
**tested.**

We sincerely appreciate your insightful suggestion. In response, we have revised the
relevant section in the main text to state the following: “*The results of these experiments*
*indicated that auditory stimulus integration depends more on the intensity and duration*
*of the stimulus rather than on the specific (frequency) characteristics.*” [Results (line
479-482)]

- **Line 533: It believe it was not specifically shown that this weak connection**
**requires an extended time to sufficiently excite downstream neurons (perhaps**

**move to discussion of proposed circuit).**

Thank you for bringing this point to our attention. As you correctly noted, we currently
lack direct experimental evidence to conclusively establish that the connection between
*sensTeA* neurons (SR-neurons) and *TeAdPAG* neurons (R-neurons) operates through weak
synaptic connections requiring prolonged temporal integration. In light of this
limitation, we have moved the corresponding text to the Discussion section, where it is
presented as a plausible hypothesis supported by our current findings and data from the
literature⁹ (Chen et al.).

**- Line 406-413: formulate as hypothesis. Only part of the claims here is shown.**

We confirm that the requested revisions have been incorporated into the main text “*On*
*the basis of these results, we hypothesized that at least three types of neurons are*
*involved in information processing from sensation to running behaviour. unSR-neurons*
*respond to sensory stimuli and are localized upstream of SR-neurons, which translate*
*sensory information into a running signal and relay this signal to R-neurons, which*
*then are responsible for driving running behaviours. R-neurons might be TeAdPAG*
*neurons, whereas SR-neurons might be those that receive projections from input sources,*
*hereafter referred to as *sensTeA*. Both types of neurons are located in L5, but they play*
*different roles in the sensory-to-motor transformation process.” [Results (line 347-354)]*

All modifications align with the suggestions provided and have been carefully

implemented to improve clarity and precision.

**- Line 376: Biologists love categorization, but in this scenario, it is unclear whether**
**it is justified and what is gained. Do the authors believe the categorization into 14**
**neuron types is biologically important and future research should focus on the**
**differences between subtypes? We know from earlier work (Raposo et al. 2014)**
**that neurons in higher order mouse cortex show mixed selectivity that defy**
**categorization. It is likely that the reported sensory subtypes come from**
**continuous distributions. It is hard to judge, the Methods lack any description of**
**how significant responsiveness was calculated, which statistics were used. If**
**similar to running analyses the standard deviation of the baseline was used, which**
**time frames served as baseline? Moreover, how was running and sensory**
**responsiveness dissociated? (If sensory stimulation causes the mice to run, then to**
**what does the neuron respond?). The presented single neurons are not convincing**
**and highlight the problem. For example, the neuron in Ext. Data Fig 4c, is it**
**responding to light stimulation or running? And the neuron in Ext. Data Fig 4d,**
**responding to air puffs, or increasing its firing rate upon running? Misinterpreting**
**behavioral state related signals as and dissociating them from sensory signals has**
**recently been an important topic in (multi)sensory research (Zagha et al. 2022,**
**Bimbard et al. 2024, Oude Lohuis et al. 2024) Supporting the statement that there**
**are meaningful categories of multisensory responsive neurons would thus require**
**showing distributions of a metric of sensory responsiveness + low-latency sensory-**

evoked responses that are independent of behavioral state.

Thank you for bringing this important point to our attention. We fully acknowledge that

our initial classification of neurons into 14 subtypes on the basis solely of their

responses to different sensory modalities was an overly simplistic approach. To address

this limitation, we have revised our classification system throughout the manuscript

(see updated Fig. 4e-g). Neurons are now divided into two primary categories: R-

neurons (nonsensory-responsive, running-related) and S-neurons (sensory-responsive).

S-neurons are further distinguish into unSR-neurons (sensory-responsive but running-

unrelated) and SR-neurons (sensory-responsive and running-related).

**Fig. 4 | Sensation-related and running-related neurons in the TeA. e-g,** Summary of the

correlation coefficients between the firing rate and running speed (**e**, $F_{5,286} = 971.4$, $P < 10^{-8}$), depth

location (**f**, $F_{2,101} = 18.588$, $P < 10^{-8}$, unSR vs. SR: $P < 10^{-8}$; SR vs. R: $P = 0.465$), and peak spike

rate (**g**, $F_{2,186} = 51.552$, $P < 10^{-8}$, unSR vs. SR: $P < 10^{-8}$; SR vs. R: $P = 1.1 \times 10^{-5}$), one-way ANOVA

with Bonferroni post hoc correction.

**Further notes:**

**Line 64: two sensory information, information has no plural. Similarly, line 221,**
**various information.**

Due to the revision of the article, the two errors you mentioned have been deleted.

**Line 158: retrospectively -> retrogradely?**

We confirm that the requested modifications have been incorporated into the
manuscript. Please refer to the revised text “*To further confirm the projections of the*
*TeA to the dPAG, we injected AAVretro-hSyn-mCherry into the dPAG, which retrograde*
*labelled TeA cells specifically in layer 5a (L5a) (Fig. 2b).*” [Results (line 185-188)]

**Fig. 1k, inset of waveform missing for Mus+? 7 TeA cells in 3 animals. Why 7**
**animals for muscimol, but only 3 animals have cells recorded? Some more clarity**
**in the methods about how many cells were recorded across animals and whether**
**there were exclusion criteria would help interpretability.**

The Mus+ waveform diagrams have been incorporated into the main text. Additionally,
the erroneous statement regarding “*Statistics for 7 TeA cells (at depths of 250-800 μ m)*
*in three animals*” has been corrected. Please refer to the revised text “*Statistics for 7*
*TeA cells (at depths of 250–800 μ m) in 7 animals.*” [Main figure legends

**(line 1448-1449)]**

**From line 192: consistency, why PAG and not dPAG?**

We confirm that the experimental intervention specifically targeted the dPAG region
and have made the corresponding changes to the text. Please refer to the updated
description “*Specific inhibition of the TeA-dPAG circuit blocks running behaviours.*”

**[Results (line 219)]**

**Line 281, ‘distinguish’ is a verb, not a noun**

We confirm that the requested modifications have been incorporated into the
manuscript. Please refer to the revised text “*Although TeA neurons seemed to receive*
*multiple sensory inputs, including auditory (AuI, AuD, AuV, MGB), visual (V1, V2L,*
*DLG), and somatosensory air puffs (S1) inputs, without distinction, we still believe that*
*different types of sensory information in the TeA exert varying degrees of influence on*
*driving running behaviours.*” **[Results (line 419-423)]**

**Line 290, an article is missing ‘AC-TeA projection loop’**

To clarify, the experimental intervention specifically targeted TeA neurons within the
AC-TeA circuit. We have revised the manuscript to reflect a more precise description,
now stating: “*ACTeA neurons were suppressed.*” **[Results (line 428)]**

**Figure 3, legend missing for panel f**

In the revised manuscript, the original Fig. 3f-h has been relocated to Supplementary
Fig. 3e-g, with corresponding changes to the figure legends. Specifically, the original
legend for Fig. 3f (now Supplementary Fig. 3e) has been edited accordingly to
correspond with the new figure organization. All the panel labels and methodological
descriptions in the supplementary figure legends have been rigorously revised to ensure
consistency with the new figures themselves.

**Fig. 3d: how anterograde? Or retrograde?**

In the revised manuscript, we clarify that the AAV2/1-Cre and AAV2/1-Flp vectors in
Figure 3d are anterograde transsynaptic viral vectors. Specifically, AAV2/1-Cre and
AAV2/1-Flp (anterograde transsynaptic tracers) were coinjected into distinct cortical
and subcortical regions (AC, S1, VC, MGB, DLG) in different mice. Concurrently,
AAV2/9-DIO and AAV2/9-fDIO were injected into the TeA. Leveraging the Cre/DIO
and Flp/fDIO orthogonal recombination systems, this strategy enables specific labelling
of TeA neuron populations receiving anterograde projections from distinct sensory-
related brain regions. The revised methodology now explicitly emphasizes the
anterograde tracing capability of these viral vectors and their combined use for the
pathway-specific investigation of the circuitry of the TeA.

**Fig 3j: The resolution is too poor to judge the overlap in expression. Make sure the**
**resolution is sufficiently high and/or include a closeup.**

We confirm that the figures have been updated with higher-resolution images of the
specified regions to improve the visual clarity and detail.

**Fig. 5 | Diverse sensory input characteristics of the TeA. e, TeA neurons receiving multisensory**
**inputs.**

**Line 324: ‘Even within a single modality’... this is hardly a surprising statement**
**with rich worlds of sensory signals in each modality.**

We fully agree with your perspective and have revised the relevant descriptions in the
main text accordingly. For details, please refer to the revised text “*The related stimuli*
*can also vary in terms of characteristics such as type, intensity, and duration within a*
*single sensation.*” [Results (line 463-464)]

**Line 357: Specify for the lazy reader how this stimulation intensity compares to**
**earlier experiments and why this intensity was used.**

The present study employed attenuated sensory stimulation paradigms (sound, light,
and air-puff stimuli) during in vivo electrophysiological recordings in awake, behaving
mice to investigate how TeA neurons encode sensory processing and running. By
systematically reducing the stimulus intensity from that in prior protocols, we achieved
a critical balance: while such stimuli remained effective in eliciting movement, they
resulted in reduced success rates and lower locomotor velocities. This approach allowed
1423 us to address two critical aspects: 1. Sensory-evoked neuronal responses under
1424 behaviourally relevant conditions. 2. Running-associated firing patterns during low-
1425 intensity running. The attenuated stimulus paradigm facilitated simultaneous
acquisition of TeA neuronal activity during running over long periods, allowing the
precise dissection of sensory-induced versus motor-related neural dynamics. This
methodological refinement significantly improved our ability to classify neuronal
subtypes (e.g., S-neurons versus R-neurons) on the basis of their dual modulation by
sensory inputs and behavioural outputs.

**Line 363: ‘Connections’, poor word usage as it might be interpreted anatomically**
**and can be described sticking closer to the results: ‘whose firing rate correlated**
**with running speed’**

We acknowledge that the term “connections” might have been ambiguous in our
original description. We should clarify that not all recorded neuronal activities can be

characterized as running-related, as we identified a subset of neurons in the TeA (unSR-
neurons) whose firing patterns were not correlated with any quantitative aspects of the
running behaviours. In response to your concern, we have revised the relevant section
in the main text to state the following: “*We recorded action potentials (APs) from a total*
*of 104 single neurons and recorded the generation of flight behaviours in the mice*
*during neuronal firing.*” This change emphasizes the observational nature of our data
while avoiding overinterpretation of causal relationships. **[Results (line 294-296)]**

**Line 401: S-neuron, here and below I suggest to use plural (S-neurons) or S-neuron**
**type.**

We confirm that all the suggested revisions have been thoroughly implemented
throughout the manuscript following a comprehensive review.

*Line 411: ‘correspond to’ -> activate?*

Thank you for bringing this important clarification to our attention. We have clarified
this idea by revising the text to the following: “*R-neurons might be TeA_{dPAG} neurons,*”
**[Results (line 351)]** This terminology reflects the functional identity of these neurons
as motor command neurons, defined by their direct control of running behaviour and
speed-associated firing patterns, as demonstrated in our experiments. We have removed
the ambiguous phrase "correspond to" and revised all related descriptions to ensure that

the text regarding this concept is suitably precise.

**Line 441: the virus was retrograde, but was not injected retrograde.**

We confirm that the requested modifications have been incorporated into the
manuscript. Please refer to the revised text “*Concurrently, we injected AAVretro-CMV-*
*WGA [wheat germ agglutinin]-Cre into the dPAG.*” [Results (line 516-517)]

**Line 477. I confused the results of ChR2- cells in the experimental groups with the**
**control animals. Perhaps the authors can include additional phrasing that help**
**distinguish the behavioral results of the optogenetic control cohort with the**
**distinct analysis of ChR2 pos and neg cells.**

We apologize for any confusion caused by our abbreviated nomenclature and have now
provided more thorough explanations of the neuronal designations in the revised text:
ChR2⁺-_{AC}TeA neurons refer to TeA neurons that receive anterograde projections from
the auditory cortex (AC) in the experimental paradigm depicted in Fig. 7a and express
ChR2. ChR2⁻-TeA_{dPAG} neurons denote the downstream targets of ChR2⁺-_{AC}TeA
neurons in the experimental paradigm illustrated in Fig. 7a. As these neurons do not
receive direct anterograde inputs from the AC, they do not express ChR2. ChR2⁺-
TeA_{dPAG} neurons, depicted in the experimental paradigm shown in Fig. 7b, represent
TeA_{dPAG} projection neurons and express ChR2.

As shown in Fig. 7g, optogenetic stimulation in control animals (lacking ChR2
expression) failed to evoke running behaviours. Consequently, we did not record
neuronal activity from ChR2-negative cells in control animals during optical
stimulation, as these cells were functionally irrelevant to the observed behavioural
paradigm.

**Fig. 7 | Firing and running evoked by activation of $_{AC}TeA$ and TeA_{dPAG} cells. a, b,** Virus injection
and LED light delivery sites. **g,** Running speed ($F_{2,15} = 96.822$, $P < 10^{-8}$, $n = 6$ animals/group). One-
way ANOVA with Bonferroni post hoc correction.

**Fig 6h: the presence of at least one merged neuron can be observed, but a**
**quantification is missing, similar to earlier in the manuscript (Fig 3k).**

Thank you for your valuable suggestion. We confirm that the requested content has
been incorporated into the main figures. For details, please refer to Fig. 6h.

**Figure 6 | *SensTeA* and *TeAdPAG* neurons. h**, Left, images showing *SensTeA* neurons (EYFP),
 *TeAdPAG* neurons and neurons labelled in the upper layer (mCherry). Right, percentage of *TeA*
 neurons with overlapping labelled inputs from different sources. n = 9, 9, and 9 slices from 3, 3, and
 3 animals, respectively.

**Line 564: adjust word order.**

Due to the revision of the article, the sentence you mentioned have been deleted.

**Line 578: What are the differences between the *TeA* circuit and SC circuit. Do they**
 **overlap, play a role in different circumstances? How come *TeA* inhibition can block**
 **the running behavior with intact SC circuit?**

These specific questions regarding neural circuit mechanisms are systematically
 addressed in our companion manuscript (Chen et al., Figs. 4-6). Our findings revealed
 that dPAG neurons receiving projections from the *TeA* (*TeAdPAG*) and superior
 colliculus (*scdPAG*) constitute two largely nonoverlapping neuronal populations (Chen
 et al., Fig. 4a-c). Optogenetic activation of the SC-dPAG circuit induces backwards
 movement and stopping behaviours during stimulation, followed by rebound running

upon cessation, whereas TeA-dPAG activation elicits forwards running (Chen et al., Fig.
5g-h). Mechanistically, we demonstrated that *scdPAG* CaMKII neurons mediate
backwards running, whereas *scdPAG* SOM neurons transiently inhibit *TeAdPAG*
CaMKII neurons during activation (causing stopping behaviours) and disinhibit them
postactivation (triggering rebound running). Critically, inhibition of the TeA-dPAG
circuit does not impair SC-dPAG-mediated behaviours, confirming the functional
independence of these parallel pathways.

[FIGURE REDACTED]

**Chen et al.—Figure 4 a-c**, Virus injection protocol (**a**), *TeAdPAG* (green) and *scdPAG* neuronal
labelling (red) (**b**), and statistics of merged cells (yellow) (**c**). **Figure 5 g**, Behavioural responses for
**f**, respectively, plotted like **b. h**, Statistics for **f**, respectively, plotted like **c**. Rebound running ($F_{(2, 12)} = 112.013, P < 0.0001$) and backing away ($F_{(2, 12)} = 1.780, P = 0.210$). Control group rebound
running ($F_{(2, 12)} = 2.547, P = 0.120$) and backing away ($F_{(2, 12)} = 0.476, P = 0.633$).

**Line 591: ‘involved in’... Involved in what?**

We appreciate your feedback and apologize for the lack of clarity. The relevant text
has been revised as follows: “*This study represents the first observation of these three*
*distinct types of neurons, providing new insights into and methodologies for future*
*motor and behaviour research.*” **[Discussion (line 706-707)]**

**Line 586-592: I find the positioning of this study in the literature a bit off. In**
**comparison to the cited references, this study also epitomizes on circuit mapping**
**techniques to discover pathways and cell types involved in specific sensorimotor**
**transformations. No proper cellular mechanisms are described. Moreover, the**
**phrasing ‘special technics’ (techniques) suggest a newly introduced technique**
**whereas the study uses commonly available tools and techniques, albeit a**
**comparably sophisticated level of circuit investigation.**

We have revised the text to state the following: “*Previous studies have focused mainly*
*on the role of neural circuits, wherein different functions are typically performed by*
*distinct neural nuclei, forming pathways. However, whether a single neural nucleus in*
*the mouse brain can directly convert sensory input into motor output, particularly the*
*distributions of sensory neurons, sensory–motor decision neurons, and motor*
*command neurons at the cellular level, remains unknown.*” **[Discussion (line 677-**
**682)]**

**In comparison to the cited references, this study also epitomizes on circuit**

**mapping techniques to discover pathways and cell types involved in specific**
**sensorimotor transformations.**

We confirm that the term “*special techniques*” has been removed from the main text to
ensure precise terminology and alignment with standard scientific nomenclature.

**Line 881: degrees -> percentage?**

We confirm that the requested modifications have been incorporated into the
manuscript. Please refer to the revised text “*Numbers: percentage reduction in the peak*
*speed and running probability for CNO+ animals relative to CNO- animals.*” [**Main**
**figure legends (line 1561-1562)**]

**Methods:**

**Line 1000: a retro-pseudotyped virus was used for anterograde tracing?**

We have moved text describing the viral construct AAV2/retro-CMV-WGA-NLS-Cre
(2.6×10^{13} vg/mL, OBiO), which functions as a retrograde tracer, to **line 796** in the
revised manuscript. This revision ensures accurate categorization of the viruses on the
basis of their functional properties and experimental applications.

**Line 1074: What does ‘maintaining a stable state on the recording platform’ mean?**

**Were the mice spontaneously running? What percentage of animals does ‘most**
**animals’ mean and what was done if animals did not ‘maintain a stable state’?**

The term “stable state” refers to the ability of head-fixed mice to continue running on a
rotating disk voluntarily. A small subset of mice (approximately 5% of the cohort)
exhibited persistent motor incoordination or complete inability to run despite training,
likely due to incomplete adaptation to head fixation. These mice were excluded from
the study analyses to ensure behavioural consistency and data validity.

**Line 1289: What does randomized batch processing refer to?**

We have clarified the corresponding text, which now reads as follows: “*The recorded*
*running speeds were pooled and processed in randomized batches after alignment with*
*the pseudorandom sequence of sensory stimuli (designed to induce heterogeneous*
*running behaviours across animals). The batch-processed data were subsequently*
*mapped to their corresponding stimuli and categorized according to the experimental*
*condition.*” Please refer to the updated manuscript. [**Methods (line 1144-1148)**]

**Line 1276: No fiber photometry data is presented in the study.**

The text regarding fibre photometry recording has been removed.

**Line 1296: ‘produced apparent’ -> ‘were’?**

We confirm that the requested modifications have been incorporated into the
manuscript. Please refer to the revised text “*We defined “running behaviour” as*
*movements in which the running speed of the mouse exceeded 0.5 cm/s, and the onset*
*of the running behaviour was defined as the point at which this speed threshold was*
*first exceeded.*” [Methods (line 1157-1160)]

**Line 1129: Pharmacal -> pharmacological? + revise sentence grammar**

We confirm that the requested modifications have been incorporated into the
manuscript. Please refer to the revised text “*For pharmacological manipulation of the*
*GABA receptor agonist (Muscimol-M1523, Sigma, USA),”* [Methods (line 941)]

**Line 1152: 800 mm -> um?**

We confirm that the requested modifications have been incorporated into the
manuscript. Please refer to the revised text “*no hChR2- or eNpHR3.0-expressing neural*
*fibres or structures other than the target structure were present in the light pathway*
*within 800 μ m from the end of the optic fibre.*” [Methods (line 963-965)]

**Line 1156: Were animals excluded based on expression? If so, on what criteria and**
**how many?**

Yes, as previously described, animals for whom the injections were unsuccessful were
excluded from the study. Following each experimental session, we rigorously
confirmed viral expression in all the animals and excluded those for whom the
injections were unsuccessful. While we did not systematically quantify the exact
amount of excluded data, we confirm that these animals represented a small percentage
of the total.

**Line 1383: ‘The ordinates.’ What about the ordinates?**

Due to the revision of the article, the text you mentioned have been deleted.

**Line 1315: What is meant by that the raw data is available in the main text?**

The text has been revised to “*All relevant data are within the paper and its Supporting*
*Information files.*” Please refer to the updated manuscript (**line 1181-1182**) for details.

**Line 1321: Where is the code available?**

We confirm that the relevant code will be made publicly available on GitHub upon

publication, with the repository link to be inserted here: [GitHub URL:
[https://github.com/LiHe0606?tab=repositories.](https://github.com/LiHe0606?tab=repositories)]. Please refer to the updated manuscript
**(lines 1186).**

**References:**

Xiong, X. R. *et al.* Auditory cortex controls sound-driven innate defense
behaviour through corticofugal projections to inferior colliculus. *Nat Commun*
**6**, 7224,(2015).

Evans, D. A. *et al.* A synaptic threshold mechanism for computing escape
decisions. *Nature* **558**, 590-594,(2018).

Wang, H. *et al.* Direct auditory cortical input to the lateral periaqueductal gray
controls sound-driven defensive behavior. *PLoS Biol* **17**, e3000417,(2019).

Deng, H., Xiao, X. & Wang, Z. Periaqueductal Gray Neuronal Activities
Underlie Different Aspects of Defensive Behaviors. *J Neurosci* **36**, 7580-
7588,(2016).

Shapson-Coe, A. *et al.* A petavoxel fragment of human cerebral cortex
reconstructed at nanoscale resolution. *Science* **384**, eadk4858,(2024).

**J.C., H.L. et al. “Neural circuits and speed coding of running, rebound running**
**and backing away behaviours” . Submitted as a companion paper.**

Dear Reviewer,

The line numbers referenced in our point-by-point responses correspond to the
"Revised manuscript without track changes-He Li.pdf" file.

**Reviewer #1-Li (Remarks to the Author):**

**This revised version of the manuscript has been greatly improved in terms of**
**readability and clarity, although the wording of some sentences remains a little**
**odd. The current version presents a clear and elegant description of a cortical**
**circuit transforming sensory input into the decision to initiate running (escape).**
**Unfortunately, despite this improvement in the form, many of my concerns about**
**the robustness of the data and analyses remain. In fact, the authors have largely**
**ignored many of my concerns on these points. Another problem is that the**
**"Methods" section still lacks essential information or provides incomplete**
**information, making it difficult to truly assess the relevance of the analyses and**
**results.**

In the newly submitted version, we have expanded the Methods section to include
details on neuronal recording, encompassing the number of animals and neurons
utilized. We have also introduced specific criteria for distinguishing sensory neurons
from motor neurons and provided a more comprehensive description of the correlation
analysis between neuronal firing rates and the movement speed of the animals.

**Here are examples of critical issues that have not been carefully addressed:**

**- It is still unclear to me whether the authors have adequately addressed the issue**
**of disentangling the sensory- and movement-related neuronal activity**
**(juxtacellular recordings). I understand that a complex approach like GLM might**
**not be suited for the type of data in this study, with low number of trials but at**
**least the sensory-evoked response should be assessed from purely sensory trials**
**(no evoked running) and the running-related activity should be assessed from**
**spontaneous running not preceded by any sensory stimulus. And the significance**
**of the responses should be assessed for well-defined time-windows and appropriate**
**statistical test. PSTHs aligned to sensory stimulus AND running onset times should**
**be presented for single neurons and groups of neurons (See my Major comments**
**4 and 5 from the previous version).**

First, thank you for your questions. Some of these issues were actually raised in your
initial review, and we sincerely apologize for not fully understanding your points at that
time. In this revised version, we have first clarified the issue of time windows, including
the three distinct types: stimulus–running, stimulus–firing, and firing–running. Your
suggestion is excellent and highly important, and we believe that establishing these
three windows is crucial for subsequent experimental analyses and processing. Detailed
explanations regarding the establishment of these time windows have been added to the

Methods section of the revised manuscript, “*Definitions of the Response Windows for*
*Evoked Behaviours. We defined a response window for each experimental paradigm*
*based on a comparative analysis of pre- and poststimulus running latencies to*
*objectively distinguish sensory-evoked escape behaviour from spontaneous running.*
*We first confirmed that the distribution of running latencies following sensory*
*stimulation significantly differed from the distribution of spontaneous running intervals*
*during the pre-stimulus baseline period using a two-sample Kolmogorov–Smirnov test.*
*This significant difference confirmed that the post-stimulus running events constituted*
*a distinct, time-locked population. Given this clear separation, we defined the response*
*window to capture the specific temporal characteristics of the evoked response. The*
*upper bound of the window was set to the 99th percentile of the post-stimulus-evoked*
*latency distribution. This approach ensures that the window is tailored to the actual*
*temporal profile of the stimulus-locked behaviour, providing a sensitive measure of*
*elicitation efficacy. The spontaneous latency distribution was used as a null model to*
*statistically confirm the specificity of the evoked behaviour within the defined window.*
*We performed a binomial test to confirm that these responses were specifically evoked*
*by the stimulus rather than occurring by chance. For all three experimental models,*
*binomial tests were conducted against the null hypothesis that post-stimulus running*
*occurred at the spontaneous rate. The tests revealed that the number of runs within the*
*defined response windows significantly exceeded the number expected by chance ($P <$*
*10^{-4} in each model), providing strong statistical evidence that running was specifically*
*evoked by the sensory stimulus.*

*Quantification of Response Latency. The onset of each sensory stimulus was aligned to*
*time zero. Spike times from all trials for a given stimulus condition were pooled. Peri-*
*stimulus time histograms (PSTHs) were constructed with a bin width of 10 ms to*
*visualize the temporal dynamics of neuronal firing rates in response to each stimulus.*
*The neuronal response latency was determined for each effective stimulus using a*
*threshold-based method. The mean firing rate (baseline rate) and its standard deviation*
*(SD) were calculated from a 2 s pre-stimulus epoch. The response threshold was defined*
*as the baseline rate plus two times the standard deviation (baseline + 2 × SD). The*
*response latency was identified as the first bin in the post-stimulus period (starting from*
*time zero) where the firing rate in the PSTH exceeded this threshold. Only neurons that*
*showed a significant excitatory response to at least one stimulus type were included in*
*the latency analysis.*

*Quantification of Running-Related Firing Activity. We defined a running epoch to*
*identify neurons whose firing was modulated by spontaneous running. Spontaneous*
*running onsets were aligned to time zero. Peri-event time histograms (PETHs) were*
*constructed for each neuron using a 10-ms bin width centred on the running onset. The*
*baseline firing rate was defined as the average firing rate when the mouse was not*
*running. The standard deviation (SD) of the firing rate during this baseline period was*
*also calculated. The onset of running-related neuronal activity was determined using a*
*threshold-crossing method. The threshold was set as the baseline mean firing rate plus*

*two times its standard deviation ($\text{Baseline} + 2 \times \text{SD}$). The start of the running-related*
*activity window was identified as the first time bin where the firing rate in the PETH*
*exceeded this threshold. The temporal window of running-related activity was then*
*defined as the period from this identified start time until movement onset (time zero).*
*This window represents the period of a significant increase in the firing rate preceding*
*observable running.” [Methods (line 1380-1429)]*

Additionally, you mentioned that peri-stimulus time histograms (PSTHs) aligned to
both sensory stimulus and running onset times should be provided for single neurons
and groups of neurons. In this revision, we have added PSTHs for different types of
neurons in response to sensory stimuli, as well as their running speed trajectories. For
neurons that do not respond to sensory stimuli, we observed their firing patterns by
aligning the peri-event time histogram (PETH) to the onset of running. The construction
of both single-neuron and population PSTHs or PETHs further clarified the
relationships between different types of neurons and sensory or running events.
Through the construction of these PSTHs or PETHs and the cross-correlation function
(CCF) analysis, we are able to distinguish between different types of neurons.

Extended Figure 2

Figure 4

**Extended Data Fig. 2 | S- and R-neurons in the TeA.** a, Responses of the R-neurons to three types
of sensory stimuli: auditory (sound), visual (light), and tactile (air puff). Top panel, Population raster
plot and the corresponding PSTH of firing activity. Bottom panel, Superimposed running speed
curves from all trials (n = 156 trials). The blue bars indicate the stimulus period (duration: 5 s).
**b**, Spontaneous running events for the R-neurons. Top panel, Population raster plot and the
corresponding PETH of firing activity. Bottom panel, Superimposed running speed curves from all
trials (n = 24 trials).

**Fig. 4 | Sensation-related and running-related neurons in the TeA.** h1–2, S neuron responses to
the three sensory stimuli. Top panel, Population raster plot and the corresponding PSTH of firing
activity. Bottom panel, Superimposed running speed curves from all trials (n = 65 trials of **h1**, n =
15 trials of **h2**).

**- I am also still concern by the possibility to achieve good-quality 3D single-cell**
**reconstructions (Figure 6b-c) from high-density labelling obtained from viral**
**expression (Figure 6a). Again, no information provided in the Methods section.**

As you observed, performing 3D reconstruction in regions with a high viral expression
intensity (as in Fig. 6a) presents challenges. When conducting single-cell 3D
reconstructions, the ideal scenario is one in which only a single neuron is present (as

shown in **Fig. a** and **c** below). The next best option is to select areas with relatively
sparse viral labelling (within the white dashed boxes in **Fig. b** and **c**, left panel). When
two or more neurons are located in close proximity, determining the dendritic
arborization trajectories in two-dimensional planes is often difficult. In such cases, we
prioritized the use of 3D reconstructions by rotating the model to roughly assess
neuronal morphology and endeavoured to select neurons whose structures can be
clearly discerned through rotation (as indicated by white arrows in the right panels of
**Fig. b** and **c**). Only after identifying suitable neurons did we proceed with single-cell
reconstructions.

In the newly submitted version, we have added a detailed description in the Methods
section regarding the specific procedures for reconstructing the 3D morphology of
neurons. The specific operational steps have been supplemented in the Methods section.

*“Confocal image stacks were imported into FIJI software. Background noise was*
*reduced using the “Subtract Background” function, and contrast was appropriately*
*enhanced to improve neurite visibility. The soma was identified either in the maximum*
*intensity projection or in the optical slice where it appeared largest and most clearly*
*defined. With the “Freehand selection” tool, the somatic contour was carefully traced,*
*and its cross-sectional area (in μm^2) was measured using the “Measure” command.*
*The “Simple Neurite Tracer” plugin was used for semiautomated 3D reconstruction of*
*dendritic and axonal processes. Tracing began at the soma, and paths were manually*
*followed through the z-stack by placing nodes along the neurites. The plugin*

*interpolates between nodes to create a 3D structure. Tracing continued until all visible*
*neuronal processes were fully reconstructed. Sholl analysis was performed using the*
*“Sholl Analysis” plugin. The plugin calculates the number of intersections between the*
*neurites and each concentric circle, generating a profile of the dendritic density as a*
*function of radial distance from the soma. All measured data, including the somal area*
*and number of Sholl intersections, were exported for further statistical analysis and*
*graphing using the appropriate software (e.g., GraphPad Prism, SPSS).” [Methods*
**(line 1319-1335)]**

**a:** Three-dimensional reconstruction of a single labelled TeA_{Sens} neuron (left panel) and a magnified
view showing its morphological details (right panel). **b:** Three-dimensional reconstruction of
TeA_{Sens} neurons in a region with a high viral labelling density (left panel). The white dashed area
indicates a sparsely labelled region selected for morphological reconstruction. The neuron chosen
for reconstruction is marked with a white arrow (right panel). **c, d:** The same as **a** and **b**, respectively,
but showing a TeA_{dPAG} neuron. Scale bar, 100 μm .

- According to the rebuttal of the other paper (NCOMMS-24-78436A) the
anatomical location of the recorded neurons was achieved either through single-
cell juxtacellular staining with biocytin (no example picture provided in this
paper!) or by injection of a dye at the same location after the recording. There is
no mention of that in the Method section!

In the newly submitted version, we have added a detailed description in the Methods
section regarding the labelling and staining of recorded neurons using biocytin. Notably,
not all recorded neurons were subjected to neuronal staining. More frequently, we
determined whether the recording site was located within the TeA region by injecting
dye at the same location. These procedures have been thoroughly elaborated in the
current revision. *“In accordance with the methods reported in a previous article¹,
following the establishment of a seal resistance ranging from 0.2 to 1 GΩ, the amplifier
was switched from voltage-clamp to current-clamp mode. A positive direct current (3–
10 nA, 1 Hz) was applied for 20 minutes to iontophoretically deliver biocytin for an
assessment of neuronal morphology. The biocytin tracer was prepared as a 1% (wt/vol)
solution to enable the subsequent visualization of neuronal morphology. A total of 20
mice were used for the intracellular recordings, with an average of 6–10 neurons
recorded per mouse.”*

*“We primarily employed biocytin labelling to localize the recorded neurons and obtain
their coordinates during recording; however, the labelling efficiency was suboptimal.*

*Accordingly, for a subset of neurons, an alternative approach was adopted. Following*
*the electrophysiological recordings, glass micropipettes with larger tip diameters than*
*those of standard recording electrodes were filled with an eosin dye solution. This dye*
*was then pressure-ejected (~2 psi) at the recording site. Subsequent frozen sectioning*
*and histological examinations allowed us to confirm whether the recording site was*
*located within the TeA.” [Methods (line 1238-1254)]*

**- There still is not explanation about how ChR2⁺ neurons were identified. There**
**should be clear criteria to assign neurons as ChR2⁺ or ChR2⁻. Also, it is (almost)**
**correct to assume that ChR2⁺ neurons are either AC-TeA or TeA-PAG neurons**
**depending on the labeling procedure but it is absolutely not correct to assign**
**ChR2⁻ neurons to any category!**

The criteria for distinguishing between ChR2⁺ and ChR2⁻ neurons are based on their
response latency and firing rate following optogenetic stimulation. ChR2⁺ neurons,
which express channelrhodopsin, exhibit very short response latencies (within 5 ms)
and significantly higher firing rates upon light stimulation. In contrast, ChR2⁻ neurons
do not express ChR2. If such neurons are synaptically connected to ChR2⁺ neurons,
they may fire in response to the input from directly activated cells, resulting in both a
longer response latency and a lower firing rate than those of ChR2⁺ neurons. These
differences in latency and firing rate can be observed in **Fig. 7f** and **7i**, respectively.

*“The firing rates of AC-TeA-ChR2⁺ and TeA_{dPAG}-ChR2⁺ neurons were significantly*

greater than those of *TeA_{dPAG}-ChR2⁻* neurons (Fig. 7f).” [Results (line 643-645)]

“The average latency of the firing of *ACTeA-ChR2⁺* and *TeA_{dPAG}-ChR2⁺* neurons was

4.36 ± 0.574 ms and 4.157 ± 0.476 ms, respectively, indicating direct effects of blue

light stimulation on *ChR2⁺* neurons (Fig. 7i).” [Results (line 649-652)]

Regarding the terminology, we fully acknowledge the reviewer's concern. We realize

that referring to these populations simply as “*ChR2⁺* neurons” or “*ChR2⁻* neurons”

without specifying their anatomical identity was imprecise. Instead, neurons should be

accurately described based on their circuit-specific definition, for example, as *ACTeA-*

*ChR2⁺* neurons or *TeA_{dPAG}-ChR2⁺* neurons, depending on the viral strategy used. We

thank the reviewer for highlighting this issue, and we have revised the entire manuscript

accordingly to ensure consistent and accurate descriptions.

**Fig. 7 | Firing and running evoked by the activation of *ACTeA* and *TeA_{dPAG}* neurons. f, Firing**

**rate ($F_{2,24} = 74.87$, $P < 10^{-6}$; $n = 12$, 7 , and 8 cells). One-way ANOVA with the Tukey's post hoc**

**correction. i, Firing latency. $F_{2,24} = 468.9227$, $P < 10^{-6}$; one-way ANOVA with the Dunntee T3 post**

**hoc correction. $n = 12$, 7 , and 8 cells.**

**Other less critical issues:**

- **Regarding response latencies, in their rebuttal the authors claim that “The**
**latencies of sensory-induced behaviours are almost universally on the order of**
**seconds[REFs 1-3]”. This is absolutely wrong, in many goal-directed behaviors in**
**head-fixed mice, response latencies are in the range of 100-300 ms! Also, the study**
**by Evans et al. (Nature 2018 <https://doi.org/10.1038/s41586-018-0244-6> cited by**
**the authors, REF 2) reports response latencies of 0.5 s to sound or high-contrast**
**visual stimulus (see figure 1). Considering the high-response probability in the**
**current study (Figure 1k), one could expect much shorter latencies. Which**
**questions the way response probability, peak speed and latency were computed:**
**ideally, one should define a response window during which one may consider that**
**the running onset is indeed initiated in response to the sensory stimulus and not**
**just a spontaneous running event (see my Major comments 1 and 3 from the**
**previous version).**

First, we apologize for our previous incorrect statement. We would like to further
explain the calculation of the response latency. As you suggested, delays in the response
of many goal-directed behaviours of head-fixed mice typically fall within the range of
100–300 ms. However, importantly, these behaviours are diverse and include but are
not limited to whisking and/or facial/jaw/paw movements. Some of these actions do not
cause significant body displacement or postural changes and often involve rapid
subcortical pathways (e.g., the superior colliculus and brainstem) that bypass higher

levels of cognitive processing. In this study, we defined the onset of escape running as
follows: “running behaviour” was defined as movements in which the running speed of
the mouse exceeded 0.5 cm/s, and the onset of running was identified as the first time
point at which this speed threshold was reached. For sensory-evoked flight, the onset
was defined as the moment when the speed exceeded the baseline level (mean speed
during the 5 s prestimulus period) by 2 standard deviations (SDs). We believe that
differences in how movement onset is defined may contribute to the longer latencies
observed in our study than in conventional measurements. Whole-body behaviours such
as escape running require the coordinated engagement of multiple muscle groups and
often involve motivation, decision-making, and expectation. These complex processes
engage more elaborate neural circuits—from cortical regions (e.g., the auditory cortex
and prefrontal cortex) to subcortical motor centres (e.g., the basal ganglia)—which are
inherently more time-consuming.

You also referenced the study by Evans et al. (Nature
2018, <https://doi.org/10.1038/s41586-018-0244-6>, Ref. 2 in our manuscript), which
reported response latencies of approximately 0.5 seconds to auditory or high-contrast
visual stimuli (see their Fig. 1). Given the high response probability in our study (**Fig.**
**1n**), one might expect shorter latencies. However, as Evans et al. noted, reducing the
stimulus contrast gradually increased the reaction time and decreased the escape
probability, whereas escape vigour (speed) increased with contrast (their Fig. 1f),
indicating that response latency is also influenced by the escape speed.

Notably, our second manuscript (Chen et al., **Fig. 3j and k**) revealed that the latency of
sensory-evoked flight behaviour is not fixed but decreases progressively as running
speed increases. During high-speed running, latencies remain very short. A study
published in 2019¹ using a paradigm similar to ours also reported prolonged response
latencies (exceeding 1 s) even under high response probability conditions, which we
attributed to the lower running speeds of the animals in that study.

We agree that establishing a stimulus–response window is valuable. In accordance with
your suggestions, we not only defined a stimulus–running response window but also
introduced a stimulus–firing window and a firing–running window to better
characterize the neuron types recorded in **Fig. 4**. The stimulus–firing window was
analysed using pure sensory trials (without evoked running), and the firing–running
window was assessed during spontaneous running without any sensory stimulation. We
believe that this approach effectively dissociates neuronal response types and
behavioural states.

[FIGURE REDACTED]

**Chen et al- Figure 3 | Firing and running evoked by optogenetic stimulation of the TeA-dPAG**
**circuit. j, Running latency versus St. Fre. (every 10 trials, error bars: SDs). k, Running latency**

versus running speed.

[FIGURE REDACTED]

**Wang H, et al- Fig 1. Involvement of the ACx and IPAG in noise-evoked defensive behaviors.**

(A) Time line of the behavioral protocol. (B) Schematic showing the sound-evoked escape behavior.

(C) Probability of noise-evoked escape behavior ($t(6) = 6.874, P = 0.0005$, paired t test, $n = 7$ mice).

(D) Schematic showing how running speed is recorded in a head-fixed mouse. (E) Representative

running speed traces. Arrowheads denote initiation of noise. (F) Running speed ($t(6) = 8.282, P =$

0.0002 , paired t test, $n = 7$ mice).

**- From the revised version, I understood that in fact, each sensory stimulus is only**

**presented 10 times, which does not provide many trials to analyze. Was it to avoid**

**habituation and decrease in behavioral response (see my Major comment 2 from**

**the previous version)?**

You also correctly noted that each sensory stimulus was presented only 10 times,

providing limited trials for analysis. We intentionally used a low number of trials to

avoid habituation and reduce behavioural adaptation, which is why we employed

randomized alternations of auditory, visual, and air puff stimuli. We have now added a
detailed explanation of this rationale in the Methods section of the newly submitted
version. *“The three types of sensory stimuli were presented in a pseudorandom order.
Each stimulus was repeated only 10 times to prevent habituation in mice, which could
diminish behavioural responses, while also ensuring the validity of our results.”*

**[Methods (line 1075-1078)]**

**- Importantly, the extracellular recording experiments were conducted with lower-**
**intensity sensory stimuli (to avoid large and fast behavioral responses). And indeed,**
**when looking at example recordings, it appears that the sensory stimuli rarely**
**evoke running and running appears more like spontaneous than sensory-driven**
**events. The behavioral response (running probability and latency) during the**
**recording session should have been quantify and reported. If the sensory stimuli**
**do not evoke running with significantly higher probability that spontaneous**
**running, it does limit the relevance of the extracellular recordings for this study.**

As you noted, the extracellular recordings were conducted under low-intensity sensory
stimulation. Our goal was indeed to avoid eliciting large and rapid behavioural
responses in the animals, as such reactions are unfavourable for stable extracellular
recordings. Additionally, the attenuated stimulus paradigm facilitated the simultaneous
acquisition of TeA neuronal activity during running over extended periods, enabling the
precise dissection of sensory-induced versus motor-related neural dynamics. This

methodological refinement significantly improved our ability to classify neuronal
subtypes (e.g., S-neurons vs. R-neurons) based on their dual modulation by sensory
inputs and behavioural outputs.

Regarding neuronal classification, you previously suggested—and reiterated in this
comment—that establishing temporal windows would help clarify the neuronal types.
We fully agree with this excellent suggestion. As mentioned earlier, we implemented
both a stimulus–firing window and a firing–running window. The stimulus–firing
window was analysed using trials with pure sensory input (without evoked running),
whereas the firing–running window was evaluated during spontaneous running
episodes without any sensory stimulation. We are confident that this approach
effectively dissociates neuronal response profiles and behavioural states.

Your primary concern seems to be that sensory stimuli rarely trigger running, which
appears more spontaneous than evoked. We directly addressed this concern by
quantifying the probability and latency of both sensory-evoked and spontaneous
running. Based on the time window defined in **Fig. 4b**, the probability of sensory-
evoked running was 51.58%. Although this value is lower than the running probability
shown in **Fig. 1n**, it remains significantly higher than the probability of spontaneous
running. Furthermore, as shown in Extended **Fig. 5a–f**, reducing the stimulus intensity
abolished the evoked running behaviour. This result confirms that the behaviour is
stimulus dependent. Notably, in trained, head-fixed animals, the state of nonrunning

typically occurs >70% of the time, indicating a low baseline probability of initiating
 spontaneous running. Finally, the quantified latency of sensory-evoked running (2.574
 ± 0.259 s) clearly demonstrated a time-locked relationship between the stimulus and
 behavioural onset.

Figure 1

Figure 4

Extended Figure 5

**Fig. 1 | The TeA controls flight behaviours.** n, Probability ($F_{2,12} = 11.75$, $P = 0.001$) of flight.

One-way repeated-measures ANOVA with the Bonferroni post hoc correction. $n = 7$ animals.

**Fig. 4 | Sensation-related and running-related neurons in the TeA.** b, The same as in Fig. 1m,

analysis of the turntable running model in a ($D = 0.487$, $P < 10^{-4}$). Kolmogorov–Smirnov test.

**Extended Data Fig. 5 | Multisensory integration.** a–f, Paradigm of the running model (a) with

subthreshold or suprathreshold unisensory (b, sound: 30 dB, 12 k; light: 1 lux and air puff: 0.4 L/min)

or multisensory stimuli in different combinations for inducing running shown as rasters (c) and

running speed traces (d, 5 trials) and corresponding plots of the probability of running (e, $F_{2,12} =$

176.4602 , $P < 10^{-6}$) and peak speed (f, $F_{2,12} = 35.3831$, $P = 9 \times 10^{-6}$). One-way repeated-measures

ANOVA with the Bonferroni post hoc correction. $n = 7$ animals.

- The correlation between neuronal activity and running speed would be better
assessed by simply computing the cross-correlogram between the two, which
would provide both peak correlation and time lag.

We believe your suggestion is excellent, and we have adopted this analytical approach
in both the present study and a separate manuscript to evaluate the correlation between
neuronal activity and running speed.

Extended Figure 2

Figure 4

**Extended Data Fig. 2 | S- and R-neurons in the TeA.** **c–e**, CCF analyses of R-neurons (**c**, n = 29
neurons), SR-neurons (**d**, n = 66 neurons) and unSR-neurons (**e**, n = 9 neurons). The black line
represents the mean CCF curve; grey lines indicate individual neuronal CCF traces.

**Fig. 4 | Sensation-related and running-related neurons in the TeA.** **f**, Left panel, Distribution of
the maximum correlation coefficient (r_{max}) from the analysis of R-neurons (**Extended Data Fig. 2c**).
The red dashed line represents the Gaussian fit curve ($R^2 = 0.7464$, $P = 0.0097$). Right panel,
Distribution of the optimal time lag from the analysis of R-neurons ($R^2 = 0.7437$, $P = 0.0021$). **j**,
The same as **f**, left panel, analysis of SR- neurons (**Extended Data Fig. 2d**) (n = 66 cells, $R^2 =$

0.6075, $P = 2.1947 \times 10^{-6}$). Right panel, Analysis of SR-neurons ($R^2 = 0.7285$, $P = 3.8492 \times 10^{-6}$). **m**,
The same as **j**, analysis of the unSR-neurons ($n = 9$ cells) in **Extended Data Fig. 2e**.

- **The text mention 29 R-Neurons, 66 SR-Neurons and 9 unSR-Neurons (104 cells)**
**but figure 4h-j shows more many unSR-Neurons (at least 15) and different number**
**of cells from panel to panel (see unSR/LS).**

We apologize for the lack of clarity in our original description, which led to a
misunderstanding. For clarity, **Fig. 4q–s**, respectively, present the following findings:
firing latency of neurons in response to sensory stimulation; running latency of animals
in response to sensory stimulation; and difference in time between neuronal firing and
animal running. Taking **Fig. 4q** as an example, although we previously mentioned that
9 unSR neurons were recorded, the number of data points in the figure exceeds 9. This
discrepancy is because each point does not represent a single neuron but rather the
response latency of a neuron to a single stimulus trial. For instance, if one recorded
neuron responded to sound stimulation and was tested with 5 auditory trials, 5 distinct
latency values were calculated and plotted, accounting for the greater number of data
points compared with the number of neurons recorded.

We separated the response latencies to different sensory stimuli to align with **Fig. 4q**,
which analyses the difference in time between neuronal firing and movement events
across neuron types. Accordingly, **Fig. 4q** specifically evaluates the latency of neuronal
responses to various sensory stimuli.

We have now provided a detailed explanation of this analysis in the Results section of
the revised manuscript. “*The firing latency of S-neurons to each individual sensory*
*stimulus (sound, light, and air puff) was calculated (Extended Data Fig. 2f and g)*”

[Results (line 408-409)]

**Fig. 4 | Sensation-related and running-related neurons in the TeA.** **q**, Summary of the stimulus–
firing latency for each type of neuron. unSR-SS: 21.76 ± 3.46 ms, SR-SS: 31.86 ± 13.19 ms, $U =$
194 , $z = -3.1944$, $P = 0.0014$, Mann–Whitney U test. unSR-LS: 414.8 ± 37.13 ms, SR-LS: $568.3 \pm$
70.11 ms, $t_{23} = -5.4872$, $P = 1.4 \times 10^{-5}$, two-sample t test. unSR-AP: 37.62 ± 5.71 ms, SR-AP: 56.87
± 11.24 ms, $t_{55} = -5.7133$, $P < 10^{-6}$, two-sample t test. **r**, Summary of the stimulus–running
latency for each type of neuron. unSR-SS: 2.51 ± 0.11 s; SR-SS: 2.49 ± 0.18 s; $U = 125.5$, $z = -0.085$, $P =$
0.9326 ; Mann–Whitney U test. unSR-LS: 2.87 ± 0.15 s, SR-LS: 2.92 ± 0.19 s, $U = 19$, $z = -0.014$,
$P = 0.8875$, Mann–Whitney U test. unSR-AP: 2.60 ± 0.21 s, SR-AP: 2.52 ± 0.19 s, $t_{32} = 0.774$, $P =$
0.4446 , two-sample t test. **s**, Summary of the difference in firing–running time for each type of
neuron. unSR-SS: 2.49 ± 0.11 s, unSR-LS: 2.45 ± 0.13 s, unSR-AP: 2.56 ± 0.21 s, SR-SS: $2.45 \pm$
0.18 s, SR-LS: 2.38 ± 0.15 s, SR-AP: 2.47 ± 0.19 s, R: 2.12 ± 0.24 s. $K-W = 45.319$, $P < 10^{-6}$,
Kruskal–Wallis one–way ANOVA with the Bonferroni post hoc correction.

- **How many mice were used for juxtacellular recordings? How many neurons /**

**mouse were recorded ?**

This information has been added to the Methods section. “A total of 20 mice were used
for the intracellular recordings, with an average of 6–10 neurons recorded per mouse.”

[Methods (line 1244-1245)]

- In figure 6g, the example images show 1 or 2 double-labeled neurons (yellow) for
single-labeled neurons (green or red). That does not really fit with the reported %
of overlap (60% for AC).

We sincerely apologize for our oversight in the previously presented AC images, where
not all yellow neurons were accurately marked. In the newly submitted figures, we have
corrected the number of labelled neurons and recalculated the proportion of double-
labelled cells.

**Figure 6 | SensTeA and TeA_{dPAG} neurons.** g. Schematic of the virus injection sites. h, Left panel,
Images showing SensTeA neurons (EYFP), TeA_{dPAG} neurons and neurons labelled in the upper layer
(mCherry). Right panel, Percentage of TeA neurons with overlapping labelled inputs from different
sources. n = 9, 9, and 9 slices from 3, 3, and 3 animals, respectively.

- Results figure 6l: there seems to be a scaling mismatch between the left and the
right panels: the grand-average traces for EPSCs on the left peak at about 200 pA

(black) and 125 pA (red) whereas the bar graph on the right indicate averaged
values of 500 pA (black) and 300 pA (red) ?! (see my Minor comments from the
previous version).

We sincerely thank you for raising this question. We have carefully re-examined the
relevant data and, as you correctly observed, both the raw trajectories and the statistical
results are correct; the issue was solely due to an incorrect scale bar. This error has been
corrected in the newly submitted version. For accuracy, we have also double-checked
all the scale bars throughout the manuscript.

**Figure 6 | SensTeA and TeAdPAG neurons. I,** Left panel, Inhibitory postsynaptic current (IPSC) and
excitatory postsynaptic current (EPSC) of a TeAdPAG neuron following the delivery of blue light to
the slice to activate $_{AC}$ TeA or $_{SI}$ TeA neurons expressing ChR2. Right panel, LED-evoked IPSCs and
EPSCs before (black) and after (red) the TTX (1 μ M) + 4-aminopyridine (1 mM) infusion. EPSCs:
$t_9 = -8.6468, P = 1.2 \times 10^{-5}$. IPSCs: $t_9 = 12.2603, P = 1 \times 10^{-6}$. Paired t test. n = 10 cells from 10 slices.
Data were obtained from both the right and left hemispheres.

**Other not-addressed comments from the previous version:**

- **An important claim is that TeA is critical to elicit running in response to the**
**sensory-cue. And indeed, TeA inactivation blocks running induced by all 3 sensory**

**stimuli. But the authors now also show that TeA inactivation a has an effect per se**
**on spontaneous locomotor activity, i.e. TeA inactivation reduces locomotion thus**
**questioning a specific role of TeA in sensory-evoked running.**

We sincerely thank you for this important question. In fact, other reviewers have raised
similar concerns, and we have therefore revised and clarified the relevant descriptions
throughout the manuscript.

As you rightly noted, the inactivation of TeA or TeA_{dPAG} neurons affects spontaneous
motor activity, indicating that the TeA region plays a broad role in movement regulation
and is not exclusively involved in sensory-evoked running. We agree that our initial
conclusion that TeA is “essential for sensory-triggered running” may have been
overstated. While TeA inactivation only partially reduces, rather than completely
abolishes, spontaneous movement, the results from both **Fig. 1** and **Fig. 3** show that it
fully blocks sensory stimulus-evoked running. These findings suggest that although the
TeA contributes to spontaneous locomotion, it also plays a necessary role in running
triggered by sensory signals.

We fully acknowledge the concerns raised by the reviewers. Indeed, the results in **Fig.**
**3c–d** weaken our earlier emphasis on the TeA as uniquely critical for processing
sensory-induced running. In the revised version, we have modified all related assertions:
the TeA is involved in sensory-evoked running, but it also participates in running driven

by other signals or nonsensory inputs.

We have toned down the strong claim made in the original manuscript that the TeA–

dPAG circuit is a specialized pathway for integrating sensory stimuli to initiate escape.

Accordingly, we have adjusted the language intensity from “specifically responsible for”

to “participates in” throughout the text.

Figure 1

**Fig. 1 | The TeA controls flight behaviours.** **r**, Top panel, Slices containing the bilateral TeA

showing virus expression. Scale bar, 100 μ m, unless specified otherwise. Bottom left panel, Raw

traces of current-clamp recordings from an hM4D(Gi)-expressing TeA cell in a slice preparation.

Bottom right panel, Average spontaneous spike frequencies recorded before and after perfusion with

CNO and after CNO washout. $F_{1,504,4.218} = 260.973$, $P < 10^{-6}$. One-way repeated-measures ANOVA

with the Bonferroni post hoc correction; $n = 5$ cells from two mice. **s**, Left panel, Representative

flight speed traces before (CNO-) and 30 min after the intraperitoneal injection of CNO (i.p., CNO+)

in hM4D(Gi)-expressing animals. Right panel, Peak speed induced by S ($t_6 = 18.266$, $P = 2 \times 10^{-6}$),

L ($t_6 = 14.327$, $P = 7 \times 10^{-6}$), and A (A: $t_6 = 10.849$, $P = 3.6 \times 10^{-5}$) before and after the CNO

injection. Paired t test. $n = 7$ animals. **t**, Left panel, The same as in **s**, analysis of mCherry-expressing

control animals. Right panel, Peak speed induced by S ($t_6 = -0.523, P = 0.620$), L ($t_6 = 1.127, P =$
 0.303), and A (A: $t_6 = 0.922, P = 0.392$) before and after the CNO injection. Paired t test. $n = 7$
 animals. **u**, Top panel, Bilateral TeA slices showing virus expression. Bottom panel, PSTHs (100
 trials) for the responses of a TeA neuron to noise (80 dB SPL) without (LED-) and with (LED+) LED
 illumination. The data correspond to 7 TeA cells (at depths of 250–800 μm) from 7 animals. Calibration:
 50 μV , 0.5 ms. $t_6 = 10.405, P = 1.41 \times 10^{-4}$. Paired t test. **v**, Left panel, Representative
 flight speed traces without (LED-) and with (LED+) LED illumination for eNpHR-expressing
 animals. Right panel, Peak speed induced by S ($t_6 = 12.178, P = 1.9 \times 10^{-5}$), L ($t_6 = 6.209, P = 0.001$),
 and A (A: $t_6 = 8.191, P = 1.78 \times 10^{-4}$) without and with LED illumination. Paired t test. $n = 7$ animals.
 **w**, Left panel, The same as in **v**, analysis of EGFP-expressing control animals. Right panel, Peak
 speed induced by S ($t_6 = 0.018, P = 0.9862$), L ($t_6 = -0.5252, P = 0.6183$), and A (A: $t_6 = -0.2696,$
 $P = 0.7965$) without and with LED illumination. Paired t test. $n = 7$ animals.

Figure 3

**Fig. 3 | A TeA-dPAG circuit determines running behaviours. b,** Blockade of the TeA-dPAG
pathway in mice running on the turntable: TeA_{dPAG} CaMKII neurons. **1,** Virus injection site and
optogenetic (LED) interventions. **2,** Virus expression. **3,** Representative flight speed trace following
random presentation of S, L, and A stimuli before (LED-) and after (LED+) interventions in eNpHR-
expressing animals. Thick lines: average flight speed (30 trials). **4,** Peak flight speeds ($F_{1,12} = 62.159$,
$P = 4 \times 10^{-6}$), two-way ANOVA with the Bonferroni post hoc correction. $n = 7$ animals/group. **c,**
Blockade of TeA_{dPAG} CaMKII neurons. **1,** Virus injection site and chemogenetic (CNO) intervention.
**2,** Virus expression. **3,** Representative traces before (CNO-) and after (CNO+) intraperitoneal (i.p.)
injections of CNO in hM4D(Gi)-expressing animals. **4,** Performance of hM4D-expressing and
control mice in the open field test: distance travelled ($F_{1,18} = 5.192$, $P = 0.0351$), speed ($F_{1,18} =$
15.679 , $P = 9.19 \times 10^{-4}$), immobility time ($F_{1,18} = 8.107$, $P = 0.0107$), and time spent exploring the
centre zone ($F_{1,18} = 0.9949$, $P = 0.3318$) before (CNO-) and after (CNO+) intraperitoneal injections
of CNO. Two-way ANOVA with the Bonferroni post hoc correction. $n = 10$ animals/group. **d,**
Blockade of TeA_{dPAG} CaMKII neurons. **1-3,** The same as in (c1-3). **4,** Performance of hM4D-
expressing and control mice in the open-field test: distance travelled ($F_{1,18} = 27.705$, $P = 5 \times 10^{-5}$),
speed $F_{1,18} = 51.165$, $P = 1 \times 10^{-6}$), immobility time ($F_{1,18} = 78.193$, $P < 10^{-6}$), and time spent
exploring the centre zone ($F_{1,18} = 0.7787$, $P = 0.3892$) before (CNO-) and after (CNO+) the
intraperitoneal injection of CNO. Two-way ANOVA with the Bonferroni post hoc correction. $n =$
10 animals/group.

**- Extended Figure 1f: I am really surprised by the distribution of VIP+ and PV+**
**cells in TeA. I would have expected much less VIP+ cells (especially in deep layers)**
**and much more many PV+ cells. See [https://www.frontiersin.org/journals/neural-](https://www.frontiersin.org/journals/neural-circuits/articles/10.3389/fncir.2021.781928/full)**
**circuits/articles/10.3389/fncir.2021.781928/full for a possible explanation. By the**
**way, the JAX reference of the PV-Cre line (Methods, line 977) is missing.**

We sincerely thank you for providing the relevant references, which have helped us

better understand the potential limitations of the PV-Cre mouse model we are currently
using. As indicated in the cited study, the PV-IRES-Cre mouse line exhibits low
efficiency in labelling PV-INs in the association cortex, which may be a common
characteristic across regions of the association cortex.

Notably, according to the Allen Brain Atlas, the number of PV neurons in the
association cortex of mice remains considerable. For clarity, we have now included the
JAX stock numbers of the PV-Cre mice in the Methods section. “*PV-Cre (RRID:*
*IMSR_JAX: 008069)*” [Methods (line 957)]

[FIGURE REDACTED]

**Nigro MJ, et al- Figure 1. (A)** Representative immunofluorescence staining for PV of a slice at
about bregma –3 mm of a PVcre/Ai9 mouse. Dotted boxes highlight the lateral part of the secondary
visual cortex (V2L) and the hippocampus (Hipp). The perirhinal cortex (PER) is delineated by the
dotted line on the lateral side of the brain. **(B)** Representative image of the immunofluorescence for
tdTomato in the same section as **(A)**. Upper insert shows the merged signals in the cortex, and lower
insert shows merged signals in the hippocampus (SLM, stratum lacunosum-moleculare; ML,
molecular layer; GL, granule layer of the dentate gyrus; HI, hilus; CA3pyr, pyramidal layer of the
CA3 region).

- **Methods, line 1143: Provide more details about the CNO solution and**
**preparation. How was CNO dissolved?**

Detailed information regarding the preparation of the CNO solution has been
thoroughly described in the Methods section. “*Mice that expressed hm4D(Gi) were*
*intraperitoneally injected with CNO (0.33 mg/ml, 1 mg/kg). Prior to the formal*
*experiments, a high-concentration stock solution of CNO (10 mg/ml) was prepared by*
*dissolving CNO powder (BrainVTA, China) in sterile dimethyl sulfoxide (DMSO) (HY-*
*Y0320; Med-Chem Express, USA). Briefly, sterile DMSO was added to the tube*
*containing the CNO powder. The tube was tightly capped and vortexed vigorously for*
*30–60 seconds to achieve initial dispersion of the powder. The tube was then subjected*
*to sonication in an ultrasonicator for 5–10 minutes to facilitate complete dissolution.*
*The tube was inspected visually to confirm complete dissolution, indicated by a clear*
*and transparent appearance without any undissolved particles at the bottom or on the*
*walls of the tube. If the particles remained, the cycle of vortexing and sonication was*
*repeated until the solution was entirely clear. Once fully dissolved, the stock solution*
*was immediately aliquoted into single-use portions to avoid repeated freeze–thaw*
*cycles and stored at -20°C until use. The stock solution was thawed and diluted in sterile*
*physiological saline (0.9% NaCl) to the desired working concentration (0.33 mg/ml)*
*immediately prior to intraperitoneal (i.p.) injection into the animals.” [Methods (line*
**1119-1134)]**

- **Methods, line 1202-1203: By definition in voltage-clamp mode, the V_m is**
**clamped to a given value! What was this value during the recordings?**

We thank the reviewer for this valuable comment. You are absolutely correct that the
clamping voltage must be clearly specified in voltage-clamp mode. In our cellular
recordings, the amplifier indeed applied a holding voltage. We used the default
amplifier setting, which clamped the electrode potential at 0 mV (i.e., $V_{cmd} = 0$ mV).
Therefore, the actual voltage applied to the recorded patch (V_{patch}) equals the cell's
intrinsic membrane potential (V_{cell}). The currents we recorded reflect the ionic flows
driven by the cell's own physiological activity, such as resting potential fluctuations
and action potentials. We have now clarified and corrected this description in the
Methods section of the manuscript. "*Cellular recordings were performed in voltage-*
*clamp mode ($V_{cmd} = 0$ mV) using a MultiClamp 700B amplifier (Axon, USA).*"

**[Methods (line 1234-1235)]**

- **Last page: it would help to have the abbreviations in alphabetical order.**

We thank you for your suggestion and have revised the manuscript accordingly in the
newly submitted version.

**Sylvain Crochet**

**References:**

1 Wang, H. *et al.* Direct auditory cortical input to the lateral periaqueductal gray
controls sound-driven defensive behavior. *PLoS Biol* **17**, e3000417, (2019).

**J.C., H.L. et al. “Neural circuits and speed coding of running, rebound running**
**and backing away behaviours”.** Submitted as a companion paper.

Dear Reviewer,

The line numbers referenced in our point-by-point responses correspond to the
"Revised manuscript without track changes-He Li.pdf" file.

**Reviewer #2-Li (Remarks to the Author):**

**The study by He et al. aims to investigate the role of the TeA (temporal association**
**area) as a hub for multisensory integration driving escape behavior through**
**microcircuit computations. While the breadth and depth of the experiments—**
**encompassing behavior, anatomy, physiology, and causal manipulations—are**
**impressive, and some findings do indeed support the authors' claims, the overall**
**narrative suffers from significant shortcomings.**

**The manuscript is plagued by numerous inconsistencies in experimental design,**
**logical reasoning, and the interpretation of results. The narrative lacks cohesion,**
**with critical connections between key experiments and conclusions often left**
**underdeveloped or unclear. Instead of building a compelling and logical story, the**
**sheer volume of data and experiments overwhelm and obscure the central message,**
**detracting from the paper's impact.**

**The revised manuscript has largely improved in term of clarity. The writing**
**remains a bit odd sometimes and the paper still contains a large number of**
**experiments that are not all of critical importance or are even clearly redundant.**

We sincerely appreciate your feedback. In response to your comment that the writing
occasionally seemed unclear or awkward, we have carefully re-examined the entire
manuscript and refined inaccuracies in the descriptions based on the suggestions from
all the reviewers. Additionally, the paper has been professionally edited and polished to
improve its clarity and precision.

Regarding your concern that the manuscript contains a substantial number of noncritical
or apparently redundant experiments, we suspect that you may be referring to the
behavioural experiments in **Fig. 1–3** and the characterization of the neuronal
distribution within the TeA–dPAG circuit. We addressed this concern by further
streamlining our results, optimizing the behavioural inhibition experiments, and
removing nonessential or repetitive data in the revised version.

However, we have retained the analysis of the neuronal distribution in the TeA–dPAG
circuit, as, to our knowledge, no previous reports have described the projection patterns
between the TeA and dPAG in detail. We believe that these findings must be presented
in sufficient detail to fill this gap in the literature.

**Although a few specific examples are highlighted below, these issues are pervasive**
**throughout the manuscript. As it stands, the paper is disjointed and challenging to**
**follow, making it difficult to discern a coherent scientific narrative. A thorough**
**and thoughtful revision is required to address these issues before the manuscript**

can be considered suitable for peer review.

Below are a few examples of major concerns with the manuscript (minor
comments are far too numerous to list):

The central theme of “running” as highlighted in the title and abstract feels
misaligned with the actual findings, which seem to emphasize sensory-evoked
escape behavior instead.

This issue has been addressed in the revised version.

Furthermore, the broader significance of the study is poorly articulated. The
authors must clarify why this research is interesting and important within the
larger context of behavioral neuroscience.

This issue has not been fully addressed. The introduction remains vague with
many sentences stating the obvious, and lacks logical progression.

We sincerely appreciate your comment. In the newly submitted version, we have further
revised the Introduction accordingly. *“Animals’ perceptions of their external and
internal environments determine and modulate their behaviours, as evidenced by
studies exploring sensory modalities such as auditory²⁻⁷, visual⁸⁻¹², olfactory¹³, and
somatosensory stimuli^{5,6}, all of which can elicit innate flight responses such as escape,
which are essential for animals to survive in the face of life-threatening environmental
cues¹⁴. A critical function of neural centres is to establish neuronal circuits that perceive
these cues, process them as sensory signals, make decisions, transform these decisions
into motor commands, and subsequently initiate appropriate defensive behaviours¹⁵.*

*The neuronal circuits in mice that underlie the detection of and escape from threats*
*have been extensively investigated recently. However, whether a specific neural nucleus*
*acts as a central hub to coordinate cross-sensory escape behaviours remains unclear.*
*Such a nucleus might integrate inputs from sensory and decision-making neurons to*
*orchestrate motor commands, potentially within a localized microcircuit.*

*A widely accepted hypothesis is that sensory regions process sensory information^{2,3};*
*motor regions generate motor signals (e.g., in circuits related to innate escape*
*behaviours, which ultimately project to the deep periaqueductal grey (dPAG))^{2,3,16}; and*
*higher-order cortical areas (such as the association cortex), striatum, superior*
*colliculus, and various midbrain regions generate decision-related information,*
*facilitating the transformation of sensory signals into motor signals^{9,17-21}. However,*
*sensory regions also encode decision-making and reward-related information²²⁻²⁵, and*
*motor-related signals have been detected in nonmotor-related nuclei²⁶⁻³¹. These*
*observations suggest that sensory neurons, sensory–motor decision neurons, and motor*
*command neurons capable of directly converting sensory input into motor output may*
*all be present within some nuclei of the mouse brain. As a region that receives sensory*
*signals and connects to motor-related areas, the association cortex appears to be the*
*most likely candidate for fulfilling this function.*

*The temporal association cortex (TeA), which is part of the association cortex, is a high-*
*level nucleus that receives diverse sensory inputs³², processing auditory³³, visual³⁴⁻³⁶,*

*and tactile³⁷ signals through projections from different sensory nuclei and enables the*
*integration of multimodal sensory information³⁸⁻⁴⁰. Previous research has revealed the*
*role of the TeA in processing auditory signals⁴¹ and influencing both innate⁴² and*
*learned behaviours^{41,43}. Moreover, the TeA sends projections to the dPAG, a region that*
*controls defensive behaviours^{32,42}. While emerging evidence has implicated the TeA in*
*behavioural responses such as the retrieval of pups in maternal mice (an innate care*
*behaviour)⁴² and auditory-cued fear conditioning (learned behaviour)^{41,43}, its specific*
*role in the defensive behaviour circuit remains poorly defined. This critical knowledge*
*gap, coupled with the robust anatomical connectivity of the TeA to key command*
*neurons, renders this region a highly compelling and potentially pivotal node worthy of*
*in-depth investigation. We hypothesize that the TeA likely serves not only as a simple*
*relay station but also as a critical hub coordinating diverse sensory-induced escape*
*behaviours. It may integrate sensory information and participate in driving*
*downstream motor command pathways, constituting a microcomputational hub that*
*encompasses sensory integration, decision-making, and motor command functions. It*
*plays an indispensable role in triggering adaptive escape behaviours.*

*In a mouse model of flight behaviours driven by sound, light, and air puffs, along with*
*a combination of in vivo and in vitro electrophysiological recordings and chemogenetic*
*and optogenetic interventions, our study elucidated the contribution of the TeA–dPAG*
*pathway to sensory-induced escape behaviours and characterized the features of*
*neurons within the TeA nucleus that are responsible for sensory response, sensorimotor*

*decision-making, and motor command. Furthermore, information on diverse sensory*
*stimuli is integrated into layer 5 (L5) intratelencephalic (IT) neurons as running signals;*
*these IT neurons activate L5a critical pyramidal tract (PT) neurons, which project to*
*the dPAG, and, together, these neurons form an intralayer IT–PT microcircuit for*
*sensory–motor decision-making and inducing running behaviours.” [Introduction*
**(line 49-105)]**

**The text is frequently unclear, with superficial or anecdotal explanations. The**
**English language must undergo significant editing for grammar, style, and overall**
**readability to meet minimal academic standards.**

**These issues have been partially addressed in the revised version. But despite a**
**significant improvement, the writing remains sometimes suboptimal with odd**
**sentences or expressions.**

We sincerely appreciate your comment. As English is not our native language, some of
the descriptions of the results may still be unclear. In response, we have carefully
reviewed the entire manuscript and refined inaccurate statements based on specific
suggestions from all the reviewers. Additionally, the full text has been professionally
edited and polished to ensure greater clarity and accuracy of the language.

**There are innumerable inconsistencies between figure labeling and the main text,**
**making interpretation challenging.**

**As far as I could tell, this issue has been addressed in the revised version.**

**Some core experimental designs lack clarity and justification. For instance, the**
**claim that AAV-Cre (source) and DIO-virus (target) represent transsynaptic**
**labeling is fundamentally questionable. This combination typically labels neurons**
**projecting from the target to the source region, not synaptically connected**
**downstream cells. This issue alone casts doubt on the interpretation and validity**
**of conclusions drawn from these experiments.**

**I think the authors have missed the point raised by the reviewer. There is no**
**question that some serotypes of AAV virus can jump one synapse and infect**
**connected neurons downstream to the source (injection site) region. But the**
**question is about the specificity. There is also clear evidence that these viruses also**
**result in retrograde labelling (i.e. neurons projecting TO the source area). I**
**suppose that should not be much of a problem when considering TeAdPAG**
**neurons as I doubt that dPAG neurons project directly to the TeA (as far as I know,**
**but should be verified) but it is clearly more of an issue when considering TeA**
**neurons receiving inputs from other cortical areas since reciprocal cortico- cortical**
**projections are largely expected.**

We fully agree with your perspective. Regarding the TeA–dPAG circuit, the impact
of this issue is likely minimal. Our preliminary retrograde tracing experiments (as
shown in the figure below) revealed that after AAV-retro was injected into the TeA
(**Fig. a**), no labelled neuronal somata were observed in the dPAG (**Fig. b**), whereas

other regions (such as the AC, S1, etc.) were clearly visible (**Fig. 5a**). In conjunction
with the Allen Brain Atlas, currently, no evidence supports the existence of neurons
in the dPAG that project directly to the TeA (**Fig. c and d**).

However, when TeA neurons receive inputs from other cortical regions, as you noted,
the risk of retrograde labelling cannot be overlooked due to the extensive mutual
projections between cortical regions. When a Cre-DIO strategy is used, in addition to
labelling TeA neurons that receive projections from other cortical areas, labelling TeA
neurons that project to other cortical regions is possible. As shown in the figure below
(**Fig. e**), after retrograde tracing of viruses into A1 was performed, retrograde
labelling could be observed in the TeA. However, based on the morphology of the
retrogradely labelled neurons (only the somata are clearly labelled), the morphology
of TeA neurons receiving AC projections that we labelled using the Cre + DIO method
differs (**Fig. 6**). Therefore, the neurons we labelled should still primarily be those that
receive projections from other cortical regions.

Second, in terms of the functional impact, let us take the validation of AC projections
to the TeA observed in our experiments as an example. A more complex circuit (e.g.,
TeA→AC→other nuclei) could theoretically contribute to running behaviours.
However, our experimental evidence does not support its functional necessity. As
shown in **Fig. 7k–n**, activating the AC→TeA input fails to induce running if the
TeA→dPAG output is blocked. These findings suggest that even if such alternative

circuits exist, their capacity to drive behaviour is negligible without the key
TeA→dPAG link. Additionally, our results in **Fig. 6i–l** confirm that TeA neurons
receiving AC projections directly project to TeA_{dPAG} neurons. Thus, the circuit we
originally investigated remains valid.

Therefore, although the theoretical risks mentioned above exist, we believe that this
limitation is unlikely to undermine the core findings of this study. The key aspect of
this research lies in revealing that specific populations within the TeA region, defined
by viral strategies (i.e., neurons functionally connected to other cortical areas), are
functionally distinct from other neurons. Regardless of whether this population is
absolutely “pure”, the discovery of this functional heterogeneity is robust. Second,
all the behavioural results were derived from direct optogenetic stimulation applied
to the TeA region. This stimulation simultaneously affects all ChR2-labelled neurons.
Therefore, the observed behavioural-modulating effects are the net result of
manipulating the entire labelled population. Our conclusion that "manipulating this
population is sufficient to influence behaviour" remains valid. We have proposed in
the Discussion that future use of more specific tools, such as the rabies virus system,
will be the gold standard for addressing this issue.

We have also addressed this aspect in the Discussion section. *“This study utilized an*
*AAV-based anterograde tracing strategy to label input neurons projecting from other*
*sensory cortices to the TeA, as well as to label output neurons projecting from the TeA*

[revised manuscript text omitted]

**a**, AAVretro-hSyn-mCherry injection into the TeA. **b**, No retrograde viral expression was observed
 in the dPAG following AAVretro-hSyn-mCherry injection in the TeA. **c**, Anterograde tracing from
 the dPAG. **d**, No anterograde labelling was observed in the TeA (Allen Brain Atlas). **e**, AAVretro-
 hSyn-EYFP injection in A1. Retrogradely labelled neuronal somata were observed in the TeA.

**Fig. 5 | Diverse sensory input characteristics of the TeA.** a, Virus injection site in the TeA (left
 panel) and expression in the target cortical (middle panel, AC, S1, and V2L) and subcortical
 regions (right panel, DLG, and MGB).

**Figure 6 | Sens TeA and TeA_{dPAG} neurons.** a, Top left panel, Schematic of the virus injection in
 the AC, S1, V2L, MGB, DLG, TeA, and dPAG. Images showing Sens TeA neurons (EYFP, green)

**Figure 6 | Sens TeA and TeA_{dPAG} neurons.** a, Top left panel, Schematic of the virus injection in
 the AC, S1, V2L, MGB, DLG, TeA, and dPAG. Images showing Sens TeA neurons (EYFP, green)

in L2-5b, including $_{AC}TeA$, $_{S1}TeA$, $_{V2L}TeA$, $_{MGB}TeA$, and $_{DLG}TeA$, and TeA_{dPAG} neurons in L5a
 (mCherry, red) in the TeA. **i**, Virus injection sites. **j**, Representative firing patterns of $_{Sens}TeA$ and
 TeA_{dPAG} neurons in response to a depolarizing current (240 pA). **k**, Top panel, Comparison of
 action potential half widths ($t_{12.838} = -3.2759$, $P = 0.0061$). Bottom panel, F/I (firing rate/current)
 slope ($t_{18} = 4.8353$, $P = 1.33 \times 10^{-4}$). Two-sample t test. $n = 10$ cells from 10 slices. **l**, Left panel,
 Inhibitory postsynaptic current (IPSC) and excitatory postsynaptic current (EPSC) of a TeA_{dPAG}
 neuron following the delivery of blue light to the slice to activate $_{AC}TeA$ or $_{S1}TeA$ neurons
 expressing ChR2. Right panel, LED-evoked IPSCs and EPSCs before (black) and after (red) the
 TTX (1 μ M) + 4-aminopyridine (1 mM) infusion. EPSCs: $t_9 = -8.6468$, $P = 1.2 \times 10^{-5}$. IPSCs: $t_9 =$
 12.2603, $P = 1 \times 10^{-6}$. Paired t test. $n = 10$ cells from 10 slices. Data were obtained from both the
 right and left hemispheres.

**Fig. 7 | Firing and running evoked by the activation of $_{AC}TeA$ and TeA_{dPAG} neurons.** **k**, Virus
 injection and chemogenetic (CNO) and optogenetic (LED) interventions. $_{AC}TeA$ and $_{S1}TeA$
 neurons expressing ChR2-EYFP; TeA_{dPAG} neurons expressing hM4D(Gi). **l**, **m**, Representative
 LED light (20 Hz, 5 s, 9 trials)-induced speed raster (**l**) and running speed traces (**m**) (10 trials)
 before (Pre), during (CNO), and recovery (Rec) from the effect of i.p. injections of CNO. **n**,
 Induced peak speed for the Pre, CNO, and Rec periods. Pre: 6.149 ± 1.179 cm/s, CNO: $0.9806 \pm$
 0.6357 cm/s, Rec: 6.27 ± 1.366 cm/s, $F_{2,12} = 142.4438$, $P < 10^{-6}$, one-way repeated-measures
 ANOVA with the Bonferroni post hoc correction. $n = 7$ animals.

**The framing suggests a focus on multisensory integration, yet the experiments**
 **alternate inconsistently between unisensory and multisensory paradigms, leaving**
 **the narrative fragmented.**

**I think this has been clarified in the revised version.**

**The analysis of interneuron anatomy and connectivity feels redundant and poorly**
**integrated into the overarching story.**

**This part has been moved to supplementary figures.**

**These problems, combined with the sheer volume of data and experiments, result**
**in a manuscript that is difficult to follow and comprehend. The connections**
**between experiments, results, and conclusions are weak, and the narrative lacks**
**cohesion.**

**In summary, the manuscript requires substantial revision to address these**
**conceptual, technical, and narrative flaws. Without these changes, it remains**
**disjointed and exceedingly difficult to read as a coherent scientific manuscript.**

**There still are many – somewhat redundant – experiments in the paper but the**
**overall clarity of the manuscript has been largely improved. Some technical**
**aspects remain insufficiently described, in particular regarding the data analysis.**

We sincerely thank you for your previous feedback regarding the inclusion of
noncritical or repetitive experiments in our manuscript. In direct response to this
comment, we have streamlined the Results section in the revised version by removing
several peripheral datasets. This revision significantly improves the overall coherence
and narrative flow of the paper.

From a technical perspective, we have also expanded the Methods section to provide

more detailed descriptions, including the specific protocol for preparing the CNO
solution and the criteria for neuronal labelling during *in vivo* patch-clamp recordings.

With respect to data analysis, we have now included comprehensive details on how the
temporal windows for different types of neuronal responses (e.g., stimulus–running,
stimulus–firing, and firing–running relationships) were defined. Furthermore, we
addressed your point on statistical clarity by providing the full set of p values from all
post hoc paired comparisons that were part of our original analysis. Given the large
number of comparisons, we have included the complete results in a supplementary file
to ensure transparency.

Dear Reviewer,

The line numbers referenced in our point-by-point responses correspond to the
"Revised manuscript without track changes-He Li.pdf" file.

**Reviewer #3-Li (Remarks to the Author):**

**This manuscript by Li et. al presents the interesting and somewhat provocative**
**claim that multi- sensory integration, decision-making, and generation of motor**
**command to initiate walking all occur within an intra-laminar circuit in layer 5 of**
**temporal association cortex (TeA). The work does a careful anatomical**
**characterization of the projection from TeA to periaqueductal grey (PAG) and a**
**series of manipulations of this pathway, with increasing specificity as the**
**manuscript progresses.**

**However, the only readout is stimulus-evoked locomotion. Locomotion in other**
**contexts would need to be assayed to determine specificity.**

**Indeed, the behavioral response could have been analyzed more carefully. It**
**remains unclear whether the response to the sensory stimuli is acquired during the**
**training, to what extent the sensory stimuli evoke running compared to**
**spontaneous running, whether the mice show any sign of habituation across the**
**repetition of the sensory stimuli.**

The animals' responses to sensory stimuli were not acquired during training. Briefly,

since we applied aversive stimuli, the movement behaviour following stimulation
constitutes an innate defensive response—an experimental paradigm that has been
extensively documented in previous studies.

Furthermore, according to earlier reports, in head-fixed experimental setups, the
probability of mice engaging in spontaneous running is approximately 10%, while they
remain stationary or exhibit whisking and/or facial/jaw/paw movements more than 70%
of the time⁴⁵. Our data are consistent with these findings.

In contrast, sensory stimuli—particularly high-intensity aversive stimuli—can evoke
running with a probability of approximately 70%, which is significantly higher than
that during spontaneous running. Although adaptive behaviour may occur during
repeated sensory stimulation, especially with short interstimulus intervals, we
minimized such adaptation by employing three different sensory modalities delivered
in a pseudorandom order.

**Furthermore, analyses of locomotor responses to each of the stimuli are not**
**sufficiently presented to interpret the effects.**

**This issue has been clarified.**

**More generally, the results are sloppily presented, with inconsistencies in**
**presentation, a lack of justification for selection of data for illustrations, a lack of**

**detail needed to interpret statistical comparisons, and numerous saturated**
**anatomical images.**

**The presentation of the results has been largely improved although some aspects**
**of the data analyses and statistics remain unclear.**

Thank you for your question. With respect to data analysis, we have now included
comprehensive details on how the temporal windows for different types of neuronal
responses (e.g., stimulus–running, stimulus–firing, and firing–running relationships)
were defined. Furthermore, we addressed your point on statistical clarity by providing
the full set of p values from all post hoc paired comparisons that were part of our
original analysis. Given the large number of comparisons, we have included the
complete results in a supplementary file to ensure transparency.

**The intersectional anatomical characterization of input in Figure 3 is valuable, but**
**for a more specialized audience.**

**Not addressed but I tend to disagree with the reviewer on that point. I think these**
**are important experiments to understand how different sensory inputs may**
**converge to the same behavioral response in TeA.**

**Key experiments for necessity are missing, which would rule out other pathways,**
**such as ACC, sensory-to-PAG, mesencephalic locomotor region, or cortico-striatal**
**pathways. Along these lines, more broadly, many circuits and pathways are known**

**to be involved in sensory-driven locomotion and thus the results of this paper are**
**a modest advance that would require more careful experiments, analysis,**
**presentation, and interpretation to be persuasive.**

**This point was not addressed by the authors. I believe it has more to do with the**
**conceptual aspect of the study and its relevance. I think the reviewer is correct,**
**but, to be fair, this criticism could apply to many other studies published in high-**
**profile journals. Ultimately, it all comes down to a narrative problem rather than**
**a study design issue: it remains unclear why the authors want to focus on TeA,**
**rather than another input to dPAG.**

We appreciate your question. Although the reasons for our focus on the TeA rather than
other inputs were mentioned in the Introduction, we recognize that the description may
not have been sufficiently clear. Therefore, in the newly submitted version, we have
revised the text to more explicitly state our rationale for concentrating on the TeA.
*“Moreover, the TeA sends projections to the dPAG, a region that controls defensive*
*behaviours^{32,42}. While emerging evidence has implicated the TeA in behavioural*
*responses such as the retrieval of pups in maternal mice (an innate care behaviour)⁴²*
*and auditory-cued fear conditioning (learned behaviour)^{41,43}, its specific role in the*
*defensive behaviour circuit remains poorly defined. This critical knowledge gap,*
*coupled with the robust anatomical connectivity of the TeA to key command neurons,*
*renders this region a highly compelling and potentially pivotal node worthy of in-depth*
*investigation. We hypothesize that the TeA likely serves not only as a simple relay station*

*but also as a critical hub coordinating diverse sensory-induced escape behaviours. It*
*may integrate sensory information and participate in driving downstream motor*
*command pathways, constituting a microcomputational hub that encompasses sensory*
*integration, decision-making, and motor command functions. It plays an indispensable*
*role in triggering adaptive escape behaviours.” [Introduction (line 82-94)]*

**Major concerns:**

**Organization of the results lacks introductory and other transitional sentences, so**
**it is not clear how one section relates to the next. For example, first the first two**
**sections with headers ‘Open field running model.’ And ‘turntable running model’**
**describe two behaviors but it is not clear where things are going. Are both going**
**to be used later? What is the purpose of the open field model given the turntable**
**model is described as having some advantages but no disadvantages?**

**The rational for the two behavioral conditions appears clear to me in the revised**
**version.**

**The framing around multisensory may create confusion, since the paper does not**
**look at multisensory integration (e.g. stimuli/cues with components/features from**
**multiple senses). Rather, it presents sound, light, or air puff on separate trials. As**
**such, they are studying a set of related associations for the 3 senses in isolation.**
**From this perspective, there are gross overstatements of the results, such as this:**
**“With the above experiments, we have established that the TeA-dPAG pathway is**

**the neural circuit associated with multisensory-induced movement.” And**
**“Therefore, the TeA must have the function of integrating various sensory**
**information.” This is partially addressed by experiments in Figure 4g-l; most of**
**the paper is about uni-sensory motor responses and framing should be adjusted,**
**accordingly.**

**This is not an issue in my opinion. The authors demonstrate the multi-sensory**
**aspect of TeA and sensory-induced escape behavior. They also suggest a form of**
**multisensory ‘integration’ in the extended Figure 4, but without overemphasizing**
**it. So, I think their claims are legitimate on this point.**

**The descriptions of analyses and selection of data to be presented are not clearly**
**motivated or described to give confidence in what is being compared and why.**

**The analyses of the data remain poorly explained.**

Thank you for your question. With respect to data analysis, we have now included
comprehensive details on how the temporal windows for different types of neuronal
responses (e.g., stimulus–running, stimulus–firing, and firing–running relationships)
were defined. Furthermore, we addressed your point on statistical clarity by providing
the full set of p values from all post hoc paired comparisons that were part of our
original analysis. Given the large number of comparisons, we have included the
complete results in a supplementary file to ensure transparency.

**The results are sloppily presented, with inconsistencies in presentation, a lack of**
**justification for selection of data for illustrations, and a lack of detail needed to**
**interpret statistical comparisons. Numerous examples are given below in ‘minor**
**concerns’ section, as well as a few ‘big ones’ here in the major concerns section.**

**The presentation of the results has been largely improved but the explanation**
**about statistical analyses remains elusive and insufficient. As an example, in figure**
**1 the responses to Sound, Light and Air-puff are compared. The figure legend does**
**not report the p values for the paired comparisons from the Bonferroni Post-hoc-**
**Test.**

Thank you for your question. This point has been addressed in our previous response.

**It is not clear if ‘Silencing TeA blocks running’ experiments are done in the freely**
**moving or head- fixed paradigm. Similarly, it is not clear in panel l and m of Figure**
**1 if these speed calculations are done for sound, light, or air puff. There is a smillar**
**issue with the optogenetic experiments, where it is not clear if the running under**
**investigation is triggered by sound, light, or air puff.**

**This issue has been clarified.**

**Physiological recording for the CNO experiments in Figure 1 are lacking, so it is**
**not clear what effect the CNO has had on TeA and other structures.**

**Done. See Figure 1o.**

**AAVretro is known to leak into neighboring cells. It would be helpful to confirm**
**the projection from TeA layer 5a to PAG using a confirmatory method, such as**
**pseudorabies or fluoro-gold.**

**The authors have performed retrograde labelling with CTB injected into the**
**dPAG. An example picture of the injection site in the dPAG and retrogradely**
**labelled neurons in TeA is shown only for the reviewers in the rebuttal. There**
**seems to be some neurons labelled in the layer V of TeA but clearly much fewer**
**compared to AAVretro labelling.**

**NB: I have noticed a possible error in figure 3d2 that shows two pictures of cortex**
**(TeA), left and right, instead of cells labelled in the cortex and injection site in**
**dPAG.**

You observed that following a CTB injection into the dPAG for retrograde labelling,
some neurons in layer V of the TeA appeared to be labelled, although the number was
clearly much fewer than those labelled with AAVretro. We also observed this
phenomenon and speculated that it may be due to the diluted concentration of CTB we
used, resulting in a lower labelling efficiency relative to the high titre of the viral vector.
Additionally, the error mentioned in **Fig. 3b2** has been corrected in the newly submitted
version.

**Fig. 3 | A TeA–dPAG circuit determines running behaviours. b,** Blockade of the TeA–dPAG
 pathway in mice running on the turntable: TeA_{dPAG} CaMKII neurons. **1,** Virus injection site and
 optogenetic (LED) interventions. **2,** Virus expression.

**Some images are saturated (e.g. 1k,q; 2a1; 3a,c; e1b; e2g; e3a) or show signs of**
 **injection damage (2a2), raising concerns about data quality. Images should be**
 **adjusted so they are not saturated and extent of tissue damage at injection site**
 **should be discussed and/or addressed as needed with new experiments.**

**I do not see much (if any) difference in the anatomical images in the revised version.**

We have re-examined the manuscript for overexposed images. The specific instances
 previously noted have been corrected; furthermore, any other images that were deemed
 nonessential have been removed as part of our broader effort to streamline the Results
 section.

**It seems possible that the CNO and optogenetic experiments in Figures 1 and 2 are**
 **a ‘hammer’ and are shutting to locomotion and behavior rather broadly, rather**
 **than indicating the specific multimodal integration and transformation to a motor**
 **signal that the paper suggests. It would be helpful to determine if locomotion in**

**other contexts (e.g. plus maze exploration) is impacted. Also, CNO injections into**
**virus control animals would be needed to mitigate e.g. tissue damage concerns.**

**The revised version provides control experiments for CNO and optogenetic**
**experiments, as well as inactivation experiments on freely-moving mice in open-**
**field. The problem is that in fact, the later experiments show that inactivation of**
**TeA_{dPAG} neurons or TeA_{dPAG} neurons markedly reduces spontaneous locomotion**
**thus questioning the overall interpretation of this circuit as being responsible for**
**sensory-induced escape response.**

We sincerely thank you for this important question. In fact, other reviewers have raised
similar concerns, and we have therefore revised and clarified the relevant descriptions
throughout the manuscript.

As you rightly noted, the inactivation of TeA or TeA_{dPAG} neurons affects spontaneous
motor activity, indicating that the TeA plays a broad role in movement regulation and
is not exclusively involved in sensory-evoked running. We agree that our initial
conclusion that TeA is “essential for sensory-triggered running” may have been
overstated. While TeA inactivation only partially reduces, rather than completely
abolishes, spontaneous movement, the results from both **Fig. 1** and **Fig. 3** show that it
fully blocks sensory stimulus-evoked running. These findings suggest that although the
TeA contributes to spontaneous locomotion, it also plays a necessary role in running
triggered by sensory signals.

We fully acknowledge the concerns raised by the reviewers. Indeed, the results in **Fig.**
**3c–d** weaken our earlier emphasis on the TeA as uniquely critical for processing sensory
stimulus-induced running. In the revised version, we have modified all related
assertions: the TeA is involved in sensory stimulus-evoked running, but it also
participates in running driven by other signals or nonsensory inputs.

We have toned down the strong claim made in the original manuscript that the TeA–
dPAG circuit is a specialized pathway for integrating sensory stimuli to initiate escape.
Accordingly, we have adjusted the language from “specifically responsible for” to
“participates in” throughout the text.

**In Figure 2, the responses to the 3 different stimuli should be parsed (walking to**
**sound, light, and airpuff).**

**Done.**

**The characterization of neuronal responses in TeA to running versus sound stimuli,**
**although intriguing, do not demonstrate much about the integration process.**

**Furthermore, running related signals are extremely prominent across neocortex,**
**and thus not surprising or very informative regarding this study.**

**Claims about multisensory integration have been removed.**

**The critical experiment of silencing the TeA-dPAG neurons and determining if**

**that blocks sound/light/air-puff induced locomotion appears to be missing. This is**
**the experiment that would test if those neurons are necessary for the behavior.**
**Without this experiment, it seems likely that many other pathways are also**
**involved and/or sufficient.**

**Done.**

**Minor concerns:**

**Figure 1f shows peak running speed and the authors claim the stimulation**
**intensities were carefully chosen to evoke similar running speeds. It would also**
**help to see average, rather than peak, running speed, given that this peak running**
**speed may be insensitive to differences in evoked running patterns by the stimuli**
**used.**

**OK.**

**How were 10 trials shown in 1c selected? What does these plots look like for**
**different animals and single trial?**

**What the authors forgot to explain is that there is no ‘selection’ of the trials since**
**only 10 repetitions are presented for each stimulus. See M&M (lines 1016-1018):**
**“ The three sensory stimuli were presented in a pseudorandom manner with a 25-**
**s interstimulus spacing, and this process was repeated 10 times. Each recording**
**session lasted 15 minutes.”**

Yes, you are correct that our experiment did not include a choice paradigm. We have
supplemented and refined the relevant description in the Methods section in the newly
submitted version. *“Each stimulus was repeated only 10 times to prevent habituation in*
*mice, which could diminish behavioural responses, while also ensuring the validity of*
*our results.”* [Methods (line 1076-1078)]

**It is not clear in Fig. 1d-f how the stats were done. There are 7 animals and 7 data**
**points, suggesting each point is an animal. This should be stated. The type of post**
**hoc test is not stated. And the F values are reported as 2,18 for degrees of freedom.**
**How do these numbers relate to the 7 mice and 7 data points?**

**This problem appears to have been partially corrected. However, the authors do**
**not report p-values for paired comparisons resulting from the Bonferroni post-hoc**
**test. Overall, the authors' poor explanation of this makes me question their general**
**understanding and appropriate use of statistics. Furthermore, nonparametric**
**tests are generally recommended for small sample sizes.**

We addressed your point on statistical clarity by providing the full set of p values from
all post hoc paired comparisons that were part of our original analysis. Given the large
number of comparisons, we have included the complete results in a supplementary file
to ensure transparency. For all the statistical analyses, we initially assessed the
assumptions of normality and homogeneity of variance. Although the sample sizes were
relatively small, we opted for parametric tests because of their generally higher

statistical power. However, when these assumptions were not met, we appropriately
switched to nonparametric tests.

**Why are walk bout durations not shown for the freely-moving task in Figure 1?**

**Done.**

**It is not clear if it is the same, or different, 7 animals used in the freely moving**
**versus head-fixed behavior.**

**Clarified.**

**This statement “However, the characteristics of the running, such as the**
**probability, delay, and peak speed (Fig. 1j), were not significantly different**
**between the turntable model and the open field model (Fig. 1c-f), except for the**
**running duration.” Is not justified based on what is shown in panel ‘j’, at least as**
**described. Again, how the F values were determined, and what the groups were,**
**etc., is not clear.**

**Resolved.**

**The co-localization rate reported in Fig. S1 is ambiguous; they should report both**
**false positive and false negative rates (i.e. were there any retrogradely labeled cells**
**that were negative for camKII/SOM/VIP/and PV? Also, why was AAVretro used**
**for the CaMKII labeling?**

**I think the explanations from the authors are reasonable. They could indeed have**
**reported the two proportions of overlap for each labeling (for instance the**
**proportion of CaMKII+&EYFP+ / CaMKII+ and CaMKII+&EYFP+ / EYFP+**
**neurons). But I do not think this is an important issue for this study.**

**The purpose of this sentence is unclear: “Due to the scarcity of cholinergic and**
**dopaminergic neurons in the cortex, we did not perform corresponding tests on**
**these neuron types.” Why are cholinergic and dopaminergic neurons singled out**
**for mention? And in fact there are cholinergic neurons in neocortex (Von**
**Engelhardt, J., Eliava, M., Meyer, A. H., Rozov, A., & Monyer, H. (2007).**
**Functional characterization of intrinsic cholinergic interneurons in the cortex.**
**Journal of Neuroscience, 27(21), 5633-5642.).**

**Resolved.**

**Statistical quantification of significant reduction for Figure 4d-f is lacking.**

**Done.**

**Experiments in Figure 4m-r would benefit from a CNO or optogenetic**
**manipulation. More broadly, some form of psychometric manipulation to**
**understand how graded responses varies with graded manipulation would**
**strengthen the manuscript.**

**This part has been moved to supplementary which I think is fine.**

Dear Reviewer,

The line numbers referenced in our point-by-point responses correspond to the
"Revised manuscript without track changes-He Li.pdf" file.

**Reviewer #4 (Remarks to the Author):**

**I want to thank the authors for the considerable amount of work that has**
**improved the manuscript in significant ways. The points were addressed with**
**attention, references to other work, the companion paper and explanation of**
**reasoning.**

**The manuscript is improved with additional controls, panels and supplementary**
**data to aid interpretation of the data. I maintain that with revision the manuscript**
**offers exciting data for the neuroscience community to move forward.**

**I have 2 major comments and a few minor comments. Line numbers are for the**
**markup document.**

**1) Onset latency to running**

**Although the authors provide more information about the neural and behavioral**
**latencies, and discussion of their relationships, I am still puzzled by the proposed**
**circuit and behavior.**

**Line 204 says “triggered immediate escape behaviors”, but the data show a latency**

**of 1-3 seconds until mice start running to the other side. Auditory startle responses**
**are within 50-100 ms, so this onset latency can hardly be called immediate.**

**In the rebuttal, the authors say that the mice often changed their behavior after**
**the aversive stimuli: “Furthermore, our experimental video recordings revealed**
**that mice typically exhibit brief vigilance (e.g., head turning, ear twitching) after**
**sensory stimulation, followed by the initiation of running, rather than entering a**
**motionless state.”**

**I believe these observations are instrumental. They show that running speed is an**
**incomplete description of the behavior and that changes in behavior can be**
**observed before any changes in locomotion. This means that in parallel to the**
**proposed circuit there are likely pathways that process the sensory stimuli and**
**participate in motor programs involved in vigilance and orientation behaviors.**

**I strongly suggest the authors to discuss (1) whether they consistently observe**
**other behavioral changes before locomotion onset and this explains the latency to**
**running, and (2) whether they believe that sensory to motor processing in TeA is**
**central to all sensory-induced behaviors observed in this paradigm, or specific to**
**locomotion (with faster circuits operating in parallel to this slow synaptic**
**integration in TeA to induce locomotion?).**

We fully concur with your perspective. In the newly submitted version, we have revised
the description of the movement state of the mice following stimulus perception in the
Results section. “*When the mice were located in one compartment of the experimental*

*chamber, the random application of sound, light, or air puff stimuli at their current*
*position invariably caused them to move away from the stimulus source to the opposite*
*compartment, thus resulting in escape behaviour. No measurable freezing behaviour*
*was detected either during stimulus presentation or during the execution of escape*
*movements¹ (Fig. 1a).” [Results (line 114-115)]*

Furthermore, we have added explanations addressing the two main points you raised to
the Discussion section in the new version of our manuscript.

Discussion Section: “*Our data reveal that sensory stimulation in mice triggers robust*
*escape running but with a significant delay of 1–3 s (Figs. 1d, g, m, o and 4b, r).*
*Although decreasing the stimulus intensity prolongs the response latency (Fig. 4b), this*
*latency is markedly longer than that of subcortically controlled startle reflexes, such as*
*the auditory startle response, where rapid reactions typically occur on a millisecond*
*timescale^{46,47}. A delay of several seconds suggests a decision-making process involving*
*more complex neural integration rather than simple, reflexive, “immediate” escape.*
*Our observations revealed that the mice were not unresponsive during this period.*
*Although they did not exhibit freezing behaviour, we observed brief vigilant and*
*orienting behaviours (e.g., head turning and ear twitching). This finding indicates that*
*the sensory stimulus likely triggers a cascade of neural processes and behaviours*
*before culminating in full-flight escape. We quantified the frequency and latency of*
*these alert behaviours. They occurred with a high probability (probability of head*

*turning in the open field escape model: $89.02\% \pm 15.35\%$; probability of sound-induced*
*ear twitch in the turntable running model: $96\% \pm 8.944\%$) and a short latency (latency*
*of head turning: 298.1 ± 94.7 ms; latency of ear twitch: 194.6 ± 53.6 ms).*

*Therefore, we propose that the identified TeA pathway is not the sole initiator of all*
*sensory-evoked behaviours in this paradigm but may play a more specific role in the*
*final commitment to and execution of coordinated escape locomotion. The initial, faster*
*components of the reaction (e.g., vigilance and orientation) are likely mediated by*
*parallel subcortical pathways that process the threat rapidly and prime the animal for*
*action. The longer latency of the TeA-driven escape is consistent with the polysynaptic*
*nature of this cortico-amygdalar pathway, which may allow for more integrated*
*processing of sensory context before triggering a major locomotor action⁴⁸. In this*
*model, the TeA pathway acts as a critical gate, translating the integrated assessment of*
*a threat into the decisive motor program of running to shelter.*

*This behavioural sequence suggests the existence of a parallel processing architecture*
*within the brain. We speculate that a fast subcortical pathway (e.g., potentially*
*involving the superior colliculus and/or amygdala) might be responsible for the initial*
*rapid vigilance and orienting responses to such aversive stimuli. Simultaneously, the*
*slower integration process we observed in the TeA might be crucial for constructing a*
*more complex threat representation—perhaps by integrating the sensory stimulus with*
*contextual information—and ultimately contributes to the decision to coordinate and*

*execute a coordinated escape plan rather than merely eliciting a simple startle response.*
*Thus, our study suggests that the TeA is not the exclusive pathway for initiating*
*perceptually guided behaviour but is a key node involved in the translation of integrated*
*threat signals into sustained escape motor commands.” [Discussion (line 752-790)]*

**2) TeA-dPAG specificity to sensory-induced flight behavior**

**The inclusion of new results regarding the role of the proposed circuit in an open**
**field arena are powerful, but bring new interpretations to the table. The fact that**
**inhibition of the TeA-dPAG neurons also affects non-sensory induced locomotion**
**generally degrades the claim that this pathway “integrates information from**
**diverse sensory stimuli into running signals for inducing escape” (from Abstract).**
**These results need to be discussed in the main text and – based also on the opinion**
**of the other reviewers – the authors should attenuate the claim that the TeAdPAG**
**is specifically involved in the integration of sensory stimuli to evoke flight**
**throughout the manuscript.**

Thank you for your suggestion. We have toned down the strong claims in the original
manuscript that emphasized the specific role of the TeA–dPAG neural circuit in
integrating sensory stimuli to trigger escape. We have reframed the language to be more
moderate, shifting from phrases such as “specifically responsible for” to “plays a role
in”. Additionally, we have expanded the Discussion section to include further
clarification of these points. *“However, chemogenetic and optogenetic inhibition of TeA*

*completely blocked the flight behaviours induced by multiple sensory stimuli (Fig. 1r–*
*w). The critical role of the TeA in this process was further underscored by the specific*
*inhibition of the TeA–dPAG circuit, which also abolished escape responses (Figs. 3a–b*
*and 7k–n). Conversely, its sufficiency in driving running was demonstrated by the direct*
*activation of this circuit (Fig. 7b), positioning it as a key efferent pathway for escape.*
*Importantly, the finding that inhibition of the TeA–dPAG circuit reduces spontaneous*
*locomotion (Fig. 3c–d) suggests that its function extends beyond threat-evoked escape*
*to include a more general role in modulating locomotor activity. This finding positions*
*the TeA–dPAG pathway not only as a dedicated escape circuit but also as a key*
*regulator of motor output that can be powerfully engaged by threatening stimuli. In*
*addition to the SC^{49,50} and prefrontal cortex, our findings provide a new brain region*
*and model for quantitative and qualitative studies of multisensory integration and*
*behaviour initiation.” [Discussion (line 737-750)]*

**Minor comments:**

• **Line 2953, smoothing: what is the unit for the window width of 5?**

The unit is 0.1 seconds, meaning that the window width is 0.5 seconds.

• **The exclusion criteria and fraction of data excluded should be included in the**
**manuscript for transparency and reproducibility (based on replies to my**
**comments on line 1585-1589, 1629-1634 in rebuttal).**

We thank you for this question and apologize for the oversight in our previous point-
by-point response. Although the exclusion criteria and the proportion of excluded data
are clearly stated in the Methods section, we inadvertently omitted this information in
our replies, which may have caused confusion. The relevant details can be found in the
Methods section. *“Following three consecutive days of daily training (1–2 hours/day),*
*95% of the animals successfully adapted to the head-fixed experimental paradigm,*
*demonstrating stable, self-initiated locomotion on the rotary disc platform with*
*coordinated limb movements. Animals that failed to maintain such locomotor stability*
*(5%), characterized by limb coordination deficits or complete immobility, were*
*excluded from subsequent recordings and analysis to ensure experimental consistency.”*

**[Methods (line 1058-1063)]**

• **Line 1593 in rebuttal: The updated methods regarding the ‘batch processing’ are**
**still quite unclear to me. Data was ‘pooled’ across what? Why was data processed**
**in ‘randomized batches’? Was the running speed data simply aligned to stimulus**
**onset and averaged across stimulus conditions?**

We apologize for the confusion caused by our nonprofessional and even incorrect
description. We would like to reorganize our explanation to clarify how we processed
the recorded animal movement speeds and their relationships with different stimuli in
response to your current and previous questions. During the behavioural tests, the

rotation speed of the turntable was measured with a rotary encoder and recorded in real
time. For each animal, we delivered three types of sensory stimuli—sound, light, and
air puff—in a pseudorandom order, with each stimulus repeated 10 times. Thus, a total
of 30 stimulus–movement trials were recorded per animal. After each animal was
recorded using the rotary encoder, we obtained its respective stimulus–movement
dataset. After all the experiments were completed, we compiled data from multiple
animals. These datasets were then processed for each animal using custom MATLAB
scripts. The specific procedure involved classifying each animal’s 30 stimulus–
movement trials by stimulus type, resulting in three groups: 10 sound–speed curves, 10
light–speed curves, and 10 air puff–speed curves. Each group of curves was aligned to
the stimulus onset and averaged, yielding a single averaged movement speed curve per
stimulus type for each animal.

We have revised the Methods section in the newly submitted version to provide a clearer
description of this process. *“The recorded running speeds were aggregated and
processed for each animal individually using custom MATLAB code. The 30 stimulus–
movement trials per animal were categorized by stimulus type, resulting in three groups:
10 sound–speed curves, 10 light–speed curves, and 10 air puff–speed curves. Each set
of curves was then aligned to the stimulus onset and averaged, producing a mean
movement speed curve for each animal in response to each type of sensory stimulus.”*

**[Methods (line 1356-1361)]**

• **Line 1440: ‘extractive’, what does this mean?**

We apologize for the unclear expression due to our nonnative English proficiency. We
originally intended to convey that the speed raster plot in the left panel of **Fig. 1k**
represents responses to three distinct sensory stimuli—sound, light, and air puff—each
delivered in a pseudorandom order and repeated 10 times, resulting in a total of 30
stimulus–movement trials. The term “extracted” was used to indicate that we classified
the 30 mixed trials by stimulus type (10 trials per type) and grouped them for
visualization.

In the newly submitted version, we have revised the description in the figure caption to
clarify this process. “*k*, Example speed raster plots showing flight responses evoked by
three stimuli presented randomly (left panel) and regrouped by stimulus type for
comparison (right panel). Each stimulus was repeated 10 times.” [Main figure legends
**Fig1. (line 1709-1711)**]

Dear Reviewer #1,

The line numbers referenced in our point-by-point responses correspond to the "Revised manuscript without track changes-He Li.pdf" file.

Reviewer #1 (Remarks to the Author):

In this revised manuscript, the authors have addressed most of my comments. But there remains a few changes to be made before this paper can be accepted for publication:

- I would suggest a more specific title that includes explicit reference to the TeA and mouse model.

Thank you for your valuable suggestion. As suggested, we have revised the title to explicitly include reference to both TeA and the mouse model used. The new title is: "*An intralayer microcircuit in the temporal association cortex underlies sensory-induced escape in mice*". **(line 1–2)**

The running title: "*Running title: A TeA intralayer microcircuit for escape*". **(line 4)**

- line 286-287: "Three types of neurons in the TeA may regulate signalling that running behaviour" ???

We have revised this subheading in light of the subsequent results. The updated heading: "*Identification and characterization of three functional neuron types in the TeA region*". **[Results line (286-287)]**

- Figure 4: report real mean or median values, not the ones computed from Gaussian fitting.

We have updated Figure 4 to present the real mean values. All relevant results are now reported using the actual mean values rather than those derived from Gaussian fitting.

Fig. 4 | Sensation-related and running-related neurons in the TeA. **f**, Left panel, Distribution of the maximum correlation coefficient (r_{\max}) from the analysis of all recorded TeA R-neurons (**Extended Data Fig. 2d**), where the r_{\max} for each neuron exceeded the 95% confidence interval derived from permutation testing ($P < 0.05$ for all cells). The red dashed line represents the Gaussian fit curve ($R^2 = 0.9324$, $P = 3.66 \times 10^{-6}$). Right panel, Distribution of the optimal time lag from the analysis of R-neurons ($R^2 = 0.9165$, $P = 1.125 \times 10^{-14}$). **j**, The same as (**f**), left panel, analysis of all recorded SR-neurons (**Extended Data Fig. 2e**), where the r_{\max} for each neuron exceeded the 95% confidence interval derived from permutation testing ($P < 0.05$ for all cells). ($n = 66$ cells, $R^2 = 0.6075$, $P = 2.1947 \times 10^{-6}$). Right panel, Analysis of SR-neurons ($R^2 = 0.7285$, $P = 3.8492 \times 10^{-6}$).

- Lines 381-397: I maintain that it is not correct to realign each running event to the preceding neuronal event with variable delays! This section is flawed and unnecessary, and should be removed. The report of the max correlation coefficients (r_{\max}) and lags from the cross-correlations is sufficient.

Thank you for your valuable comment. For the sake of clarity and focus, we have removed the section describing the realignment of each running event to its preceding neuronal event with variable delays. The revised manuscript now only reports the key outcomes, specifically the maximum correlation coefficients (r_{\max}) and the corresponding lag values obtained from the cross-correlation analyses.

- As indicated by the authors “the low overlap rate between hSyn and CaMKII could result from differences in the efficiency of the expression of the two types of

viruses". Indeed, the images in the Extended Data Figure 4d clearly show a significant number pyramidal cells only labeled with EYFP (green, hSyn promoter) and not by mCherry (CaMKIIa promoter). Therefore, I do not think that this part of the data is really informative.

We thank you for this insightful comment. While we had complementary data from both GAD67 immunostaining and the dual-virus labeling, the key conclusion that these projections are predominantly glutamatergic is primarily supported by the definitive GAD67 staining. The dual-virus experiment, although illustrative, introduces unnecessary complexity given the known technical variations. To streamline the narrative and focus on the core findings, we have restructured this section. We now first directly establish that "the projections are predominantly glutamatergic" using the conclusive GAD67 immunostaining data. The dual-virus imaging and its statistics are retained but presented as a supplementary observation, accompanied by a note on its technical limitations. This entire section has been significantly condensed and serves specifically to justify our rationale for selecting the hSyn promoter for subsequent functional manipulations.

The revised text in the Results section: "*We also investigated the neuronal types of TeA neurons receiving sensory projections (SensTeA neurons). AAV-Cre was injected into different sensory cortices, and AAV-hSyn-DIO-mCherry was injected into the TeA to label neurons receiving distinct sensory inputs in the TeA (Extended Data Fig. 4c). Direct identification of these neurons was performed via GAD67 (a marker for GABAergic neurons) immunostaining (Extended Data Fig. 4d). The results revealed an extremely low proportion of GABAergic neurons among these SensTeA neurons. The colocalization rates were 0% for the AC, 0% for S1, 0% for V2L, 0.40% for the MGB, and 0.90% for the DLG projections (Extended Data Fig. 4e). Since the cerebral cortex is primarily composed of glutamatergic and GABAergic neurons, these findings suggest that the SensTeA neurons receiving projections from different sensory areas, are mostly glutamatergic neurons.*

We also attempted parallel labeling using a dual-virus strategy (CaMKII α -DIO-mCherry and hSyn-fDIO-EYFP) (Extended Data Fig. 4f, g). We observed that the colocalization ratio of EYFP (green, broad spectrum) and mCherry (red, glutamatergic) signals in labeled TeA neurons varied depending on the input region, with rates of AC: 50%, S1: 60%, V2L: 66%, MGB: 48%, and DLG: 51% (Extended Data Fig. 4e). The low overlap rate is likely attributable to differences in the expression efficiency of the two viral vectors. Therefore, based on the conclusive evidence from GAD67 staining (showing a near absence of GABAergic neurons) and to circumvent the technical limitations of dual-virus co-labeling efficiency, we selected the broadly expressed AAV-hSyn promoter for all subsequent labeling and manipulations. This approach allows us to confidently and comprehensively target the *Sens*TeA neuron population, which we have established as predominantly glutamatergic." [Results line (449-472)]

Extended Data Fig. 4 | Main source regions projecting to the TeA. c, f, Virus injection sites in sensory source regions (AC, S1, V2L, MGB, and DLG) and the TeA. d, Images showing mCherry- and anti-GAD 67-labelled neurons. e, Percentages of neurons with overlapping labelling with GAD 67 in d and CaMKII with in g. GAD labelling: n = 4, 4, 5, 4, and 5 slices from 3, 4, 3, 3, and 3 animals. CaMKII labelling: n = 6, 5, 5, 6, and 6 slices from 4, 4, 3, 4, and 4 animals. g, Images showing mCherry- and EYFP-labelled *Sens*TeA neurons.

- Lines 506-508: Incorrect phrasing. In these experiments the inhibitory receptor

(hM4D-Gi) is expressed in TeA neurons receiving inputs from sensory cortical or thalamic areas. So technically, the CNO will not block the sensory inputs from cortex or thalamus but instead block the activity downstream to SENS-TeA neurons.

We thank you for this insightful comment. We have corrected all related phrasing in this section. The revised description:

*"CNO was injected i.p. to selectively inhibit the activity of these targeted *SensTeA* neurons (Fig. 5g). As an example, we show hM4D-mCherry expression in *ACTeA* neurons (Fig. 5h). When activated by CNO, *ACTeA* neurons were inhibited, and both the running speed and the probability of running behaviours driven by the three sensory stimuli decreased (Fig. 5i). We systematically assessed the behavioral effects of inhibiting *SensTeA* neurons defined by inputs from the AC, MGB, S1, V2L, and DLG on the peak running speed and the probability of inducing running behaviours. Chemogenetic inhibition of these neuronal populations significantly reduced the probability of sound-induced flight behaviors. This effect was more pronounced when the TeA neurons targeted by AC and MGB projections were inhibited (Fig. 5j). In addition, inhibition of TeA neurons receiving inputs from V2L, DLG, or AC significantly affected light-induced flight behaviours (Fig. 5k), while inhibition of those receiving S1 input significantly affected air puff-induced flight behaviours (Fig. 5l). These results demonstrate that *SensTeA* neurons exhibit modality-specific functional roles in driving flight behaviors. While TeA neurons receiving AC or MGB inputs are crucial for sound-induced flight, those receiving V2L/DLG inputs are crucial for light-induced flight, and those receiving S1 inputs are crucial for air puff-induced flight. This input-defined functional specificity ensures appropriate behavioral responses to distinct threats."*

[Results line (491-509)]

- Lines 510-511: No. These results show that SENS-TeA neurons are somewhat modality specific (at least partially).

We have revised the relevant text in the Results section as suggested. The revised description: "*These results demonstrate that $SensTeA$ neurons exhibit modality-specific functional roles in driving flight behaviors. While TeA neurons receiving AC or MGB inputs are crucial for sound-induced flight, those receiving V2L/DLG inputs are crucial for light-induced flight, and those receiving SI inputs are crucial for air puff-induced flight. This input-defined functional specificity ensures appropriate behavioral responses to distinct threats. Despite this specificity, all three sensory inputs contributed to the overall regulation of flight behaviours. The presence of bisensory responsive neurons partly explains the influence of different stimuli on movement speed (Fig. 5e and f), although the impact of multisensory integration cannot be excluded.*"

[Results line (504-513)]

- Lines 570-584: In the end only 5 cells of each group were reconstructed. The authors should clearly indicate that they could only reconstruct the dendritic arborization from a small subset of neurons that were isolated enough. The authors should include the images provided in the rebuttal (line 157-158) as Extended Figure.

We have added a detailed description of the neuronal morphology reconstruction process in this section of the Results. The images provided in the initial rebuttal have now been incorporated as a new Extended Data Figure (Extended Data Fig. 6). The revised description: "*We characterized the morphology of these two types of neurons. Performing three-dimensional reconstructions in regions with high viral expression intensity (Fig. 6a) posed certain challenges. For single-cell 3D reconstruction, we preferentially selected images containing only one isolated neuron (Extended Data Fig. 6a and c). A secondary strategy was to choose regions with relatively sparse viral labeling (Extended Data Fig. 6b and c, the white dashed boxes in the left panels). When two or more neurons were located in close proximity, it often became difficult to trace dendritic arbors accurately in two-dimensional planes. In such cases, we prioritized 3D reconstruction, rotating the model to broadly assess neuronal morphology, and*

made efforts to select neurons whose structures remained discernible after rotation (as indicated by the white arrows in the right panels of Extended Data Fig. 6b and d). Following these selection criteria, five cells from each group were ultimately chosen for reconstruction. The reconstruction results revealed that both neuronal types exhibited a pyramidal morphology characterized by an apical dendritic tuft and basal dendrites around the soma." [Results line (569-583)]

Extended Data Fig. 6 | Three-dimensional reconstructed images showing the morphology of individual SensTeA and TeA_{dPAG} neurons. a, 3D reconstruction of an individual labeled SensTeA neuron (left panel) and a magnified view highlighting its morphological details (right panel). **b**, 3D reconstruction of SensTeA neurons in a region with higher viral labeling density (left panel). The white dashed area indicates a sparsely labeled region selected for morphological reconstruction. The neuron used for reconstruction is marked by a white arrow (right panel). **c**, **d**: Same as **(a)** and **(b)**, respectively, but shown for TeA_{dPAG} neurons. Scale bars: 100 μm .

- Lines 634-636 and Figure 7: Considering ChR2- cells in the experiments labelling the cells receiving inputs from AC, it is quite speculative to consider them as TeA_{dPAG} neurons. It would be more rigorous and less confusing to call them 'AC-

TeA-ChR2-'

We thank you for your insightful suggestion. We agree that our original terminology could be misleading. In the revised manuscript, we have adopted your recommended naming convention. Additionally, we present a more cautious discussion regarding whether these neurons can be considered TeA_{dPAG} neurons, based on the results of their recordings and statistical analysis. The revised text in the Results section is as follows:

"We then performed in vivo recordings in the TeA to identify neurons that were activated by this pathway but were not directly expressing ChR2. We identified the locally recorded, light-responsive, ChR2-negative neurons as AC- TeA -ChR2⁻ responsive neurons (hereafter referred to as AC-TeA-ChR2⁻ neurons). These cells represent the functional postsynaptic targets of the AC to TeA pathway in our experimental paradigm. These AC-TeA-ChR2⁻ neurons also responded to photostimulation. Their firing rates gradually increased across successive light pulses until reaching a plateau (Fig. 7d). This pattern was mirrored in the animal's behavioral output: the evoked running speed also increased progressively across trials before stabilizing (Fig. 7d). In contrast, the firing of the presynaptic AC-TeA-ChR2⁺ neurons remained stable throughout the stimulation train (Fig. 7c), indicating that the observed facilitation originated within the local TeA circuit. The observed facilitation of both postsynaptic firing and behavior suggests that the AC to TeA pathway engages a local microcircuit that exhibits short-term plasticity. Crucially, this microcircuit is positioned to directly influence running behavior." **[Results line (645-659)]**

"Notably, we observed a highly similar time difference between firing and the initiation of running in AC-TeA-ChR2⁻ neurons (2.062 ± 0.2133 s) and TeA_{dPAG}-ChR2⁺ neurons (2.076 ± 0.1480 s) (Fig. 7j). Combined with the finding that the facilitatory profile of AC-TeA-ChR2⁻ neuron firing precisely matched the facilitation of running speed evoked by AC-TeA-ChR2⁺ stimulation positions the activity of AC-TeA-ChR2⁻ neurons as a direct predictor of behavioral plasticity. The convergence of these congruent temporal and dynamic response properties provides evidence that AC-TeA-ChR2⁻ neurons are

functionally equivalent to, and likely encompass, the TeA_{dPAG} neuron population that directly governs running behavior." [Results line (686-694)]

Fig. 7 | Firing and running evoked by the activation of $AC\ TeA$ and TeA_{dPAG} neurons. a, b, Virus injection and LED light delivery sites. **c–e,** Firing rates of $AC\ TeA$ neurons expressing ChR2 (**c**), $AC\ TeA$ neurons not expressing ChR2 (**d**) and TeA_{dPAG} neurons expressing ChR2 (**e**) that responded to the speed of running behaviours evoked by blue LED light stimulation (20 Hz, 5 s, 10 trials). **f, g,** Firing rate (**f**, $F_{2,24} = 74.87$, $P < 10^{-6}$; $n = 12, 7$, and 8 cells) and running speed (**g**, $F_{2,15} = 98.0402$, $P < 10^{-6}$, $n = 6$ animals/group). One-way ANOVA with the Tukey's post hoc correction. **h,** Correlations between the firing rate and running speed. $AC\ TeA\ -ChR2^+$: 0.8980 ± 0.0737 ; $AC\ TeA\ -ChR2^-$: 0.8301 ± 0.0879 ; $TeA_{dPAG}\ -ChR2^+$: 0.8537 ± 0.0792 ; $n = 12, 7$, and 8 cells, respectively. **i,** Firing latency. $F_{2,24} = 468.9227$, $P < 10^{-6}$; one-way ANOVA with the Dunnett T3 post hoc correction. $n = 12, 7$, and 8 cells. **j,** Difference in time between firing events and running events. $F_{2,72} = 45.1575$, $P < 10^{-6}$; one-way ANOVA with the Bonferroni post hoc correction. $AC\ TeA\ -ChR2^+$: 2.483 ± 0.1661 ;

AC-TeA-ChR2⁻: 2.075 ± 0.2017 ; TeA_{dPAG}-ChR2⁺: 2.059 ± 0.1474 ; n = 12, 7, and 8 cells.

Dear Reviewer #4,

The line numbers referenced in our point-by-point responses correspond to the "Revised manuscript without track changes-He Li.pdf" file.

Reviewer #4 (Remarks to the Author):

This revised version of the manuscript presents clear improvements. However, the manuscript remains problematic in the disentanglement of sensory and motor related signals. In sum, I am not convinced about the presence of clear sensory-evoked activity, which presents a key aspect of the proposed model where within TeA diverse sensory signals are transformed into motor command signals.

First, the new raster plots are unclear to me. For example:

Extended Data Fig. 2 | S- and R-neurons in the TeA. a, Responses of the R-neurons to three types of sensory stimuli: auditory (sound), visual (light), and tactile (air puff). Top panel, Population raster plot and the corresponding PSTH of firing activity. Bottom panel, Superimposed running speed curves from all trials (n = 156 trials). The blue bars indicate the stimulus period (duration: 5 s).

In the legend it says response of the R-neurons and Population raster plot, but these must be individual example neurons (not a population of neurons) with each row representing individual trials, is that correct? A figure that shows the responses across all neurons of a certain category is then lacking. (If it is not, and each row is the mean activity across trials for one neuron and it is indeed a population plot, then there is just one spike for some neurons, and the legend is incorrect as these must be different numbers of trials for neurons recorded on different days.)

We sincerely thank you for this insightful critique regarding the presentation of

neuronal responses in the original Extended Data Fig. 2. We fully recognize the lack of clarity in our previous data presentation and terminology. The "raster plots" were indeed intended to illustrate the activity of a single, representative R-neuron, with each row representing an individual trial for that neuron. Consequently, the labels "Population raster plot" and "Responses of the R-neurons" for that specific panel were inaccurate and misleading. We acknowledge this error and apologize for the confusion. We have now addressed these issues comprehensively in the revision, as detailed below.

To clarify, the raster plots in Fig. 4d1 depict individual example neuron (Fig. 4c). For R-neuron (Fig. 4c), the relationship between sensory stimuli and neuronal firing was presented using raster plots combined with PSTH (Fig. 4d1). To determine whether R-neuron responded to sensory stimuli, only events where the sensory stimulus did not trigger running behavior were selected for analysis. These sensory stimulation events were aligned to the onset of the stimulus.

Following your suggestion, we have replaced the original raster plots in Extended Data Fig. 2 with new heatmaps. We have now generated population-level heatmaps for R-neurons (Extended Data Fig. 2a) in response to all three sensory stimuli (sound, light, air puff). In these new heatmaps, each row represents a single neuron, and the displayed activity is the neuron's average firing rate across all valid trials. The criteria for event selection and alignment for generating these population heatmaps are identical to those used for the single-neuron analyses presented in Fig. 4.

Figure 4

Extended Figure 2

Fig. 4 | Sensation-related and running-related neurons in the TeA. **c**, Characteristics of a representative running-related neuron (R-neuron) in the TeA: firing rates (black); sound (S), light (L), and air puff (A) waveforms (blue); and running speed (red). The ordinates for the firing rate and running speed are on the right and left, respectively. **d1**, Responses to the three sensory stimuli in (c). Top panel, Raster plots of individual trials and their corresponding running speed curves. Bottom panel, PSTH of firing activity ($n = 6$ trials). Blue bars: stimulus period (duration: 5 s).

Extended Data Fig. 2 | S- and R-neurons in the TeA. **a**, Responses of the R-neurons to three types of sensory stimuli: auditory (sound), visual (light), and tactile (air puff). Top, Heatmaps depicting neural activity of all R-neurons to sound, light, and air puff stimuli, aligned to stimulus onset (2 s) and averaged across trials without behavioral movement (0.5 cm/s). Each row represents the

averaged activity of a single R-neuron across all trials. The color bar on the right indicates firing rate intensity (Hz). (n = 29 neurons for sound, light and air puff, respectively). Middle, Population peri-stimulus time histograms (PSTHs) for all R-neurons in response to sound, light, and air puff stimuli, respectively. Bottom, Running speed aligned to stimulus onset for each neuron (each trace is the average across all trials presented during that neuron's recording; n = 29 neurons for sound, light, air puff, respectively). The blue bars indicate the stimulus period (duration: 5 s).

Extended data figure 2f and 2g show 2 example S neurons that do not show stimulus-locked onset latency, arguing against strongly sensory driven activity. Similarly, Extended Data Fig 3 shows only example neurons and only for 1-3 individual stimulus presentations, too little to judge stimulus-locked sensory-evoked activity that is independent of behavioral changes.

Thank you for this comment. We acknowledge that the initial presentation of the data was not correct. We agree that the original selection and presentation of examples were insufficient to effectively demonstrate stimulus-locked sensory responses. We have taken the following corrective measures:

The original examples in Extended Data Fig. 2f and 2g were not suitable for clearly illustrating stimulus-synchronized responses. Upon re-evaluation, we identified a more fundamental issue: these figures employed a population-level raster format which inadvertently conflated responses to different sensory stimuli. This was misleading and failed to effectively convey the intended message. Given these deficiencies and the focus of the core population-level analysis, we have completely removed the original Extended Data Figs. 2f and 2g.

Similarly, to analyze whether S-neurons respond to sensory stimuli, the raster plots were intended to illustrate the activity of a single, representative S-neuron, with each row representing an individual trial for that neuron. Therefore, when presenting individual example S-neuron in Fig. 4 (Fig. 4g, k), we used raster plots combined with

PSTH to demonstrate the relationship between sensory stimuli and neuronal firing for the S-neuron (Fig. 4h1, h2). To determine whether S-neuron responded to sensory stimuli, we selected only sensory stimulation events that did not elicit running behavior for analysis. Each sensory stimulation event was aligned to the onset of the stimulus.

In the revised Extended Data Fig. 2, and in accordance with your suggestion, we have generated population-level heatmaps for all recorded S-neurons (Extended Data Fig. 2c) in response to the three sensory stimuli (sound, light, air puff). In these new heatmaps, each row represents a single neuron, and the displayed activity corresponds to the neuron's average firing rate across all trials. The criteria for event selection and alignment are identical to those used for the single-neuron analyses presented in Fig. 4.

Figure 4

g

h1

k

h2

Extended Figure 2

c

Fig. 4 | Sensation-related and running-related neurons in the TeA. g, The same as (c), characteristics for a representative SR-neuron (responds to the sound stimulation (SS) of TeA). The black dashed lines indicate events that can be used to analyze the correlation between neuronal firing and sensory stimuli, where the sensory stimuli did not induce running. **h1**, Example SR-neuron from

(g) responding to SS. Top panel, Raster plots of individual trials and their corresponding running speed curves. Bottom panel, PSTH of firing activity (n = 5 trials). Blue bars: stimulus period (duration: 5 s). **h2**, The same as (h1), analysis of (k). **k**, The same as (g), characteristics of a representative unSR-neuron (responds to SS) of TeA.

Extended Data Fig. 2 | S- and R-neurons in the TeA. **c**, Responses of S-neurons (including 66 SR-neurons and 9 unSR-neurons) to three sensory stimuli. Among them, the numbers of neurons responsive to sound, light, and air-puff stimuli were 51, 15, and 50, respectively. Top: Heatmaps showing responses of all S-neurons to sound, light, and air puff stimuli, respectively. Each row represents the averaged activity of one S-neuron, aligned to stimulus onset (2 s) and averaged across all trials without running. The color bar on the right indicates firing rate intensity (Hz). Middle: Population PSTHs showing the response of all S-neurons to each of the three stimuli. Bottom: Running speed aligned to each stimulus onset for each neuron (each trace is the average across all trials presented during that neuron's recording). The blue bars indicate the stimulus period (duration: 5 s).

Regarding the revision of Extended Data Fig. 3: We recognize that the original examples in Extended Data Fig. 3 (each neuron shown with only 1-3 stimulus presentations) were insufficient for assessing the reliability and stimulus-locking of sensory-evoked activity. To address this, we have replaced all example neurons in this figure. The new examples are selected from our population to more clearly demonstrate reliable and stimulus-synchronized onset responses. Crucially, each example neuron is now presented with a significantly number of individual stimulus presentations (trials), thereby more clearly illustrating the neuron's response consistency and temporal relationship across repeated presentations of the same stimulus. More importantly, among the presented multiple stimulus presentations, there are a substantial number of events where the stimulus failed to successfully induce movement. This allows us to more clearly establish that the neuron's firing is a response to the sensory stimulus itself, independent of motor-related activity.

Extended Data Fig. 3 | SR-neurons and unSR-neurons respond to different sensory stimuli. a–d, Representative firing patterns of SR-neurons in response to light stimulation (LS) (a), air puff stimulation (AP) (b), sound and air puff stimulation (SS+AP) (c), sound and light stimulation (SS+LS) (d) related to running speed (red). The black dashed lines indicate events that can be used to analyze the correlation between neuronal firing and sensory stimuli, where the sensory stimuli did not induce running. **e–h,** Representative firing patterns of unSR-neurons in response to light stimulation (LS) (e), air puff stimulation (AP) (f), sound and air puff stimulation (SS+AP) (g), and air puff and light stimulation (AP + LS) (h) unrelated to running speed (red). A representative SR-

neuron that responded to sound stimulation (SS) is shown in **Fig. 4g**. A representative unSR-neuron that responded to sound stimulation (SS) is shown in **Fig. 4k**.

I understand from the Results section that the identification of SR and S neuron is assessed based on trials without any movement, but shockingly I can not find a good explanation of how S, SR and unSR neurons were categorized in the Methods section, including this information about trial selection.

We appreciate the request for clarification regarding the classification criteria. A more thorough description of how S, SR, and unSR neurons were categorized, including the crucial details of trial selection, has now been added to the Methods section. We have now added a detailed subsection titled "*Neuronal Classification*." in the Methods section to fully address this issue. The revised text is as follows:

***"Neuronal Classification.** Recorded neurons were classified based on their responses to sensory stimuli and the correlation of their spiking activity with running speed. To dissociate sensory responses from running-related activity, analyses of sensory responses were performed exclusively using trials in which sensory stimulation failed to elicit running behavior.*

1. Assessment of Sensory Responses.

Trial Selection: To isolate pure sensory responses, we analyzed only trials where sensory stimulation did not successfully evoke running. A trial was defined as a "non-running" trial if the running speed remained below a threshold of 0.5 cm/s throughout the posts-timulus-running window [0, 3.4 s].

Response Definition: For each neuron and each sensory modality (sound, light, air puff), a stimulus-aligned peri-stimulus time histogram (PSTH; bin size = 10 ms) was constructed. A neuron was considered responsive to a given stimulus if the mean firing rate within the post-stimulus response window [0, 1] s showed a significant increase

compared to the mean firing rate during a 2-s pre-stimulus baseline window (Wilcoxon signed-rank test, $P < 0.05$). The response threshold was defined as the mean baseline rate plus two times the standard deviation ($\text{baseline} + 2 \times \text{SD}$). The mean baseline rate and its standard deviation (SD) were calculated from a 2 s pre-stimulus baseline window.

Definition of S-neurons: Neurons that exhibited a significant response to at least one type of sensory stimulus were classified as sensory-related neurons (S-neurons). Neurons not meeting this criterion proceeded to the assessment for running-related activity.

2. Assessment of Running Correlation.

Trial Selection: Running-related activity was assessed using all spontaneous running events (i.e., running initiated in the absence of any sensory stimulus). Run onset was defined as the time point when running speed first exceeded a threshold of 0.5 cm/s.

Response Definition: Running onset times were aligned to time zero for each neuron. Peri-event time histograms (PETHs) were constructed (bin size = 10 ms) centered on running onset. A neuron was considered to exhibit a significant running-related change in activity if its mean firing rate during the defined pre-running window showed a significant increase compared to the mean firing rate during a quiescent baseline period (Wilcoxon signed-rank test, $P < 0.05$). The activity threshold for defining the onset of this change was set as the mean baseline firing rate plus two times its standard deviation ($\text{baseline} + 2 \times \text{SD}$). The mean baseline rate and standard deviation were calculated from periods when the mouse was not running.

3. Final Classification

Cross-Correlation Function (CCF) Analysis: For each neuron, the CCF between its firing rate and running speed was computed over a time lag range of $[-10, 10]$ s with 100 ms bins. The maximum correlation coefficient (r_{max}) and its corresponding time lag

(τ) were recorded.

Significance Testing: The statistical significance of the observed correlation was assessed using a permutation test. A null distribution was generated by performing $F = 1000$ random circular shuffles of the running speed trace relative to the firing rate. The correlation was deemed statistically significant if the absolute value of the observed $|r_{max}|$ exceeded the 95th percentile of this null distribution ($P < 0.05$).

Neurons were categorically assigned as follows: R-neurons (Running-related neurons): Neurons that were not classified as S-neurons (i.e., no significant sensory response) but showed a significant positive correlation ($P < 0.05$) between their firing rate and running speed. SR-neurons: Neurons that were classified as S-neurons (significant sensory response) and also exhibited a significant positive correlation ($P < 0.05$) between their firing rate and running speed. unSR-neurons: Neurons that were classified as S-neurons either show no significant correlation between firing rate and running speed ($P \geq 0.05$) or exhibit a correlation ($P < 0.05$) with a maximum correlation coefficient $r_{max} < 0.3$." [Methods line (1448-1509)]

By now, claiming strictly sensory evoked activity in awake behaving mice is not easy, and the default null hypothesis is widespread activity changes related to running, but also whisking, postural changes and other facial motion indices.

I am happy to see the discussion. Given that the stimuli used cause behaviors other than locomotion such as head twitching and orienting behaviors at least in the open area, it would be a natural interpretation that the activity observed is related to non-locomotion related behavioral changes.

In sum, with more and more methods and new figures coming to the table in this MS, I am not assured by the presented unconvincing example neurons that there are clear auditory, tactile and visual responses.

A convincing result for me would be a simple heatmap where each row is the activity of one S neuron (66 SR and 9 unSR cells) aligned to stimulus onset and

averaged for all trials without locomotion. This should show clear stimulus locked increase in activity across both populations.

We appreciate the constructive suggestion and have incorporated this recommendation into the revised manuscript. As we mentioned in our previous responses and have now fully implemented in the revised manuscript, we have directly and accurately followed the exact analytical approach.

Specifically, as you suggested, we have created a simple heatmap to unambiguously demonstrate the stimulus-locked responses of the S-neurons. This analysis is presented in the new Extended Data Figure 2c (As shown in the figure displayed above). This population-level analysis now clearly demonstrates the sensory-evoked responses across the S-neurons population.

We are confident that directly adopting your suggestion has significantly strengthened the core argument of our paper, and we are grateful for the guidance that led to this improvement.

Additionally, it would be even more convincing to show that these responses also occur in the absence of other movements, or at least have different temporal dynamics. However, given the high rate of sound-induced behavioral changes reported in the discussion there might be too few trials without any sensory-induced behavioral changes.

In that case it is warranted that the authors add a statement in the Discussion that they can not fully exclude that the reported sensory-related activity is (partially) related to observed movements other than locomotion.

In principle, the ideal experimental paradigm would involve demonstrating sensory responses in the complete absence of any stimulus-induced behavioral changes. As you anticipated, given the prevalence of head turns, ear pricks, orienting movements,

freezing, or other subtle postural adjustments reported in response to sound (and to a lesser extent other modalities) in our study and the relevant literature, the number of trials truly devoid of any movement is extremely limited. This limitation makes a statistically robust, purely "no-movement" analysis challenging.

Therefore, fully in line with your suggestion, we have added an explicit statement to the Discussion section acknowledging this important caveat. The revised text in the Discussion section: *"While our study establishes a functional distinction between sensory-related neurons (S-neurons) and running-related neurons (R-neurons) in TeA, some limitations should be considered. To isolate sensory-evoked activity, we strictly selected trials without overt locomotion (running speed < 0.5 cm/s). However, as noted in the preceding discussion, sensory stimuli can elicit brief orienting behaviors such as head turns or ear twitches. Although our experimental setup and analysis focused on dissociating neural activity from gross locomotion, we did not employ high-resolution, multi-parametric behavioral tracking (e.g., videography or pose estimation) to systematically quantify these subtle movements. Consequently, we cannot fully exclude the possibility that the sensory-driven activity we report is partially related to, or modulated by, such unmeasured, stimulus-induced behavioral changes. This represents a common and recognized challenge in studying neural coding in awake, behaving animals. Future work that combines dense electrophysiological or imaging recordings with comprehensive behavioral phenotyping will be crucial to further disentangle "pure" sensory signals from activity related to specific motor outputs or internal states."* **[Discussion line (982-996)]**

REVIEWER #2:

The study by He et al. aims to investigate the role of the TeA (temporal association area) as a hub for multisensory integration driving escape behavior through microcircuit computations. While the breadth and depth of the experiments—encompassing behavior, anatomy, physiology, and causal manipulations—are impressive, and some findings do indeed support the authors' claims, the overall narrative suffers from significant shortcomings.

The manuscript is plagued by numerous inconsistencies in experimental design, logical reasoning, and the interpretation of results. The narrative lacks cohesion, with critical connections between key experiments and conclusions often left underdeveloped or unclear. Instead of building a compelling and logical story, the sheer volume of data and experiments overwhelm and obscure the central message, detracting from the paper's impact.

The revised manuscript has largely improved in term of clarity. The writing remains a bit odd sometimes and the paper still contains a large number of experiments that are not all of critical importance or are even clearly redundant.

Although a few specific examples are highlighted below, these issues are pervasive throughout the manuscript. As it stands, the paper is disjointed and challenging to follow, making it difficult to discern a coherent scientific narrative. A thorough and thoughtful revision is required to address these issues before the manuscript can be considered suitable for peer review.

Below are a few examples of major concerns with the manuscript (minor comments are far too numerous to list):

The central theme of “running” as highlighted in the title and abstract feels misaligned with the actual findings, which seem to emphasize sensory-evoked escape behavior instead.

This issue has been addressed in the revised version.

Furthermore, the broader significance of the study is poorly articulated. The authors must clarify why this research is interesting and important within the larger context of behavioral neuroscience.

This issue has not been fully addressed. The introduction remains vague with many sentences stating the obvious, and lacks logical progression.

The text is frequently unclear, with superficial or anecdotal explanations. The English language must undergo significant editing for grammar, style, and overall readability to meet minimal academic standards.

These issues have been partially addressed in the revised version. But despite a significant improvement, the writing remains sometimes suboptimal with odd sentences or expressions.

There are innumerable inconsistencies between figure labeling and the main text, making interpretation challenging.

As far as I could tell, this issue has been addressed in the revised version.

Some core experimental designs lack clarity and justification. For instance, the claim that AAV-Cre (source) and DIO-virus (target) represent transsynaptic labeling is fundamentally questionable. This combination typically labels neurons projecting from the target to the source region, not synaptically connected downstream cells. This issue alone casts doubt on the interpretation and validity of conclusions drawn from these experiments.

I think the authors have missed the point raised by the reviewer. There is no question that some serotypes of AAV virus can jump one synapse and infect connected neurons downstream to the source (injection site) region. But the question is about the specificity. There is also clear evidence that these viruses also result in retrograde labelling (i.e. neurons projecting TO the source area). I suppose that should not be much of a problem when considering $_{TeA}dPAG$ neurons as I doubt that dPAG neurons project directly to the TeA (as far as I know, but should be verified) but it is clearly more of an issue when considering TeA neurons receiving inputs from other cortical areas since reciprocal cortico-cortical projections are largely expected.

The framing suggests a focus on multisensory integration, yet the experiments alternate inconsistently between unisensory and multisensory paradigms, leaving the narrative fragmented.

I think this has been clarified in the revised version.

The analysis of interneuron anatomy and connectivity feels redundant and poorly integrated into the overarching story.

This part has been moved to supplementary figures.

These problems, combined with the sheer volume of data and experiments, result in a manuscript that is difficult to follow and comprehend. The connections between experiments, results, and conclusions are weak, and the narrative lacks cohesion.

In summary, the manuscript requires substantial revision to address these conceptual, technical, and narrative flaws. Without these changes, it remains disjointed and exceedingly difficult to read as a coherent scientific manuscript.

There still are many – somewhat redundant – experiments in the paper but the overall clarity of the manuscript has been largely improved. Some technical aspects remain insufficiently described, in particular regarding the data analysis.

REVIEWER #3:

This manuscript by Li et. al presents the interesting and somewhat provocative claim that multi-sensory integration, decision-making, and generation of motor command to initiate walking all occur within an intra-laminar circuit in layer 5 of temporal association cortex (TeA). The work does a careful anatomical characterization of the projection from TeA to periaqueductal grey (PAG) and a series of manipulations of this pathway, with increasing specificity as the manuscript progresses.

However, the only readout is stimulus-evoked locomotion. Locomotion in other contexts would need to be assayed to determine specificity.

Indeed, the behavioral response could have been analyzed more carefully. It remains unclear whether the response to the sensory stimuli is acquired during the training, to what extent the sensory stimuli evoke running compared to spontaneous running, whether the mice show any sign of habituation across the repetition of the sensory stimuli.

Furthermore, analyses of locomotor responses to each of the stimuli are not sufficiently presented to interpret the effects.

This issue has been clarified.

More generally, the results are sloppily presented, with inconsistencies in presentation, a lack of justification for selection of data for illustrations, a lack of detail needed to interpret statistical comparisons, and numerous saturated anatomical images.

The presentation of the results has been largely improved although some aspects of the data analyses and statistics remain unclear.

The intersectional anatomical characterization of input in Figure 3 is valuable, but for a more specialized audience.

Not addressed but I tend to disagree with the reviewer on that point. I think these are important experiments to understand how different sensory inputs may converge to the same behavioral response in TeA.

Key experiments for necessity are missing, which would rule out other pathways, such as ACC, sensory-to-PAG, mesencephalic locomotor region, or cortico-striatal pathways. Along these lines, more broadly, many circuits and pathways are known to be involved in sensory-driven locomotion and thus the results of this paper are a modest advance that would require more careful experiments, analysis, presentation, and interpretation to be persuasive.

This point was not addressed by the authors. I believe it has more to do with the conceptual aspect of the study and its relevance. I think the reviewer is correct, but, to be fair, this criticism could apply to many other studies published in high-profile journals. Ultimately, it all comes down to a narrative problem rather than a study design issue: it remains unclear why the authors want to focus on TeA, rather than another input to dPAG.

Major concerns:

Organization of the results lacks introductory and other transitional sentences, so it is not clear how one section relates to the next. For example, first the first two sections with headers 'Open field running model.' And 'turntable running model' describe two behaviors but it is not clear where things are going. Are both going to be used later? What is the purpose of the open field model given the turntable model is described as having some advantages but no disadvantages?

The rationale for the two behavioral conditions appears clear to me in the revised version.

The framing around multisensory may create confusion, since the paper does not look at multisensory integration (e.g. stimuli/cues with components/features from multiple senses). Rather, it presents sound, light, or air puff on separate trials. As such, they are studying a set of related associations for the 3 senses in isolation. From this perspective, there are gross overstatements of the results, such as this: "With the above experiments, we have established that the TeA-dPAG pathway is the neural circuit associated with multisensory-induced movement." And "Therefore, the TeA must have the function of integrating various sensory information." This is

partially addressed by experiments in Figure 4g-l; most of the paper is about uni-sensory motor responses and framing should be adjusted, accordingly.

This is not an issue in my opinion. The authors demonstrate the multi-sensory aspect of TeA and sensory-induced escape behavior. They also suggest a form of multisensory 'integration' in the extended Figure 4, but without overemphasizing it. So, I think their claims are legitimate on this point.

The descriptions of analyses and selection of data to be presented are not clearly motivated or described to give confidence in what is being compared and why.

The analyses of the data remain poorly explained.

The results are sloppily presented, with inconsistencies in presentation, a lack of justification for selection of data for illustrations, and a lack of detail needed to interpret statistical comparisons. Numerous examples are given below in 'minor concerns' section, as well as a few 'big ones' here in the major concerns section.

The presentation of the results has been largely improved but the explanation about statistical analyses remains elusive and insufficient. As an example, in figure 1 the responses to Sound, Light and Air-puff are compared. The figure legend does not report the p values for the paired comparisons from the Bonferroni Post-hoc-Test.

It is not clear if 'Silencing TeA blocks running' experiments are done in the freely moving or head-fixed paradigm. Similarly, it is not clear in panel l and m of Figure 1 if these speed calculations are done for sound, light, or air puff. There is a similar issue with the optogenetic experiments, where it is not clear if the running under investigation is triggered by sound, light, or air puff.

This issue has been clarified.

Physiological recording for the CNO experiments in Figure 1 are lacking, so it is not clear what effect the CNO has had on TeA and other structures.

Done. See Figure 1o.

AAVretro is known to leak into neighboring cells. It would be helpful to confirm the projection from TeA layer 5a to PAG using a confirmatory method, such as pseudorabies or fluoro-gold.

The authors have performed retrograde labelling with CTB injected into the dPAG. An example picture of the injection site in the dPAG and retrogradely labelled neurons in TeA is shown only for the reviewers in the rebuttal. There seems to be some neurons labelled in the layer V of TeA but clearly much fewer compared to AAVretro labelling.

NB: I have noticed a possible error in figure 3d2 that shows two pictures of cortex (TeA), left and right, instead of cells labelled in the cortex and injection site in dPAG.

Some images are saturated (e.g. 1k,q; 2a1; 3a,c; e1b; e2g; e3a) or show signs of injection damage (2a2), raising concerns about data quality. Images should be adjusted so they are not saturated and extent of tissue damage at injection site should be discussed and/or addressed as needed with new experiments.

I do not see much (if any) difference in the anatomical images in the revised version.

It seems possible that the CNO and optogenetic experiments in Figures 1 and 2 are a ‘hammer’ and are shutting to locomotion and behavior rather broadly, rather than indicating the specific multimodal integration and transformation to a motor signal that the paper suggests. It would be helpful to determine if locomotion in other contexts (e.g. plus maze exploration) is impacted. Also, CNO injections into virus control animals would be needed to mitigate e.g. tissue damage concerns.

The revised version provides control experiments for CNO and optogenetic experiments, as well as inactivation experiments on freely-moving mice in open-field. The problem is that in fact, the later experiments show that inactivation of TeA_{dPAG} neurons or TeAdPAG neurons markedly reduces spontaneous locomotion thus questioning the overall interpretation of this circuit as being responsible for sensory-induced escape response.

In Figure 2, the responses to the 3 different stimuli should be parsed (walking to sound, light, and airpuff).

Done.

The characterization of neuronal responses in TeA to running versus sound stimuli, although intriguing, do not demonstrate much about the integration process. Furthermore, running related signals are extremely prominent across neocortex, and thus not surprising or very informative regarding this study.

Claims about multisensory integration have been removed.

The critical experiment of silencing the TeA-dPAG neurons and determining if that blocks sound/light/air-puff induced locomotion appears to be missing. This is the experiment that would test if those neurons are necessary for the behavior. Without this experiment, it seems likely that many other pathways are also involved and/or sufficient.

Done.

Minor concerns:

Figure 1f shows peak running speed and the authors claim the stimulation intensities were carefully chosen to evoke similar running speeds. It would also help to see average, rather than peak, running speed, given that this peak running speed may be insensitive to differences in evoked running patterns by the stimuli used.

OK.

How were 10 trials shown in 1c selected? What does these plots look like for different animals and single trial?

What the authors forgot to explain is that there is no 'selection' of the trials since only 10 repetitions are presented for each stimulus. See M&M (lines 1016-1018): " The three sensory stimuli were presented in a pseudorandom manner with a 25-s interstimulus spacing, and this process was repeated 10 times. Each recording session lasted 15 minutes."

It is not clear in Fig. 1d-f how the stats were done. There are 7 animals and 7 data points, suggesting each point is an animal. This should be stated. The type of post hoc test is not stated. And the F values are reported as 2,18 for degrees of freedom. How do these numbers relate to the 7 mice and 7 data points?

This problem appears to have been partially corrected. However, the authors do not report p-values for paired comparisons resulting from the Bonferroni post-hoc test. Overall, the authors' poor explanation of this makes me question their general understanding and appropriate use of statistics. Furthermore, nonparametric tests are generally recommended for small sample sizes.

Why are walk bout durations not shown for the freely-moving task in Figure 1?

Done.

It is not clear if it is the same, or different, 7 animals used in the freely moving versus head-fixed behavior.

Clarified.

This statement "However, the characteristics of the running, such as the probability, delay, and peak speed (Fig. 1j), were not significantly different between the turntable model and the open field model (Fig. 1c-f), except for the running duration." Is not justified based on what is shown in panel 'j', at least as described. Again, how the F values were determined, and what the groups were, etc., is not clear.

Resolved.

The co-localization rate reported in Fig. S1 is ambiguous; they should report both false positive and false negative rates (i.e. were there any retrogradely labeled cells that were negative for camKII/SOM/VIP/and PV? Also, why was AAVretro used for the CaMKII labeling?

I think the explanations from the authors are reasonable. They could indeed have reported the two proportions of overlap for each labeling (for instance the proportion of CaMKII+&EYFP+ / CaMKII+ and CaMKII+&EYFP+ / EYFP+ neurons). But I do not think this is an important issue for this study.

The purpose of this sentence is unclear: “Due to the scarcity of cholinergic and dopaminergic neurons in the cortex, we did not perform corresponding tests on these neuron types.” Why are cholinergic and dopaminergic neurons singled out for mention? And in fact there are cholinergic neurons in neocortex (Von Engelhardt, J., Eliava, M., Meyer, A. H., Rozov, A., & Monyer, H. (2007). Functional characterization of intrinsic cholinergic interneurons in the cortex. *Journal of Neuroscience*, 27(21), 5633-5642.).

Resolved.

Statistical quantification of significant reduction for Figure 4d-f is lacking.

Done.

Experiments in Figure 4m-r would benefit from a CNO or optogenetic manipulation. More broadly, some form of psychometric manipulation to understand how graded responses varies with graded manipulation would strengthen the manuscript.

This part has been moved to supplementary which I think is fine.